# How Does Sharpness-Aware Minimization Minimize Sharpness?

**Kaiyue Wen**
Institute for Interdisciplinary Information Sciences
Tsinghua University
wenky20@mails.tsinghua.edu.cn

**Tengyu Ma, Zhiyuan Li**
Computer Science Department
Stanford University
{tengyuma,zhiyuanli}@stanford.edu

## ABSTRACT

Sharpness-Aware Minimization (SAM) is a highly effective regularization technique for improving the generalization of deep neural networks for various settings. However, the underlying working of SAM remains elusive because of various intriguing approximations in the theoretical characterizations. SAM intends to penalize a notion of sharpness of the model but implements a computationally efficient variant; moreover, a third notion of sharpness was used for proving generalization guarantees. The subtle differences in these notions of sharpness can indeed lead to significantly different empirical results. This paper rigorously nails down the exact sharpness notion that SAM regularizes and clarifies the underlying mechanism. We also show that the two steps of approximations in the original motivation of SAM individually lead to inaccurate local conclusions, but their combination accidentally reveals the correct effect, when full-batch gradients are applied. Furthermore, we also prove that the stochastic version of SAM in fact regularizes the third notion of sharpness mentioned above, which is most likely to be the preferred notion for practical performance. The key mechanism behind this intriguing phenomenon is the alignment between the gradient and the top eigenvector of Hessian when SAM is applied.

## 1 INTRODUCTION

Modern deep nets are often overparametrized and have the capacity to fit even randomly labeled data (Zhang et al., 2016). Thus, a small training loss does not necessarily imply good generalization. Yet, standard gradient-based training algorithms such as SGD are able to find generalizable models. Recent empirical and theoretical studies suggest that generalization is well-correlated with the sharpness of the loss landscape at the learned parameter (Keskar et al., 2016; Dinh et al., 2017; Dziugaite et al., 2017; Neyshabur et al., 2017; Jiang et al., 2019). Partly motivated by these studies, Foret et al. (2021); Wu et al. (2020); Zheng et al. (2021); Norton et al. (2021) propose to penalize the sharpness of the landscape to improve the generalization. We refer this method to *Sharpness-Aware Minimization* (SAM) and focus on the version of Foret et al. (2021).

Despite its empirical success, the underlying working of SAM remains elusive because of the various intriguing approximations made in its derivation and analysis. There are three different notions of sharpness involved — SAM intends to optimize the first notion, the sharpness along the worst direction, but actually implements a computationally efficient notion, the sharpness along the direction of the gradient. But in the analysis of generalization, a third notion of sharpness is actually used to prove generalization guarantees, which admits the first notion as an upper bound. The subtle difference between the three notions can lead to very different biases (see Figure 1 for demonstration).

More concretely, let $L$ be the training loss, $x$ be the parameter and $\rho$ be the *perturbation radius*, a hyperparameter requiring tuning. The first notion corresponds to the following optimization problem (1), where we call $R_\rho^{\mathrm{Max}}(x) = L_\rho^{\mathrm{Max}}(x) - L(x)$ the *worst-direction sharpness* at $x$. SAM intends to minimize the original training loss plus the worst-direction sharpness at $x$.

$$\min_x L_\rho^{\mathrm{Max}}(x), \quad \text{where} \quad L_\rho^{\mathrm{Max}}(x) = \max_{\|v\|_2 \le 1} L(x + \rho v). \tag{1}$$

However, even evaluating $L_\rho^{\mathrm{Max}}(x)$ is computationally expensive, not to mention optimization. Thus Foret et al. (2021); Zheng et al. (2021) have introduced a second notion of sharpness, which approximates the worst-case direction in (1) by the direction of gradient, as defined below in (2). We call $R_\rho^{\mathrm{Asc}}(x) = L_\rho^{\mathrm{Asc}}(x) - L(x)$ the *ascent-direction sharpness* at $x$.

$$\min_x L_\rho^{\mathrm{Asc}}(x), \quad \text{where} \quad L_\rho^{\mathrm{Asc}}(x) = L\left(x + \rho \nabla L(x)/\|\nabla L(x)\|_2\right). \tag{2}$$

| Type of Sharpness-Aware Loss | Notation | Definition | Biases (among minimizers) |
|---|---|---|---|
| Worst-direction | $L_\rho^{\text{Max}}$ | $\max_{\|v\|_2 \le 1} L(x + \rho v)$ | $\min_x \lambda_1(\nabla^2 L(x))$ (Thm G.3) |
| Ascent-direction | $L_\rho^{\text{Asc}}$ | $L\left(x + \rho \frac{\nabla L(x)}{\|\nabla L(x)\|_2}\right)$ | $\min_x \lambda_{\min}(\nabla^2 L(x))$ (Thm G.4) |
| Average-direction | $L_\rho^{\text{Avg}}$ | $\mathbb{E}_{g \sim N(0,I)} L(x + \rho \frac{g}{\|g\|_2})$ | $\min_x \text{Tr}(\nabla^2 L(x))$ (Thm G.5) |

Table 1: **Definitions and biases of different notions of sharpness-aware loss**. The corresponding sharpness is defined as the difference between sharpness-aware loss and the original loss. Here $\lambda_1$ denotes the largest eigenvalue and $\lambda_{\min}$ denotes the smallest *non-zero* eigenvalue.

For further acceleration, Foret et al. (2021); Zheng et al. (2021) omit the gradient through other occurrence of $x$ and approximate the gradient of ascent-direction sharpness by gradient taken after one-step ascent, *i.e.*, $\nabla L_\rho^{\text{Asc}}(x) \approx \nabla L\left(x + \rho\nabla L(x)/\|\nabla L(x)\|_2\right)$ and derive the update rule of SAM, where $\eta$ is the learning rate.

$$\text{Sharpness-Aware Minimization (SAM):} \quad x(t+1) = x(t) - \eta\nabla L\left(x + \rho\nabla L(x)/\|\nabla L(x)\|_2\right). \quad (3)$$

Intriguingly, the generalization bound of SAM upperbounds the generalization error by the third notion of sharpness, called *average-direction sharpness*, $R_\rho^{\text{Avg}}(x)$ and defined formally below.

$$R_\rho^{\text{Avg}}(x) = L_\rho^{\text{Avg}}(x) - L(x), \text{ where } L_\rho^{\text{Avg}}(x) = \mathbb{E}_{g \sim N(0,I)} L\left(x + \rho g/\|g\|_2\right). \quad (4)$$

The worst-case sharpness is an upper bound of the average case sharpness and thus it is a looser bound for generalization error. In other words, according to the generalization theory in Foret et al. (2021); Wu et al. (2020) in fact motivates us to directly minimize the average case sharpness (as opposed to the worst-case sharpness that SAM intends to optimize).

In this paper, we analyze the biases introduced by penalizing these various notions of sharpness as well as the bias of SAM (Equation 3). Our analysis for SAM is performed for small perturbation radius $\rho$ and learning rate $\eta$ under the setting where the minimizers of loss form a manifold following the setup of Fehrman et al. (2020); Li et al. (2021). In particular, we make the following theoretical contributions.

1. We prove that full-batch SAM indeed minimizes worst-direction sharpness. (Theorem 4.5)
2. Surprisingly, when batch size is 1, SAM minimizes average-direction sharpness. (Theorem 5.4)
3. We provide a characterization (Theorems 4.2 and 5.3) of what a few sharpness regularizers bias towards among the minimizers (including all the three notions of the sharpness in Table 1), when the perturbation radius $\rho$ goes to zero. Surprisingly, both heuristic approximations made for SAM lead to inaccurate conclusions: (1) Minimizing worst-direction sharpness and ascent-direction sharpness induce different biases among minimizers, and (2) SAM doesn't minimize ascent-direction sharpness.

The key mechanism behind this bias of SAM is the alignment between gradient and the top eigenspace of Hessian of the original loss in the latter phase of training—the angle between them decreases gradually to the level of $O(\rho)$. It turns out that the worst-direction sharpness starts to decrease once such alignment is established (see Section 4.3). Interestingly, such an alignment is not implied by the minimization problem (2), but rather, it is an implicit property of the specific update rule of SAM. Interestingly, such an alignment property holds for SAM with full batch and SAM with batch size one, but does not necessarily hold for the mini-batch case.

## 2 RELATED WORKS

**Sharpness and Generalization.** The study on the connection between sharpness and generalization can be traced back to Hochreiter et al. (1997). Keskar et al. (2016) observe a positive correlation between the batch size, the generalization error, and the sharpness of the loss landscape when changing the batch size. Jastrzebski et al. (2017) extend this by finding a correlation between the sharpness and the ratio between learning rate to batch size. Dinh et al. (2017) show that one can easily construct networks with good generalization but with arbitrary large sharpness by reparametrization. Dziugaite et al. (2017); Neyshabur et al. (2017); Wei et al. (2019a;b) give theoretical guarantees on the generalization error using sharpness-related measures. Jiang et al. (2019) perform a large-scale empirical study on various generalization measures and show that sharpness-based measures have the highest correlation with generalization.

**Background on Sharpness-Aware Minimization.** Foret et al. (2021); Zheng et al. (2021) concurrently propose to minimize the loss at the perturbed from current parameter towards the worst direction to improve generalization. Wu et al. (2020) propose an almost identical method for a different purpose, robust generalization of adversarial training. Kwon et al. (2021) propose a different metric for SAM to fix the rescaling problem pointed out by Dinh et al. (2017). Liu et al. (2022) propose a more computationally efficient version

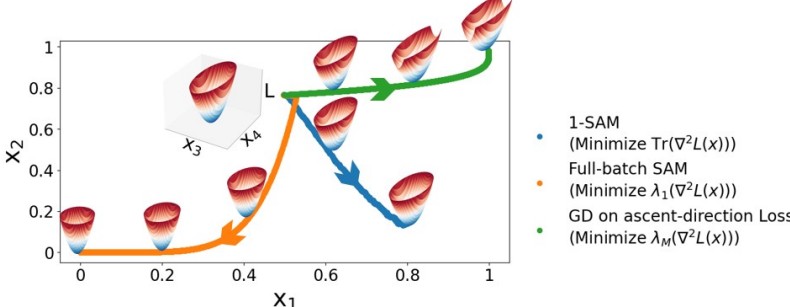

Figure 1: **Visualization of the different biases of different sharpness notions on a 4D-toy example.** Let $F_1, F_2 : \mathbb{R}^2 \to \mathbb{R}^+$ be two positive functions satisfying that $F_1 > F_2$ on $[0, 1]^2$. For $x \in \mathbb{R}^4$, consider loss $L(x) = F_1(x_1, x_2)x_3^2 + F_2(x_1, x_2)x_4^2$. The loss $L$ has a zero loss manifold $\{x_3 = x_4 = 0\}$ of codimension $M = 2$ and the two non-zero eigenvalues of $\nabla^2 L$ of any point $x$ on the manifold are $\lambda_1(\nabla^2 L(x)) = F_1(x_1, x_2)$ and $\lambda_2(\nabla^2 L(x)) = F_2(x_1, x_2)$. We test three optimization algorithms on this 4D-toy model with small learning rates. They all quickly converge to zero loss, *i.e.*, $x_3(t), x_4(t) \approx 0$, and after that $x_1(t), x_2(t)$ still change slowly, *i.e.*, moving along the zero loss manifold. We visualize the loss restricted to $(x_3, x_4)$ as the 3D shape at various $(x_1, x_2)$'s where $x_1 = x_1(t), x_2 = x_2(t)$ follows the trajectories of the three algorithms. In other words, each of the 3D surface visualize the function $g(x_3, x_4) = L(x_1(t), x_2(t), x_3, x_4)$. As our theory predicts, (1) **Full-batch SAM** (Equation 3) finds the minimizer with **the smallest top eigenvalue**, $F_1(x_1, x_2)$; (2) **GD on ascent-direction loss** $L_\rho^{\text{Asc}}$ (Equation 2) finds the minimizer with **the smallest bottom eigenvalue**, $F_2(x_1, x_2)$; (3) **1-SAM** (Equation 13) (with $L_0(x) = F_1(x_1, x_2)x_3^2$ and $L_1(x) = F_2(x_1, x_2)x_4^2$) finds the minimizer with **the smallest trace of Hessian**, $F_1(x_1, x_2) + F_2(x_1, x_2)$. See more details in Appendix B.

of SAM. Zhuang et al. (2022) proposes a variant of SAM, which improves generalization by simultaneously optimizing the surrogate gap and the sharpness-aware loss. Zhao et al. (2022) propose to improve generalization by penalizing gradient norm. Their proposed algorithm can be viewed as a generalization of SAM. Andriushchenko et al. (2022) study a variant of SAM where the step size of ascent step is $\rho$ instead of $\frac{\rho}{\|\nabla L(x)\|_2}$. They show that for a simple model this variant of SAM has a stronger regularization effect when batch size is 1 compared to the full-batch case and argue that this might be the explanation that SAM generalizes better with small batch sizes. More related works are discussed in Appendix A.

## 3    NOTATIONS AND ASSUMPTIONS

For any natural number $k$, we say a function is $\mathcal{C}^k$ if it is $k$-times continuously differentiable and is $\overline{\mathcal{C}}^k$ if its $k$th order derivatives are locally lipschitz. We say a subset of $\mathbb{R}^D$ is compact if each of its open covers has a finite subcover. It is well known that a subset of $\mathbb{R}^D$ is compact if and only if it is closed and bounded. For any positive definite symmetric matrix $A \in \mathbb{R}^{D \times D}$, define $\{\lambda_i(A), v_i(A)\}_{i \in [D]}$ as all its eigenvalues and eigenvectors satisfying $\lambda_1(A) \geq \lambda_2(A)... \geq \lambda_D(A)$ and $\|v_i(A)\|_2 = 1$. For any mapping $F$, we define $\partial F(x)$ as the Jacobian where $[\partial F(x)]_{ij} = \partial_j F_i(x)$. Thus the directional derivative of $F$ along the vector $u$ at $x$ can be written as $\partial F(x)u$. We further define the second order directional derivative of $F$ along the vectors $u$ and $v$ at $x$, $\partial^2 F(x)[u, v]$, $\partial(\partial F \cdot u)(x)v$, that is, the directional derivative of $\partial F \cdot u$ along the vector $v$ at $x$.

Given a $\mathcal{C}^1$ submanifold (Definition C.1) $\Gamma$ of $\mathbb{R}^D$ and a point $x \in \Gamma$, define $P_{x, \Gamma}$ as the projection operator onto the manifold of the normal space of $\Gamma$ at $x$ and $P_{x, \Gamma}^\perp = I_D - P_{x, \Gamma}$. We fix our initialization as $x_{\text{init}}$ and our loss function as $L : \mathbb{R}^D \to \mathbb{R}$. Given the loss function, its gradient flow is denoted by mapping $\phi : \mathbb{R}^D \times [0, \infty) \to \mathbb{R}^D$. Here, $\phi(x, \tau)$ denotes the iterate at time $\tau$ of a gradient flow starting at $x$ and is defined as the unique solution of $\phi(x, \tau) = x - \int_0^\tau \nabla L(\phi(x, t))dt, \forall x \in \mathbb{R}^D$. We further define the limiting map $\Phi$ as $\Phi(x) = \lim_{\tau \to \infty} \phi(x, \tau)$, that is, $\Phi(x)$ denotes the convergent point of the gradient flow starting from $x$. When $L(x)$ is small, $\Phi(x)$ and $x$ are near. Hence in our analysis, we regularly use $\Phi(x(t))$ as a surrogate to analyze the dynamics of $x(t)$. Lemma 3.1 is an important property of $\Phi$ from Li et al. (2021) (Lemma C.2), which is repeatedly used in our analysis. The proof is shown in Appendix F.

**Lemma 3.1.** *For any $x$ at which $\Phi$ is defined and differentiable, we have that $\partial \Phi(x) \nabla L(x) = 0$.*

Recent empirical studies have shown that there are essentially no barriers in loss landscape between different minimizers, that is, the set of minimizers are path-connected (Draxler et al., 2018; Garipov et al., 2018). Motivated by this empirical discovery, we make the assumption below following Fehrman et al. (2020); Li et al. (2021); Arora et al. (2022), which is theoretically justified by Cooper (2018) under a generic setting.

**Assumption 3.2.** *Assume loss $L : \mathbb{R}^D \to \mathbb{R}$ is $\mathcal{C}^4$, and there exists a $\mathcal{C}^2$ submanifold $\Gamma$ of $\mathbb{R}^D$ that is a $(D - M)$-dimensional for some integer $1 \leq M \leq D$, where for all $x \in \Gamma$, $x$ is a local minimizer of $L$ and $\text{rank}(\nabla^2 L(x)) = M$.*

Though our analysis for the full-batch setting is performed under the general and abstract setting, Assumption 3.2, our analysis for the stochastic setting uses a more concrete one, Setting 5.1, where we can prove that Assumption 3.2 holds. (see Theorem 5.2)

**Definition 3.3** (Attraction Set). *Let $U$ be the attraction set of $\Gamma$ under gradient flow, that is, a neighborhood of $\Gamma$ containing all points starting from which gradient flow w.r.t. loss $L$ converges to some point in $\Gamma$, or mathematically, $U \triangleq \{x \in \mathbb{R}^D | \Phi(x) \text{ exists and } \Phi(x) \in \Gamma\}$.*

It can be shown that for a minimum loss manifold, the rank of Hessian plus the dimension of the manifold is at most the environmental dimension $D$, and thus our assumption about Hessian rank essentially says the rank is maximal. Assumption 3.2 implies that $U$ is open and $\Phi$ is $\overline{\mathcal{C}}^2$ on $U$ (Arora et al., 2022, Lemma B.15).

## 4 Explicit and Implicit Bias in the Full-Batch Setting

In this section, we present our main results in the full-batch setting. Section 4.1 provides characterization of explicit bias of *worst-direction*, *ascent-dircetion*, and average-direction sharpness. In particular, we show that ascent-direction sharpness and worst-direction sharpness have different explicit biases. However, it turns out the explicit bias of ascent-direction sharpness is not the effective bias of SAM (that approximately optimizes the ascent-direction sharpness), because the particular implementation of SAM imposes additional, different biases, which is the main focus of Section 4.2. We provide our main theorem in the full-batch setting, that SAM implicitly minimizes the worst-direction sharpness, via characterizing its limiting dynamics as learning rate $\rho$ and $\eta$ goes to 0 with a Riemmanian gradient flow with respect to the top eigenvalue of the Hessian of the loss on the manifold of local minimizers. In Section 4.3, we sketch the proof of the implicit bias of SAM and identify a key property behind the implicit bias, which we call the *implicit alignment* between the gradient and the top eigenvector of the Hessian.

### 4.1 Worst- and Ascent-direction Sharpness Have Different Explicit Biases

In this subsection, we show that the explicit biases of three notions of sharpness are all different under Assumption 3.2. We first recap the heuristic derivation of ascent-direction sharpness $R_\rho^{\text{Asc}}$.

The intuition of approximating $R_\rho^{\text{Max}}$ by $R_\rho^{\text{Asc}}$ comes from the following Taylor expansions (Foret et al., 2021; Wu et al., 2020). Consider any compact set, for sufficiently small $\rho$, the following holds uniformly for all $x$ in the compact set:

$$R_\rho^{\text{Max}}(x) = \sup_{\|v\|_2 \leq 1} L(x + \rho v) - L(x) = \sup_{\|v\|_2 \leq 1} \left(\rho v^\top \nabla L(x) + \frac{\rho^2}{2} v^\top \nabla^2 L(x) v + O(\rho^3)\right), \quad (5)$$

$$R_\rho^{\text{Asc}}(x) = L\left(x + \rho \frac{\nabla L(x)}{\|\nabla L(x)\|_2}\right) - L(x) = \rho \|\nabla L(x)\|_2 + \frac{\rho^2}{2} \frac{\nabla L(x)^\top \nabla^2 L(x) \nabla L(x)}{\|\nabla L(x)\|_2^2} + O(\rho^3). \quad (6)$$

Here, the preference among the local or global minima is what we are mainly concerned with. Since $\sup_{\|v\|_2 \leq 1} v^\top \nabla L(x) = \|\nabla L(x)\|_2$ when $\|\nabla L(x)\|_2 > 0$, the leading terms in Equations 5 and 6 are both the first order term, $\rho \|\nabla L(x)\|_2$, and are the same. However, it is erroneous to think that the first order term decides the explicit bias, as the first order term $\|\nabla L(x)\|_2$ vanishes at the local minimizers of the loss $L$ and thus the second order term becomes the leading term. Any global minimizer $x$ of the original loss $L$ is an $O(\rho^2)$-approximate minimizer of the sharpness-aware loss because $\nabla L(x) = 0$. Therefore, the sharpness-aware loss needs to be of order $\rho^2$ so that we can guarantee the second-order terms in Equation 5 and/or Equation 6 to be non-trivially small. Our main result in this subsection (Theorem 4.2) gives an explicit characterization for this phenomenon. The corresponding explicit biases for each type of sharpness is given below in Definition 4.1. As we will see later, they can be derived from a general notion of *limiting regularizer* (Definition 4.3).

**Definition 4.1.** *For $x \in \mathbb{R}^D$, we define $S^{\text{Max}}(x) = \lambda_1(\nabla^2 L(x))/2$, $S^{\text{Asc}}(x) = \lambda_M(\nabla^2 L(x))/2$ and $S^{\text{Avg}}(x) = \text{Tr}(\nabla^2 L(x))/(2D)$.*

**Theorem 4.2.** *Under Assumption 3.2, let $U'$ be any bounded open set such that its closure $\overline{U'} \subseteq U$ and $\overline{U'} \cap \Gamma \subseteq \overline{U' \cap \Gamma}$. For any $\text{type} \in \{\text{Max}, \text{Asc}, \text{Avg}\}$ and any optimality gap $\Delta > 0$, there is a function $\epsilon : \mathbb{R}^+ \to \mathbb{R}^+$ with $\lim_{\rho \to 0} \epsilon(\rho) = 0$, such that for all sufficiently small $\rho > 0$ and all $u \in U'$ satisfying that*

$$L(u) + R_\rho^{\text{type}}(u) - \inf_{x \in U'} \left(L(x) + R_\rho^{\text{type}}(x)\right) \leq \Delta \rho^2, [1]$$

*it holds $L(u) - \inf_{x \in U'} L(x) \leq (\Delta + \epsilon(\rho))\rho^2$ and that $S^{\text{type}}(u) - \inf_{x \in U' \cap \Gamma} S^{\text{type}}(x) \in [-\epsilon(\rho), \Delta + \epsilon(\rho)]$.*

---

[1]We note that $R_\rho^{\text{Asc}}(x)$ is undefined when $\|\nabla L(x)\|_2 = 0$. In such cases, we set $R_\rho^{\text{Asc}}(x) = \infty$.

Theorem 4.2 suggests a sharp phase transition of the property of the solution of $\min_x L(x) + R_\rho(x)$ when the optimization error drops from $\omega(\rho^2)$ to $O(\rho^2)$. When the optimization error is larger than $\omega(\rho^2)$, no regularization effect happens and any minimizer satisfies the requirement. When the error becomes $O(\rho^2)$, there is a non-trivial restriction on the coefficients in the second-order term.

Next we give a heuristic derivation for the above defined $S^{\text{type}}$. First, for worst- and average-direction sharpness, the calculations are fairly straightforward and well-known in literature (Keskar et al., 2016; Kaur et al., 2022; Zhuang et al., 2022; Orvieto et al., 2022), and we sketch them here. In the limit of perturbation radius $\rho \to 0$, we know that the minimizer of the sharpness-aware loss will also converges to $\Gamma$, the manifold of minimizers of the original loss $L$. Thus to decide to which $x \in \Gamma$ the minimizers will converge to as $\rho \to 0$, it suffices to take Taylor expansion of $L_\rho^{\text{Asc}}$ or $L_\rho^{\text{Avg}}$ at each $x \in \Gamma$ and compare the second-order coefficients, *e.g.*, we have that $R_\rho^{\text{Avg}}(x) = \frac{\rho^2}{2D}\text{Tr}(\nabla^2 L(x)) + O(\rho^3)$ and $R_\rho^{\text{Max}}(x) = \frac{\rho^2}{2}\lambda_1(\nabla^2 L(x)) + O(\rho^3)$ by Equation 5.

However, the analysis for ascent-direction sharpness is more tricky because $R_\rho^{\text{Asc}}(x) = \infty$ for any $x \in \Gamma$ and thus is not continuous around such $x$. Thus we have to aggregate information from neighborhood to capture the explicit bias of $R_\rho$ around manifold $\Gamma$. This motivates the following definition of *limiting regularizer* which allows us to compare the regularization strength of $R_\rho$ around each point on manifold $\Gamma$ as $\rho \to 0$.

**Definition 4.3** (Limiting Regularizer). *We define the* limiting regularizer *of $\{R_\rho\}$ as the function*[2]

$$S : \Gamma \to \mathbb{R}, \quad S(x) = \lim_{\rho \to 0} \lim_{r \to 0} \inf_{\|x'-x\|_2 \le r} R_\rho(x')/\rho^2.$$

To minimize $R_\rho^{\text{Asc}}$ around $x$, we can pick $x' \to x$ satisfying that $\|\nabla L(x')\|_2 \to 0$ yet strictly being non-zero. By Equation 6, we have $R_\rho^{\text{Asc}}(x') \approx \frac{\rho^2}{2} \frac{\cdot \nabla L(x')^\top \nabla^2 L(x) \nabla L(x')}{\|\nabla L(x')\|_2^2}$. Here the crucial step of the proof is that because of Assumption 3.2, $\nabla L(x)/\|\nabla L(x)\|_2$ must almost lie in the column span of $\nabla^2 L(x)$, which implies that $\inf_{x'} \nabla L(x')^\top \nabla^2 L(x) \nabla L(x')/\|\nabla L(x')\|_2^2 \overset{\rho \to 0}{\to} \lambda_M(\nabla^2 L(x))$, where $\text{rank}(\nabla^2 L(x)) = M$ by Assumption 3.2. The above alignment property between the gradient and the column space of Hessian can be checked directly for any non-negative quadratic function. The maximal Hessian rank assumption in Assumption 3.2 ensures that this property extends to general losses.

We defer the proof of Theorem 4.2 into Appendix G.1, where we develop a sufficient condition where the notion of limiting regularizer characterizes the explicit bias of $R_\rho$ as $\rho \to 0$.

## 4.2 SAM PROVABLY DECREASES WORST-DIRECTION SHARPNESS

Though ascent-direction sharpness has different explicit bias from worst-direction sharpness, in this subsection we will show that surprisingly, SAM (Equation 3), a heuristic method designed to minimize ascent-direction sharpness, provably decreases worst-direction sharpness. The main result here is an exact characterization of the trajectory of SAM (Equation 3) via the following ordinary differential equation (ODE) (Equation 7), when learning rate $\eta$ and perturbation radius $\rho$ are small and the initialization $x(0) = x_{\text{init}}$ is in $U$, the attraction set of manifold $\Gamma$.

$$X(\tau) = X(0) - \frac{1}{2} \int_{s=0}^{\tau} P_{X(s),\Gamma}^{\perp} \nabla \lambda_1(\nabla^2 L(X(s))) ds, \quad X(0) = \Phi(x_{\text{init}}). \tag{7}$$

We assume ODE (Equation 7) has a solution till time $T_3$, that is, Equation 7 holds for all $t \le T_3$. We call the solution of Equation 7 the *limiting flow* of SAM, which is exactly the Riemannian Gradient Flow on the manifold $\Gamma$ with respect to the loss $\lambda_1(\nabla^2 L(\cdot))$. In other words, the ODE (Equation 7) is essentially a projected gradient descent algorithm with loss $\lambda_1(\nabla^2 L(\cdot))$ on the constraint set $\Gamma$ and an infinitesimal learning rate. Note $\lambda_1(\nabla^2 L(x))$ may not be differentiable at $x$ if $\lambda_1(\nabla^2 L(x)) = \lambda_2(\nabla^2 L(x))$, thus to ensure Equation 7 is well-defined, we assume there is a positive eigengap for $L$ on $\Gamma$.[3]

**Assumption 4.4.** *For all $x \in \Gamma$, there exists a positive eigengap, i.e., $\lambda_1(\nabla^2 L(x)) > \lambda_2(\nabla^2 L(x))$.*

Theorem 4.5 is the main result of this section, which is a direct combination of Theorems I.1 and I.3. The proof is deferred to Appendix I.3.

---

[2]Here we implicitly assume the zeroth and first order term varnishes, which holds for all three sharpness notions.

[3]In fact we only need to assume the positive eigengap along the solution of the ODE. If $\Gamma$ doesn't satisfy Assumption 4.4, we can simply perform the same analysis on its submanifold $\{x \in \Gamma \mid \text{eigengap is positive at } x\}$.

**Theorem 4.5** (Main). *Let $\{x(t)\}$ be the iterates of full-batch SAM (Equation 3) with $x(0) = x_{init} \in U$. Under Assumptions 3.2 and 4.4, for all $\eta, \rho$ such that $\eta \ln(1/\rho)$ and $\rho/\eta$ are sufficiently small, the dynamics of SAM can be characterized in the following two phases:*

- *Phase I: (Theorem I.1) Full-batch SAM (Equation 3) follows Gradient Flow with respect to $L$ until entering an $O(\eta\rho)$ neighborhood of the manifold $\Gamma$ in $O(\ln(1/\rho)/\eta)$ steps;*
- *Phase II: (Theorem I.3) Under a mild non-degeneracy assumption (Assumption I.2) on the initial point of phase II, full-batch SAM (Equation 3) tracks the solution $X$ of Equation 7, the Riemannian Gradient Flow with respect to the loss $\lambda_1(\nabla^2 L(\cdot))$ in an $O(\eta\rho)$ neighborhood of manifold $\Gamma$. Quantitatively, the approximation error between the iterates $x$ and the corresponding limiting flow $X$ is $O(\eta \ln(1/\rho))$, that is,*

$$\|x(\lceil T_3/(\eta\rho^2)\rceil) - X(T_3)\|_2 = O(\eta \ln(1/\rho)).$$

*Moreover, the angle between $\nabla L\left(x(\lceil \frac{T_3}{\eta\rho^2}\rceil)\right)$ and the top eigenspace of $\nabla^2 L(x(\lceil \frac{T_3}{\eta\rho^2}\rceil))$ is $O(\rho)$.*

Theorem 4.5 shows that SAM decreases the largest eigenvalue of Hessian of loss locally around the manifold of local minimizers. Phase I uses standard approximation analysis as in Hairer et al. (2008). In Phase II, as $T_3$ is arbitrary, the approximation and alignment properties hold simultaneously for all $X(t)$ along the trajectory, provided that $\eta \ln(1/\rho)$ and $\rho/\eta$ are sufficiently small. The subtlety here is that the threshold of being "sufficiently small" on $\eta \ln(1/\rho)$ and $\rho/\eta$ actually depends on $T_3$, which decreases when $T_3 \to 0$ or $\to \infty$. We defer the proof of Theorem 4.5 to Appendix I.

As a corollary of Theorem 4.5, we can also show that the largest eigenvalue of the limiting flow closely tracks the worst-direction sharpness.

**Corollary 4.6.** *In the setting of Theorem 4.5, the difference between the worst-direction sharpness of the iterates and the corresponding scaled largest eigenvalues along the limiting flow is at most $O(\eta\rho^2 \ln(1/\rho))$. That is,*

$$\left| R_\rho^{\text{Max}}(x(\lceil T_3/\eta\rho^2\rceil)) - \rho^2 \lambda_1(\nabla^2 L(X(T_3))/2 \right| = O(\eta\rho^2 \ln(1/\rho)). \tag{8}$$

Since $\eta \ln(1/\rho)$ is assumed to be sufficiently small, the error $O(\eta \ln(1/\rho) \cdot \rho^2)$ is only $o(\rho^2)$, meaning that penalizing the top eigenvalue on the manifold does lead to non-trivial reduction of worst-direction sharpness, in the sense of Section 4.1.

Hence we can show that full-batch SAM (Equation 3) provably minimizes *worst-direction sharpness* around the manifold if we additionally assume the limiting flow converges to a minimizer of the top eigenvalue of Hessian in the following Corollary 4.7.

**Corollary 4.7.** *Under Assumptions 3.2 and 4.4, define $U'$ as in Theorem 4.2 and suppose $X(\infty) = \lim_{t\to\infty} X(t)$ exists and is a minimizer of $\lambda_1(\nabla^2 L(x))$ in $U' \cap \Gamma$. Then for all $\epsilon > 0$, there exists $T_\epsilon > 0$, such that for all $\rho, \eta$ such that $\eta \ln(1/\rho)$ and $\rho/\eta$ are sufficiently small, we have that*

$$L_\rho^{\text{Max}}(x(\lceil T_\epsilon/(\eta\rho^2)\rceil)) \leq \epsilon\rho^2 + \inf_{x \in U'} L_\rho^{\text{Max}}(x).$$

We defer the proof of Corollaries 4.6 and 4.7 to Appendix I.4.

### 4.3 ANALYSIS OVERVIEW FOR SHARPNESS REDUCTION IN PHASE II OF THEOREM 4.5

Now we give an overview of the analysis for the trajectory of full-batch SAM (Equation 3) in Phase II (in Theorem 4.5). The framework of the analysis is similar to Arora et al. (2022); Lyu et al. (2022); Damian et al. (2021), where the high-level idea is to use $\Phi(x(t))$ as a proxy for $x(t)$ and study the dynamics of $\Phi(x(t))$ via Taylor expansion. Following the analysis in Arora et al. (2022) we can show Equation 9 using Taylor expansion, starting from which we will discuss the key innovation in this paper regarding implicit Hessian-gradient alignment. We defer its intuitive derivation into Appendix I.5.

$$\Phi(x(t+1)) - \Phi(x(t)) = -\frac{\eta\rho^2}{2} \partial\Phi(x(t)) \partial^2(\nabla L)(x(t)) \left[ \frac{\nabla L(x(t))}{\|\nabla L(x(t))\|_2}, \frac{\nabla L(x(t))}{\|\nabla L(x(t))\|_2} \right] + O(\eta^2\rho^2 + \eta\rho^3). \tag{9}$$

Now, to understand how $\Phi(x(t))$ moves over time, we need to understand what the direction of the RHS of Equation 9 corresponds to—we will prove that it corresponds to the Riemannian gradient of the loss function $\nabla\lambda_1(\nabla^2 L(x))$ at $x = \Phi(x(t))$. To achieve this, the key is to understand the direction $\frac{\nabla L(x(t))}{\|\nabla L(x(t))\|_2}$. It turns out that we will prove $\frac{\nabla L(x(t))}{\|\nabla L(x(t))\|_2}$ is close to the top eigenvector of the Hessian up to sign flip, that is

$\|\frac{\nabla L(x(t))}{\|\nabla L(x(t))\|_2} - s \cdot v_1(\nabla^2 L(x))\|_2 \leq O(\rho)$ for some $s \in \{-1, 1\}$. We call this phenomenon Hessian-gradient alignment and will discuss it in more detail at the end of this subsection.

Using this property, we can proceed with the derivation (detailed in Appendix I.5):

$$\Phi(x(t+1)) - \Phi(x(t)) = -\frac{\eta\rho^2}{2}\partial\Phi(\Phi(x(t)))\nabla\lambda_1(\nabla^2 L(\Phi(x(t)))) + O(\eta^2\rho^2 + \eta\rho^3), \qquad (10)$$

**Implicit Hessian-gradient Alignment.** It remains to explain why the gradient implicitly aligns to the top eigenvector of the Hessian, which is the key component of the analysis in Phase II. The proof strategy here is to first show alignment for a quadratic loss function, and then generalize its proof to general loss functions satisfying Assumption 3.2. Below we first give the formal statement of the implicit alignment on quadratic loss, Theorem 4.8 and defer the result for general case (Lemma I.19) to appendix. Note this alignment property is an implicit property of the SAM algorithm as it is not explicitly enforced by the objective that SAM is intended to minimize, $L_\rho^{Asc}$. Indeed optimizing $L_\rho^{Asc}$ would rather explicitly align gradient to the smallest non-zero eigenvector (See proofs of Theorem G.5)!

**Theorem 4.8.** *Suppose $A$ is a positive definite symmetric matrix with unique top eigenvalue. Consider running full-batch SAM (Equation 3) on loss $L(x) := \frac{1}{2}x^T Ax$ as in Equation 11 below.*

$$x(t+1) = x(t) - \eta A(x(t) + \rho Ax(t)/\|Ax(t)\|_2). \qquad (11)$$

*Then, for almost every $x(0)$, we have $x(t)$ converges in direction to $v_1(A)$ up to a sign flip and $\lim_{t\to\infty}\|x(t)\|_2 = \frac{\eta\rho\lambda_1(A)}{2-\eta\lambda_1(A)}$ with $\eta\lambda_1(A) < 1$.*

The proof of Theorem 4.8 relies on a two-phase analysis of the behavior of Equation 11, where we first show that $x(t)$ enters an invariant set from any initialization and in the second phase, we construct a potential function to show alignment. The proof is deferred to Appendix H.

Below we briefly discuss why the case with general loss is closely related to the quadratic loss case. We claim that, in the general loss function case, the analog of Equation 11 is the update rule for the gradient:

$$\nabla L(x(t+1))=\nabla L(x(t))-\eta\nabla^2 L(x(t))\big(\nabla L(x(t)) + \rho\nabla^2 L(x(t))\frac{\nabla L(x(t))}{\|\nabla L(x(t)))\|_2}\big) + O(\eta\rho^2). \qquad (12)$$

We first note that indeed in the quadratic case where $\nabla L(x) = Ax$ and $\nabla^2 L(x) = A$, Equation 12 is equivalent to Equation 11 because they only differ by a multiplicative factor $A$ on both sides. We derive its intuitive derivation into Appendix I.5.

## 5 EXPLICIT AND IMPLICIT BIASES IN THE STOCHASTIC SETTING

In practice, people usually use SAM in the stochastic mini-batch setting, and the test accuracy improves as the batch size decreases (Foret et al., 2021). Towards explaining this phenomenon, Foret et al. (2021) argue intuitively that stochastic SAM minimizes stochastic worst-direction sharpness. Given our results in Section 4, it is natural to ask if we can justify the above intuition by showing the Hessian-gradient alignment in the stochastic setting. Unfortunately, such alignment is not possible in the most general setting. Yet when the batch size is 1, we can prove rigorously in Section 5.2 that stochastic SAM minimizes *stochastic worst-direction sharpness*, which is the expectation of the worst-direction sharpness of loss over each data (defined in Section 5.1), which is the main result in this section. We stress that the stochastic worst-direction sharpness has a different explicit bias to the worst-direction sharpness, which full-batch SAM implicitly penalizes. When perturbation radius $\rho \to 0$, the former corresponds to $\text{Tr}(\nabla^2 L(\cdot))$, the same as average-direction sharpness, and the latter corresponds to $\lambda_1(\nabla^2 L(\cdot))$.

Below we start by introducing our setting for SAM with batch size 1, or 1-*SAM*. We still need Assumption 3.2 in this section. We first analyze the explicit bias of the stochastic ascent- and worst-direction sharpness in Section 5.1 via the tools developed in Section 4.1. It turns out they are all proportional to the trace of hessian as $\rho \to 0$. In Section 5.2, we show that 1-SAM penalizes the trace of Hessian. Below we formally state our setting for stochastic loss of batch size one (Setting 5.1).

**Setting 5.1.** *Let the total number of data be $M$. Let $f_k(x)$ be the model output on the $k$-th data where $f_k$ is a $C^4$-smooth function and $y_k$ be the $k$-th label, for $k = 1, \ldots, M$. We define the loss on the $k$-th data as $L_k(x) = \ell(f_k(x), y_k)$ and the total loss $L = \sum_{k=1}^{M} L_k/M$, where function $\ell(y', y)$ is $C^4$-smooth in $y'$. We also assume for any $y \in \mathbb{R}$, it holds that $\arg\min_{y'\in\mathbb{R}} \ell(y', y) = y$ and that $\frac{\partial^2 \ell(y',y)}{(\partial y')^2}|_{y'=y} > 0$. Finally, we denote the set of global minimizers of $L$ with full-rank Jacobian by $\Gamma$ and assume that it is non-empty, that is,*

$$\Gamma \triangleq \{x \in \mathbb{R}^D \mid f_k(x) = y_k, \forall k \in [M] \text{ and } \{\nabla f_k(x)\}_{k=1}^{M} \text{ are linearly independent}\} \neq \emptyset.$$

We remark that given training data (*i.e.*, $\{f_k\}_{k=1}^M$), $\Gamma$ defined above is just equal to the set of global minimizers, $\{x \in \mathbb{R}^D \mid f_k(x) = y_k, \forall k \in [M]\}$, except for a zero measure set of labels $(y_k)_{k=1}^M$ when $f_k$ are $\mathcal{C}^\infty$ smooth, by Sard's Theorem. Thus Cooper (2018) argued that the global minimizers form a differentiable manifold generically if we allow perturbation on the labels. In this work we do not make such an assumption for labels. Instead, we consider the subset of the global minimizers with full-rank Jacobian, $\Gamma$. A standard application of implicit function theorem implies that $\Gamma$ defined in Setting 5.1 is indeed a manifold. (See Theorem 5.2, whose proof is deferred into Appendix E.1)

**Theorem 5.2.** *Loss $L$, set $\Gamma$ and integer $M$ defined in Setting 5.1 satisfy Assumption 3.2.*

**1-SAM:** We use 1-*SAM* as a shorthand for SAM on a stochastic loss with batch size 1 as below Equation 13, where $k_t$ is sampled i.i.d from uniform distribution on $[M]$.

$$1\text{-SAM}: \qquad x(t+1) = x(t) - \eta \nabla L_{k_t}\left(x + \rho \nabla L_{k_t}(x)/\|\nabla L_{k_t}(x)\|_2\right). \qquad (13)$$

### 5.1 STOCHASTIC WORST-, ASCENT- AND AVERAGE- DIRECTION SHARPNESS HAVE THE SAME EXPLICIT BIASES AS AVERAGE DIRECTION SHARPNESS

Similar to the full-batch case, we use $L_{k,\rho}^{\text{Max}}, L_{k,\rho}^{\text{Asc}}, L_{k,\rho}^{\text{Avg}}$ to denote the corresponding sharpness-aware loss for $L_k$ and $R_{k,\rho}^{\text{Max}}, R_{k,\rho}^{\text{Asc}}, R_{k,\rho}^{\text{Avg}}$ to denote corresponding sharpness for $L_k$ respectively (defined as Equations 1, 2 and 4 with $L$ replaced by $L_k$). We further use *stochastic worst-, ascent- and average-direction sharpness* to denote $\mathbb{E}_k[R_{k,\rho}^{\text{Max}}], \mathbb{E}_k[R_{k,\rho}^{\text{Asc}}]$ and $\mathbb{E}_k[R_{k,\rho}^{\text{Avg}}]$. Unlike the full-batch setting, these three sharpness notions have the same explicit biases, or more precisely, they have the same limiting regularizers (up to some scaling factor).

**Theorem 5.3.** *The limiting regularizers of three notions of stochastic sharpness, denoted by $\widetilde{S}^{\text{Max}}, \widetilde{S}^{\text{Asc}}, \widetilde{S}^{\text{Avg}}$, satisfy that $\widetilde{S}^{\text{Max}}(x) = \widetilde{S}^{\text{Asc}}(x) = D \cdot \widetilde{S}^{\text{Avg}}(x) = \text{Tr}(\nabla^2 L(x))/2$. Furthermore, define $U'$ in the same way as in Theorem 4.2 . For any $\text{type} \in \{\text{Max}, \text{Asc}, \text{Avg}\}$, it holds that if for some $u \in U'$, $L(u) + \mathbb{E}_k[R_{k,\rho}^{\text{type}}(u)] \leq \inf_{x \in U'}\left(L(x) + \mathbb{E}_k[R_{k,\rho}^{\text{type}}(x)]\right) + \epsilon\rho^2$,[4] then we have that $L(u) - \inf_{x \in U'} L(x) \leq \epsilon\rho^2 + o(\rho^2)$ and that $\left|\widetilde{S}^{\text{type}}(u) - \inf_{x \in U' \cap \Gamma} \widetilde{S}^{\text{type}}(x)\right| \leq \epsilon + o(1)$.*

We defer the proof of Theorem 5.3 to Appendix G.4. Unlike in the full-batch setting where the implicit regularizer of ascent-direction sharpness and worst-direction sharpness have different explicit bias, here they are the same because there is no difference between the maximum and minimum of its non-zero eigenvalue for rank-1 Hessian of each individual loss $L_k$, and that the average of limiting regularizers is equal to the limiting regularizer of the average regularizer by definition.

### 5.2 STOCHASTIC SAM MINIMIZES AVERAGE-DIRECTION SHARPNESS

This subsection aims to show that the implicit bias of 1-SAM (Equation 13) is minimizing the average-direction sharpness for small perturbation radius $\rho$ and learning rate $\eta$, which has the same implicit bias as all three notions of stochastic sharpness do (Theorem 5.3). As an analog of the analysis in Section 4.3, which shows full-batch SAM minimizes worst-direction sharpness, analysis in this section conceptually shows that 1-SAM minimizes the stochastic worst-direction sharpness.

Mathematically, we prove that the trajectory of 1-SAM tracks the following Riemannian gradient flow (Equation 14) with respect to their limiting regularize $\text{Tr}(\nabla^2 L(\cdot))$ on the manifold for sufficiently small $\eta$ and $\rho$ and thus penalizes stochastic worst-direction sharpness (of batch size 1). We assume the ODE (Equation 14) has a solution till time $T_3$.

$$X(\tau) = X(0) - \frac{1}{2}\int_{s=0}^{\tau} P_{X(s),\Gamma}^{\perp} \nabla \text{Tr}(\nabla^2 L(X(s)))ds, \quad X(0) = \Phi(x_{\text{init}}). \qquad (14)$$

**Theorem 5.4.** *Let $\{x(t)\}$ be the iterates of 1-SAM (Equation 13) and $x(0) = x_{\text{init}} \in U$, then under Setting 5.1, for almost every $x_{\text{init}}$, for all $\eta$ and $\rho$ such that $(\eta + \rho)\ln(1/\eta\rho)$ is sufficiently small, with probability at least $1 - O(\rho)$ over the randomness of the algorithm, the dynamics of 1-SAM (Equation 13) can be split into two phases:*

- *Phase I (Theorem J.1): 1-SAM follows Gradient Flow with respect to $L$ until entering an $\tilde{O}(\eta\rho)$ neighborhood of the manifold $\Gamma$ in $O(\ln(1/\rho\eta)/\eta)$ steps;*

---

[4] We note that $R_\rho^{\text{Asc}}(x)$ is undefined when $\|\nabla L(x)\|_2 = 0$. In such cases, we set $R_\rho^{\text{Asc}}(x) = \infty$.

- *Phase II (Theorem J.2): 1-SAM tracks the solution of Equation 14, X, the Riemannian gradient flow with respect to $\mathrm{Tr}(\nabla^2 L(\cdot))$ in an $\tilde{O}(\eta\rho)$ neighborhood of manifold $\Gamma$. Quantitatively, the approximation error between the iterates $x$ and the corresponding limiting flow $X$ is $\tilde{O}(\eta^{1/2} + \rho)$, that is,*

$$\|x(\lceil T_3/(\eta\rho^2)\rceil) - X(T_3)\|_2 = \tilde{O}(\eta^{1/2} + \rho).$$

The high-level intuition for the Phase II result of Theorem 5.4 is that Hessian-gradient alignment holds true for every stochastic loss $L_k$ along the trajectory of 1-SAM and therefore by Taylor expansion (the same argument in Section 4.3), at each step $\Phi(x(t))$ moves towards the negative (Riemannian) gradient of $\lambda_1(\nabla^2 L_{k_t})$ where $k_t$ is the index of randomly sampled data, or the limiting regularizer of the worst-direction sharpness of $L_{k_t}$. Averaging over a long time, the moving direction becomes the negative (Riemannian) gradient of $\mathbb{E}_{k_t}[\lambda_1(\nabla^2 L_{k_t})]$, which is the limiting regularizer of stochastic worst-direction sharpness and equals to $\mathrm{Tr}(\nabla^2 L)$ by Theorem 5.3.

The reason that Hessian-gradient alignment holds under Setting 5.1 is that the Hessian of each stochastic loss $L_k$ at minimizers $p \in \Gamma$, $\nabla^2 L_k(p) = \frac{\partial^2 \ell(y', y_k)}{(\partial y')^2}\big|_{y'=f_k(p)} \nabla f_k(p)(\nabla f_k(p))^\top$ (Lemma J.15), is exactly rank-1, which enforces the gradient $\nabla L_k(x) \approx \nabla^2 L_k(\Phi(x))(x - \Phi(x))$ to (almost) lie in the top (which is also the unique) eigenspace of $\nabla^2 L_k(\Phi(x))$. Lemma 5.5 formally states this property.

**Lemma 5.5.** *Under Setting 5.1, for any $p \in \Gamma$ and $k \in [M]$, it holds that $\nabla f_k(p) \neq 0$ and that there is an open set $V$ containing $p$, satisfying that*

$$\forall x \in V, \nabla L_k(x) \neq 0 \implies \exists s \in \{-1, 1\}, \frac{\nabla L_k(x)}{\|\nabla L_k(x)\|} = s\frac{\nabla f_k(p)}{\|\nabla f_k(p)\|_2} + O(\|x - p\|_2).$$

Corollaries 5.6 and 5.7 below are stochastic counterparts of Corollaries 4.6 and 4.7, saying that the trace of Hessian are close to the stochastic worst-direction sharpness along the limiting flow (14), and therefore when the limiting flow converges to a local minimizer of trace of Hessian, 1-SAM (Equation 13) minimizes the average-direction sharpness. We defer the proofs of Corollaries 5.6 and 5.7 to Appendix J.4.

**Corollary 5.6.** *Under the condition of Theorem 5.4, we have that with probability $1 - O(\sqrt{\eta} + \sqrt{\rho})$, the difference between the stochastic worst-direction sharpness of the iterates and the corresponding scaled trace of Hessian along the limiting flow is at most $O\big((\eta^{1/4} + \rho^{1/4})\rho^2\big)$, that is,*

$$\left|\mathbb{E}_k[R_{k,\rho}^{\mathrm{Max}}(x(\lceil T_3/(\eta\rho^2)\rceil))] - \rho^2\mathrm{Tr}(\nabla^2 L(X(T_3)))/2\right| = O\big((\eta^{1/4} + \rho^{1/4})\rho^2\big).$$

**Corollary 5.7.** *Define $U'$ as in Theorem 4.2, suppose $X(\infty) = \lim_{t\to\infty} X(t)$ exists and is a minimizer of $\mathrm{Tr}(\nabla^2 L(x)))$ in $U' \cap \Gamma$. Then for all $\epsilon > 0$, there exists a constant $T_\epsilon > 0$, such that for all $\rho, \eta$ such that $(\eta + \rho)\ln(1/\eta\rho)$ are sufficiently small, we have that with probability $1 - O(\sqrt{\eta} + \sqrt{\rho})$,*

$$\mathbb{E}_k[L_{k,\rho}^{\mathrm{Max}}(x(\lceil T_\epsilon/(\eta\rho^2)\rceil))] \leq \epsilon\rho^2 + \inf_{x \in U'}\mathbb{E}_k[L_{k,\rho}^{\mathrm{Max}}(x)].$$

# 6 CONCLUSION

In this work, we have performed a rigorous mathematical analysis of the explicit bias of various notions of sharpness when used as regularizers and the implicit bias of the SAM algorithm. In particular, we show the explicit biases of worst-, ascent- and average-direction sharpness around the manifold of minimizers are minimizing the largest eigenvalue, the smallest nonzero eigenvalue, and the trace of Hessian of the loss function. We show that in the full-batch setting, SAM provably decreases the largest eigenvalue of Hessian, while in the stochastic setting when batch size is 1, SAM provably decreases the trace of Hessian.

The most interesting future work is to generalize the current analysis for stochastic SAM to arbitrary batch size. This is challenging because, without the alignment property which holds automatically with batch size 1, such an analysis essentially requires understanding the stationary distribution of the gradient direction along the SAM trajectory. It is also interesting to incorporate other features of modern deep learning like normalization layers, momentum, and weight decay into the current analysis.

Another interesting open question is to further bridge the difference between generalization bounds and the implicit bias of the optimizers. Currently, the generalization bounds in Wu et al. (2020); Foret et al. (2020) only work for the randomly perturbed model. Moreover, the bound depends on the average sharpness with finite $\rho$, whereas the analysis of this paper only works for infinitesimal $\rho$. It's an interesting open question whether the generalization error of the model (without perturbation) can be bounded from above by some function of the training loss, norm of the parameters, and the trace of the Hessian.

## ACKNOWLEDGEMENTS

We thank Jingzhao Zhang for helpful discussions. The authors would like to thank the support from NSF IIS 2045685.

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

CONTENTS

## A   ADDITIONAL RELATED WORKS

**Implicit Bias of Sharpness Minimization.** Recent theoretical works (Blanc et al., 2019; Damian et al., 2021; Li et al., 2021) show that SGD with label noise implicitly biased toward local minimizers with a smaller trace of Hessian under the assumption that the minimizers locally connect as a manifold. Arora et al. (2022) show that normalized GD implicitly penalizes the largest eigenvalue of the Hessian. Ma et al. (2022) argues that such sharpness reduction phenomena can also be caused by a multi-scale loss landscape. Lyu et al. (2022) show that GD with weight decay on a scale invariant loss function implicitly decreases penalize the spherical sharpness, *i.e.*, the largest eigenvalue of the Hessian evaluated at the normalized parameter.

Another line of works study the sharpness minimization effect of large learning rate assuming the (stochastic) gradient descent converges in the end of training, where the analysis is mainly based on linear stability (Wu et al., 2018; Cohen et al., 2021; Ma et al., 2021; Cohen et al., 2022). Recent theoretical analysis (Damian et al., 2022; Li et al., 2022) show that the sharpness minimization effect of large learning rate in gradient descent do not necessarily rely on the convergence assumption and linear stability via a four-phase characterization of the dynamics at the so-called Edge of Stability regime (Cohen et al., 2021).

**Comparison with concurrent work   Bartlett et al. (2022).** Bartlett et al. (2022) prove that on quadratic loss, the iterate of SAM (Equation 11) and its gradient converges to the top eigenvector of Hessian, which is almost the same as our Theorem 4.8. Assuming such alignment for a general loss, the work of Bartlett et al.

(2022) shows that the largest eigenvalue of Hessian decreases in the next step. This paper also proves such a Hessian-gradient alignment for general loss functions (Lemma I.19) and an end-to-end theorem showing that the largest eigenvalue of Hessian and worst-direction sharpness decrease along the trajectory of SAM (Theorem 4.5), which are not shown in Bartlett et al. (2022). Moreover, this paper also characterize implicit bias of stochastic SAM with batch size 1, which is minimizing the average-direction sharpness, while Bartlett et al. (2022) only considers the deterministic case.

**Comparison with Arora et al. (2022).** Our proof uses a similar framework as Arora et al. (2022). However, our analysis has its own difficulty for the following reasons. First, Arora et al. (2022) only deal with the deterministic case, while our analysis extends to stochastic SAM as well (Section 5). Second, our analysis for the deterministic case is different from that of Arora et al. (2022) in the following two aspects. First, the alignment analysis is more complicated because we have two hyperparameters, learning rate $\eta$ and perturbation radius $\rho$, while Arora et al. (2022) only needs to deal with one hyperparameter, learning rate $\eta$. Second, the mechanism of penalizing worst-direction sharpness is different, which can be seen from the dependency of the sharpness-reduction rate over learning rate $\eta$. In Arora et al. (2022), normalized GD reduces the sharpness via a second-order effect of GD and thus the sharpness is reduced by $O(\eta^2)$ per step. In our analysis, for fixed small perturbation radius $\rho$, the sharpness is reduced by $O(\rho^2\eta)$ per step, which is linear in $\eta$.

**Analyzing Discrete-time Dynamics via Continuous-time Approaches.** There is a long line of research that shows the trajectory of stochastic discrete iterations with *decaying step size* eventually tracks the solution of some ODE (see Kushner et al. (2003); Borkar et al. (2009); Duchi et al. (2018) and the reference therein). However, those results mainly focus on the convergence property of the stochastic iterates (e.g., convergence to stationary points), while we are interested in characterizing the trajectory especially when the process is running for *a long time* even after the iterate reaches the neighborhood of the manifold of stationary points.

Recently there has been an effort of modeling the discrete-time trajectory of (stochastic) gradient methods by continuous-time approximation (Su et al., 2014; Mandt et al., 2017; Li et al., 2017; 2019). Notably, Li et al. (2019) presents a general and rigorous mathematical framework to prove such continuous-time approximation. More specifically, Li et al. (2019) proves for various stochastic gradient-based methods, the discrete-time weakly converges to the continuous-time one when LR $\eta \to 0$ in $\Theta(1/\eta)$ steps. The main difference between our results with these results (e.g., Theorem 9 in Li et al. (2019)) is that we focus on a much longer training regime, *i.e.*, $T = \Theta(\eta^{-1}\rho^{-2})$ steps where the previous continuous-time approximation results no longer holds throughout the entire training. As a result, their continuous approximation is only equivalent to the Phase I dynamics in our Theorems 4.5 and 5.4 and cannot capture the dynamics of SAM in Phase II, when the sharpness-reduction implicit bias happens. The latter requires a more fine-grained analysis to capture the effects of higher-order terms in $\eta$ and $\rho$ in SAM Equation 3.

## B EXPERIMENTAL DETAILS FOR FIGURE 1

In Figure 1, we choose $F_1(x) = x_1^2 + 6x_2^2 + 8$ and $F_2(x) = 4(1-x_1)^2 + (1-x_2)^2 + 1$. The loss $L$ has a zero loss manifold $\{x = 0\}$ and the eigenvalues of its Hessian on the manifold are $F_1(x)$ and $F_2(x)$ with $F_1(x) \geq 8 > 6 \geq F_2(x)$ on $[0,1]^2$. The loss $L$ has a zero loss manifold $\{x_3 = x_4 = 0\}$ of codimension $M = 2$ and the two non-zero eigenvalues of $\nabla^2 L$ of any point $x$ on the manifold are $\lambda_1(\nabla^2 L(x)) = F_1(x_1, x_2)$ and $\lambda_2(\nabla^2 L(x)) = F_2(x_1, x_2)$.

As our theory predicts,

1. Full-batch SAM (Equation 3) finds the minimizer with the smallest top eigenvalue $F_1(x)$, which is $x_1 = 0, x_2 = 0, x_3 = 0, x_4 = 0$;
2. GD on ascent-direction loss $L_\rho^{\text{Asc}}$ (2) finds the minimizer with the smallest bottom eigenvalue, $F_2(x)$, which is $x_1 = 1, x_2 = 1, x_3 = 0, x_4 = 0$;
3. Stochastic SAM (Equation 13) (with $L_0(x,y) = F_1(x)y_0^2$, $L_1(x,y) = F_2(x)y_1^2$) finds the minimizer with smallest trace of Hessian, which is $x_1 = 4/5, x_2 = 1/7, x_3 = 0, x_4 = 0$.

## C ADDITIONAL PRELIMINARY

In this section, we introduce some additional notations and clarification before the proof.

We will first give the detailed definition of differentiable submanifold.

**Definition C.1** (Differentiable Submanifold of $\mathbb{R}^D$)**.** *We call a subset $\Gamma \subset \mathbb{R}^D$ a $\mathcal{C}^k$ submanifold of $\mathbb{R}^D$ if and only if for every $x \in \Gamma$, there exists a open neighborhood $U$ of $x$ and an invertible $\mathcal{C}^k$ map $\psi : U \to \mathbb{R}^D$, such that $\psi(\Gamma \cap U) = (\mathbb{R}^n \times \{0\}) \cap \psi(U)$.*

**Necessity of Manifold Assumption.** The connectivity of the set of local minimizers implied by the manifold assumption above allows us to take limits of perturbation radius $\rho \to 0$ while still yield interesting and insightful implicit bias results in the end-to-end analysis. So far almost all analysis of implicit bias for general model parameterizations relies on Taylor expansion, *e.g.* Blanc et al. (2019); Damian et al. (2021); Li et al. (2021); Arora et al. (2022), so does the derivation of the SAM algorithm Foret et al. (2020); Wu et al. (2020). Thus it's crucial to consider small perturbation size $\rho$. On the contrary, if the set of global minimizers are a set of discrete points, then with small perturbation radius $\rho$, implicit bias of optimizers is not sufficient to drive the iterate from global minimum to the other one.

**Implicit versus Explicit Bias.** If an algorithm or optimizer has a bias towards certain type of global/local minima of the loss over other minima of the loss, and this bias is not encoded in the loss function, then we call such bias an *implicit bias*. On the other hand, a bias emerges as solely a consequence of successfully minimizing certain regularized loss regardless of the optimizers (as long as the optimzers minimize the loss), we say such bias is an *explicit bias* of the regularized loss (or the regularizer).

As a concrete example, we will prove that full-batch SAM (Equation 3) prefers local minima with certain sharpness property. The bias stems from the particular update rule of full-batch SAM (Equation 3), and *not* all optimizers for the intended target loss function $L_\rho^{\mathrm{Asc}}$ (Equation 2) has this bias. Therefore, it's considered as an implicit bias. As an example for explicit bias, all optimizers minimizing a loss combined with $\ell_2$ regularization will prefer model with smaller parameter norm and this is considered as an explicit bias of $\ell_2$ regularization.

**Usage of $O(\cdot)$ Notation:** Our analysis assumes small $\eta$ and $\rho$ while treating all other problem-dependent parameters as constants, such as the dimension of parameter space and the maximum possible value of derivatives (of different orders) of loss function $L$ and the limit map $\Phi$. In $O(\cdot), \Omega(\cdot), o(\cdot), \omega(\cdot), \Theta(\cdot)$, we hide all the dependency related to the problem, e.g., the (unique) initialization $x_{\mathrm{init}}$, the manifold $\Gamma$, compact set $\overline{U'}$ in Theorem 4.2, and the continuous time $T_3$ in Theorems 4.5 and 5.4, and only keep the dependency on $\rho$ and $\eta$. For example, $O(f(\rho))$ is a placeholder for some function $g(\rho)$ such that there exists problem-dependent constant $C > 0, \forall \rho > 0, |g(\rho)| \leq C|f(\rho)|$. In informal equations such as Equation 31 in the proof sketch section, we are a bit more sloppy and hide dependency on $x(t)$ in $O(\cdot)$ notation as well. But these will be formally dealt with in the proofs.

**Ill-definedness of SAM with Zero Gradient.** The update rule of SAM (Equations 3 and 13) is ill-defined when the gradient is zero. However, our analysis in Appendix D shows that when the stationary point of loss $L$, $\{x \mid \nabla L(x) = 0\}$, is a zero-measure set, for any perturbation radius $\rho$, except for countably many learning rates, full-batch SAM is well-defined for almost all initialization and all steps (Theorem D.1). A similar result is shown for stochastic SAM if the stationary points of each stochastic loss form a zero-measure set (Theorem D.2). Thus SAM is generically well-defined. For the sake of rigorousness, when SAM encountering zero gradients, we modify the algorithm via replacing the ill-defined normalized gradient by an arbitrary vector with unit norm and our analysis for implicit bias of SAM still holds.

## D  WELL-DEFINEDNESS OF SAM

In this section, we discuss the well-definedness of SAM. When $\nabla L(x) = 0$, SAM (Equation 3) is not well-defined, because the normalized gradient $\frac{\nabla L(x)}{\|\nabla L(x)\|_2}$ is not well-defined. The main result of this section are Theorems D.1 and D.2, which say that (stochastic) SAM starting from random initialization only has zero probability to reach points that SAM is undefined (*i.e.*, points with zero gradient), for *all except countably many learning rates*. These results follow from Theorem D.3, which is a more general theorem also applicable to other discrete update rules as well, like SGD. Note results in this section does not rely on the manifold assumption, *i.e.*, Assumption 3.2. We end this section with a concrete example where SAM is undefined with constant probability, suggesting that the exclusion of countably many learning rates are necessary in Theorems D.1 and D.2.

**Theorem D.1.** *Consider any $\mathcal{C}^2$ loss $L$ with zero-measure stationary set $\{x \mid \nabla L(x) = 0\}$. For every $\rho > 0$, except countably many learning rates, for almost all initialization and all $t$, the iterate of full-batch SAM (Equation 3) $x(t)$ has non-zero gradient and is thus well-defined.*

**Theorem D.2.** *Consider any $\mathcal{C}^2$ losses $\{L_k\}_{k=1}^M$ with zero-measure stationary set $\{x \mid \nabla L_k(x) = 0\}$ for each $k \in [M]$. For every $\rho > 0$, except countably many learning rates $\eta$, for almost all initialization and all $t$, with probability one of the randomness of the algorithm, the iterate of stochastic SAM (Equation 13) $x(t)$ has non-zero gradient and is thus well-defined.* [5]

Before present our main theorem (Theorem D.3), we need to introduce some notations first. For a map $F$ mapping from $\mathbb{R}^D \setminus Z \to \mathbb{R}^D$, we define that $F_\eta : \mathbb{R}^D \setminus Z \to \mathbb{R}^D$ as $F_\eta(x) \triangleq x - \eta F(x)$ for any $\eta \in \mathbb{R}^+$. Given a sequence of functions $\{F^n\}_{n=1}^\infty$, we define $F_\eta^n(x) \triangleq x - \eta F^n(x)$, for any $x \in \mathbb{R}^D$. We further define that $\overline{F}_\eta^n(x) \triangleq F_\eta^n(\overline{F}_\eta^{n-1}(x))$ for any $n \geq 1$ and that $\overline{F}_\eta^0(x) = x$.

**Theorem D.3.** *Let $Z$ be a closed subset of $\mathbb{R}^D$ with zero Lebesgue measure and $\mu$ be any probability measure on $\mathbb{R}^D$ that is absolutely continuous to the Lesbegue measure. For any sequence of $\mathcal{C}^1$ functions $F^n : \mathbb{R}^D \setminus Z \to \mathbb{R}^D, n \in \mathbb{N}^+$, the following claim holds for all except countably many $\eta \in \mathbb{R}^+$:*

$$\mu\left(\{x \in \mathbb{R}^D \mid \exists n \in \mathbb{N}, \overline{F}_\eta^n(x) \in Z \text{ and } \forall 0 \leq i \leq n-1, \overline{F}_\eta^i(x) \notin Z\}\right) = 0.$$

*In other words, for almost all $\eta$ (except countably many positive numbers), iteration $x(t+1) = x(t) - \eta F(x(t)) = \overline{F}_\eta^t(x(0))$ will not enter $Z$ almost surely, provided that $x(0)$ is sampled from $\mu$.*

Theorem D.1 and Theorem D.2 follows immediately from Theorem D.3.

*Proof of Theorem D.1.* Let $F(x) = \nabla L(x + \rho \frac{\nabla L(x)}{\|\nabla L(x)\|_2})$ and $Z = \{x \in \mathbb{R}^D \mid \nabla L(x) = 0\}$. We can easily check $F$ is $\mathcal{C}^1$ on $\mathbb{R}^D \setminus Z$ and by assumption $Z$ is a zero-measure set. Applying Theorem D.3 with $F^n \equiv F$ for all $n \in \mathbb{N}^+$, we get the desired results. $\qquad\square$

*Proof of Theorem D.2.* Let $G^k(x) = \nabla L(x + \rho \frac{\nabla L_k(x)}{\|\nabla L_k(x)\|_2})$ and $Z = \cup_{k=1}^M \{x \in \mathbb{R}^D \mid \nabla L_k(x) = 0\}$. We can easily check $F_k$ is $\mathcal{C}^1$ on $\mathbb{R}^D \setminus Z$ and by assumption $Z$ is a zero-measure set. Applying Theorem D.3 with $F^n = G^{k_n}$ for all $n \in \mathbb{N}^+$ where $k_n$ is the $n$th data/batch sampled by the algorithm, we get the desired results. $\qquad\square$

Now we will turn to the proof of Theorem D.3, which is based on the following two lemmas.

**Lemma D.4.** *Let $Z$ be a closed subset of $\mathbb{R}^D$ with zero Lebesgue measure and $F : \mathbb{R}^D \setminus Z \to \mathbb{R}^D$ be a continuously differentiable function. Then except countably many $\eta \in \mathbb{R}^+$, $\{x \in \mathbb{R}^d \setminus Z \mid \det(\partial F_\eta(x)) = 0\}$ is a zero-measure set under Lebesgue measure.*

**Lemma D.5.** *Let $Z$ be a closed subset of $\mathbb{R}^D$ with zero Lebesgue measure and $H : \mathbb{R}^D \setminus Z \to \mathbb{R}^D$ be a continuously differentiable function. If $\{x \in \mathbb{R}^d \setminus Z \mid \det(\partial H(x)) = 0\}$ is a zero-measure set, then for any zero-measure set $Z'$, $H^{-1}(Z')$ is a zero-measure set.*

*Proof of Theorem D.3.* It suffices to prove that for every $N \in \mathbb{N}^+$, at most for countably many $\eta$:

$$\mu\left(\{x \in \mathbb{R}^D \mid \overline{F}_\eta^N(x) \in Z \text{ and } \forall 0 \leq i \leq N-1, \overline{F}_\eta^i(x) \notin Z\}\right) = 0. \tag{15}$$

The desired results is immediately implied by the above claim because the countable union of countable set is still countable, and countable union of zero-measure set is still zero measure.

To prove *Equation 15*, we first introduce some notations. For any $\eta > 0, 0 \leq n \leq N-1$, and $x \in \mathbb{R}^D$, we define $\overline{F}_\eta^{-(n+1)}(x) \triangleq (F_\eta^{N-n})^{-1}(\overline{F}_\eta^{-n}(x))$, where $\overline{F}_\eta^0(x) = x$. We extend the definition to set in a natural way, namely $\overline{F}_\eta^{-n}(S) \triangleq \cup_{x \in S} \overline{F}_\eta^{-n}(x)$ for any $S \subseteq \mathbb{R}^D$. Under this notation, we have that

$$F_\eta^{-N}(Z) = \mu\left(\{x \in \mathbb{R}^D \mid \overline{F}_\eta^N(x) \in Z \text{ and } \forall 0 \leq i \leq N-1, \overline{F}_\eta^i(x) \notin Z\}\right)$$

We will prove by induction. We claim that for each $0 \leq n \leq N$ except for countably many $\eta \in \mathbb{R}^+$, $\overline{F}_\eta^{-n}(Z)$ has zero Lebesgue measure. The base case $n = 0$ is by trivial as $Z$ is assumed to be zero-measure. Suppose

---

[5]Though we call Equation 13 1-SAM, but our result here applies to any batch size where $L_k$ can be regarded as the loss for $k$-th possible batch and $M$ is the number of the total number of batches.

this holds for $n$. By Lemma D.4, except countably many $\eta \in \mathbb{R}^+$, $\{x \in \mathbb{R}^d \setminus \overline{F}_\eta^{-n}(Z) \mid \det(\partial F_\eta^{N-n-1}(x)) = 0\}$ is a zero-measure set. Next by Lemma D.5 if for some $\eta \in \mathbb{R}^+$, $\{x \in \mathbb{R}^d \setminus \overline{F}_\eta^{-n}(Z) \mid \det(\partial F_\eta^{N-n-1}(x)) = 0\}$ is a zero-measure set, then $\overline{F}_\eta^{-n-1}(Z) = (F_\eta^{N-n-1})^{-1}(\overline{F}_\eta^{-n}(Z))$ is a zero-measure set. Then by induction, we know that except countably many $\eta \in \mathbb{R}^+$, for all integer $0 \le n \le N$, $\overline{F}_\eta^{-n}(Z)$ is zero-measure. Since $\mu$ is absolutely continuous to Lebesgue measure, $\mu(F_\eta^{-N}(Z)) = 0$. $\qquad\square$

We end this section with the proofs of Lemmas D.4 and D.5.

*Proof of Lemma D.4.* We use $\lambda_i(x)$ to denote that the real part of the $i$th eigenvalue of the matrix $\partial F(x)$ in the descending order. Since $\partial F(x)$ is continuous in $x$, $\lambda_i(x)$ is continuous in $x$ as well, for any $i \in [D]$, and thus $\{x \in \mathbb{R}^D \setminus Z \mid \lambda_i(x) = 1/\eta\}$ is a measurable set. Note that for a fixed $i \in [D]$, for each positive integer $n$, let $I_n$ be the set of $\eta$ where $\mu(\{x \in \mathbb{R}^D \setminus Z \mid \lambda_i(x) = 1/\eta\}) > 1/n$, then $|I_n| \le n$, because

$$\frac{|I_n|}{n} \le \sum_{\eta \in I_n} \mu((\{x \in \mathbb{R}^D \setminus Z \mid \lambda_i(x) = 1/\eta\}) \le \mu((\{x \in \mathbb{R}^D \setminus Z \mid 1/\lambda_i(x) \in I_n\}) \le 1.$$

Therefore, there are at most countably many $\eta \in \mathbb{R}^+$, such that $\mu(\{x \in \mathbb{R}^D \setminus Z \mid \lambda_i(x) = 1/\eta\}) > 0$. Further note that $\det(\partial F_\eta(x)) = 0 \iff \exists i \in [D], \lambda_i(x) = 1/\eta$, we know that there are at most countably many $\eta \in \mathbb{R}^+$, such that $\mu(\{x \in \mathbb{R}^D \setminus Z \mid \det(\partial F_\eta(x)) = 0\}) = 0$. This completes the proof. $\qquad\square$

*Proof of Lemma D.5.* Denote $\{x \in \mathbb{R}^D \setminus Z \mid \det(\partial H(x)) = 0\}$ by $Z''$, since $\det(\partial H(x))$ is continuous in $x$ as $F$ is $\mathcal{C}^1$, $Z''$ is relatively closed in $\mathbb{R}^D \setminus Z$. Since $Z'$ is a closed set, $\mathbb{R}^D \setminus (Z' \cup Z'')$ is open. Thus for all $x \in \mathbb{R}^D \setminus (Z' \cup Z'')$ with $\det(\partial H(x)) \ne 0$, there exists a open neighborhood of $x$, $U$, where for all $x' \in U$, $\det(\partial H(x')) \ne 0$, since thus $\det(\partial H(x))$ is continuous. This further implies $H$ is invertible on $U$ and its inverse $(H|_U)^{-1}$ is differentiable on $F(U)$. Therefore, $(H|_U)^{-1}$ maps any zero-measure set to a zero-measure set. In particular, $(H|_U)^{-1}(Z' \cap H(U))$ is zero measure, so is $(H)^{-1}(Z') \cap U \subset (H|_U)^{-1}(Z' \cap H(U))$. Now for every $x \in \mathbb{R}^D \setminus Z$ we take an open neighborhood $U_x \subseteq \mathbb{R}^D \setminus (Z' \cup Z'')$. Since $\mathbb{R}^D$ is a separable metric space, the open cover of $\mathbb{R}^D$, $\{U_x\}_{x \in \mathbb{R}^D \setminus (Z' \cup Z'')}$ has a countable subcover, $\{U_x\}_{x \in I}$, where $I$ is a countable set of $\mathbb{R}^D \setminus (Z' \cup Z'')$. Therefore we have that $H^{-1}(Z') \setminus (Z' \cup Z'') = H^{-1}(Z') \cap (\mathbb{R}^D \setminus (Z' \cup Z'')) = \cup_{x \in I} H^{-1}(Z') \cap U_x$ is a zero-measure set. Thus $H^{-1}(Z')$ is also zero-measure since $Z', Z''$ are both zero-measure. This completes the proof. $\qquad\square$

We end this section with an example where SAM is undefined with constant probability.

**Theorem D.6.** *For any $\eta, \rho > 0$, there is a $\mathcal{C}^2$ loss function $L : \mathbb{R} \to \mathbb{R}$ satisfying that (1) $L$ has a unique stationary point and (2) the set of initialization that makes SAM with learning rate $\eta$ and perturbation radius $\rho$ to reach the unique stationary point has positive Lebesgue measure.*

*Proof of Theorem D.6.* We first consider the case with $\rho = \eta = 1$ with

$$L(x) = \begin{cases} x^2/2 + x + 1/2, & \text{for } x \in (-\infty, -2); \\ x^4/64 + x^2/8, & \text{for } x \in [-2, 2]; \\ x^2/2 - x + 1/2, & \text{for } x \in (2, \infty). \end{cases} \tag{16}$$

We first check $L$ is indeed $\mathcal{C}^1$: $L(2) = L(-2) = 1/2$, $L'(2) = -L'(-2) = 1$ and $L''(2) = L''(-2) = 1$. Now we claim that for all $|x(0)| > 2$, $x(1) = 0$, which is a stationary point. Note that $L$ is even and monotone increasing on $[0, \infty)$, we have $\nabla L(x)/|\nabla L(x)| = \text{sign}(x)$. Thus for $|x(t)| > 1$, it holds that $|x(t) + \text{sign}(x(t)| > 2$ and therefore

$$\begin{aligned} x(t+1) &= x(t) - \eta L'(x(t) + \rho \nabla L(x)/|\nabla L(x)|) \\ &= x(t) - L'(x(t) + \text{sign}(x(t))) \\ &= x(t) - (x(t) + \text{sign}(x(t)) - \text{sign}(x(t) + \text{sign}(x(t)))) \\ &= x(t) - x(t) = 0. \end{aligned} \tag{17}$$

Now we turn to the case with arbitrary positive $\eta, \rho$. It suffices to consider $L_{\eta,\rho}(x) \triangleq \frac{\rho}{\eta} L(\frac{x}{\rho})$. We can use the calculation for $\rho = \eta = 1$ to verify for any $|x| > 2\rho$,

$$L'_{\eta,\rho}(x + \rho \text{sign}(L'_{\eta,\rho}(x))) = L'_{\eta,\rho}(x + \rho \text{sign}(x)) = \frac{1}{\eta} L(x/\rho + \text{sign}(x)) = \frac{x}{\eta},$$

namely $x - L'_{\eta,\rho}(x + \rho \operatorname{sign}(L'_{\eta,\rho}(x))) = 0$. This completes the proof. $\qquad\square$

A common (but wrong) intuition here is that, for a continuously differentiable update rule, as long as the points where the update rule is ill-defined (here it means the points with zero gradient) has zero measure, then almost surely for all initialization, gradient-based optimization algorithms like SAM will not reach exactly at any stationary point. However the above example negate this intuition. The issue here is that though a differentiable map (like SAM $x \mapsto x - \eta\nabla L(x + \rho \frac{\nabla L(x)}{\|\nabla L(x)\|_2})$) always maps the zero-measure set to zero-measure set, the preimage of zero-measure set is not necessarily zero-measure, as the map $x \mapsto x - \eta\nabla L(x + \rho \frac{\nabla L(x)}{\|\nabla L(x)\|_2})$ is not necessarily invertible. The update rule of SAM is not invertible at $0$ is exactly the reason of why preimage of $0$ has a positive measure.

## E  PROOF SETUPS

In this section we provide details of our proof setups, including notations and assumptions/settings.

We first introduce some additional notations that will be used in the proofs. For any subset $S \in \mathbb{R}^D$, we define $\operatorname{dist}(x, S) \triangleq \inf_{y \in S} \|x - y\|_2$. For any $d > 0$ and any subset $S \in \mathbb{R}^D$, we define $S^d \triangleq \{x \in \mathbb{R}^D \mid \operatorname{dist}(x, S) \leq d\}$. Our convention is to use $K$ to denote a compact set and $U$ to denote an open set.

Below we restate our main assumption in the full-batch case and related notations in Section 3. Throughout the analysis, we fix our initialization as $x_{\mathrm{init}}$, our loss function as $L : \mathbb{R}^D \to \mathbb{R}$.

**Assumption 3.2.** *Assume loss $L : \mathbb{R}^D \to \mathbb{R}$ is $\mathcal{C}^4$, and there exists a $\mathcal{C}^2$ submanifold $\Gamma$ of $\mathbb{R}^D$ that is a $(D - M)$-dimensional for some integer $1 \leq M \leq D$, where for all $x \in \Gamma$, $x$ is a local minimizer of $L$ and $\operatorname{rank}(\nabla^2 L(x)) = M$.*

**Notations for Full-Batch Setting:** Given any point $x \in \Gamma$, define $P_{x,\Gamma}$ as the projection operator onto the manifold of the normal space of $\Gamma$ at $x$ and $P_{x,\Gamma}^{\perp} = I_D - P_{x,\Gamma}$. Given the loss function $L$, its gradient flow is denoted by mapping $\phi : \mathbb{R}^D \times [0, \infty) \to \mathbb{R}^D$. Here, $\phi(x, \tau)$ denotes the iterate at time $\tau$ of a gradient flow starting at $x$ and is defined as the unique solution of $\phi(x, \tau) = x - \int_0^\tau \nabla L(\phi(x, t))dt$, $\forall x \in \mathbb{R}^D$. We further define the limiting map of $\phi(x, \cdot)$ as $\Phi(x) = \lim_{\tau \to \infty} \phi(x, \tau)$, that is, $\Phi(x)$ denotes the convergent point of the gradient flow starting from $x$. For convenience, we define $\lambda_i(x), v_i(x)$ as $\lambda_i(\nabla^2 L(\Phi(x))), v_i(\nabla^2 L(\Phi(x)))$ whenever the latter is well defined. When $x(t)$ and $\Gamma$ is clear from context, we also use $\lambda_i(t) := \lambda_i(x(t)), v_i(t) := v_i(x(t)), P_{t,\Gamma}^{\perp} := P_{\Phi(x(t)),\Gamma}^{\perp}, P_{t,\Gamma} := P_{\Phi(x(t)),\Gamma}$.

**Definition 3.3** (Attraction Set). *Let $U$ be the attraction set of $\Gamma$ under gradient flow, that is, a neighborhood of $\Gamma$ containing all points starting from which gradient flow w.r.t. loss $L$ converges to some point in $\Gamma$, or mathematically, $U \triangleq \{x \in \mathbb{R}^D | \Phi(x) \text{ exists and } \Phi(x) \in \Gamma\}$.*

Below we restate the setting for stochastic loss of batch size one in Section 5.

**Setting 5.1.** *Let the total number of data be $M$. Let $f_k(x)$ be the model output on the $k$-th data where $f_k$ is a $\mathcal{C}^4$-smooth function and $y_k$ be the $k$-th label, for $k = 1, \dots, M$. We define the loss on the $k$-th data as $L_k(x) = \ell(f_k(x), y_k)$ and the total loss $L = \sum_{k=1}^{M} L_k/M$, where function $\ell(y', y)$ is $\mathcal{C}^4$-smooth in $y'$. We also assume for any $y \in \mathbb{R}$, it holds that $\arg\min_{y' \in \mathbb{R}} \ell(y', y) = y$ and that $\frac{\partial^2 \ell(y', y)}{(\partial y')^2}|_{y'=y} > 0$. Finally, we denote the set of global minimizers of $L$ with full-rank Jacobian by $\Gamma$ and assume that it is non-empty, that is,*

$$\Gamma \triangleq \left\{ x \in \mathbb{R}^D \mid f_k(x) = y_k, \forall k \in [M] \text{ and } \{\nabla f_k(x)\}_{k=1}^{M} \text{ are linearly independent} \right\} \neq \emptyset.$$

**Theorem 5.2.** *Loss $L$, set $\Gamma$ and integer $M$ defined in Setting 5.1 satisfy Assumption 3.2.*

In our analysis, we prove our main theorems in the stochastic setting under a more general condition than Setting 5.1, which is Condition E.1 (on top of Assumption 3.2). The only usage of Setting 5.1 in the proof is Theorems 5.2 and E.2.

**Condition E.1.** *Total loss $L = \frac{1}{M}\sum_{k=1}^{M} L_k$. For each $k \in [M]$, $L_k$ is $\mathcal{C}^4$, and there exists a $(D - 1)$-dimensional $\mathcal{C}^2$-submanifold of $\mathbb{R}^D$, $\Gamma_k$, where for all $x \in \Gamma_k$, $x$ is a global minimizer of $L_k$, $L_k(x) = 0$ and $\operatorname{rank}(\nabla^2 L_k(x)) = 1$. Moreover, $\Gamma = \cap_{k=1}^{M} \Gamma_k$ for $\Gamma$ defined in Assumption 3.2.*

**Theorem E.2.** *Setting 5.1 implies Condition E.1.*

**Notations for Stochastic Setting:** Since $L_k$ is rank-1 on $\Gamma_k$ for each $k \in [M]$, we can write it as $L_k(x) = \Lambda_k(x)w_k(x)w_k^\top(x)$ for any $x \in \Gamma$, where $w_k$ is a continuous function on $\Gamma$ with pointwise unit norm. Given the loss function $L_k$, its gradient flow is denoted by mapping $\phi_k : \mathbb{R}^D \times [0, \infty) \to \mathbb{R}^D$. Here, $\phi_k(x, \tau)$ denotes the iterate at time $\tau$ of a gradient flow starting at $x$ and is defined as the unique solution of $\phi_k(x, \tau) = x - \int_0^\tau \nabla L_k(\phi_k(x, t))dt, \forall x \in \mathbb{R}^D$. We further define the limiting map $\Phi_k$ as $\Phi_k(x) = \lim_{\tau \to \infty} \phi_k(x, \tau)$, that is, $\Phi_k(x)$ denotes the convergent point of the gradient flow starting from $x$. Similar to Definition 3.3, we define $U_k = \{x \in \mathbb{R}^D | \Phi(x) \text{ exists and } \Phi_k(x) \in \Gamma_k\}$ be the attraction set of $\Gamma_i$. We have that each $U_k$ is open and $\Phi_k$ is $\overline{\mathcal{C}}^2$ on $U_k$ by Lemma B.15 in Arora et al. (2022).

**Definition E.3.** *A function $L$ is $\mu$-PL in a set $U$ iff $\forall x \in U$, $\|\nabla L(x)\|_2^2 \geq 2\mu(L(x) - \inf_{x \in U} L(x))$.*

**Definition E.4.** *The spectral 2-norm of a $k$-order tensor $X_{i_1,\dots,i_k} \in R^{d_1 \times \dots \times d_k}$ is defined as*
$$\|X\|_2 = \max_{x_i \in R^{d_i}, \|x_i\|_2 = 1} X[x_1, \dots, x_k].$$

**Lemma E.5** (Arora et al. (2022) Lemma B.2)**.** *Given any compact set $K \subseteq \Gamma$, there exist $r(K), \mu(K), \Delta(K) \in \mathbb{R}^+$ such that*

1. $K^{r(K)} \cap \Gamma$ is compact.
2. $K^{r(K)} \subset U \cap (\cap_{k \in [M]} U_k)$.
3. $L$ is $\mu(K)$-PL on $K^{r(K)}$.
4. $\inf_{x \in K^{r(K)}} (\lambda_1(\nabla^2 L(x)) - \lambda_2(\nabla^2 L(x))) \geq \Delta(K) > 0$.
5. $\inf_{x \in K^{r(K)}} \lambda_M(\nabla^2 L(x)) \geq \mu(K) > 0$.
6. $\inf_{x \in K^{r(K)}} \lambda_1(\nabla^2 L_k(x)) \geq \mu(K) > 0$.

Given compact set $K \subset \Gamma$, we further define
$$\zeta(K) = \sup_{x \in K^{r(K)}} \|\nabla^2 L(x)\|_2, \ \nu(K) = \sup_{x \in K^{r(K)}} \|\nabla^3 L(x)\|_2, \ \Upsilon(K) = \sup_{x \in K^{r(K)}} \|\nabla^4 L(x)\|_2,$$
$$\xi(K) = \sup_{x \in K^{r(K)}} \|\nabla^2 \Phi(x)\|_2, \ \chi(K) = \sup_{x,y \in K^{r(K)}} \frac{\|\nabla^2 \Phi(x) - \nabla^2 \Phi(y)\|_2}{\|x - y\|_2}.$$

Similarly, we use notations like $\zeta_k(K), \nu_k(K), \Upsilon_k(K), \xi_k(K), \chi_k(K)$ to denote the counterpart of the above quantities defined for stochastic loss $L_k$ and its limiting map $\Phi_k$ for $k \in [M]$.

**Lemma E.6** (Arora et al. (2022), Lemma B.5 and B.7)**.** *Given any compact subset $K \subset \Gamma$, let $r(K)$ be defined in Lemma E.5, there exist $0 < h(K) < r(K)$ such that*

1. $\sup_{x \in K^{h(K)}} L(x) - \inf_{x \in K^{h(K)}} L(x) \leq \frac{\mu(K)\rho^2(K)}{8}$.
2. $\forall x \in K^{h(K)}, \Phi(x) \in K^{r(K)/2}$.
3. $\forall x \in K^{h(K)}, \|x - \Phi(x)\|_2 \leq \frac{8\mu(K)^2}{\zeta(K)\nu(K)}$.
4. *The whole segment $\overline{x\Phi(x)}$ lies in $K^{r(K)}$, so does $\overline{x\Phi_k(x)}$, for any $k \in [D]$.*

The proof of the lemmas above can be found in Arora et al. (2022). Readers should note that although Arora et al. (2022) only prove these lemmas when $K$ is a special compact set (the trajectory of an ODE), all the proof does not use any property of $K$ other than it is a compact subset of $\Gamma$, and thus our Lemmas E.5 and E.6 hold for general compact subsets of $\Gamma$.

In the rest part of the appendix, for convenience we will drop the dependency on $K$ in various constants when there is no ambiguity.

### E.1 Proofs of Theorems 5.2 and E.2

*Proof of Theorem 5.2.* Define $F : \mathbb{R}^D \to \mathbb{R}^M$ as $[F(x)]_k = f_k(x), \forall k \in [M]$. Let $T_x \triangleq \text{span}(\{\nabla f_k(x)\}_{k=1}^M)$ and $T_x^\perp$ be the orthogonal complement of $T_x$ in $\mathbb{R}^D$. Now we apply implicit function theorem on $F$ at each $x \in \Gamma$. Without loss of generality (e.g. by rotating the coordinate system), we can assume that $x = 0$, $T_x = \mathbb{R}^{D-M} \times \{0\}$, and that $T_x^\perp = \{0\} \times \mathbb{R}^M$. Implicit function theorem ensures that there are two open sets $0 \in U \subset \mathbb{R}^{D-M}$ and $0 \in V \subset \mathbb{R}^M$ and an invertible $\mathcal{C}^4$ map $g : U \to V$ such that
$$F^{-1}(Y) \cap (U \times V) = \{(u, g(u)) \mid u \in U\},$$
where $Y \triangleq [y_1, \dots, y_M] \in \mathbb{R}^M$. Moreover, $\{\nabla f_k(x)\}_{k=1}^M$ is linearly independent for every $x' \in U \times V$. Thus by definition of $\Gamma$, it holds that $\Gamma \cap (U \times V) = F^{-1}(y) \cap (U \times V) = \{(u, g(u)) \mid u \in U\}$. Now for

$x = (u,v) \in U \times V$, we define $\psi : U \times V \to \mathbb{R}^D$ by $\psi(u,v) \triangleq (u, v - g(u))$. We can check that $\psi$ is $\mathcal{C}^4$ and $\psi(\Gamma \cap (U \times V)) = \{(u, v - g(u)) \mid v = g(u), u \in U)\} = \{(u, 0) \mid u \in U)\} = U \times \{0\} = (\mathbb{R}^{D-M} \times \{0\}) \cap \psi(U)$. This proves that $\Gamma$ is a $\mathcal{C}^4$ submanifold of $\mathbb{R}^D$ of dimension $D - M$. (c.f. Definition C.1) Since $\arg\min_{y' \in \mathbb{R}} \ell(y', y) = y$ for any $y \in \mathbb{R}$, it is clear that $\forall x \in \Gamma$, $x$ is a global minimizer of $L$. Finally we check the rank of Hessian of loss $L$. Note that for any $x \in \Gamma$, $\nabla^2 L_k(x) = \frac{\partial^2 \ell(y', y_k)}{(\partial y')^2}|_{y'=y_k} \nabla f_k(x)(\nabla f_k(x))^\top$ and that $\frac{\partial^2 \ell(y', y_k)}{(\partial y')^2}|_{y'=y_k} > 0$, $\operatorname{rank}(\nabla^2 L(x)) = \operatorname{rank}(\partial F(x)) = M$. This completes the proof. □

*Proof of Theorem E.2.*

1. $L = \frac{1}{M} \sum_{k=1}^M L_k$ by definition.
2. $\forall k \in [M]$, $L_k(x) = \ell(f_k(x), y_k)$ is $\mathcal{C}^4$ as $\ell$ and $f_k$ are both $\mathcal{C}^4$.
3. For any $x \in \Gamma$, by Lemma 5.5, we have $\nabla f_k(x) \neq 0$. Then there exists an open neighborhood $V_k$ such that $\Gamma \subset V_k$ and $\nabla f_k(x) \neq 0$ for any $x \in V_k, k \in [M]$. Then applying implicit function theorem as in the proof of Theorem 5.2, for any $k \in M$ there exists a $(D-1)$-dimensional $\mathcal{C}^4$-manifold $\Gamma'_k \subset V_k$, such that for any $x' \in V$, $f_k(x') = y_k$ if and only if $x' \in \Gamma'_k$. As for any $x \in \Gamma \subset V_k$, $f_k(x') = y_k$, we can infer that $\Gamma \subset \Gamma'_k$. Then $\Gamma \subset \cup_{k=1}^M \Gamma_k$.
4. For any $x \in \Gamma_k$, we have $f_k(x) = y_k$, which implies $L_k(x) = 0$. Also as $x \in V, \nabla f_k(x) \neq 0$. By Lemma J.15, we have $\operatorname{rank}(\nabla^2 L(x)) = 1$.

□

# F  PROPERTIES OF LIMITING MAP OF GRADIENT FLOW, $\Phi$

In our analysis, the property of $\Phi$ will be heavily used. In this section, we will recap some related lemmas from Arora et al. (2022), and then introduce some new lemmas for the stochastic setting with batch size one.

**Lemma F.1** (Arora et al. (2022) Lemma B.6)**.** *Given any compact set $K \subset \Gamma$, for any $x \in K^h$,*

$$\|x - \Phi(x)\|_2 \leq \int_0^\infty \|\frac{d\phi(x,t)}{dt}\|_2 \leq \sqrt{\frac{2(L(x) - L(\Phi(x)))}{\mu}} \leq \frac{\|\nabla L(x)\|_2}{\mu}.$$

**Lemma F.2.** *Given any compact set $K \subset \Gamma$, for any $x \in K^h$,*

$$\|\nabla L(x)\|_2 \leq \zeta \|x - \Phi(x)\|_2 \leq \zeta \sqrt{\frac{2(L(x) - L(\Phi(x)))}{\mu}}.$$

*Proof of Lemma F.2.* The first inequality is by Lemma E.5 and Taylor Expansion. The second inequality is by Lemma F.1. □

**Lemma F.3** (Arora et al. (2022) Lemmas B.16 and B.22)**.**
$$\partial \Phi(x) \nabla L(x) = 0, x \in U;$$
$$\partial \Phi(x) \nabla^2 L(x) \nabla L(x) = -\partial^2 \Phi(x) [\nabla L(x), \nabla L(x)], x \in U;$$
$$\partial \Phi(x) \partial^2(\nabla L)(x)[v_1, v_1] = P_{x,\Gamma}^\perp \nabla(\lambda_1(\nabla^2(L(x)))), x \in \Gamma.$$

**Lemma F.4** (Arora et al. (2022) Lemmas B.8 and B.9)**.** *Given any compact set $K \subset \Gamma$, for any $x \in K^h$,*

$$\|P_{\Phi(x),\Gamma}^\perp(x - \Phi(x))\|_2 \leq \frac{\zeta\nu}{4\mu^2} \|x - \Phi(x)\|_2^2;$$

$$\|\nabla L(x) - \nabla^2 L(\Phi(x))(x - \Phi(x))\|_2 \leq \frac{\nu}{2} \|x - \Phi(x)\|_2^2;$$

$$\left|\frac{\|\nabla L(x)\|_2}{\|\nabla^2 L(\Phi(x))(x - \Phi(x))\|_2} - 1\right| \leq \frac{2\nu}{\mu} \|x - \Phi(x)\|_2;$$

$$\frac{\nabla L(x)}{\|\nabla L(x)\|} = \frac{\nabla^2 L(\Phi(x))(x - \Phi(x))}{\|\nabla^2 L(\Phi(x))(x - \Phi(x))\|_2} + O(\frac{\nu}{\mu} \|x - \Phi(x)\|_2).$$

**Lemma F.5** (Li et al. (2021), Lemma 4.3)**.** *For $x \in \Gamma$, $\partial \Phi(x) = P_{x,\Gamma}^\perp$, the orthogonal projection matrix onto the tangent space of $\Gamma$ at $x$. Since $d\, \partial \Phi(x) \nabla^2 L(x) = 0$.*

The proof of above lemmas can be found in Arora et al. (2022); Li et al. (2021). In the following, we will first show the proof of Lemma 3.1

*Proof of Lemma 3.1.* Since $\Phi$ is defined the limit map of gradient flow, it holds that for any $t \geq 0$, $\Phi(\phi(x, t)) = \Phi(x)$. Differentiating both sides at $t = 0$, we have $\partial\Phi(\phi(x, 0))\frac{\partial\phi(x,t)}{\partial t} = 0$. The proof is completed by noting that $\frac{\partial\phi(x,t)}{\partial t} = -\nabla L(\phi(x, t))$ by definition of $\phi$. □

**Lemma F.6.** *Given any compact set $K \subset \Gamma$, for any $x \in K^h$,*

$$\|\partial\Phi(x)\nabla L_k(x)\|_2 \leq (\nu_k + \zeta_k \xi)\|x - \Phi(x)\|_2^2$$

$$\|\partial\Phi(x)\nabla^2 L_k(x)\frac{\nabla L_k(x)}{\|\nabla L_k(x)\|}\|_2 \leq (\nu_k + \zeta_k \xi)\|x - \Phi(x)\|_2$$

*Proof of Lemma F.6.* By Lemma E.6 and Taylor Expansion,

$$\begin{aligned}
\|\partial\Phi(x)\nabla L_k(x)\|_2 &\leq \|\partial\Phi(x)\nabla^2 L_k(\Phi(x))(x - \Phi(x))\|_2 + \nu_k\|x - \Phi(x)\|_2^2 \\
&\leq \|\partial\Phi(\Phi(x))\nabla^2 L_k(\Phi(x))(x - \Phi(x))\|_2 + \nu_k\|x - \Phi(x)\|_2^2 + \zeta_k \xi\|x - \Phi(x)\|_2^2 \\
&= \|P_{x,\Gamma}^\perp \partial\Phi(\Phi(x))\nabla^2 L_k(\Phi(x))(x - \Phi(x))\|_2 + \nu_k\|x - \Phi(x)\|_2^2 + \zeta_k \xi\|x - \Phi(x)\|_2^2 \\
&= (\nu_k + \zeta_k \xi)\|x - \Phi(x)\|_2^2,
\end{aligned}$$

this proves the first claim.

Again by Lemma E.5 and Taylor Expansion,

$$\begin{aligned}
\|\partial\Phi(x)\nabla^2 L_k(x)\frac{\nabla L_k(x)}{\|\nabla L_k(x)\|}\|_2 &\leq \|\partial\Phi(x)\nabla^2 L_k(\Phi(x))\frac{\nabla L_k(x)}{\|\nabla L_k(x)\|}\|_2 + \nu_k\|x - \Phi(x)\|_2 \\
&\leq \|\partial\Phi(\Phi(x))\nabla^2 L_k(\Phi(x))\frac{\nabla L_k(x)}{\|\nabla L_k(x)\|}\|_2 + (\nu_k + \zeta_k \xi)\|x - \Phi(x)\|_2 \\
&= (\nu_k + \zeta_k \xi)\|x - \Phi(x)\|_2,
\end{aligned}$$

this proves the second claim. □

**Lemma F.7.** *Suppose $x \in K^h$ and $y = x - \eta\nabla L\left(x + \rho\frac{\nabla L(x)}{\|\nabla L(x)\|}\right)$,*

$$\|y - x\|_2 \leq \eta\|\nabla L(x)\|_2 + \eta\zeta\rho$$

$$\begin{aligned}
\|\Phi(x) - \Phi(y)\|_2 &\leq \xi\eta\rho\|\nabla L(x)\|_2 + \nu\eta\rho^2 + \xi\eta^2\|\nabla L(x)\|_2^2 + \xi\zeta^2\eta^2\rho^2 \\
&\leq \zeta\xi\eta\rho\|x - \Phi(x)\|_2 + \zeta^2\xi\eta^2\|x - \Phi(x)\|_2^2 + \nu\eta\rho^2 + \xi\zeta^2\eta^2\rho^2
\end{aligned}$$

*Proof of Lemma F.7.* For sufficient small $\rho$, $x + \rho\frac{\nabla L(x)}{\|\nabla L(x)\|} \in K^r$. By Taylor Expansion,

$$\|y - x\|_2 = \eta\|\nabla L\left(x + \rho\frac{\nabla L(x)}{\|\nabla L(x)\|}\right)\|_2 \leq \eta\|\nabla L(x)\|_2 + \eta\zeta\rho$$

This further implies that for sufficiently small $\eta$ and $\rho$, $\overline{xy} \in K^r$.

Again by Taylor Expansion,

$$\|\partial\Phi(x)(y - x)\|_2 \leq \eta\|\partial\Phi(x)\nabla L(x) + \rho\partial\Phi(x)\nabla^2 L(x)\frac{\nabla L(x)}{\|\nabla L(x)\|}\|_2 + \eta\rho^2\nu/2.$$

By Lemma F.3, $\partial\Phi(x)\nabla L(x) = 0$ and $\partial\Phi(x)\nabla^2 L(x)\nabla L(x) = -\partial^2\Phi(x)[\nabla L(x), \nabla L(x)]$. Hence,

$$\begin{aligned}
\|\partial\Phi(x)(y - x)\|_2 &\leq \eta\rho\|\nabla L(x)\|_2\|\partial^2\Phi(x)\left[\frac{\nabla L(x)}{\|\nabla L(x)\|}, \frac{\nabla L(x)}{\|\nabla L(x)\|}\right]\|_2 + \eta\rho^2\nu/2 \\
&\leq \xi\eta\rho\|\nabla L(x)\|_2 + \eta\rho^2\nu/2.
\end{aligned}$$

As $\overline{xy} \in K^r$, by Taylor Expansion,

$$\|\Phi(y) - \Phi(x)\|_2 \leq \|\partial\Phi(x)(y - x)\|_2 + \xi\|y - x\|_2^2/2$$

Putting together we have
$$\|\Phi(x) - \Phi(y)\|_2 \leq \xi\eta\rho\|\nabla L(x)\|_2 + \eta\rho^2\nu + \xi\eta^2\|\nabla L(x)\|_2^2 + \xi\zeta^2\eta^2\rho^2.$$

Finally, by Lemma F.2, we have
$$\|\Phi(x) - \Phi(y)\|_2 \leq \xi\eta\rho\|\nabla L(x)\|_2 + \nu\eta\rho^2 + \xi\eta^2\|\nabla L(x)\|_2^2 + \xi\zeta^2\eta^2\rho^2$$
$$\leq \zeta\xi\eta\rho\|x - \Phi(x)\|_2 + \zeta^2\xi\eta^2\|x - \Phi(x)\|_2^2 + \nu\eta\rho^2 + \xi\zeta^2\eta^2\rho^2.$$

This completes the proof. □

**Lemma F.8.** *Suppose* $x \in K^h$ *and* $y = x - \eta\nabla L_k\left(x + \rho\frac{\nabla L_k(x)}{\|\nabla L_k(x)\|}\right)$,
$$\|y - x\|_2 \leq \eta\|\nabla L_k(x)\|_2 + \eta\zeta\rho,$$
$$\|\Phi(x) - \Phi(y)\|_2 \leq O(\eta\|\nabla L(x)\|_2^2 + \eta\rho\|\nabla L(x)\|_2 + \eta\rho^2).$$

*Proof of Lemma F.8.* For sufficient small $\rho$, $x + \rho\frac{\nabla L_k(x)}{\|\nabla L_k(x)\|} \in K^r$. By Taylor Expansion,
$$\|y - x\|_2 = \eta\|\nabla L_k\left(x + \rho\frac{\nabla L_k(x)}{\|\nabla L_k(x)\|}\right)\|_2 \leq \eta\|\nabla L_k(x)\|_2 + \eta\zeta\rho.$$

This further implies that for sufficiently small $\eta$ and $\rho$, $\overline{xy} \in K^r$.

Again by Taylor Expansion,
$$\|\partial\Phi(x)(y - x)\|_2 \leq \eta\|\partial\Phi(x)\nabla L_k(x) + \rho\partial\Phi(x)\nabla^2 L_k(x)\frac{\nabla L_k(x)}{\|\nabla L_k(x)\|}\|_2 + \eta\rho^2\nu/2.$$

We further have by Lemma F.1,
$$\|\partial\Phi(x)\nabla L_k(x)\|$$
$$\leq \|\partial\Phi(\Phi(x))\nabla L_k(x)\| + \xi\|\nabla L_k(x)\|_2\|x - \Phi(x)\|$$
$$\leq \|\partial\Phi(\Phi(x))\nabla^2 L_k(\Phi(x))(x - \Phi(x))\| + \nu\|x - \Phi(x)\|_2^2 + \zeta\xi\|x - \Phi(x)\|_2^2$$
$$\leq \frac{\nu}{\mu}\|\nabla L(x)\|_2^2 + \frac{\zeta\xi}{\mu^2}\|\nabla L(x)\|_2^2.$$

Similarly,
$$\|\rho\partial\Phi(x)\nabla^2 L_k(x)\frac{\nabla L_k(x)}{\|\nabla L_k(x)\|}\|_2$$
$$\leq \|\rho\partial\Phi(\Phi(x))\nabla^2 L_k(x)\frac{\nabla L_k(x)}{\|\nabla L_k(x)\|}\|_2 + \rho\zeta\xi\|x - \Phi(x)\|_2$$
$$\leq \|\rho\partial\Phi(\Phi(x))\nabla^2 L_k(\Phi(x))\frac{\nabla L_k(x)}{\|\nabla L_k(x)\|}\|_2 + \rho\zeta\xi\|x - \Phi(x)\|_2 + \rho\nu\|x - \Phi(x)\|_2$$
$$\leq \rho\frac{\zeta\xi}{\mu^2}\|\nabla L(x)\|_2 + \rho\frac{\nu}{\mu}\|\nabla L(x)\|_2.$$

This completes the proof. □

## G  ANALYSIS FOR EXPLICIT BIAS

Throughout this section, we assume that Assumption 3.2 holds.

### G.1  A GENERAL THEOREM FOR EXPLICIT BIAS IN THE LIMIT CASE

In this subsection we provide the proof details for section 4.1, which shows that the explicit biases of three notions of sharpness are all different, using our new mathematical tool, Theorem G.6.

**Notation for Regularizers.** Let $R_\rho : \mathbb{R}^D \to \mathbb{R} \cup \{\infty\}$ be a family of regularizers parameterized by $\rho$. If $R_\rho$ is not well-defined at some $x$, then we let $R_\rho(x) = \infty$. This convention will be useful when analyzing

ascent-direction sharpness $R_\rho^{\text{Asc}} = L_\rho^{\text{Asc}} - L$ which is not defined when $\nabla L(x) = 0$. This convention will not change the minimizers of the regularized loss. Intuitively, a regularizer should always be non-negative, but however, when far away from manifold, there are regularizers $R_\rho(x)$ of our interest that can actually be negative, *e.g.*, $R_\rho^{\text{Avg}}(x) \approx \frac{\rho^2}{2D}\text{Tr}(\nabla^2 L(x))$. Therefore we make the following assumption to allow the regularizer to be mildly negative.

**Condition G.1.** *Suppose for any bounded closed set $B \subset U$, there exists $C > 0$, such that for sufficiently small $\rho$, $\forall x \in B, R_\rho(x) \geq -C\rho^2$.*

**Definition 4.3** (Limiting Regularizer). *We define the* limiting regularizer *of $\{R_\rho\}$ as the function[6]*

$$S : \Gamma \to \mathbb{R}, \quad S(x) = \lim_{\rho \to 0} \lim_{r \to 0} \inf_{\|x' - x\|_2 \leq r} R_\rho(x')/\rho^2.$$

The high-level intuition is that we want to use the notion of limiting regularizer to capture the explicit bias of $R_\rho$ among the manifold of minimizers $\Gamma$ as $\rho \to 0$, which is decided by the second order term in the Taylor expansion, *e.g.*, Equation 5 and Equation 6. In other words, the hope is that whenever the regularized loss is optimized, the final solution should be in a neighborhood of minimizer $x$ with smallest value of limiting regularizer $S(x)$. However, such hope cannot be true without further assumptions, which motivates the following definition of *good limiting regularizer*.

**Definition G.2** (Good Limiting Regularizer). *We say the limiting regularizer $S$ of $\{R_\rho\}$ is* good *around some $x^* \in \Gamma$, if $S$ is non-negative and continuous at $x^*$ and that there is an open set $V_{x^*}$ containing $x^*$, such that for any $C > 0$, $\inf_{x':\|x'-x\|_2 \leq C\rho} R_\rho(x')/\rho^2$ converges uniformly to $S(x)$ in for all $x \in \Gamma \cap V_{x^*}$ as $\rho \to 0$.*

*In other words, a good limiting regularizer satisfy that for any $C, \epsilon > 0$, there is some $\rho_{x^*} > 0$,*

$$\forall x \in \Gamma \cap V_{x^*} \text{ and } \rho \leq \rho_{x^*}, \quad \left| S(x) - \inf_{\|x'-x\|_2 \leq C \cdot \rho} R_\rho(x')/\rho^2 \right| < \epsilon.$$

*We say the limiting regularizer $S$ is* good *on $\Gamma$, if $S$ is good around every point $x \in \Gamma$. In such case we also say $R_\rho$ admits $S$ as a good limiting regularizer on $\Gamma$.*

The intuition of the concept of a good limiting regularizer is that, the value of the regularizer should not drop too fast when moving away from a minimizer $x$ in its $O(\rho)$ neighborhood. If so, the minimizer of the regularized loss may be $\Omega(\rho)$ away from any minimizer to reduce the regularizer at the cost of increasing the original loss, which makes the limiting regularizer unable to capture the explicit bias of the regularizer. (See Appendix G.2 for a counter example) We emphasize that the conditions of good limiting regularizer is natural and covers a large family of regularizers, including worst-, ascent- and average-direction sharpness. See Theorems G.3 to G.5 below.

**Theorem G.3.** *Worst-direction sharpness $R_\rho^{\text{Max}}$ admits $\lambda_1(\nabla^2 L(\cdot))/2$ as a good limiting regularizer on $\Gamma$ and satisfies Condition G.1.*

**Theorem G.4.** *Ascent-direction sharpness $R_\rho^{\text{Asc}}$ admits $\lambda_M(\nabla^2 L(\cdot))/2$ as a good limiting regularizer on $\Gamma$ and satisfies Condition G.1.*

**Theorem G.5.** *Average-direction sharpness $R_\rho^{\text{Avg}}$ admits $\text{Tr}(\nabla^2 L(\cdot))/(2D)$ as a good limiting regularizer on $\Gamma$ and satisfies Condition G.1.*

Next we present the main mathematical tool to analyze the explicit bias of regularizers admitting good limiting regularizers, Theorem G.6.

**Theorem G.6.** *Let $U'$ be any bounded open set such that its closure $\overline{U'} \subseteq U$ and $\overline{U'} \cap \Gamma = \overline{U' \cap \Gamma}$. Then for any family of parametrized regularizers $\{R_\rho\}$ admitting a good limiting regularizer $S(x)$ on $\Gamma$ and satisfying Condition G.1, for sufficiently small $\rho$, it holds that*

$$\left| \inf_{x \in U'} \left( L(x) + R_\rho(x) \right) - \inf_{x \in U'} L(x) - \rho^2 \inf_{x \in U' \cap \Gamma} S(x) \right| \leq o(\rho^2).$$

*Moreover, for sufficiently small $\rho$, it holds uniformly for all $u \in U'$ that*

$$L(u) + R_\rho(u) \leq \inf_{x \in U'} (L(x) + R_\rho(x)) + O(\rho^2) \implies R_\rho(u)/\rho^2 - \inf_{x \in U' \cap \Gamma} S(x) \geq -o(1).$$

---

[6]Here we implicitly assume the zeroth and first order term varnishes, which holds for all three sharpness notions.

Theorem G.6 says that minimizing the regularized loss $L(u) + R_\rho(u)$ is not very different from minimizing the original loss $L(u)$ and the regularizer $R_\rho(u)$ respectively. To see this, we define the following optimality gaps

$$A(u) \triangleq L(u) + R_\rho(u) - \inf_{x \in U'} (L(x) + R_\rho(x)) \geq 0$$

$$B(u) \triangleq L(u) - \inf_{x \in U'} L(x) \geq 0$$

$$C(u) \triangleq R_\rho(u)/\rho^2 - \inf_{x \in U' \cap \Gamma} S(x),$$

and Theorem G.6 implies that $|A(u) - B(u) - \rho^2 C(u)| = o(\rho^2)$. Moreover, $A(u), B(u)$ are non-negative by definition, and $C(u) \geq -o(1)$ are almost non-negative, whenever $A(u)$ is $O(\rho^2)$-approximately optimized.

For the applications we are interested in in this paper, the good limiting regularizer $S$ can be continuously extended to the entire space $\mathbb{R}^D$. In such a case, the third optimality gap has an approximate alternative form which doesn't involve $R_\rho$, namely $S(u) - \inf_{x \in \overline{U'} \cap \Gamma} S(x)$. Corollary G.7 shows minimizing regularized loss $L(u) + R_\rho(u)$ is equivalent to minimizing the limiting regularizer, $S(u)$ around the manifold of local minimizer, $\Gamma$.

**Corollary G.7.** *Under the setting of Theorem G.6, let $\overline{S}$ be an continuous extension of $S$ to $\mathbb{R}^d$. For any optimality gap $\Delta > 0$, there is a function $\epsilon : \mathbb{R}^+ \to \mathbb{R}^+$ with $\lim_{\rho \to 0} \epsilon(\rho) = 0$, such that for all sufficiently small $\rho > 0$ and all $u \in U'$ satisfying that*

$$L(u) + R_\rho(u) - \inf_{x \in U'} \left( L(x) + R_\rho(x) \right) \leq \Delta \rho^2,$$

*it holds that $L(u) - \inf_{x \in U'} L(x) \leq (\Delta + \epsilon(\rho))\rho^2$ and that*

$$\overline{S}(u) - \inf_{x \in U' \cap \Gamma} \overline{S}(x) \in [-\epsilon(\rho), \Delta + \epsilon(\rho)].$$

### G.2 BAD LIMITING REGULARIZERS MAY NOT CAPTURE EXPLICIT BIAS

In this subsection, we provide an example where a *bad* limiting regularizer cannot capture the explicit bias of regularizer when $\rho \to 0$, to justify the necessity of Definition G.2. Here a bad limiting regularizer is a limiting regularizer which is not good.

Consider choosing $R_\rho(x) = L(x + \rho e) - L(x)$ with $\|e\| = 1$ as a fixed unit vector. We will show minimizing the regularized loss $L(x) + R_\rho(x)$ does not imply minimizing the limiting regularizer of $R_\rho(x)$ on the manifold.

By Definition 4.3 and the continuity of $R_\rho$, the limiting regularizer $S$ of $R_\rho$ is

$$\forall x \in \Gamma, \quad S(x) = \lim_{\rho \to 0} \lim_{r \to 0} \inf_{\|x' - x\|_2 \leq r} R_\rho(x')/\rho^2 = \lim_{\rho \to 0} R_\rho(x)/\rho^2 = \nabla^2 L(x)[e, e] \geq 0.$$

However, for any $x \in \Gamma$, we can choose $x' = x - \rho e$, then

$$L(x') + R_\rho(x') = L(x' + \rho e) = L(x) = 0.$$

Therefore, no matter how small $\rho$ is, minimizing $L(x) + R_\rho(x)$ can return a solution which is $\rho$-close to any point point of $\Gamma$. In other words, the explicit bias of minimizing $L(x) + R_\rho(x)$ is trivial and thus is not equivalent to minimizing the limiting regularizer $S$ on the manifold $\Gamma$.

The reason behind the inefficacy of the limiting regularizer $S$ in explaining the explicit bias of $R_\rho$ is that $S(x)$ is not a good limiting regularizer for any $x \in \Gamma$ satisfying $S(x) > 0$. To be more concrete, choose $C = 1$ and $\epsilon = S(x)/2$ in Definition G.2. For any $x \in \Gamma$ and sufficiently small $\rho > 0$, considering $x' = x - \rho e_1$, by Taylor Expansion,

$$\begin{aligned} R_\rho(x') &= L(x' + \rho e) - L(x') \\ &= \rho \langle \nabla L(x'), e \rangle + \rho^2 \nabla^2 L(x')[e, e] + o(\rho^2) \\ &= \rho \langle \nabla^2 L(x)(x' - x), e \rangle + \rho^2 \nabla^2 L(x')[e, e] + o(\rho^2) \\ &= -\rho^2 \nabla^2 L(x)[e, e] + \rho^2 \nabla^2 L(x')[e, e] + o(\rho^2) \\ &= \rho^2 e^T (\nabla^2 L(x') - \nabla^2 L(x)) e + o(\rho^2) = o(\rho^2) \end{aligned}$$

This implies $\inf_{\|x' - x\|_2 \leq C\rho} R_\rho(x') \leq R_\rho(x_1) = o(\rho^2)$. Hence,

$$S(x) - \inf_{\|x' - x\|_2 \leq C\rho} R_\rho(x')/\rho^2 \geq S(x) - o(1) > S(x)/2 = \epsilon.$$

### G.3 Proof of Theorem G.6

This subsection aims to prove Theorem G.6. We start with a few lemmas that will be used later.

**Lemma G.8.** $\Gamma = U \cap \overline{\Gamma}$.

*Proof of Lemma G.8.* For any point $x \in U \cap \overline{\Gamma}$, there exists $\{x_k\}_{k=1}^{\infty} \in \Gamma$ such that $\lim_{k \to \infty} x_k = x$. Since $x \in U$ and $\Phi$ is continuous in $U$, it holds that $\Phi$ is continuous at $x$, thus $\lim_{k \to \infty} \Phi(x_k) = \Phi(x) \in \Gamma$. However $\Phi(x_k) = x_k$ because $x_k \in \Gamma, \forall k$. Thus we know $x = \Phi(x) \in \Gamma$. Hence $U \cap \overline{\Gamma} \subset \Gamma$. The other side is clear because $\Gamma \subset U$ and $\Gamma \subset \overline{\Gamma}$. $\square$

**Lemma G.9.** *Let $U'$ be any bounded open set such that its closure $\overline{U'} \subseteq U$. If $\overline{U'} \cap \Gamma \subseteq \overline{U' \cap \Gamma}$, then $\overline{U'} \cap \Gamma = \overline{U' \cap \Gamma}$.*

*Proof of Lemma G.9.* By Lemma G.8, it holds that $\overline{U'} \cap \Gamma = \overline{U'} \cap U \cap \overline{\Gamma} = \overline{U'} \cap \overline{\Gamma}$. Note that $\overline{U' \cap \Gamma} \subseteq \overline{U'}, \overline{U' \cap \Gamma} \subseteq \overline{\Gamma}$, we have that $\overline{U' \cap \Gamma} \subseteq \overline{U'} \cap \overline{\Gamma} = \overline{U'} \cap \Gamma$, which completes the proof. $\square$

**Lemma G.10.** *Let $U'$ be any bounded open set such that its closure $\overline{U'} \subseteq U$ and $\overline{U'} \cap \Gamma \subseteq \overline{U' \cap \Gamma}$. Then for all $h_2 > 0, \exists \rho_0 > 0$ if $x \in U', \mathrm{dist}(x, \Gamma) \leq \rho_0 \Rightarrow \mathrm{dist}(x, \overline{U' \cap \Gamma}) \leq h_2$.*

*Proof of Lemma G.10.* We will prove by contradiction. Suppose there exists $h_2 > 0$ and $\{x_k\}_{k=1}^{\infty} \in U'$, such that $\lim_{k \to \infty} \mathrm{dist}(x_k, \Gamma) = 0$ but $\forall k > 0, \mathrm{dist}(x_k, \overline{U' \cap \Gamma}) \geq h_2$. Since $U'$ is bounded, $\overline{U'}$ is compact and thus $\{x_k\}_{k=1}^{\infty}$ has at least one accumulate point $x^*$ in $\overline{U'} \subseteq U$. Since $U$ is the attraction set of $\Gamma$ under gradient flow, we know that $\Phi(x^*) \in \Gamma$. Now we claim $x^* \in \Gamma$. This is because $\lim_{k \to \infty} \mathrm{dist}(x_k, \Gamma) = 0$ and thus there exists a sequence of points on $\Gamma, \{y_k\}_{k=1}^{\infty}$, where $\lim_{k \to \infty} \|x_k - y_k\| = 0$. Thus we have that $x^* = \lim_{k \to \infty} y_k = \lim_{k \to \infty} \Phi(y_k) = \Phi(x^*)$, where the last step we used that $x^* \in U$ and $\Phi$ is continuous on $U$. By the definition of $U$, $x^* \in U \iff \Phi(x^*) \in \Gamma$, thus $x^* \in \Gamma$. Then we would have $x^* \in \overline{U'} \cap \Gamma$, which is contradictory to $\mathrm{dist}(x_k, \overline{U'} \cap \Gamma) \geq \mathrm{dist}(x_k, \overline{U' \cap \Gamma}) \geq h_2, \forall k > 0$. This completes the proof. $\square$

**Lemma G.11.** *Let $U'$ be any bounded open set such that its closure $\overline{U'} \subseteq U$ and $\overline{U'} \cap \Gamma \subseteq \overline{U' \cap \Gamma}$. Then for all $h_2 > 0, \exists \rho_1 > 0$ if $x \in U', L(x) \leq \inf_{x \in U'} L(x) + \rho_1 \Rightarrow \mathrm{dist}(x, \overline{U' \cap \Gamma}) \leq h_2$.*

*Proof of Lemma G.11.* We will prove by contradiction. If there exists a list of $\rho_1, ..., \rho_k, ...$, such that $\rho_k \to 0$ and there exists $x_k \in U'$, such that $L(x_k) \leq \inf_{x \in U'} L(x) + \rho_k$ and $\mathrm{dist}(x_k, \overline{U' \cap \Gamma}) \geq h_2$. Since $U'$ is bounded, $\overline{U'}$ is compact and thus $\{x_k\}_{k=1}^{\infty}$ has at least one accumulate point $x^*$ in $\overline{U'} \subseteq U$. Since $L$ is continuous in $U$, $L(x^*) = \lim_{k \to \infty} L(x_k) = \inf_{x \in U'} L(x)$. Thus $x^*$ is a local minimizer of $L$ and thus has zero gradient, which further implies that $x^* = \Phi(x^*)$. Thus $x^* \in \overline{U'} \cap \Gamma$, which is contradictory to $\mathrm{dist}(x_k, \overline{U'} \cap \Gamma) \geq \mathrm{dist}(x_k, \overline{U' \cap \Gamma}) \geq h_2, \forall k > 0$. This completes the proof. $\square$

**Lemma G.12.** *Let $U'$ be any bounded open set such that its closure $\overline{U'} \subseteq U$ and $\overline{U'} \cap \Gamma = \overline{U' \cap \Gamma}$. Suppose regularizers $\{R_\rho\}$ admits a limiting regularizer $S$ on $\Gamma$, then*

$$\inf_{x \in U'} (L(x) + R_\rho(x)) \leq \rho^2 \inf_{x \in U' \cap \Gamma} S(x) + \inf_{x \in U'} L(x) + o(\rho^2).$$

*Proof of Lemma G.12.* First choose sufficiently small $\rho$, such that $\rho < h(\overline{U' \cap \Gamma})$. Choose an approximate minimizer of $S(x)$, $x_0 \in U' \cap \Gamma$, such that $S(x_0) \leq \inf_{x \in U' \cap \Gamma} S(x) + \rho^2$. Then by the definition of limiting regularizers (Definition 4.3) and the assumption that $U'$ is open, there exists $x_1 \in U'$ satisfying that $\|x_1 - x_0\|_2 \leq r_\rho < \rho^2$ and $R_\rho(x_1)/\rho^2 - S(x_0) \leq \rho^2$. Thus, $R_\rho(x_1) \leq \rho^2 S(x_0) + \rho^4$.

As $\|x_1 - x_0\|_2 \leq \rho^2 < h$ and $x_0 \in \overline{U' \cap \Gamma}$. This further leads to $\overline{x_0 x_1} \in \overline{U' \cap \Gamma}^h$. By Taylor expansion on $L$ at $x_0$, we would have $L(x_1) \leq L(x_0) + O(\|x_0 - x_1\|_2^2) = \inf_{x \in U' \cap \Gamma} L(x) + O(\rho^4)$. Thus it holds that

$$\inf_{x \in U'} (L(x) + R_\rho(x)) \leq L(x_1) + R_\rho(x_1) \leq \rho^2 \inf_{x \in U' \cap \Gamma} S(x) + \inf_{x \in U'} L(x) + O(\rho^4).$$

This completes the proof. $\square$

**Lemma G.13.** *Let $U'$ be any bounded open set such that its closure $\overline{U'} \subseteq U$ and $\overline{U'} \cap \Gamma = \overline{U' \cap \Gamma}$. Suppose regularizers $\{R_\rho\}$ admits a good limiting regularizer $S$ on $\Gamma$, then for all $u \in U'$,*

$$\|u - \Phi(u)\|_2 = O(\rho) \implies R_\rho(u) \geq \rho^2 \inf_{x \in U' \cap \Gamma} S(x) - o(\rho^2).$$

*Proof of Lemma G.13.* Define $r = r(K), h = h(K)$ as the constant in Lemma E.5 with $K = \overline{U' \cap \Gamma}$. Note $K$ is compact and by Lemma G.9, $K = \overline{U'} \cap \Gamma \subset \Gamma$. By Lemma E.5, we have $K^r \cap \Gamma$ is a compact set, so is $K^h \cap \Gamma$. Since $S$ is a good limiting regularizer for $\{R_\rho\}$, by Definition G.2, for any $x^* \in K^h \cap \Gamma$, there exists open neighborhood of $x^*$, $V_{x^*}$ such that for any $C, \epsilon_1 > 0$, there is a $\rho_{x^*}$ such that

$$\forall x \in V_{x^*} \text{ and } \rho \leq \rho_{x^*}, \quad \left| S(x) - \inf_{\|x'-x\|_2 \leq C \cdot \rho} R_\rho(x')/\rho^2 \right| < \epsilon_1.$$

Note that $K^h \cap \Gamma$ is compact, there exists a finite subset of $K^h \cap \Gamma$, $\{x_k\}_k$, such that $K^h \cap \Gamma \subset \cup_k V_{x_k}$. Hence for any $C, \epsilon_1 > 0$, there is some $\rho_K = \min_k \rho_{x_k} > 0$, it holds that,

$$\forall x \in K^h \cap \Gamma \text{ and } \rho \leq \rho_K, \quad \left| S(x) - \inf_{\|x'-x\|_2 \leq C \cdot \rho} R_\rho(x')/\rho^2 \right| < \epsilon_1. \tag{18}$$

We can rewrite Equation 18 as for any $C > 0$,

$$\sup_{x \in K^h \cap \Gamma} \left| S(x) - \inf_{\|x'-x\|_2 \leq C \cdot \rho} R_\rho(x')/\rho^2 \right| = o(1), \quad \text{as } \rho \to 0. \tag{19}$$

As $u \in U' \subseteq U$, we have that $\Phi(u) \in \Gamma$. If $\|u - \Phi(u)\|_2 = O(\rho)$, then $\mathrm{dist}(u, \Gamma) \leq O(\rho)$. By Lemma G.10, we have that $\mathrm{dist}(u, K) = o(1)$. This further implies $\mathrm{dist}(\Phi(u), K) \leq \mathrm{dist}(u, K) + \mathrm{dist}(\Phi(u), u) = o(1)$. Hence we have that $\Phi(u) \in K^h \cap \Gamma$ for sufficiently small $\rho$. Thus we can pick $x = \Phi(u)$ in Equation 19 and $C$ sufficiently large, which yields that

$$\rho^2 S(\Phi(u)) \leq \inf_{\|u'-\Phi(u)\|_2 \leq O(\rho)} R_\rho(u') + o(\rho^2) \leq R_\rho(u) + o(\rho^2), \tag{20}$$

where the last step is because $\|u - \Phi(u)\|_2 = O(\rho)$. On the other hand, we have that

$$S(\Phi(u)) \geq \inf_{x \in U' \cap \Gamma} S(x) - o(1). \tag{21}$$

as $S$ is continuous on $\Gamma$ and $\mathrm{dist}(\overline{U' \cap \Gamma}, \Phi(u)) = o(1)$. Combining Equations 20 and 21, we have $R_\rho(u) \geq \rho^2 \inf_{x \in U' \cap \Gamma} S(x) - o(\rho^2)$. $\square$

*Proof of Theorem G.6.* We will first lower bound $L(x) + R_\rho(x)$ for $x \in U'$. Suppose $C_{U'}$ is the constant in Condition G.1. Define $C_1 = \sqrt{2 \frac{C_{U'} + \inf_{x \in \overline{U'} \cap \Gamma} S(x) + 1}{\mu}}$. We discuss by cases. For sufficiently small $\rho$,

1. If $x \notin K^h$, then by Lemma G.11, $L(x)$ is lower bounded by a positive constant.
2. If $x \in K^h$ and $\|x - \Phi(x)\|_2 \geq C_1 \rho$, then by Lemma F.1,

$$L(x) \geq \frac{\mu \|x - \Phi(x)\|_2^2}{2} \geq (C_{U'} + \inf_{x \in U' \cap \Gamma} S(x) + 1)\rho^2.$$

   This implies $L(x) + R_\rho(x) \geq (\inf_{x \in U' \cap \Gamma} S(x) + 1)\rho^2 + \inf_{x \in U'} L(x)$.
3. If $\|x - \Phi(x)\|_2 \leq C_1 \rho$, by Lemma G.13, $R_\rho(x) \geq \rho^2 \inf_{x \in U' \cap \Gamma} S(x) - o(\rho^2)$, hence

$$L(x) + R_\rho(x) + o(\rho^2) \geq \inf_{x \in U' \cap \Gamma} S(x)\rho^2 + \inf_{x \in U'} L(x).$$

Concluding the three cases, we have

$$\inf_{x \in U'} (L(x) + R_\rho(x)) \geq \inf_{x \in U' \cap \Gamma} L(x) + \inf_{x \in U' \cap \Gamma} S(x)\rho^2 - o(\rho^2).$$

By Lemma G.12, we have that

$$\inf_{x \in U'} (L(x) + R_\rho(x)) \leq \rho^2 \inf_{x \in U' \cap \Gamma} S(x) + \inf_{x \in U' \cap \Gamma} L(x) + o(\rho^2).$$

Combining the above two inequalities, we prove the main statement of Theorem G.6.

Furthermore, if $L(u) + R_\rho(u) \leq \inf_{x \in U'}(L(x) + R_\rho(x)) + O(\rho^2)$, then by the main statement and Condition G.1, we have that

$$L(u) - \inf_{x \in U'} L(x) \leq \inf_{x \in U'} (L(x) + R_\rho(x)) - R_\rho(u) - \inf_{x \in U'} L(x) + O(\rho^2)$$

$$\leq \rho^2 \inf_{x \in U' \cap \Gamma} S(x) + C\rho^2 + O(\rho^2) = O(\rho^2).$$

Then by Lemma G.11, we have $u \in (\overline{U'} \cap \Gamma)^h$ for sufficiently small $\rho$. By Lemma F.1, we have $\|u - \Phi(u)\|_2 = O(\rho)$. By Lemma G.13, we have $R_\rho(u) \geq \rho^2 \inf_{x \in U' \cap \Gamma} S(x) - o(\rho^2)$. $\square$

### G.4 Proofs of Corollary G.7

*Proof of Corollary G.7.* Since $L(u) + R_\rho(u) - \inf_{x \in U'} \big(L(x) + R_\rho(x)\big) \le \Delta \rho^2 = O(\rho^2)$, by Theorem G.6, we have that
$$L(u) - \inf_{x \in U'} L(x) \le (\Delta + o(1))\rho^2,$$
and
$$R_\rho(x) - \inf_{x \in U' \cap \Gamma} S(x) \in [-o(1), \Delta + o(1)].$$

Thus it suffices to show $R_\rho(x) - \overline{S}(x) = o(\rho^2)$. Since $L(u) - \inf_{x \in U'} L(x) \le (\Delta + \epsilon(\rho))\rho^2 = o(1)$, by Lemma G.11, we know $\mathrm{dist}(x, \overline{U' \cap \Gamma}) = o(1)$. Thus by Lemma F.1, $\|x - \Phi(x)\| = o(1)$, which implies that $\rho^2 S(\Phi(x)) - o(\rho^2) \le R_\rho(u)$. Since $\overline{S}$ is an continuous extension, $\overline{S}(x) - \overline{S}(\Phi(x)) = \overline{S}(x) - S(\Phi(x)) = O(\|x - \Phi(x)\|_2) = o(1)$. Thus we conclude that $\overline{S}(x) \le \overline{S}(\Phi(x)) \le \inf_{x \in U' \cap \Gamma} S(x) + \Delta + o(1)$. On the other hand, $\overline{S}(x) \ge S(\Phi(x)) - o(1) \ge \inf_{x \in U' \cap \Gamma} S(x) - o(1)$, where the last step we use the fact that $\mathrm{dist}(x, \overline{U' \cap \Gamma}) = o(1)$. This completes the proof. $\qquad\square$

### G.5 Limiting Regularizers For Different Notions Of Sharpness

*Proof of Theorem G.3.*

1. We will first verify Condition G.1. For fixed compact set $B \subset U$, as $\|\nabla^3 L(x)\|_2$ is continuous, there exists constant $\nu$, such that $\forall x \in B^1, \|\nabla^3 L(x)\|_2 \le \nu$. Then by Taylor Expansion,
$$\begin{aligned} R_\rho^{\mathrm{Max}}(x) &= \max_{\|v\|_2 \le 1} L(x + \rho v) - L(x) \\ &\ge \max_{\|v\|_2 \le 1} \big(\rho\langle \nabla L(x), v\rangle + \rho^2 v^T \nabla^2 L(x) v/2\big) - \nu\rho^3/6 \\ &\ge -\nu\rho^3/6 \,. \end{aligned}$$

2. Now we verify $S^{\mathrm{Max}}(x) = \lambda_1(\nabla^2 L(\cdot))/2$ is the limiting regularizer of $R_\rho^{\mathrm{Max}}$. Let $x$ be any point in $\Gamma$, by continuity of $R_\rho^{\mathrm{Max}}$,
$$\lim_{\rho \to 0} \lim_{r \to 0} \inf_{\|x' - x\|_2 \le r} \frac{R_\rho^{\mathrm{Max}}(x')}{\rho^2} = \lim_{\rho \to 0} \frac{R_\rho^{\mathrm{Max}}(x)}{\rho^2} = \lambda_1(\nabla^2 L(x))/2 \,.$$

3. Finally we verify definition of good limiting regularizer, by Assumption 3.2, $S^{\mathrm{Max}}(x) = \lambda_1(x)/2$ is non-negative and continuous on $\Gamma$. For any $x^* \in \Gamma$, choose a sufficiently small open convex set $V$ containing $x^*$ such that $\forall x \in V^1, \|\nabla^3 L(x)\|_2 \le \nu$. For any $x \in V \cap \Gamma$, for any $x'$ satisfying that $\|x' - x\|_2 \le C\rho$, by Theorem K.3,
$$\begin{aligned} R_\rho^{\mathrm{Max}}(x') &= \max_{\|v\|_2 \le 1} L(x' + \rho v) - L(x') \ge \max_{\|v\|_2 \le 1} \big(\rho\langle \nabla L(x'), v\rangle + \rho^2 v^T \nabla^2 L(x') v/2\big) - \nu\rho^3/6 \\ &\ge \rho^2 \lambda_1(\nabla^2 L(x'))/2 - \nu\rho^3/6 \ge \rho^2 \lambda_1(\nabla^2 L(x))/2 - O(\rho^3) \,. \end{aligned}$$
This implies $\inf_{\|x' - x\|_2 \le C\rho} R_\rho^{\mathrm{Max}}(x') \ge \rho^2 \lambda_1(\nabla^2 L(x))/2 - O(\rho^3)$.
On the other hand, for any $x \in V \cap \Gamma$,
$$\begin{aligned} R_\rho^{\mathrm{Max}}(x) &= \max_{\|v\|_2 \le 1} L(x + \rho v) - L(x) \le \max_{\|v\|_2 \le 1} \big(\rho\langle \nabla L(x), v\rangle + \rho^2 v^T \nabla^2 L(x) v/2\big) + \nu\rho^3 \\ &= \max_{\|v\|_2 \le 1} \rho^2 v^T \nabla^2 L(x) v/2 + \nu\rho^3 = \rho^2 \lambda_1(\nabla^2 L(x'))/2 + O(\rho^3) \,. \end{aligned}$$
This implies $\inf_{\|x' - x\|_2 \le C\rho} R_\rho^{\mathrm{Max}}(x') \le \rho^2 \lambda_1(\nabla^2 L(x))/2 + O(\rho^3)$.

Thus, we conclude that $\left| \inf_{\|x' - x\|_2 \le C\rho} R_\rho^{\mathrm{Max}}(x')/\rho^2 - \lambda_1(\nabla^2 L(x))/2 \right| = O(\rho), \forall x \in V \cap \Gamma$, indicating $S^{\mathrm{Max}}$ is a good limiting regularizer of $R_\rho^{\mathrm{Max}}$ on $\Gamma$.

This completes the proof. $\qquad\square$

*Proof of Theorem G.4.*

1. We will first prove Condition G.1 holds. For any fixed compact set $B \subset U$, as $\lambda_1(\nabla^2 L)$ and $\|\nabla^3 L\|$ is continuous, there exists constant $C$, such that $\forall x \in B^2$, $\lambda_1(\nabla^2 L) > -\zeta$ and $\|\nabla^3 L(x)\| < \nu$. Then by Taylor Expansion,

$$
\begin{aligned}
R_\rho^{\text{Asc}}(x) &= L(x + \rho \frac{\nabla L(x)}{\|\nabla L(x)\|}) - L(x) \\
&\geq \left( \rho \|\nabla L(x)\|_2 + \rho^2 (\frac{\nabla L(x)}{\|\nabla L(x)\|})^T \nabla^2 L(x) \frac{\nabla L(x)}{\|\nabla L(x)\|} /2 \right) - \nu \rho^3/6 \\
&\geq -(\zeta + \nu/6)\rho^2.
\end{aligned}
$$

2. Now we verify $S^{\text{Asc}}(x) = \text{Tr}(\nabla^2 L(\cdot))/2$ is the limiting regularizer of $R_\rho^{\text{Asc}}$. Let $x$ be any point in $\Gamma$. Let $K = \{x\}$ and choose $h = h(K)$ as in Lemma E.5. For any $x' \in K^h \cap U'$,

$$
\begin{aligned}
R_\rho^{\text{Asc}}(x') &= L(x' + \rho \frac{\nabla L(x')}{\|\nabla L(x')\|}) - L(x') \\
&\geq \rho \|\nabla L(x')\|_2 + \rho^2 (\frac{\nabla L(x')}{\|\nabla L(x')\|})^T \nabla^2 L(x') \frac{\nabla L(x')}{\|\nabla L(x')\|} /2 - \nu \rho^3/6 \\
&\geq \rho^2 (\frac{\nabla L(x')}{\|\nabla L(x')\|})^T \nabla^2 L(\Phi(x')) \frac{\nabla L(x')}{\|\nabla L(x')\|} /2 - \nu \rho^3/6 \,.
\end{aligned}
$$

By Lemma F.4, we have $\frac{\nabla L(x')}{\|\nabla L(x')\|} = \frac{\nabla^2 L(\Phi(x'))(x' - \Phi(x'))}{\|\nabla^2 L(\Phi(x'))(x' - \Phi(x'))\|_2} + O(\frac{\nu}{\mu}\|x' - \Phi(x')\|_2)$. Hence

$$
R_\rho^{\text{Asc}}(x') \geq \rho^2 \lambda_M(\nabla^2 L(\Phi(x')))/2 - \zeta \rho^2 O(\|x' - \Phi(x')\|_2) - \nu \rho^3/6 \,.
$$

This implies $\lim_{\rho \to 0} \lim_{r \to 0} \inf_{\|x' - x\|_2 \leq r} \frac{R_\rho^{\text{Asc}}(x')}{\rho^2} \geq \lambda_M(\nabla^2 L(\Phi(x')))/2$.
We now show the above inequality is in fact equality. If we choose $x_r'' = x + r v_M$, then by Taylor Expansion,

$$
\begin{aligned}
\nabla L(x_r'') &= \nabla L(x) + \nabla^2 L(x)(x_r'' - x) + O(\|x_r'' - x\|^2) \\
&= r v_M + O(r^2)
\end{aligned}
$$

This implies $\lim_{r \to 0} \frac{\nabla L(x_r'')}{\|\nabla L(x_r'')\|} = v_M$. We also have $\lim_{r \to 0} \nabla^2 L(x_r'') = \nabla^2 L(x)$ and $\lim_{r \to 0} \nabla^L(x_r'') = 0$. Putting together,

$$
\begin{aligned}
\lim_{r \to 0} R_\rho^{\text{Asc}}(x_r'') &= \lim_{r \to 0} L(x_r'' + \rho \frac{\nabla L(x_r'')}{\|\nabla L(x_r'')\|}) - L(x_r'') \\
&= \lim_{r \to 0} \left( \rho \|\nabla L(x_r'')\|_2 + \rho^2 (\frac{\nabla L(x_r'')}{\|\nabla L(x_r'')\|})^T \nabla^2 L(x_r'') \frac{\nabla L(x_r'')}{\|\nabla L(x_r'')\|} /2 + O(\nu \rho^3) \right) \\
&= \rho^2 \lambda_M(\nabla^2 L(x))/2 + O(\rho^3).
\end{aligned}
$$

This implies $\lim_{\rho \to 0} \lim_{r \to 0} \inf_{\|x' - x\|_2 \leq r} \frac{R_\rho^{\text{Asc}}(x')}{\rho^2} \leq \lim_{\rho \to 0} \lim_{r \to 0} \frac{R_\rho^{\text{Asc}}(x_r'')}{\rho^2} = \lambda_M(\nabla^2 L(x))/2$.
Hence the limiting regularizer $S$ is exactly $\lambda_M(\nabla^2 L(\cdot))/2$.

3. Finally we verify definition of good limiting regularizer, by Assumption 3.2, $S^{\text{Max}}(x) = \lambda_M(x)/2$ is non-negative and continuous on $\Gamma$. For any $x^* \in \Gamma$, choose a sufficiently small open convex set $V$ containing $x^*$ such that $\forall x \in V^1$, $\|\nabla^3 L(x)\|_2 \leq \nu$. For any $x \in V \cap \Gamma$, for any $x'$ satisfying that $\|x' - x\|_2 \leq C\rho$,

$$
\begin{aligned}
R_\rho^{\text{Asc}}(x') &= L(x' + \rho \frac{\nabla L(x')}{\|\nabla L(x')\|}) - L(x') \\
&\geq \rho \|\nabla L(x')\|_2 + \rho^2 (\frac{\nabla L(x')}{\|\nabla L(x')\|})^T \nabla^2 L(x') \frac{\nabla L(x')}{\|\nabla L(x')\|} /2 - \nu \rho^3/6 \\
&\geq \rho^2 (\frac{\nabla L(x')}{\|\nabla L(x')\|})^T \nabla^2 L(\Phi(x')) \frac{\nabla L(x')}{\|\nabla L(x')\|} /2 - \nu \rho^3/6 \,.
\end{aligned}
$$

By Lemma F.4, we have $\frac{\nabla L(x')}{\|\nabla L(x')\|} = \frac{\nabla^2 L(\Phi(x'))(x' - \Phi(x'))}{\|\nabla^2 L(\Phi(x'))(x' - \Phi(x'))\|_2} + O(\frac{\nu}{\mu}\|x' - \Phi(x')\|_2)$. This implies $\inf_{\|x' - x\|_2 \leq C\rho} R_\rho^{\text{Asc}}(x') \geq \rho^2 \lambda_M(\nabla^2 L(x))/2 - O(\rho^3)$.

On the other hand, simillar to the proof in the second part, we have $\inf_{\|x'-x\|_2 \leq C\rho} R_\rho^{\text{Asc}}(x') \leq \rho^2 \lambda_M(\nabla^2 L(x))/2 + O(\rho^3)$.

Thus, we conclude that $\left| \inf_{\|x'-x\|_2 \leq C\rho} R_\rho^{\text{Max}}(x')/\rho^2 - \lambda_1(\nabla^2 L(x))/2 \right| = O(\rho), \forall x \in V \cap \Gamma$, indicating $S^{\text{Max}}$ is a good limiting regularizer of $R_\rho^{\text{Max}}$ on $\Gamma$.

This completes the proof. $\qquad\square$

*Proof of Theorem G.5.*

1. We will first verify Condition G.1. For fixed compact set $B \subset U$, as $\|\nabla^3 L(x)\|_2$ is continuous, there exists constant $\nu$, such that $\forall x \in B^1, \|\nabla^3 L(x)\|_2 \leq \nu$. Then by Taylor Expansion,

$$R_\rho^{\text{Avg}}(x) = \mathbb{E}_{g \sim N(0,I)} L(x + \rho \frac{g}{\|g\|}) - L(x)$$

$$\geq \mathbb{E}_{g \sim N(0,I)} \left( \rho \langle \nabla L(x), \frac{g}{\|g\|} \rangle + \rho^2 (\frac{g}{\|g\|})^T \nabla^2 L(x) \frac{g}{2\|g\|} \right) - \nu\rho^3/6$$

$$\geq -\nu\rho^3/6 \,.$$

2. Now we verify $S^{\text{Max}}(x) = \text{Tr}(\nabla^2 L(\cdot))/2D$ is the limiting regularizer of $R_\rho^{\text{Avg}}$. Let $x$ be any point in $\Gamma$, by continuity of $R_\rho^{\text{Avg}}$,

$$\lim_{\rho \to 0} \lim_{r \to 0} \inf_{\|x'-x\|_2 \leq r} \frac{R_\rho^{\text{Avg}}(x')}{\rho^2} = \lim_{\rho \to 0} \frac{R_\rho^{\text{Avg}}(x)}{\rho^2} = \text{Tr}(\nabla^2 L(x))/2D \,.$$

3. Finally we verify definition of good limiting regularizer, by Assumption 3.2, $S^{\text{Avg}}(x) = \text{Tr}(x)/2D$ is non-negative and continuous on $\Gamma$. For any $x^* \in \Gamma$, choose a sufficiently small open convex set $V$ containing $x^*$ such that $\forall x \in V^1, \|\nabla^3 L(x)\|_2 \leq \nu$. For any $x \in V \cap \Gamma$, for any $x'$ satisfying that $\|x'-x\|_2 \leq C\rho$, by Theorem K.3,

$$R_\rho^{\text{Avg}}(x') = \mathbb{E}_{g \sim N(0,I)} L(x' + \rho \frac{g}{\|g\|}) - L(x')$$

$$\geq \mathbb{E}_{g \sim N(0,I)} \left( \rho \langle \nabla L(x'), \frac{g}{\|g\|} \rangle + \rho^2 \frac{g}{\|g\|}^T \nabla^2 L(x') \frac{g}{\|2g\|} \right) - \nu\rho^3/6$$

$$\geq \rho^2 \text{Tr}(\nabla^2 L(x'))/2D - \nu\rho^3/6 \geq \rho^2 \text{Tr}(\nabla^2 L(x))/2D - O(\rho^3) \,.$$

This implies $\inf_{\|x'-x\|_2 \leq C\rho} R_\rho^{\text{Avg}}(x') \geq \rho^2 \text{Tr}(\nabla^2 L(x))/2D - O(\rho^3)$.

On the other hand, for any $x \in V \cap \Gamma$,

$$R_\rho^{\text{Avg}}(x) = \mathbb{E}_{g \sim N(0,I)} L(x + \rho \frac{g}{\|g\|}) - L(x)$$

$$\leq \mathbb{E}_{g \sim N(0,I)} \left( \rho \langle \nabla L(x), \frac{g}{\|g\|} \rangle + \rho^2 \frac{g}{\|g\|}^T \nabla^2 L(x) \frac{g}{2\|g\|} \right) + \nu\rho^3$$

$$= \mathbb{E}_{g \sim N(0,I)} \rho^2 \frac{g}{\|g\|}^T \nabla^2 L(x) \frac{g}{2\|g\|} + \nu\rho^3 = \rho^2 \text{Tr}(\nabla^2 L(x'))/2D + O(\rho^3) \,.$$

This implies $\inf_{\|x'-x\|_2 \leq C\rho} R_\rho^{\text{Avg}}(x') \leq \rho^2 \text{Tr}(\nabla^2 L(x))/2D + O(\rho^3)$.

Thus, we conclude that $\left| \inf_{\|x'-x\|_2 \leq C\rho} R_\rho^{\text{Avg}}(x')/\rho^2 - \text{Tr}(\nabla^2 L(x))/2D \right| = O(\rho), \forall x \in V \cap \Gamma$, indicating $S^{\text{Avg}}$ is a good limiting regularizer of $R_\rho^{\text{Avg}}$ on $\Gamma$.

$\qquad\square$

**Theorem G.14.** *Stochastic worst-direction sharpness $\mathbb{E}_k[R_{k,\rho}^{\text{Max}}]$ admits $\text{Tr}(\nabla^2 L(\cdot))/2$ as a good limiting regularizer on $\Gamma$ and satisfies Condition G.1.*

*Proof of Theorem G.14.* By Theorem E.2, Condition E.1 holds.

Easily deducted from Theorem G.3 $\Lambda_k(x)$ is a good limiting regularizer for $R_{k,\rho}^{\max}$ on $\Gamma_k$. Then as $\Gamma \subset \Gamma_k$, $\Lambda_k(x)$ is a good limiting regularizer for $R_{k,\rho}^{\max}$ on $\Gamma$. Hence $S(x) = \sum_k \Lambda_k(x)/2M = \text{Tr}(\nabla^2 \text{L(x)})/2$ is a good limiting regularizer of $\mathbb{E}_k[R_{k,\rho}^{\text{Max}}](x)$ on $\Gamma$. $\qquad\square$

**Theorem G.15.** *Stochastic ascent-direction sharpness $\mathbb{E}_k[R_{k,\rho}^{\text{Asc}}]$ admits $\text{Tr}(\nabla^2 L(\cdot))/2$ as a good limiting regularizer on $\Gamma$ and satisfies Condition G.1.*

*Proof of Theorem G.15.* By Theorem E.2, Condition E.1 holds.

Easily deducted from Theorem G.4 $\Lambda_k(x)$ is a good limiting regularizer for $R_{k,\rho}^{\text{asc}}$ on $\Gamma_k$ as the codimension of $\Gamma_k$ is 1. Then as $\Gamma \subset \Gamma_k$, $\Lambda_k(x)$ is a good limiting regularizer for $R_{k,\rho}^{\max}$ on $\Gamma$.Hence $S(x) = \sum_k \Lambda_k(x)/2M = \text{Tr}(\nabla^2 L(x))/2$ is a good limiting regularizer of $\mathbb{E}_k[R_{k,\rho}^{\text{Asc}}](x)$ on $\Gamma$. $\qquad\square$

**Theorem G.16.** *Stochastic average-direction sharpness $\mathbb{E}_k[R_{k,\rho}^{\text{Avg}}]$ admits $\text{Tr}(\nabla^2 L(\cdot))/(2D)$ as a good limiting regularizer on $\Gamma$ and satisfies Condition G.1.*

*Proof of Theorem G.16.* By definition, we know that $\mathbb{E}_k[R_{k,\rho}^{\text{Avg}}] = R_\rho^{\text{Avg}}$. The rest follows from Theorem G.5.
$\qquad\square$

### G.6 PROOF OF THEOREMS 4.2 AND 5.3

To end this section, we prove the two theorems presented in the main text. The readers will find the proof straight forward after we established the framework of *good limiting regularizers*.

*Proof of Theorem 4.2.* Apply Corollary G.7 on $R^{\text{type}}$. The mapping from $R$ to good limiting regularizers $S^{\text{type}}$ are characterized by Theorems G.3 to G.5. $\qquad\square$

*Proof of Theorem 5.3.* Apply Corollary G.7 on $R^{\text{type}}$. The mapping from $R$ to good limiting regularizers $\tilde{S}^{\text{type}}$ are characterized by Theorems G.14 to G.16. $\qquad\square$

## H ANALYSIS FULL-BATCH SAM ON QUADRATIC LOSS (PROOF OF THEOREM 4.8)

The goal of this section is to prove Theorem 4.8. In this section, we use $A \prec B$ to indicate $B - A$ is positive semi-definite.

**Theorem 4.8.** *Suppose $A$ is a positive definite symmetric matrix with unique top eigenvalue. Consider running full-batch SAM (Equation 3) on loss $L(x) := \frac{1}{2}x^T A x$ as in Equation 11 below.*

$$x(t+1) = x(t) - \eta A\big(x(t) + \rho A x(t)/\|A x(t)\|_2\big). \tag{11}$$

*Then, for almost every $x(0)$, we have $x(t)$ converges in direction to $v_1(A)$ up to a sign flip and $\lim_{t\to\infty} \|x(t)\|_2 = \frac{\eta\rho\lambda_1(A)}{2-\eta\lambda_1(A)}$ with $\eta\lambda_1(A) < 1$.*

*Proof of Theorem 4.8.* We first rewrite the iterate as

$$x(t+1) = x(t) - \eta A x(t) - \eta\rho\frac{A^2 x(t)}{\|A x(t)\|_2}.$$

Define $\tilde{x}(t) \triangleq \frac{\nabla L(x(t))}{\rho} = \frac{A x(t)}{\rho}$, and we have

$$\tilde{x}(t+1) = \tilde{x}(t) - \eta A \tilde{x}(t) - \eta\frac{A^2 \tilde{x}(t)}{\|\tilde{x}(t)\|_2}. \tag{22}$$

We suppose $A \in R^{D\times D}$ and use $\lambda_i, v_i$ to denote $\lambda_i(A), v_i(A)$.

Further, we define that

$$P^{(j:D)} \triangleq \sum_{i=j}^{D} v_i(A)v_i(A)^T,$$

$$\mathbb{I}_j \triangleq \{\tilde{x} \mid \|P^{(j:D)}\tilde{x}\|_2 \le \eta\lambda_j^2\},$$

$$\tilde{x}_i(t) \triangleq \langle \tilde{x}(t), v_i \rangle,$$

$$S \triangleq \{t \mid \|\tilde{x}(t)\|_2 \le \frac{\eta\lambda_1^2}{2-\eta\lambda_1}, t > T_1\}.$$

By Lemma H.1, $\mathbb{I}_j$ is an invariant set for update rule Equation 22.

Our proof consists of two steps.

(1) *Entering Invariant Set.* Lemma H.2 implies that there exists constant $T_1 > 0$, such that $\forall t > T_1, \|P^{(j:D)}\tilde{x}(t)\|_2 \leq \eta\lambda_j^2$

(2) *Alignment to Top Eigenvector.* Lemmas H.10 and H.11 show that $\|\tilde{x}(t)\|_2$ and $|\tilde{x}_1(t)|$ converge to $\frac{\eta\lambda_1^2}{2-\eta\lambda_1}$, which implies our final results.

$\square$

### H.1 ENTERING INVARIANT SET

In this subsection, we will prove the following three lemmas.

1. Lemma H.1 shows $\mathbb{I}_j$ is an invariant set for update rule (Equation 22).
2. Lemma H.2 shows that under the update rule (Equation 22), all iterates not in $\mathbb{I}_j$ will shrink exponentially in $\ell_2$ norm.
3. Lemma H.3 combines Lemmas H.1 and H.2 to show that for sufficiently large $t$, $x(t) \in \cap_j \mathbb{I}_j$.

**Lemma H.1.** *For $t \geq 0$, if $\eta\lambda_1(A) < 1$ and $\tilde{x}(t) \in \mathbb{I}_j$, then $\tilde{x}(t+1) \in \mathbb{I}_j$.*

*Proof of Lemma H.1.* By (Equation 22), we have that

$$P^{(j:D)}\tilde{x}(t+1) = (I - P^{(j:D)}\eta A - \eta\frac{P^{(j:D)}A^2}{\|\tilde{x}(t)\|_2})P^{(j:D)}\tilde{x}(t).$$

Hence we have that

$$\|P^{(j:D)}\tilde{x}(t+1)\|_2 = \|(I - P^{(j:D)}\eta A - \eta\frac{P^{(j:D)}A^2}{\|\tilde{x}(t)\|_2})P^{(j:D)}\tilde{x}(t)\|_2$$

$$\leq \|I - P^{(j:D)}\eta A - \eta\frac{P^{(j:D)}A^2}{\|\tilde{x}(t)\|_2}\|_2\|P^{(j:D)}\tilde{x}(t)\|_2.$$

Because $\tilde{x}(t) \in \mathbb{I}_j$, $\|\tilde{x}(t)\|_2 \leq \frac{\eta\lambda_j^2}{1-\eta\lambda_j}$. This implies,

$$I(1 - \eta\lambda_j - \eta\frac{\lambda_j^2}{\|P^{(j:D)}\tilde{x}(t)\|_2}) \prec I(1 - \eta\lambda_j - \eta\frac{\lambda_j^2}{\|\tilde{x}(t)\|_2}) \prec I - P^{(j:D)}\eta A - \eta\frac{P^{(j:D)}A^2}{\|\tilde{x}(t)\|_2} \prec I.$$

Hence, $\|I - P^{(j:D)}\eta A - \eta\frac{P^{(j:D)}A^2}{\|2\tilde{x}(t)\|}\|_2 \leq \max(1, \eta\lambda_j + \eta\frac{\lambda_j^2}{\|P^{(j:D)}\tilde{x}(t)\|_2} - 1)$. It holds that

$$\|P^{(j:D)}\tilde{x}(t+1)\|_2 \leq \max(\|P^{(j:D)}\tilde{x}(t)\|_2, \eta\lambda_j^2 - (1-\eta\lambda_j)\|P^{(j:D)}\tilde{x}(t)\|_2) \leq \eta\lambda_j^2,$$

where the last equality is because $1 - \eta\lambda_j \geq 0$. This above inequality is exactly the definition of $\tilde{x}(t+1) \in \mathbb{I}_j$ and thus is proof is completed. $\square$

**Lemma H.2.** *For $t \geq 0$, if $\eta\lambda_1(A) < 1$ and $\tilde{x}(t) \notin \mathbb{I}_j$, then*

$$\|P^{(j:D)}\tilde{x}(t+1)\|_2 \leq \max\left(1 - \eta\lambda_D - \eta\frac{\lambda_D^2}{\|\tilde{x}(t)\|_2}, \eta\lambda_j\right)\|P^{(j:D)}\tilde{x}(t)\|_2 \quad (23)$$

$$\leq \max(1 - \eta\lambda_D, \eta\lambda_j)\|P^{(j:D)}\tilde{x}(t)\|_2.$$

*Proof of Lemma H.2.* Note that

$$\|P^{(j:D)}\tilde{x}(t+1)\|_2 = \|(I - P^{(j:D)}\eta A - \eta\frac{P^{(j:D)}A^2}{\|\tilde{x}(t)\|_2})P^{(j:D)}\tilde{x}(t)\|_2$$

$$\leq \|P^{(j:D)} - P^{(j:D)}\eta A - \eta\frac{P^{(j:D)}A^2}{\|\tilde{x}(t)\|_2}\|_2\|P^{(j:D)}\tilde{x}(t)\|_2.$$

As $\tilde{x}(t) \notin \mathbb{I}_j$, We have $\|\tilde{x}(t)\|_2 \geq \|P^{(j:D)}\tilde{x}(t)\|_2 > \eta\lambda_j^2$, hence $\eta\frac{P^{(j:D)}A^2}{\|\tilde{x}(t)\|_2} \prec \eta\frac{P^{(j:D)}A^2}{\eta\lambda_j^2} \prec P^{(j:D)}$.

This implies that

$$-\eta\lambda_j P^{(j:D)} \prec -P^{(j:D)}\eta A \prec P^{(j:D)} - P^{(j:D)}\eta A - \eta\frac{P^{(j:D)}A^2}{\|\tilde{x}(t)\|_2},$$

and

$$P^{(j:D)} - P^{(j:D)}\eta A - \eta\frac{P^{(j:D)}A^2}{\|\tilde{x}(t)\|_2} \prec P^{(j:D)}(1-\eta\lambda_D) - \eta\frac{\lambda_D^2}{\|\tilde{x}(t)\|_2}.$$

Hence we have that

$$\|P^{(j:D)}\tilde{x}(t+1)\|_2 \le \max\left(1-\eta\lambda_D - \eta\frac{\lambda_D^2}{\|\tilde{x}(t)\|_2}, \eta\lambda_j\right)\|P^{(j:D)}\tilde{x}(t)\|_2.$$

$$\le \max(1-\eta\lambda_D, \eta\lambda_j)\|P^{(j:D)}\tilde{x}(t)\|_2$$

This completes the proof. $\qquad\square$

**Lemma H.3.** *Choosing* $T_1 = \max_j\left(-\log_{\max(1-\eta\lambda_D,\eta\lambda_j)}\max(\frac{\|\tilde{x}(0)\|_2}{\eta\lambda_j^2}, 1)\right)$, *then* $\forall t \ge T_1, D > j \ge 1, \tilde{x}(t) \in \mathbb{I}_j$

*Proof of Lemma H.3.* We will prove by contradiction. Suppose $\exists j \in [D]$ and $T > T_1$, such that $\tilde{x}(T) \notin \mathbb{I}_j$. By Lemma H.1, it holds that $\forall t < T, \tilde{x}(t) \notin \mathbb{I}_j$. Then by Lemma H.2,

$$\|P^{(j:D)}\tilde{x}(T)\|_2 \le \max(1-\eta\lambda_D, \eta\lambda_j)^T \|P^{(j:D)}\tilde{x}(0)\|_2 \le \eta\lambda_j^2,$$

which leads to a contradiction. $\qquad\square$

## H.2 ALIGNMENT TO TOP EIGENVECTOR

In this subsection, we prove the following lemmas towards showing that $\tilde{x}(t)$ converges in direction to $v_1(A)$ up to a proper sign flip.

1. Corollary H.4 show that for almost every learning rate $\eta$ and initialization $x_{init}$, $\tilde{x}_1(t) \neq 0$, for every $t \ge 0$. This condition is important because if $\tilde{x}_1(t) = 0$ at some step $t$, then for any $t' \ge t$, $\tilde{x}_1(t')$ will also be 0 and thus alignment is impossible.
2. Lemma H.5 shows that under update rule (Equation 22), $t \notin S \Rightarrow t+1 \in S$ for sufficiently large $t$, where the definition of $S$ is $\{t|\|\tilde{x}(t)\|_2 \le \frac{\eta\lambda_1^2}{2-\eta\lambda_1}, t > T_1\}$.
3. Lemma H.9, a combination of Lemmas H.6 and H.7, shows that following update rule (Equation 22), $\tilde{x}_1(t)$ increases for $t \in S$.
4. Lemma H.10 shows that $\|\tilde{x}(t)\|$ converges to $\frac{\eta\lambda_1^2}{2-\eta\lambda_1}$ under Equation 22.
5. Lemma H.11 shows that $\|\tilde{x}_1(t)\|_2$ converges to $\frac{\eta\lambda_1^2}{2-\eta\lambda_1}$ under Equation 22.

We will first prove that $\forall t, \tilde{x}_1(t) \neq 0$ happens for almost every learning rate $\eta$ and initialization $x_{init}$ (Corollary H.4), using a much more general result (Theorem D.3).

**Corollary H.4.** *Except for countably many* $\eta \in \mathbb{R}^+$, *for almost all initialization* $x_{init} = x(0)$, *it holds that for all natural number* $t$, $\tilde{x}_1(t) \neq 0$.

*Proof of Corollary H.4.* Let $F_n(x) \equiv F(x) \triangleq A(x + \rho\frac{Ax}{\|Ax\|_2})$, $\forall n \in \mathbb{N}^+, x \in \mathbb{R}^D$ and $Z = \{x \in \mathbb{R}^D \mid \langle x, v_1\rangle = 0\}$. We can easily check $F$ is $\mathcal{C}^1$ on $\mathbb{R}^D \setminus Z$ and $Z$ is a zero-measure set. Applying Theorem D.3, we have the following corollary. $\qquad\square$

**Lemma H.5.** *For* $t \ge 0$, *if* $\|\tilde{x}(t)\|_2 > \frac{\eta\lambda_1^2}{2-\eta\lambda_1}, \tilde{x}(t) \in \cap\mathbb{I}_j$, *then*

$$\|\tilde{x}(t+1)\|_2 \le \max(\frac{\eta\lambda_1^2}{2-\eta\lambda_1} - \eta\frac{\lambda_D^4}{2\lambda_1^2}, \eta\lambda_1^2 - (1-\eta\lambda_1)\|\tilde{x}(t)\|_2)$$

*Proof of Lemma H.5.* Note that

$$\tilde{x}(t+1) = (I - \eta A - \eta \frac{A^2}{\|\tilde{x}(t)\|_2})\tilde{x}(t)$$

$$= \frac{1}{\|\tilde{x}(t)\|_2} \sum_{j=1}^{D} \left((1 - \eta\lambda_j)\|\tilde{x}(t)\|_2 - \eta\lambda_j^2\right)\tilde{x}_j(t)v_j$$

Consider the following two cases.

1 If for any $i$, such that $\left|(1-\eta\lambda_1)\|\tilde{x}(t)\|_2 - \eta\lambda_1^2\right| \geq \left|(1-\eta\lambda_i)\|\tilde{x}(t)\|_2 - \eta\lambda_i^2\right|$, then we have
$$\|\tilde{x}(t+1)\|_2 \leq \left|(1-\eta\lambda_1)\|\tilde{x}(t)\|_2 - \eta\lambda_1^2\right| = \eta\lambda_1^2 - (1-\eta\lambda_1)\|\tilde{x}(t)\|_2 \,.$$

2 If there exists $i$, such that $\left|(1-\eta\lambda_1)\|\tilde{x}(t)\|_2 - \eta\lambda_1^2\right| < \left|(1-\eta\lambda_i)\|\tilde{x}(t)\|_2 - \eta\lambda_i^2\right|$, then suppose WLOG, $i$ is the smallest among such index.

As
$$\eta\lambda_i^2 - (1-\eta\lambda_i)\|\tilde{x}(t)\|_2 < \eta\lambda_1^2 - (1-\eta\lambda_1)\|\tilde{x}(t)\|_2 = \left|(1-\eta\lambda_1)\|\tilde{x}(t)\|_2 - \eta\lambda_1^2\right|$$
We have $-\eta\lambda_i^2 + (1-\eta\lambda_i)\|\tilde{x}(t)\|_2 > \eta\lambda_1^2 - (1-\eta\lambda_1)\|\tilde{x}(t)\|_2$. Equivalently,

$$\|\tilde{x}(t)\|_2 > \frac{\eta\lambda_1^2 + \eta\lambda_i^2}{2 - \eta\lambda_1 - \eta\lambda_i} \tag{24}$$

Combining with $\tilde{x}(t) \in \mathbb{I}_1 \Rightarrow \|\tilde{x}(t)\|_2 \leq \eta\lambda_1^2$, we have $\eta < \frac{\lambda_1 - \lambda_i}{\lambda_1^2}$.

Now consider the following vertors,

$$v^{(1)}(t) \triangleq (\eta\lambda_1^2 - (1-\eta\lambda_1)\|\tilde{x}(t)\|_2)\tilde{x}(t) \,,$$

$$v^{(2)}(t) \triangleq ((2 - \eta\lambda_1 - \eta\lambda_i)\|\tilde{x}(t)\|_2 - \eta\lambda_i^2 - \eta\lambda_1^2)P^{(i:D)}\tilde{x}(t) \,,$$

$$v^{(2+j)}(t) \triangleq ((\eta\lambda_{i+j-1} - \eta\lambda_{i+j})\|\tilde{x}(t)\|_2 - \eta\lambda_{i+j}^2 + \eta\lambda_{i+j-1}^2)P^{(i+j:D)}\tilde{x}(t), 1 \leq j \leq D - i \,.$$

Then we have

$$\|\tilde{x}(t+1)\|_2 = \|\frac{1}{\|\tilde{x}(t)\|_2} \sum_{j=1}^{D} \left((1 - \eta\lambda_j)\|\tilde{x}(t)\|_2 - \eta\lambda_j^2\right)\tilde{x}_j(t)v_j\|_2$$

$$\leq \|\frac{1}{\|\tilde{x}(t)\|_2} \sum_{j=1}^{i-1} \left(\eta\lambda_1^2 - (1 - \eta\lambda_1)\|\tilde{x}(t)\|_2\right)\tilde{x}_j(t)v_j\| +$$

$$\|\frac{1}{\|\tilde{x}(t)\|_2} \sum_{j=i}^{D} \left((1 - \eta\lambda_j)\|\tilde{x}(t)\|_2 - \eta\lambda_j^2\right)\tilde{x}_j(t)v_j\|_2$$

$$\leq \frac{1}{\|\tilde{x}(t)\|_2} \sum_{j=1}^{D+1-i} \|v^{(j)}\|_2$$

By assumption, we have $\tilde{x}(t) \in \cap \mathbb{I}_j$, hence we have

$$\|v^{(1)}(t)\|_2 = (\eta\lambda_1^2 - (1-\eta\lambda_1)\|\tilde{x}(t)\|_2)\|\tilde{x}(t)\|_2 \,,$$

$$\|v^{(2)}(t)\|_2 \leq \eta((2 - \eta\lambda_1 - \eta\lambda_i)\|\tilde{x}(t)\|_2 - \eta\lambda_i^2 - \eta\lambda_1^2)\lambda_i^2 \,,$$

$$\|v^{(2+j)}(t)\|_2 \leq \eta((\eta\lambda_{i+j-1} - \eta\lambda_{i+j})\|\tilde{x}(t)\|_2 - \eta\lambda_{i+j}^2 + \eta\lambda_{i+j-1}^2)\lambda_{i+j}^2, 1 \leq j \leq D - i \,.$$

Using AM-GM inequality, we have

$$\lambda_{i+j-1}\lambda_{i+j}^2 \leq \frac{\lambda_{i+j-1}^3 + 2\lambda_{i+j}^3}{3} \,,$$

$$\lambda_{i+j-1}^2\lambda_{i+j}^2 \leq \frac{\lambda_{i+j-1}^4 + \lambda_{i+j}^4}{2} \,.$$

Hence

$$\|v^{(2+j)}(t)\|_2 \leq \eta((\eta\lambda_{i+j-1} - \eta\lambda_{i+j})\|\tilde{x}(t)\|_2 - \eta\lambda_{i+j}^2 + \eta\lambda_{i+j-1}^2)\lambda_{i+j}^2$$

$$\leq \eta^2\|\tilde{x}(t)\|_2\frac{\lambda_{i+j-1}^3 - \lambda_{i+j}^3}{3} + \eta^2\frac{\lambda_{i+j-1}^4 - \lambda_{i+j}^4}{2}, 1 \leq j \leq D - i$$

$$\sum_{j=1}^{D-i} \|v^{(2+j)}(t)\|_2 \leq \eta^2\|\tilde{x}(t)\|_2\frac{\lambda_i^3 - \lambda_D^3}{3} + \eta^2\frac{\lambda_i^4 - \lambda_D^4}{2} \,.$$

Putting together,

$$\|\tilde{x}(t+1)\|_2 \leq \frac{1}{\|\tilde{x}(t)\|_2} \sum_{j=1}^{D+1-i} \|v^{(i)}\|_2$$

$$\leq \eta\lambda_1^2 + \eta\lambda_i^2(2 - \eta\lambda_1 - \eta\lambda_i) + \eta^2 \frac{\lambda_i^3 - \lambda_D^3}{3} - (1 - \eta\lambda_1)\|\tilde{x}(t)\|_2$$

$$- \eta^2\lambda_i^2(\lambda_i^2 + \lambda_1^2)\frac{1}{\|\tilde{x}(t)\|_2} + \eta^2 \frac{\lambda_i^4 - \lambda_D^4}{2}\frac{1}{\|\tilde{x}(t)\|_2}$$

$$\leq \eta\lambda_1^2 + \eta\lambda_i^2(2 - \eta\lambda_1 - \frac{2}{3}\eta\lambda_i) - (1 - \eta\lambda_1)\|\tilde{x}(t)\|_2 - \eta^2\lambda_i^2(\frac{1}{2}\lambda_i^2 + \lambda_1^2)\frac{1}{\|\tilde{x}(t)\|_2} - \eta^2\frac{\lambda_D^4}{2\|\tilde{x}(t)\|_2}$$

$$\leq \eta\lambda_1^2 + \eta\lambda_i^2(2 - \eta\lambda_1 - \frac{2}{3}\eta\lambda_i) - (1 - \eta\lambda_1)\|\tilde{x}(t)\|_2 - \eta^2\lambda_i^2(\frac{1}{2}\lambda_i^2 + \lambda_1^2)\frac{1}{\|\tilde{x}(t)\|_2} - \eta\frac{\lambda_D^4}{2\lambda_1^2}.$$

We further discuss three cases

1. If $\eta\lambda_i\sqrt{\frac{\frac{1}{2}\lambda_i^2+\lambda_1^2}{1-\eta\lambda_1}} < \frac{\eta\lambda_1^2+\eta\lambda_i^2}{2-\eta\lambda_1-\eta\lambda_i}$, we have $\|\tilde{x}(t)\|_2 > \frac{\eta\lambda_1^2+\eta\lambda_i^2}{2-\eta\lambda_1-\eta\lambda_i} > \eta\lambda_i\sqrt{\frac{\frac{1}{2}\lambda_i^2+\lambda_1^2}{1-\eta\lambda_1}}$, then

$$\|\tilde{x}(t+1)\|_2$$

$$\leq \eta\lambda_1^2 + \eta\lambda_i^2(2 - \eta\lambda_1 - \frac{2}{3}\eta\lambda_i) - (1 - \eta\lambda_1)\|\tilde{x}(t)\|_2 - \eta^2\lambda_i^2(\frac{1}{2}\lambda_i^2 + \lambda_1^2)\frac{1}{\|\tilde{x}(t)\|_2} - \eta\frac{\lambda_D^4}{2\lambda_1^2}$$

$$\leq \eta\lambda_1^2 + \eta\lambda_i^2(2 - \eta\lambda_1 - \frac{2}{3}\eta\lambda_i) - (1 - \eta\lambda_1)\frac{\eta\lambda_1^2 + \eta\lambda_i^2}{2 - \eta\lambda_1 - \eta\lambda_i}$$

$$- \eta^2\lambda_i^2(\frac{1}{2}\lambda_i^2 + \lambda_1^2)\frac{2 - \eta\lambda_1 - \eta\lambda_i}{\eta\lambda_1^2 + \eta\lambda_i^2} - \eta\frac{\lambda_D^4}{2\lambda_1^2}$$

$$\leq \frac{\eta\lambda_1^2}{2 - \eta\lambda_1} - \eta\frac{\lambda_D^4}{2\lambda_1^2}.$$

The second line is because $(1 - \eta\lambda_1)\|\tilde{x}(t)\|_2 + \eta^2\lambda_i^2(\frac{1}{2}\lambda_i^2 + \lambda_1^2)\frac{1}{\|\tilde{x}(t)\|_2}$ monotonously increase w.r.t $\|\tilde{x}(t)\|_2$ when $\|\tilde{x}(t)\|_2 > \eta\lambda_i\sqrt{\frac{\frac{1}{2}\lambda_i^2+\lambda_1^2}{1-\eta\lambda_1}}$. The last line is due to Lemma K.9.

2. If $\eta\lambda_1^2 \geq \eta\lambda_i\sqrt{\frac{\frac{1}{2}\lambda_i^2+\lambda_1^2}{1-\eta\lambda_1}} \geq \frac{\eta\lambda_1^2+\eta\lambda_i^2}{2-\eta\lambda_1-\eta\lambda_i}$, then

$$\|\tilde{x}(t+1)\|_2$$

$$\leq \eta\lambda_1^2 + \eta\lambda_i^2(2 - \eta\lambda_1 - \frac{2}{3}\eta\lambda_i) - (1 - \eta\lambda_1)\|\tilde{x}(t)\|_2 - \eta^2\lambda_i^2(\frac{1}{2}\lambda_i^2 + \lambda_1^2)\frac{1}{\|\tilde{x}(t)\|_2} - \eta\frac{\lambda_D^4}{2\lambda_1^2}$$

$$\leq \eta\lambda_1^2 + \eta\lambda_i^2(2 - \eta\lambda_1 - \frac{2}{3}\eta\lambda_i) - 2\eta\lambda_i\sqrt{(\lambda_1^2 + \frac{1}{2}\lambda_i^2)(1 - \eta\lambda_1)} - \eta\frac{\lambda_D^4}{2\lambda_1^2}$$

$$\leq \frac{\eta\lambda_1^2}{2 - \eta\lambda_1} - \eta\frac{\lambda_D^4}{2\lambda_1^2}.$$

The second line is because of AM-GM inequality. The last line is due to Lemma K.11.

3. If $\eta\lambda_1^2 < \eta\lambda_i\sqrt{\frac{\frac{1}{2}\lambda_i^2+\lambda_1^2}{1-\eta\lambda_1}}$, we have $\|\tilde{x}(t)\|_2 < \eta\lambda_1^2 < \eta\lambda_i\sqrt{\frac{\frac{1}{2}\lambda_i^2+\lambda_1^2}{1-\eta\lambda_1}}$, then

$$\|\tilde{x}(t+1)\|_2$$

$$\leq \eta\lambda_1^2 + \eta\lambda_i^2(2 - \eta\lambda_1 - \frac{2}{3}\eta\lambda_i) - (1 - \eta\lambda_1)\|\tilde{x}(t)\|_2 - \eta^2\lambda_i^2(\frac{1}{2}\lambda_i^2 + \lambda_1^2)\frac{1}{\|\tilde{x}(t)\|_2} - \eta\frac{\lambda_D^4}{2\lambda_1^2}$$

$$\leq \eta\lambda_1^2 + \eta\lambda_i^2(2 - \eta\lambda_1 - \frac{2}{3}\eta\lambda_i) - (1 - \eta\lambda_1)\eta\lambda_1^2 - \eta\lambda_i^2(\frac{1}{2}\lambda_i^2 + \lambda_1^2)\frac{1}{\lambda_1^2} - \eta\frac{\lambda_D^4}{2\lambda_1^2}$$

$$\leq \frac{\eta\lambda_1^2}{2 - \eta\lambda_1} - \eta\frac{\lambda_D^4}{2\lambda_1^2}.$$

The second line is because $(1 - \eta\lambda_1)\|\tilde{x}(t)\|_2 + \eta^2\lambda_i^2(\frac{1}{2}\lambda_i^2 + \lambda_1^2)\frac{1}{\|\tilde{x}(t)\|_2}$ monotonously decrease w.r.t $\|\tilde{x}(t)\|_2$ when $\|\tilde{x}(t)\|_2 < \eta\lambda_i\sqrt{\frac{\frac{1}{2}\lambda_i^2+\lambda_1^2}{1-\eta\lambda_1}}$. The last line is due to Lemma K.10.

$\square$

**Lemma H.6.** *if* $\|\tilde{x}(t)\|_2 \leq \frac{\eta\lambda_1^2}{2-\eta\lambda_1}$, *it holds that* $|\tilde{x}_1(t+1)| \geq |\tilde{x}_1(t)|$.

*Proof of Lemma H.6.* Nota that $|\tilde{x}_1(t+1)| = |1 - \eta\lambda_1 - \eta\frac{\lambda_1^2}{\|\tilde{x}(t)\|_2}||\tilde{x}_1(t)|$ and that $\eta\frac{\lambda_1^2}{\|\tilde{x}(t)\|_2} > 2 - \eta\lambda_1^2$. It follows that $1 - \eta\lambda_1 - \eta\frac{\lambda_1^2}{\|\tilde{x}(t)\|_2} < -1$. Hence we have that $|\tilde{x}_1(t+1)| > |\tilde{x}_1(t)|$. $\square$

**Lemma H.7.** *For any* $t \geq 0$, *if* $\|\tilde{x}(t)\|_2 \leq \frac{\eta\lambda_1^2}{2-\eta\lambda_1}, \tilde{x}(t) \in \cap\mathbb{I}_j$, *it holds that*

$$\|\tilde{x}(t+1)\|_2 \leq \eta\lambda_1^2 - (1-\eta\lambda_1)\|\tilde{x}(t)\|_2 \,.$$

*Proof of Lemma H.7.* Note that

$$\|I - \eta A - \eta\frac{A^2}{\|\tilde{x}(t)\|_2}\|_2 \leq \max_{1\leq j\leq D}\{|1 - \eta\lambda_j - \eta\frac{\lambda_j^2}{\|\tilde{x}(t)\|}|\} = \eta\frac{\lambda_1^2}{\|\tilde{x}(t)\|} - (1-\eta\lambda_j)\,.$$

The proof is completed by noting that $\|\tilde{x}(t+1)\| \leq \|I - \eta A - \eta\frac{A^2}{\|\tilde{x}(t)\|_2}\|_2\|\tilde{x}(t)\|_2$. $\square$

**Lemma H.8.** *For any* $t \geq 0$, *if* $\|\tilde{x}(t)\|_2 \leq \frac{\eta\lambda_1^2}{1-\eta\lambda_1}$, *it holds that*

$$\|\tilde{x}(t+1)\|_2$$
$$\leq(\eta\lambda_1^2 - (1+\eta\lambda_1)\|\tilde{x}(t)\|_2)\times$$
$$\sqrt{\frac{|\tilde{x}_1(t)|^2}{\|\tilde{x}(t)\|^2} + \left(\max_{j\in[2:M]}\left(\frac{|(1-\eta\lambda_j)\|\tilde{x}(t)\|_2 - \eta\lambda_j^2|}{\eta\lambda_1^2 - (1-\eta\lambda_1)\|\tilde{x}(t)\|_2}\right)\right)^2(1-\frac{|\tilde{x}_1(t)|^2}{\|\tilde{x}(t)\|^2})}.$$

*Proof of Lemma H.8.* We will discuss the movement along $v_1$ and orthogonal to $v_1$. First,

$$\|P^{(2:D)}\tilde{x}(t+1)\|_2 = \|(I - P^{(2:D)}\eta A - \eta\frac{P^{(2:D)}A^2}{\|\tilde{x}(t)\|_2})P^{(2:D)}\tilde{x}(t)\|_2$$

$$\leq \|P^{(2:D)} - P^{(2:D)}\eta A - \eta\frac{P^{(2:D)}A^2}{\|\tilde{x}(t)\|_2}\|_2\|P^{(2:D)}\tilde{x}(t)\|_2$$

$$\leq \max_{j\in[2:M]}\{|1 - \eta\lambda_j - \frac{\eta\lambda_j^2}{\|\tilde{x}(t)\|_2}|\}\|P^{(2:D)}\tilde{x}(t)\|_2 \,.$$

Second, $|\tilde{x}_1(t+1)| = (\frac{\eta\lambda_1^2}{\|\tilde{x}(t)\|_2} - 1 + \eta\lambda_1)|\tilde{x}_1(t)|$. Hence we have that

$$\|\tilde{x}(t+1)\|_2$$
$$\leq(\eta\lambda_1^2 - (1+\eta\lambda_1)\|\tilde{x}(t)\|_2)\times$$
$$\sqrt{\frac{|\tilde{x}_1(t)|^2}{\|\tilde{x}(t)\|^2} + \left(\max_{j\in[2:M]}\left(\frac{|(1-\eta\lambda_j)\|\tilde{x}(t)\|_2 - \eta\lambda_j^2|}{\eta\lambda_1^2 - (1-\eta\lambda_1)\|\tilde{x}(t)\|_2}\right)\}\right)^2(1-\frac{|\tilde{x}_1(t)|^2}{\|\tilde{x}(t)\|^2})}.$$

$\square$

**Lemma H.9.** *For* $t, t' \in S, 0 \leq t \leq t'$, *then* $|\tilde{x}_1(t)| \leq |\tilde{x}_1(t')|$.

*Proof of Lemma H.9.* For $t \in S$, by Lemma H.5, $t+1 \in S$ or $t+1 \notin S, t+2 \in S$. We will discuss by case.

1. If $t+1 \in S$, we can use Lemma H.6 to show $|\tilde{x}_1(t)| \leq |\tilde{x}_1(t+1)|$.
2. If $t+1 \notin S, t+2 \in S$, then

$$|\tilde{x}_1(t+2)| = \frac{(\eta\lambda_1^2 - (1-\eta\lambda_1)\|\tilde{x}(t)\|_2)(\eta\lambda_1^2 - (1-\eta\lambda_1)\|\tilde{x}(t+1)\|_2)}{\|\tilde{x}(t)\|_2\|\tilde{x}(t+1)\|_2}|\tilde{x}_1(t)| \,.$$

As

$$(\eta\lambda_1^2 - (1-\eta\lambda_1)\|\tilde{x}(t)\|_2)(\eta\lambda_1^2 - (1-\eta\lambda_1)\|\tilde{x}(t+1)\|_2) \geq \|\tilde{x}(t)\|_2\|\tilde{x}(t+1)\|_2$$

$$\Longleftrightarrow \eta^2\lambda_1^4 - \eta\lambda_1^2(1-\eta\lambda_1)(\|\tilde{x}(t)\|_2 + \|\tilde{x}(t+1)\|_2)$$
$$\geq (2\eta\lambda_1 - \eta^2\lambda_1^2)\|\tilde{x}(t)\|_2\|\tilde{x}(t+1)\|_2$$

$$\Longleftrightarrow \eta^2\lambda_1^4 - \eta\lambda_1^2(1-\eta\lambda_1)\|\tilde{x}(t)\|_2$$
$$\geq \left((2\eta\lambda_1 - \eta^2\lambda_1^2)\|\tilde{x}(t)\|_2 + \eta\lambda_1^2(1-\eta\lambda_1)\right)\|\tilde{x}(t+1)\|_2 \,,$$

combining with Lemma H.7, we only need to prove,

$$\eta^2\lambda_1^4 - \eta\lambda_1^2(1-\eta\lambda_1)\|\tilde{x}(t)\|_2$$
$$\geq \left((2\eta\lambda_1 - \eta^2\lambda_1^2)\|\tilde{x}(t)\|_2 + \eta\lambda_1^2(1-\eta\lambda_1)\right)\left(\eta\lambda_1^2 - (1-\eta\lambda_1)\|\tilde{x}(t)\|_2\right) \,.$$

Through some calculation, this is equivalent to

$$((2-\eta\lambda_1)\|\tilde{x}(t)\|_2 - \eta\lambda_1^2)((1-\eta\lambda_1)\|\tilde{x}(t)\|_2 - \eta\lambda_1^2) \geq 0 \,.$$

which holds for $\|\tilde{x}(t)\|_2 \leq \frac{\eta\lambda_1^2}{2-\eta\lambda_1}$.

Combining the two cases and using induction, we can get the desired result. $\qquad\square$

**Lemma H.10.** $\|\tilde{x}(t)\|$ converges to $\frac{\eta\lambda_1^2}{2-\eta\lambda_1}$ when $t \to \infty$.

*Proof of Lemma H.10.* By Lemma H.9, $|\tilde{x}_1(t)|$ increases monotonously for $t \in S$. By Lemma H.5, $S$ is infinite. By Lemma H.2, for sufficiently large $t$, $|\tilde{x}_1(t)|$ is bounded. Combining the three facts, we know $\tilde{x}_1(t)$ for $t \in S$ converges.

Formally $\forall \epsilon > 0$, there exists $T_\epsilon > 0$ such that $\forall t, t' \in S, t' > t > T_\epsilon, \frac{\|\tilde{x}_1(t')\|_2}{\|\tilde{x}_1(t)\|_2} < 1+\epsilon$.

Then by Lemma H.5, $\forall t \in S, t+1 \in S$ or $t+2 \in S$, we will discuss by case. For $t \geq T_\epsilon$,

1. If $t+1 \in S$, then

$$1 + \epsilon \geq \frac{\|\tilde{x}_1(t+1)\|_2}{\|\tilde{x}_1(t)\|_2} = \frac{\eta\lambda_1^2 - (1-\eta\lambda_1)\|\tilde{x}(t)\|_2}{\|\tilde{x}(t)\|_2} \,.$$

2. If $t+1 \notin S$ and $t+2 \in S$, then

$$1 + \epsilon \geq \frac{\|\tilde{x}_1(t+2)\|_2}{\|\tilde{x}_1(t)\|_2}$$
$$= \frac{(\eta\lambda_1^2 - (1-\eta\lambda_1)\|\tilde{x}(t)\|_2)(\eta\lambda_1^2 - (1-\eta\lambda_1)\|\tilde{x}(t+1)\|_2)}{\|\tilde{x}(t)\|_2\|\tilde{x}(t+1)\|_2}$$
$$\geq \frac{(\eta\lambda_1^2 - (1-\eta\lambda_1)\|\tilde{x}(t)\|_2)\left(\eta\lambda_1^2 - (1-\eta\lambda_1)\left(\eta\lambda_1^2 - (1-\eta\lambda_1)\|\tilde{x}(t)\|_2\right)\right)}{\|\tilde{x}(t)\|_2\left(\eta\lambda_1^2 - (1-\eta\lambda_1)\|\tilde{x}(t)\|_2\right)}$$
$$= \frac{\eta\lambda_1^2 - (1-\eta\lambda_1)\left(\eta\lambda_1^2 - (1-\eta\lambda_1)\|\tilde{x}(t)\|_2\right)}{\|\tilde{x}(t)\|_2} \,.$$

Here in the last inequality, we apply Lemma H.7.

Concluding, $\|\tilde{x}(t)\|_2 \geq \min\left(\frac{\eta\lambda_1^2}{2-\eta\lambda_1^2+\epsilon}, \frac{\eta^2\lambda_1^3}{(2-\lambda_1\eta)\lambda_1\eta+\epsilon}\right), \forall t > T_\epsilon, t \in S$. As $\forall t \notin S, t > T_\epsilon$, we have $\|\tilde{x}(t)\|_2 \geq \frac{\eta\lambda_1^2}{2-\eta\lambda_1^2}$. Hence we have $\forall t > T_\epsilon, \|\tilde{x}(t)\|_2 \geq \min\left(\frac{\eta\lambda_1^2}{2-\eta\lambda_1^2+\epsilon}, \frac{\eta^2\lambda_1^3}{(2-\lambda_1\eta)\lambda_1\eta+\epsilon}\right)$.

Further by Lemma H.7, $\forall t > T_\epsilon + 1, \|\tilde{x}(t)\|_2 \leq \eta\lambda_1^2 - (1-\eta\lambda_1)\min\left(\frac{\eta\lambda_1^2}{2-\eta\lambda_1^2+\epsilon}, \frac{\eta^2\lambda_1^3}{(2-\lambda_1\eta)\lambda_1\eta+\epsilon}\right)$.

Combining both bound, we have $\lim_{t\to\infty}\|\tilde{x}(t)\|_2 = \frac{\eta\lambda_1^2}{2-\eta\lambda_1}$. $\qquad\square$

**Lemma H.11.** $\|\tilde{x}_1(t)\|_2$ converges to $\frac{\eta\lambda_1^2}{2-\eta\lambda_1}$, when $t \to \infty$.

*Proof of Lemma H.11.* Notice that

$$\|P^{(2:D)}\tilde{x}(t+1)\|_2 \le \max\left(|1 - \eta\lambda_2 - \eta\frac{\lambda_2^2}{\|\tilde{x}(t)\|_2}|, |1 - \eta\lambda_D - \eta\frac{\lambda_D^2}{\|\tilde{x}(t)\|_2}|\right)\|P^{(2:D)}\tilde{x}(t)\|_2.$$

When $\|\tilde{x}(t)\|_2 > \frac{\eta\lambda_2^2}{2 - \eta\lambda_2 - \delta}$,

$$-1 + \delta \le 1 - \eta\lambda_2 - \eta\frac{\lambda_2^2}{\|\tilde{x}(t)\|_2} \le 1 - \eta\lambda_D - \eta\frac{\lambda_D^2}{\|\tilde{x}(t)\|_2} \le 1 - \eta\lambda_D$$

$$\|P^{(2:D)}\tilde{x}(t+1)\|_2 \le \max(1 - \eta\lambda_D, 1 - \delta)\|P^{(2:D)}\tilde{x}(t)\|_2$$

Hence for sufficiently large $t$, $\|P^{(2:D)}\tilde{x}(t)\|_2$ shrinks exponentially, showing that $\lim_{t\to\infty}\|\tilde{x}_1(t)\|_2 = \frac{\eta\lambda_1^2}{2 - \eta\lambda_1}$. $\square$

## I  ANALYSIS FOR FULL-BATCH SAM ON GENERAL LOSS (PROOF OF THEOREM 4.5)

The goal of this section is to prove the following theorem.

**Theorem 4.5** (Main). *Let $\{x(t)\}$ be the iterates of full-batch SAM (Equation 3) with $x(0) = x_{init} \in U$. Under Assumptions 3.2 and 4.4, for all $\eta, \rho$ such that $\eta\ln(1/\rho)$ and $\rho/\eta$ are sufficiently small, the dynamics of SAM can be characterized in the following two phases:*

- *Phase I: (Theorem I.1) Full-batch SAM (Equation 3) follows Gradient Flow with respect to L until entering an $O(\eta\rho)$ neighborhood of the manifold $\Gamma$ in $O(\ln(1/\rho)/\eta)$ steps;*
- *Phase II: (Theorem I.3) Under a mild non-degeneracy assumption (Assumption I.2) on the initial point of phase II, full-batch SAM (Equation 3) tracks the solution $X$ of Equation 7, the Riemannian Gradient Flow with respect to the loss $\lambda_1(\nabla^2 L(\cdot))$ in an $O(\eta\rho)$ neighborhood of manifold $\Gamma$. Quantitatively, the approximation error between the iterates $x$ and the corresponding limiting flow $X$ is $O(\eta\ln(1/\rho))$, that is,*

$$\|x(\lceil T_3/(\eta\rho^2)\rceil) - X(T_3)\|_2 = O(\eta\ln(1/\rho)).$$

*Moreover, the angle between $\nabla L(x(\lceil\frac{T_3}{\eta\rho^2}\rceil))$ and the top eigenspace of $\nabla^2 L(x(\lceil\frac{T_3}{\eta\rho^2}\rceil))$ is $O(\rho)$.*

Readers may refer to Appendix E for notation.

To prove the theorem, we will separate the dynamic of SAM on general loss $L$ to two phases.

Define

$$R_j(x) = \sqrt{\sum_{i=j}^M \lambda_i^2(x)\langle v_i(x), x - \Phi(x)\rangle^2 - \eta\rho\lambda_j^2(x)}, \forall j \in [M], x \in U,$$

which is the length projection of $x - \Phi(x)$ on button$-k$ non-zero eigenspace of $\nabla^2 L(\Phi(x))$. We will provide a fine-grained convergence bound on $R_j(x)$.

**Theorem I.1** (Phase I). *Let $\{x(t)\}$ be the iterates defined by SAM ( Equation 3) and $x(t) = x_{init} \in U$, then under Assumption 3.2 there exists a positive number $T_1$ independent of $\eta$ and $\rho$, such that for any $T_1' > T_1$, it holds for all $\eta, \rho$ such that $(\eta + \rho)\ln(1/\eta\rho)$ is sufficiently small, we have*

$$\max_{T_1\ln(1/\eta\rho)\le\eta t\le T_1'\ln(1/\eta\rho)} \max_{j\in[M]} \max\{R_j(x(t)), 0\} = O(\eta\rho^2)$$

$$\max_{T_1\ln(1/\eta\rho)\le\eta t\le T_1'\ln(1/\eta\rho)} \|\Phi(x(t)) - \Phi(x_{init})\| \le O((\eta + \rho)\ln(1/\eta\rho))$$

Theorem I.1 implies SAM will converge to an $O(\eta\rho)$ neighbor of $\Gamma$. Notice in the time frame defined by Theorem I.1, $x(t)$ effectively operates at a local regime around $\Phi(\lceil T_1\ln(1/\eta\rho)/\eta\rceil)$, this allows us to approximate $L$ with the quadratic Taylor expansion of $L$ at $\Phi(\lceil T_1\ln(1/\eta\rho)/\eta\rceil)$ and prove the following theorem Theorem I.3.

Towards proving Theorem I.3, we need to make one assumption about the trajectory of SAM, Assumption I.2.

**Assumption I.2.** *There exists step $t$, satisfying that $T_1\ln(1/\eta\rho)/\eta \le t \le O(\ln(1/\eta\rho/\eta))$, $|\langle x(t) - \Phi(x(t)), v_1(x(t))\rangle| \ge \Omega(\rho^2)$ and that $\|x(t) - \Phi(x(t))\|_2 \le \lambda_1(t)\eta\rho - \Omega(\rho^2)$, where $T_1$ is the constant defined in Theorem I.1.*

We remark that the above assumption is very mild as we only need the above two conditions in Assumption I.2 to hold for some step in $\tilde{\Theta}(1/\eta)$ steps after Phase I ends, and since then our analysis for Phase II shows that these two conditions will hold until Phase II ends.

**Theorem I.3** (Phase II). *Let $\{x(t)\}$ be the iterates defined by SAM (Equation 3) under Assumptions 3.2 and 4.4, for all $\eta, \rho$ such that $\eta \ln(1/\rho)$ and $\rho/\eta$ is sufficiently small, further assuming that (1) $\max_{j\in[M]} \max\{R_j(x(0)), 0\} = O(\eta\rho^2)$, (2) $\|\Phi(x(0)) - \Phi(x_{init})\| = O((\eta + \rho)\ln(1/\eta\rho))$, (3) $|\langle x(0) - \Phi(x(0)), v_1(x(0))\rangle| \geq \Omega(\rho^2)$ and (4) $\|x(0) - \Phi(x(t))\|_2 \leq \lambda_1(0)\eta\rho - \Omega(\rho^2)$, the iterates $x(t)$ tracks the solution $X$ of Equation 7. Quantitatively for $t = \lceil T_3/\eta\rho^2\rceil$, we have that*

$$\|\Phi(x(t)) - X(\eta\rho^2 t)\| = O(\eta \ln(1/\rho)).$$

*Moreover, the angle between $\nabla L(x(t))$ and the top eigenspace of $\nabla^2 L(\Phi(x(t)))$ is at most $O(\rho)$. Quantitatively,*

$$|\langle x(t) - \Phi(x(t)), v_1(x(t))\rangle| = \Theta(\eta\rho).$$

$$\max_{j\in[2:M]} |\langle x(t) - \Phi(x(t)), v_j(x(t))\rangle| = O(\eta\rho^2).$$

In this section we will define $K$ as $\{X(t) \mid 0 \leq t \leq T_3\}$ where $X$ is the solution of Equation 7. To simplify our proof, we assume WLOG $L(x) = 0$ for $x \in \Gamma$.

## I.1 PHASE I (PROOF OF THEOREM I.1)

*Proof of Theorem I.1.* The proof consists of three major parts.

1. *Tracking Gradient Flow.* Lemma I.4 shows the existence of step $t_{GF} = O(1/\eta)$ such that $x(t_{GF})$ is in a subset of $K^h$ and $\Phi(x(t_{GF}))$ is $O(\eta + \rho)$ close to $\Phi(x_{init})$.
2. *Decreasing Loss.* Lemma I.6 shows the existence of step $t_{DEC} = O(\ln(1/\rho)/\eta)$ such that $x(t_{DEC})$ is in $O(\rho)$ neighbor of $\Gamma$ and $\Phi(x(t_{DEC}))$ is $O((\eta + \rho)\ln(1/\rho))$ close to $\Phi(x_{init})$.
3. *Entering Invariant Set.* Lemmas I.11 and I.13 shows the existence of step $t_{INV} = O(\ln(1/\rho\eta)/\eta)$ such that for any $t$ satisfying $t_{INV} \leq t \leq t_{INV} + \Theta(\ln(1/\eta)/\eta)$, we have that $x(t) \in \cap_{k\in[M]}\mathbb{I}_k$ and $\Phi(x(t))$ is $O((\eta + \rho)\ln(1/\eta\rho))$ close to $\Phi(x_{init})$.

$\square$

### I.1.1 TRACKING GRADIENT FLOW

Lemma I.4 shows that the iterates $x(t)$ tracks gradient flow to an $O(1)$ neighbor of $\Gamma$.

**Lemma I.4.** *Under condition of Theorem I.1, there exists $t_{GF} = O(1/\eta)$, such that the iterate $x(t_{GF})$ is $O(1)$ close to the manifold $\Gamma$ and $\Phi(x(t_{GF}))$ is $O(\eta + \rho)$ is close to $\Phi(x_{init})$. Quantitatively,*

$$L(x(t_{GF})) \leq \frac{\mu h^2}{32}$$

$$\|x(t_{GF}) - \Phi(x(t_{GF}))\| \leq h/4\,,$$

$$\|\Phi(x(t_{GF})) - \Phi(x_{init})\| = O(\eta + \rho)\,.$$

*Proof of Lemma I.4.* Choose $C = \frac{1}{4}\sqrt{\frac{\mu}{\zeta}}$. Since $\Phi(x_{init}) = \lim_{T\to\infty}\phi(x_{init}, T)$, there exists $T > 0$, such that $\|\phi(x_{init}, T) - \Phi(x_{init})\|_2 \leq Ch/2$. Note that

$$x(t+1) = x(t) - \eta\nabla L(x(t) + \rho\frac{\nabla L(x(t))}{\|\nabla L(x(t))\|}) = x(t) - \eta\nabla L(x(t)) + O(\eta\rho)\,.$$

By Corollary L.3, let $b(x) = -\nabla L(x)$, $p = \eta$ and $\epsilon = O(\rho)$, we have that the iterates $x(t)$ tracks gradient flow $\phi(x_{init}, T)$ in $O(1/\eta)$ steps. Quantitatively for $t_{GF} = \lceil\frac{T}{\eta}\rceil$, we have that

$$\|x(t_{GF}) - \phi(x_{init}, T)\|_2 = O(\epsilon + p) = O(\eta + \rho)\,.$$

This implies $x(t_{GF}) \in K^h$, hence by Taylor Expansion on $\Phi$,

$$\begin{aligned}
\|\Phi(x(t_{GF})) - \Phi(x_{init})\|_2 &= \|\Phi(x(t_{GF})) - \Phi(\phi(x_{init}, T))\|_2 \\
&\leq O(\|x(t_{GF}) - \phi(x_{init}, T)\|_2) \\
&\leq O(\eta + \rho)\,.
\end{aligned}$$

This implies

$$\|x(t_{\mathrm{GF}}) - \Phi(x(t_{\mathrm{GF}}))\|_2$$
$$\leq \|x(t_{\mathrm{GF}}) - \phi(x_{\mathrm{init}}, T_0)\|_2 + \|\phi(x_{\mathrm{init}}, T_0) - \Phi(x_{\mathrm{init}})\|_2 + \|\Phi(x_{\mathrm{init}}) - \Phi(x(t_{\mathrm{GF}}))\|_2$$
$$\leq Ch/2 + O(\eta + \rho) \leq Ch \leq h/4.$$

By Taylor Expansion, we conclude that $L(x(t_{\mathrm{GF}})) \leq \zeta \|x(t_{\mathrm{GF}}) - \Phi(x(t_{\mathrm{GF}}))\|_2^2/2 \leq \frac{\mu h^2}{32}$. $\qquad\square$

### I.1.2 DECREASING LOSS

Lemma I.6 shows that the iterates $x(t)$ converges to an $O(\rho)$ neighbor of $\Gamma$ in $O(\ln(1/\rho)/\eta)$ steps.

**Lemma I.5.** *Under condition of Theorem I.1, if $x(t) \in K^h$ and $\|\nabla L(x(t))\| \geq 4\zeta\rho$, then we have that $L(x(t+1))$ decreases with respect to $L(x(t))$, quantitatively, we have that*

$$L(x(t+1)) \leq L(x(t))(1 - \eta\mu/8).$$

*Moreover the movement of the projection of the iterates on the manifold is bounded, quantitatively, we have that*

$$\|\Phi(x(t+1)) - \Phi(x(t))\| \leq O(\eta^2).$$

*Proof of Lemma I.5.* As $x(t) \in K^h$ and $L$ is $\mu$-PL in $K^h$, we have $L(x(t)) \geq 0$.

As $x(t) \in K^h$, by Lemma F.7 and Taylor Expansion, we have $\|\overline{x(t)x(t+1)}\| = O(\eta)$. hence for sufficiently small $\eta$, $\overline{x(t)x(t+1)} \subset K^r$. Using similar argument, the segment from $x(t)$ to $x(t) + \rho\frac{\nabla L(x(t))}{\|\nabla L(x(t))\|}$ is in $K^r$.

Then by Taylor Expansion on $L$,

$$L(x(t+1)) = L\left(x(t) - \eta\nabla L\left(x(t) + \rho\frac{\nabla L(x(t))}{\|\nabla L(x(t))\|}\right)\right)$$
$$\leq L(x(t)) - \eta\left\langle \nabla L(x(t)), \nabla L\left(x(t) + \rho\frac{\nabla L(x(t))}{\|\nabla L(x(t))\|}\right)\right\rangle$$
$$+ \frac{\zeta\eta^2\|\nabla L\left(x(t) + \rho\frac{\nabla L(x(t))}{\|\nabla L(x(t))\|}\right)\|^2}{2}. \tag{25}$$

By Taylor Expansion on $\nabla L$, we have that

$$\left\|\nabla L\left(x(t) + \rho\frac{\nabla L(x(t))}{\|\nabla L(x(t))\|}\right) - \nabla L(x(t))\right\| \leq \zeta\rho.$$

After plugging in Equation 25, we have that

$$L(x(t+1)) \leq L(x(t)) - \eta\|\nabla L(x(t))\|^2 + \eta\zeta\rho\|\nabla L(x(t))\| + \zeta\eta^2\|\nabla L(x(t))\|^2 + \zeta^3\eta^2\rho^2. \tag{26}$$

As $\|\nabla L(x(t))\| \geq 4\zeta\rho$, we have that the following term is bounded.

$$\zeta\eta^2\|\nabla L(x(t))\|^2 \leq \frac{1}{2}\eta\|\nabla L(x(t))\|^2,$$

$$\eta\zeta\rho\|\nabla L(x(t))\| \leq \frac{1}{4}\eta\|\nabla L(x(t))\|^2,$$

$$\zeta^3\eta^2\rho \leq \zeta^2\eta\rho^2 \leq \frac{1}{16}\eta\|\nabla L(x(t))\|^2.$$

After plugging in Equation 26, by Lemma F.2,

$$L(x(t+1)) \leq L(x(t)) - \frac{1}{16}\eta\|\nabla L(x(t))\|^2$$
$$\leq L(x(t))(1 - \eta\mu/8).$$

As $x(t) \in K^h$, by Taylor Expansion, we have

$$\|\nabla L(x(t))\| \leq \zeta h.$$

Hence by Lemma F.7 and Taylor Expansion,

$$\|\Phi(x(t+1)) - \Phi(x(t))\| \leq \xi\eta\rho\|\nabla L(x)\|_2 + \nu\eta\rho^2 + \xi\eta^2\|\nabla L(x)\|_2^2 + \xi\zeta^2\eta^2\rho^2 \leq O(\eta^2),$$

which completes the proof. $\qquad\square$

**Lemma I.6.** *Under condition of Theorem I.1, assuming there exists $t_{\mathrm{GF}}$ such that $L(x(t_{\mathrm{GF}})) \leq \frac{\mu h^2}{32}$ and $x(t_{\mathrm{GF}}) \in K^{h/4}$, then there exists $t_{\mathrm{DEC}} = t_{\mathrm{GF}} + O(\ln(1/\rho)/\eta)$, such that $x(t_{\mathrm{DEC}})$ is in $O(\rho)$ neighbor of $\Gamma$, quantitatively, we have that*

$$\|\nabla L(x(t_{\mathrm{DEC}}))\|_2 \leq 4\zeta\rho.$$

*Moreover the movement of the projection of $\Phi(x(\cdot))$ on the manifold is bounded,*

$$\|\Phi(x(t_{\mathrm{GF}})) - \Phi(x(t_{\mathrm{DEC}}))\|_2 = O(\eta\ln(1/\rho)).$$

*Proof of Lemma I.6.* Choose $t_{\mathrm{DEC}}$ as the minimal $t \geq t_{\mathrm{GF}}$ such that $\|\nabla L(x(t_{\mathrm{DEC}}))\|_2 \leq 4\zeta\rho$. Define $C = \lceil\ln_{1-\frac{\eta\mu}{8}}(64\rho^2/h^2)\rceil = O(\ln(1/\rho)/\eta)$.

We will first perform an induction on $t \leq \min\{t_{\mathrm{DEC}}, t_{\mathrm{GF}} + C\} = t_{\mathrm{GF}} + O(\ln(1/\rho)/\eta)$ to show that

$$L(x(t)) \leq (1 - \eta\mu/8)^{t-t_{\mathrm{GF}}} L(x(t_{\mathrm{GF}}))$$

$$\|\Phi(x(t)) - \Phi(x(t_{\mathrm{GF}}))\| = O(\eta^2(t - t_{\mathrm{GF}}))$$

For $t = t_{\mathrm{GF}}$, the result holds trivially. Suppose the induction hypothesis holds for $t$. Then by F.1 and Taylor Expansion,

$$\|\Phi(x(t)) - x(t)\| \leq \sqrt{\frac{2L(x(t_{\mathrm{GF}}))}{\mu}} \leq h/4.$$

Then we have that

$$\begin{aligned}
\mathrm{dist}(K, x(t)) \leq &\mathrm{dist}(K, x(t_{\mathrm{GF}})) + \|x(t_{\mathrm{GF}}) - \Phi(x(t_{\mathrm{GF}}))\|_2 \\
&+ \|\Phi(x(t_{\mathrm{GF}})) - \Phi(x(t))\| + \|\Phi(x(t)) - x(t)\| \\
\leq &3h/4 + O(\eta^2(t - t_{\mathrm{GF}})) = 3h/4 + O(\eta\ln(1/\rho)) \leq h.
\end{aligned}$$

That is $x(t) \in K^h$. Then as $t \leq t_{\mathrm{DEC}}$, $\|\nabla L(x(t))\|_2 \geq 4\zeta\rho$. Then by Lemma I.5, we have that

$$L(x(t+1)) \leq (1 - \eta\mu/8)L(x(t)) \leq (1 - \eta\mu/8)^{t+1-t_{\mathrm{GF}}} L(x(t_{\mathrm{GF}})),$$

$$\begin{aligned}
\|\Phi(x(t+1)) - \Phi(x(t_{\mathrm{GF}}))\| \leq &\|\Phi(x(t+1)) - \Phi(x(t))\| + \|\Phi(x(t)) - \Phi(x(t_{\mathrm{GF}}))\| \\
\leq &O(\eta^2(t - t_{\mathrm{GF}})),
\end{aligned}$$

which completes the induction.

Now if $t_{\mathrm{DEC}} \geq t_{\mathrm{GF}} + C = t_{\mathrm{GF}} + \Omega(\ln(1/\rho)/\eta)$, As the result of the induction, we have that

$$L(x(t_{\mathrm{GF}} + C)) \leq (1 - \frac{\eta\mu}{8})^C L(x(t_{\mathrm{GF}})) \leq \frac{64\rho^2}{h^2} L(x(t_{\mathrm{GF}})) \leq 8\rho^2\mu.$$

By Lemma F.2, we have that $\|\nabla L(x(t_{\mathrm{GF}}+C))\|_2 \leq \zeta\sqrt{\frac{2L(x(t_{\mathrm{GF}}+C))}{\mu}} = 4\zeta\rho$, which leads to a contradiction.

Hence we have that $t_{\mathrm{DEC}} \leq t_{\mathrm{GF}} + C = t_{\mathrm{GF}} + O(\ln(1/\rho)/\eta)$. By induction, we have that

$$\|\Phi(x(t_{\mathrm{DEC}})) - \Phi(x(t_{\mathrm{GF}}))\| = O(\eta^2(t_{\mathrm{DEC}} - t_{\mathrm{GF}})) = O(\eta\ln(1/\rho)).$$

This completes the proof. □

### I.1.3 ENTERING INVARIANT SET

We first introduce some notations that is required for the proof in this and following subsection.

Define

$$\begin{aligned}
\hat{x} &= x - \Phi(x), \\
A(x) &= \nabla^2 L(\Phi(x)), \\
\tilde{x} &= A(x)\hat{x}, \\
\tilde{x}_j &= \langle\tilde{x}, v_j(x)\rangle, \\
P^{(j:D)}(x) &= \sum_{i=j}^{M} v_i(x)v_i^T(x).
\end{aligned}$$

Note $\tilde{x} \approx \nabla L(x)$ for $x$ near the manifold $\Gamma$. We also use $\tilde{x}(t)$, $A(t)$ and $\hat{x}(t)$ to denote $\tilde{x(t)}$, $A(x(t))$ and $\hat{x(t)}$.

Recall the original definition of $R_j(x)$ is

$$R_j(x) = \sqrt{\sum_{i=j}^{M} \lambda_i^2(x)\langle v_i(x), x - \Phi(x)\rangle^2 - \eta\rho\lambda_j^2(x)},$$

Based on the above notions, we can rephrase the notion $R$ as

$$R_j(x) = \|P^{(j:D)}(x)\tilde{x}\| - \eta\rho\lambda_j^2(x).$$

We additionally define the approximate invariant set $\mathbb{I}_j$ as

$$\mathbb{I}_j = \{\|P^{(j:D)}(x)\tilde{x}\| \leq \eta\rho\lambda_j^2(x) + O(\eta\rho^2)\}.$$

**Lemma I.7.** *Assuming $t$ satisfy that $x(t) \in K^h$, then we have that*

$$\frac{\mu}{2}\|x(t) - \Phi(x(t))\| \leq \|\tilde{x}(t)\| \leq \zeta\|x(t) - \Phi(x(t))\|$$

*Proof of Lemma I.7.* First by Lemma F.4, $\Phi(x(t)) \in K^r$, hence

$$\|\tilde{x}(t)\| = \|\nabla^2 L(\Phi(x(t)))(x(t) - \Phi(x(t)))\| \leq \zeta\|x(t) - \Phi(x(t))\|.$$

Also

$$\|\tilde{x}(t)\| = \|\nabla^2 L(\Phi(x(t)))(x(t) - \Phi(x(t)))\| \geq \mu\|P_{\Phi(x(t)),\Gamma}^{\perp}(x(t) - \Phi(x(t)))\|.$$

By Lemma F.4 and Lemma E.6, we have

$$\|x(t) - \Phi(x(t))\| \leq \|P_{\Phi(x(t)),\Gamma}^{\perp}(x(t) - \Phi(x(t)))\| + \|P_{\Phi(x(t)),\Gamma}(x(t) - \Phi(x(t)))\|$$

$$\leq \frac{\zeta\nu}{4\mu^2}\|x(t) - \Phi(x(t))\|^2 + \frac{1}{\mu}\|\tilde{x}(t)\|$$

$$\leq \frac{1}{2}\|x(t) - \Phi(x(t))\| + \frac{1}{\mu}\|\tilde{x}(t)\|.$$

Hence $\|x(t) - \Phi(x(t))\| \leq \frac{2}{\mu}\|\tilde{x}(t)\|$. $\qquad\square$

**Lemma I.8.** *Assuming $t$ satisfy that $x(t) \in K^h$ and $\|\tilde{x}(t)\|_2 = O(\rho)$, then we have that*

$$\|\Phi(x(t+1)) - \Phi(x(t))\| = O(\eta\rho^2).$$

*Proof of Lemma I.8.* By Lemma I.7, we have $\|x(t) - \Phi(x(t))\| = O(\rho)$. By Lemma F.7, we have that

$$\|\Phi(x(t+1)) - \Phi(x(t))\| \leq \zeta\xi\eta\rho\|x - \Phi(x)\|_2 + \zeta^2\xi\eta^2\|x - \Phi(x)\|_2^2 + \nu\eta\rho^2 + \xi\zeta^2\eta^2\rho^2$$

$$\leq O(\eta\rho^2).$$

$\qquad\square$

**Lemma I.9.** *Assuming $t$ satisfy $x(t) \in K^{h/2}$ and $\|x(t) - \Phi(x(t))\|_2 = O(\rho)$, define $x'$ as $x'(t) = x(t)$ and for $\tau \geq t$,*

$$x'(\tau+1) = x'(\tau) - \eta\nabla^2 L(\Phi(x(t)))(x'(\tau) - \Phi(x(t)))$$

$$-\eta\rho\nabla^2 L(\Phi(x(t)))\frac{\nabla^2 L(\Phi(x(t)))(x'(\tau) - \Phi(x(t)))}{\|\nabla^2 L(\Phi(x(t)))(x'(\tau) - \Phi(x(t)))\|_2}$$

*Then*

$$\|x'(t+1) - x(t+1)\|_2 = O(\eta\rho^2)$$

*and further if $\|x(t+1) - \Phi(x(t+1))\|_2 = \Omega(\eta\rho)$, then*

$$\|x'(t+2) - x(t+2)\|_2 = O(\eta\rho^2).$$

*Proof of Lemma I.9.* By $\|x(t) - \Phi(x(t))\| = O(\rho)$, $x(t) \in K^{h/2}$, and Lemma F.7, we have that $\|x(t+1) - x(t)\| = O(\eta\rho)$ and hence $x(t+1) \in K^{3h/4}$. This also implies $\|x(t+1) - \Phi(x(t+1))\|_2 = O(\rho)$. Similarly we have $x(t+2) \in K^{3h/4}$.

For $k \in \{1, 2\}$, by Taylor Expansion,

$$x(t+k+1) = x(t+k) - \eta \nabla L(x(t+k)) - \eta\rho \nabla^2 L(x(t+k)) \frac{\nabla L(x(t+k))}{\|\nabla L(x(t+k))\|} + O(\eta\rho^2)$$

$$= x(t+k) - \eta \nabla^2 L(\Phi(x(t+k)))(x(t+k) - \Phi(x(t+k))) + O(\eta\rho^2)$$

$$- \eta\rho \nabla^2 L(\Phi(x(t+k))) \frac{\nabla L(x(t+k))}{\|\nabla L(x(t+k))\|} + O(\eta\rho^2)$$

$$= x(t+k) - \eta \nabla^2 L(\Phi(x(t+k)))(x(t+k) - \Phi(x(t+k)))$$

$$- \eta\rho \nabla^2 L(\Phi(x(t+k))) \frac{\nabla L(x(t+k))}{\|\nabla L(x(t+k))\|} + O(\eta\rho^2).$$

Now by Lemmas I.7 and I.8, $\|\Phi(x(t+k)) - \Phi(x(t))\|_2 = O(\eta\rho^2)$,

$$x(t+k+1) = x(t+k) - \eta \nabla^2 L(\Phi(x))(x(t+k) - \Phi(x(t)))$$

$$- \eta\rho \nabla^2 L(\Phi(x)) \frac{\nabla L(x(t+k))}{\|\nabla L(x(t+k))\|} + O(\eta\rho^2). \tag{27}$$

Now we first prove the first claim, we have for $k = 0$, $\|x(t+k) - x'(t+k)\|_2 = 0$, by Lemma F.4 and eq. 27,

$$x(t+1) = x(t) - \eta \nabla^2 L(\Phi(x))(x(t) - \Phi(x)) - \eta\rho \frac{\nabla^2 L(\Phi(x))(x(t) - \Phi(x))}{\|\nabla^2 L(\Phi(x))(x(t) - \Phi(x))\|_2} + O(\eta\rho^2)$$

$$= x'(t+1) + O(\eta\rho^2).$$

The second claim is slightly more complex. By the first claim and Lemma F.4, we have that

$$\frac{\nabla L(x(t+1))}{\|\nabla L(x(t+1))\|} = \frac{\nabla^2 L(\Phi(x(t+1)))(x(t+1) - \Phi(x(t+1)))}{\|\nabla^2 L(\Phi(x(t+1)))(x(t+1) - \Phi(x(t+1)))\|_2}$$

$$+ O(\|x(t+1) - \Phi(x(t+1))\|_2). \tag{28}$$

We first show $\|\nabla^2 L(\Phi(x(t+1)))(x(t+1) - \Phi(x(t+1)))\|_2$ is of order $\|x(t+1) - \Phi(x(t+1))\|_2 = \Omega(\rho^2)$ to show that the normalized gradient term is stable with respect to small perturbation,

$$\|\nabla^2 L(\Phi(x(t+1)))(x(t+1) - \Phi(x(t+1)))\|_2$$

$$\geq \|P_{\Phi(x(t+1)),\Gamma} \nabla^2 L(\Phi(x(t+1)))(x(t+1) - \Phi(x(t+1)))\|_2$$

$$\geq \|\nabla^2 L(\Phi(x(t+1))) P_{\Phi(x(t+1)),\Gamma}(x(t+1) - \Phi(x(t+1)))\|_2$$

$$\geq \mu \|P_{\Phi(x(t+1)),\Gamma}(x(t+1) - \Phi(x(t+1)))\|_2$$

$$\geq \mu(\|(x(t+1) - \Phi(x(t+1)))\|_2 - \|P_{\Phi(x(t+1)),\Gamma}^{\perp}(x(t+1) - \Phi(x(t+1)))\|_2)$$

$$\geq \mu(\|(x(t+1) - \Phi(x(t+1)))\|_2 - \frac{\nu\zeta}{4\mu^2}\|x(t+1) - \Phi(x(t+1))\|_2^2)$$

$$\geq \frac{\mu}{2}\|(x(t+1) - \Phi(x(t+1)))\|_2 = \Omega(\eta\rho).$$

Based on Lemma F.7, we have

$$\Phi(x(t+1)) - \Phi(x(t)) = O(\eta\rho^2).$$

We further have by the first claim and Lemma I.8,

$$\nabla^2 L(\Phi(x(t+1)))(x(t+1) - \Phi(x(t+1))) - \nabla^2 L(\Phi(x))(x'(t+1) - \Phi(x(t)))$$

$$= \nabla^2 L(\Phi(x))(x(t+1) - \Phi(x(t+1))) - \nabla^2 L(\Phi(x))(x'(t+1) - \Phi(x(t)))$$

$$+ O(\|x(t+1) - \Phi(x(t+1))\|_2 \|\Phi(x(t+1)) - \Phi(x)\|_2)$$

$$= \nabla^2 L(\Phi(x))(x(t+1) - \Phi(x(t+1))) - \nabla^2 L(\Phi(x))(x'(t+1) - \Phi(x(t))) + O(\eta\rho^3)$$

$$= \nabla^2 L(\Phi(x))(x(t+1) - x'(t+1)) + \nabla^2 L(\Phi(x))(\Phi(x(t+1)) - \Phi(x(t))) + O(\eta\rho^3)$$

$$= O(\eta\rho^2)$$

This implies

$$\frac{\nabla^2 L(\Phi(x(t+1)))(x(t+1) - \Phi(x(t+1)))}{\|\nabla^2 L(\Phi(x(t+1)))(x(t+1) - \Phi(x(t+1)))\|_2} = \frac{\nabla^2 L(\Phi(x))(x'(t+1) - \Phi(x))}{\|\nabla^2 L(\Phi(x))(x'(t+1) - \Phi(x))\|_2} + O(\rho)$$

Combining with Equation 28, we have

$$\frac{\nabla L(x(t+1))}{\|\nabla L(x(t+1))\|} = \frac{\nabla^2 L(\Phi(x))(x'(t+1) - \Phi(x))}{\|\nabla^2 L(\Phi(x))(x'(t+1) - \Phi(x))\|_2} + O(\rho)$$

By the above approximation and Equation 27,

$$x(t+2) = x'(t+2) + O(\eta\rho^2).$$

$\square$

**Lemma I.10.** *Assuming $t$ satisfy that $x(t) \in K^{3h/4}$ and $\|\tilde{x}(t)\|_2 = O(\rho)$, then we have that*

$$\|\tilde{x}(t+1) - \tilde{x}(t) + \eta A(t)\tilde{x}(t) + \eta\rho A^2(t)\frac{\tilde{x}(t)}{\|\tilde{x}(t)\|}\|_2 = O(\eta\rho^2).$$

*Proof of Lemma I.10.* By Lemma I.9, we know

$$\|x(t+1) - x(t) + \eta\tilde{x}(t) + \eta\rho A(t)\frac{\tilde{x}(t)}{\|\tilde{x}(t)\|}\| \le O(\eta\rho^2).$$

This implies

$$\|A(t)(x(t+1) - \Phi(x(t))) - \tilde{x}(t) + \eta A(t)\tilde{x}(t) + \eta\rho A^2(t)\frac{\tilde{x}(t)}{\|\tilde{x}(t)\|}\|$$

$$=\|A(t)(x(t+1) - x(t) + \eta\tilde{x}(t) + \eta\rho A(t)\frac{\tilde{x}(t)}{\|\tilde{x}(t)\|})\|$$

$$\le\zeta\|x(t+1) - x(t) + \eta\tilde{x}(t) + \eta\rho A(t)\frac{\tilde{x}(t)}{\|\tilde{x}(t)\|}\| = O(\eta\rho^2). \qquad (29)$$

We also have

$$\tilde{x}(t+1) - A(t)(x(t+1) - \Phi(x(t)))$$
$$=(A(t+1) - A(t))(x(t+1) - \Phi(x(t+1))) - A(t)(\Phi(x(t)) - \Phi(x(t+1)))$$
$$=O(\eta\rho^2).$$

Plugging in Equation 29, we have that

$$\|\tilde{x}(t+1) - \tilde{x}(t) + \eta A(t)\tilde{x}(t) + \eta\rho A^2(t)\frac{\tilde{x}(t)}{\|\tilde{x}(t)\|}\|_2 = O(\eta\rho^2).$$

$\square$

**Lemma I.11.** *Under condition of Theorem I.1, assuming there exists $t_{\mathrm{DEC}}$ such that $x(t_{\mathrm{DEC}}) \in K^{h/2}$ and $\|\nabla L(x(t_{\mathrm{DEC}}))\| \le 4\zeta\rho$, then there exists $t_{\mathrm{DEC2}} = t_{\mathrm{DEC}} + O(\ln(1/\eta)/\eta)$, such that $x(t_{\mathrm{DEC2}})$ is in $\mathbb{I}_1 \cap K^{3h/4}$.*

*Furthermore, for any $t$ satisfying $t_{\mathrm{DEC2}} \le t \le t_{\mathrm{DEC2}} + \Theta(\ln(1/\eta)/\eta)$, we have that $x(t) \in \mathbb{I}_1 \cap K^{3h/4}$ and $\|\Phi(x(t)) - \Phi(x(t_{\mathrm{DEC}}))\| = O(\rho^2\ln(1/\eta))$.*

*Proof of Lemma I.11.* For simplicity, denote $C = \lceil\ln_{1-\eta\mu}\frac{\eta\mu^3}{4\zeta^2}\rceil + \Theta(\ln(1/\rho)/\eta) = O(\ln(1/\eta)/\eta)$. Here the quantity $\Theta(\ln(1/\rho)/\eta)$ is the same quantity in the statement of the lemma.

We will prove the induction hypothesis for $t_{\mathrm{DEC}} \le t \le t_{\mathrm{DEC}} + 2C$,

$$\begin{cases} \|\tilde{x}(t-1)\| \ge \eta\rho\lambda_1^2(t), t > t_{\mathrm{DEC}} \Rightarrow \|\tilde{x}(t)\| \le (1 - \eta\mu)\|\tilde{x}(t-1)\|, \\ \|\tilde{x}(t-1)\| \le \eta\rho\lambda_1^2(t-1), t > t_{\mathrm{DEC}} \Rightarrow \|\tilde{x}(t)\| \le \eta\rho\lambda_1^2(t) + O(\eta\rho^2), \\ \qquad\qquad\qquad \|\Phi(x(t)) - \Phi(x(t_{\mathrm{DEC}}))\| \le O(\eta\rho^2(t - t_{\mathrm{DEC}})), \\ \qquad\qquad\qquad\qquad\qquad x(t) \in K^{3h/4}. \end{cases}$$

The induction hypothesis holds trivially for $t = t_{\text{DEC}}$.

Assume the induction hypothesis holds for $t' \leq t$. By Lemmas F.1 and I.7, $\|\tilde{x}(t_{\text{DEC}})\|_2 \leq \zeta\|x(t_{\text{DEC}}) - \Phi(x(t_{\text{DEC}}))\| \leq \frac{\zeta}{\mu}\|\nabla L(x(t_{\text{DEC}}))\| \leq \frac{4\zeta^2}{\mu}\rho$. Combining with the induction hypothesis, we have $\|\tilde{x}(t)\| \leq \frac{4\zeta^2}{\mu}\rho$.

By $x(t) \in K^{3h/4}$ and Lemma I.8, we have that

$$\|\Phi(x(t+1)) - \Phi(x(t))\| \leq O(\eta\rho^2).$$

Hence we have that

$$\|\Phi(x(t+1)) - \Phi(x(t_{\text{DEC}}))\| \leq \|\Phi(x(t+1)) - \Phi(x(t))\| + \|\Phi(x(t)) - \Phi(x(t_{\text{DEC}}))\|$$
$$\leq O(\eta\rho^2(t+1-t_{\text{DEC}})). \tag{30}$$

This proves the third statement of the induction hypothesis.

By $\|\tilde{x}(t)\| = O(\rho)$ and Lemma I.10, we have that

$$\|\tilde{x}(t+1) - \tilde{x}(t) + \eta A(t)\tilde{x}(t) + \eta\rho A^2(t)\frac{\tilde{x}(t)}{\|\tilde{x}(t)\|}\|_2 = O(\eta\rho^2).$$

Analogous to the proof of Lemmas H.1 and H.2, we have

1. If $\|\tilde{x}(t)\| > \eta\rho\lambda_1^2(t)$, we would have

$$\|\tilde{x}(t) - \eta A(t)\tilde{x}(t) - \eta\rho A^2(t)\frac{\tilde{x}(t)}{\|\tilde{x}(t)\|}\|$$

$$\leq\|\tilde{x}(t)\|\|I - \eta A(t) - \eta\rho A^2(t)\frac{1}{\|\tilde{x}(t)\|}\|$$

$$\leq\|\tilde{x}(t)\|\max\{\eta\lambda_1, 1 - \eta\lambda_D - \eta\rho\lambda_D^2\frac{1}{\|x(t)\|}\}$$

$$\leq\max\{(1-\eta\lambda_D)\|\tilde{x}(t)\| - \eta\rho\lambda_D^2, \eta\lambda_1\|\tilde{x}(t)\|\}$$

$$\leq\max\{(1-\eta\mu)\|\tilde{x}(t)\| - \eta\rho\mu^2, \eta\zeta\|\tilde{x}(t)\|\}$$

Hence we have

$$\|\tilde{x}(t+1)\| \leq \max\{(1-\eta\mu)\|\tilde{x}(t)\| - \eta\rho\mu^2, \eta\zeta\|\tilde{x}(t)\|\} + O(\eta\rho^2) \leq (1-\eta\mu)\|\tilde{x}(t)\|.$$

2. If $\|\tilde{x}(t)\|_2 \leq \eta\rho\lambda_1^2(t)$, then by Lemma H.1, we have that

$$\|\tilde{x}(t) - \eta A(t)\tilde{x}(t) - \eta\rho A^2(t)\frac{\tilde{x}(t)}{\|\tilde{x}(t)\|}\|_2 \leq \eta\rho\lambda_1^2(t).$$

Hence by Lemma K.1

$$\|\tilde{x}(t+1)\| \leq \eta\rho\lambda_1^2(t) + O(\eta\rho^2) \leq \eta\rho\lambda_1^2(t+1) + O(\eta\rho^2).$$

Concluding the two cases, we have shown the first and second claim of the induction hypothesis holds. Hence we can show that $\|\tilde{x}(t+1)\| \leq \frac{4\zeta^2}{\mu}\rho$. Then by Lemma I.7, we have that $\|x(t+1) - \Phi(x(t+1))\| \leq \frac{8\zeta^2}{\mu^2}\rho$.

As $t \leq t_{\text{DEC}} + 2C = t_{\text{DEC}} + O(\ln(1/\eta)/\eta)$, by Equation 30,

$$\|\Phi(x(t+1)) - \Phi(x(t_{\text{DEC}}))\| \leq O(-\rho^2 \ln \eta).$$

This implies

$$\text{dist}(x(t+1), K) \leq \text{dist}(x(t_{\text{DEC}}), K) + \|x(t_{\text{DEC}}) - \Phi(x(t_{\text{DEC}}))\|$$
$$+ \|\Phi(x(t_{\text{DEC}})) - \Phi(x(t+1))\| + \|x(t+1) - \Phi(x(t+1))\|$$
$$= h/2 + O(\rho^2 \ln(1/\eta)) + O(\rho) \leq 3h/4.$$

This proves the fourth claim of the inductive hypothesis.

The induction is complete.

Now define $t_{\text{DEC2}}$ the minimal $t \geq t_{\text{DEC}}$, such that $\|\tilde{x}(t)\| \leq \eta\rho\lambda_1^2(t)$.

If $t_{\text{DEC2}} > t_{\text{DEC}} + C$, then by the induction, Lemmas F.1 and I.7,

$$\|\tilde{x}(t_{\text{DEC}} + C)\| \leq (1 - \eta\mu)^C \|\tilde{x}(t_{\text{DEC}})\|$$

$$\leq \frac{\eta\mu^3}{4\zeta^2}\|\tilde{x}(t_{\text{DEC}})\|$$

$$\leq \frac{\eta\mu^3}{4\zeta^2}\zeta\|x(t_{\text{DEC}}) - \Phi(x(t_{\text{DEC}}))\|$$

$$\leq \frac{\eta\mu^2}{4\zeta}\|\nabla L(t_{\text{DEC}})\|$$

$$\leq \mu^2\eta\rho$$

$$\leq \lambda_1^2(t_{\text{DEC}} + C)\eta\rho.$$

This is a contradiction. Hence we have $t_{\text{DEC2}} \leq t_{\text{DEC}} + C$. By the induction hypothesis $x(t_{\text{DEC2}}) \in \mathbb{I}_1 \cap K^{3h/4}$.

Furthermore by induction, for any $t$ satisfying $t_{\text{DEC2}} \leq t \leq t_{\text{DEC}} + 2C$, we have that

$$\|\tilde{x}(t)\| \leq \eta\rho\lambda_1^2(t) + O(\eta\rho^2).$$

By the induction hypothesis $x(t) \in \mathbb{I}_1 \cap K^{3h/4}$ and $\|\Phi(x(t)) - \Phi(x(t_{\text{DEC}}))\| = O(\rho^2 \ln(1/\rho))$. □

**Lemma I.12.** *Under condition of Theorem I.1, assuming $t$ satisfy that $x(t) \in \mathbb{I}_1 \cap K^{3h/4}$, then we have that*

$$\begin{cases} R_k(x(t)) \geq 0 \Rightarrow R_k(x(t+1)) + \lambda_k^2(t+1)\eta\rho \leq (1 - \eta\mu)(R_k(x(t)) + \lambda_k^2(t)\eta\rho), \\ R_k(x(t)) \leq 0 \Rightarrow R_k(x(t+1)) \leq O(\eta\rho^2). \end{cases}$$

*Proof of Lemma I.12.* As $x(t) \in \mathbb{I}_1$, $\|\tilde{x}(t)\|_2 \leq \zeta\eta\rho + O(\eta\rho^2)$.

As $\|\tilde{x}(t)\|_2 = O(\rho)$, we have $\overline{x(t)x(t+1)} \subset K^h$ and $\overline{\Phi(x(t))\Phi(x(t+1))} \subset K^r$.

We will begin with a quantization technique separating $[M]$ into disjoint continuous subset $S_1, ..., S_p$ such that $\forall i \neq j$,

$$\min_{k \in S_i, l \in S_j} |\lambda_k(t) - \lambda_l(t)| \geq \rho.$$

By Lemmas I.8 and K.1, we have that for any $n \in [M]$,

$$|\lambda_k(t) - \lambda_k(t+1)| = O(\|\nabla^2 L(\Phi(x(t))) - \nabla^2 L(\Phi(x(t+1)))\|)$$

$$= O(\|\Phi(x(t)) - \Phi(x(t+1))\|)$$

$$= O(\eta\rho^2).$$

This implies

$$\min_{k \in S_i, l \in S_j} |\lambda_k(t+1) - \lambda_l(t+1)| \geq \rho - O(\eta\rho^2) \geq 0.99\rho.$$

Define

$$P_{S^{(i)}}^{(t)} \triangleq \sum_{k \in S_i} v_n(t)v_n(t)^T.$$

By Theorem K.3, for any $k$,

$$\|P_{S_k}^{(t)} - P_{S_k}^{(t+1)}\| \leq O\left(\frac{\|\nabla^2 L(\Phi(x(t))) - \nabla^2 L(\Phi(x(t+1)))\|}{\rho}\right) = O(\eta\rho).$$

By Lemma I.10, we have that

$$\|\tilde{x}(t+1) - \tilde{x}(t) + \eta A(t)\tilde{x}(t) + \eta\rho A^2(t)\frac{\tilde{x}(t)}{\|\tilde{x}(t)\|}\|_2 = O(\eta\rho^2).$$

We will write $x'(t+1)$ as shorthand of $\tilde{x}(t) - \eta A(t)\tilde{x}(t) - \eta\rho A^2(t)\frac{\tilde{x}(t)}{\|\tilde{x}(t)\|}$.

Now we discuss by cases,

1. If $\sqrt{\sum_{i=j}^{p} \|P_{S^{(i)}}^{(t)} \tilde{x}(t)\|^2} > \max_{k \in S_j} \lambda_k^2(t) \eta \rho > \mu^2 \eta \rho$, by Lemma H.3,

$$\sqrt{\sum_{i=j}^{p} \|P_{S^{(i)}}^{(t)} \tilde{x}(t+1)\|^2} \leq \sqrt{\sum_{i=j}^{p} \|P_{S^{(i)}}^{(t)} x'(t+1)\|^2} + O(\eta \rho^2)$$

$$\leq \max\{(1 - \eta \lambda_D(t+1))\| \sum_{i=j}^{p} P_{S^{(i)}}^{(t)} \tilde{x}(t)\| - \eta \rho \lambda_D(t+1)^2 \frac{\| \sum_{i=j}^{p} P_{S^{(i)}}^{(t)} \tilde{x}(t)\|}{\|\tilde{x}(t)\|},$$

$$\eta \max_{k \in S_j} \lambda_k(t+1)\| \sum_{i=j}^{p} P_{S^{(i)}}^{(t)} \tilde{x}(t)\|\} + O(\eta \rho^2)$$

$$\leq \max\{(1 - \eta \mu)\| \sum_{i=j}^{p} P_{S^{(i)}}^{(t)} \tilde{x}(t)\| - \eta \rho \frac{\mu^3}{2\zeta}, \eta \zeta \| \sum_{i=j}^{p} P_{S^{(i)}}^{(t)} \tilde{x}(t)\|\} + O(\eta \rho^2).$$

This further implies

$$\sqrt{\sum_{i=j}^{p} \|P_{S^{(i)}}^{(t+1)} \tilde{x}(t+1)\|^2} \leq \sqrt{\sum_{i=j}^{p} \|P_{S^{(i)}}^{(t)} \tilde{x}(t+1)\|^2} + O(\eta \rho \|\tilde{x}(t+1)\|)$$

$$\leq \max\{(1 - \eta \mu)\| \sum_{i=j}^{p} P_{S^{(i)}}^{(t)} \tilde{x}(t)\| - \eta \rho \frac{\mu^3}{2\zeta}, \eta \zeta \| \sum_{i=j}^{p} P_{S^{(i)}}^{(t)} \tilde{x}(t)\|\} + O(\eta \rho^2)$$

$$\leq (1 - \eta \mu)\| \sum_{i=j}^{p} P_{S^{(i)}}^{(t)} \tilde{x}(t)\|.$$

2. If $\sqrt{\sum_{i=j}^{p} \|P_{S^{(i)}}^{(t)} \tilde{x}(t)\|^2} \leq \max_{k \in S_j} \lambda_k^2(t) \eta \rho$, then by Lemma H.1, we have that

$$\| \sum_{i=j}^{p} P_{S^{(i)}}^{(t)} x'(t+1)\|_2 \leq \eta \rho \max_{k \in S_j} \lambda_k^2(t).$$

Hence we have that

$$\sqrt{\sum_{i=j}^{p} \|P_{S^{(i)}}^{(t)} \tilde{x}(t+1)\|^2} \leq \sqrt{\sum_{i=j}^{p} \|P_{S^{(i)}}^{(t)} x'(t+1)\|^2} + O(\eta \rho^2)$$

$$\leq \max_{k \in S_j} \lambda_k^2(t) \eta \rho + O(\eta \rho^2)$$

$$\leq \max_{k \in S_j} \lambda_k^2(t+1) \eta \rho + O(\eta \rho^2).$$

This further implies

$$\sqrt{\sum_{i=j}^{p} \|P_{S^{(i)}}^{(t+1)} \tilde{x}(t+1)\|^2} \leq \sqrt{\sum_{i=j}^{p} \|P_{S^{(i)}}^{(t)} \tilde{x}(t+1)\|^2} + O(\eta \rho \|\tilde{x}(t+1)\|)$$

$$\leq \max_{k \in S_j} \lambda_k^2(t) \eta \rho + O(\eta \rho^2)$$

$$\leq \max_{k \in S_j} \lambda_k^2(t+1) \eta \rho + O(\eta \rho^2).$$

Finally taking into quantization error, as all the eigenvalue in the same group at most differ $D\rho$, for any $i \in S_j$, we have that $-\lambda_i^2(t+1) + \max_{k \in S_j} \lambda_k^2(t+1) \leq 2D\zeta \rho + D^2 \rho^2$.

Hence the previous discussion concludes as

1. If $R_k(x(t)) \geq 0$

$$R_k(x(t+1)) + \lambda_k^2(t+1) \eta \rho \leq (1 - \eta \mu)(R_k(x(t)) + \lambda_k^2(t).\eta \rho)$$

2. If $R_k(x(t)) < 0$

$$R_k(x(t+1)) \leq O(\eta\rho^2).$$

□

**Lemma I.13.** *Under condition of Theorem I.1, assuming there exists $t_{\mathrm{DEC2}}$ such that for any $t$ satisfying $t_{\mathrm{DEC2}} \leq t \leq t_{\mathrm{DEC2}} + \Theta(\ln(1/\eta)/\eta)$, we have that $x(t) \in \mathbb{I}_1 \cap K^{3h/4}$. Then there exists $t_{\mathrm{INV}} = t_{\mathrm{DEC2}} + O(\ln(1/\eta)/\eta))$ such that for any $t$ satisfying $t_{\mathrm{INV}} \leq t \leq t_{\mathrm{INV}} + \Theta(\ln(1/\eta)/\eta)$, we have that*

$$x(t) \in (\cap_{k \in [M]} \mathbb{I}_k) \cap K^{7h/8}.$$

$$\|\Phi(x(t)) - \Phi(x(t_{\mathrm{DEC2}}))\| = O(\rho^2 \ln(1/\eta)).$$

*Proof of Lemma I.13.* The proof is almost identical with Lemma I.11 replacing the first two iterative hypothesis to Lemma I.12 and is omitted here. □

### I.2 PHASE II (PROOF OF THEOREM I.3)

*Proof of Theorem I.3.* Let $t_{\mathrm{ALIGN}} = O(\ln(1/\rho)/\eta)$ be the quantity defined in Lemma I.19.

We will inductively prove the following induction hypothesis $\mathcal{P}(t)$ holds for $t_{\mathrm{ALIGN}} \leq t \leq T_3/\eta\rho^2 + 1$,

$$x(t) \in K^{h/2}, t_{\mathrm{ALIGN}} \leq \tau \leq t$$

$$|\langle x(\tau) - \Phi(x(\tau)), v_1(x(\tau)) \rangle| = \Theta(\eta\rho), t_{\mathrm{ALIGN}} \leq \tau \leq t$$

$$\max_{j \in [2:M]} |\langle x(\tau) - \Phi(x(\tau)), v_j(x(\tau)) \rangle| = O(\eta\rho^2), t_{\mathrm{ALIGN}} \leq \tau \leq t$$

$$\|\Phi(x(\tau)) - X(\eta\rho^2\tau)\| = O(\eta\ln(1/\rho)), t_{\mathrm{ALIGN}} \leq \tau \leq t$$

$\mathcal{P}(t_{\mathrm{ALIGN}})$ holds due to Lemma I.19. Now suppose $\mathcal{P}(t)$ holds, then $x(t+1) \in K^h$. By Lemma I.19 again, $|\langle x(t+1) - \Phi(x(t+1)), v_1(x(t+1)) \rangle| = \Theta(\eta\rho)$ and $\max_{j \in [2:M]} |\langle x(t+1) - \Phi(x(t+1)), v_j(x(t+1)) \rangle| = O(\eta\rho^2)$ holds.

Now by Lemma I.20,

$$\|\Phi(x(\tau+1)) - \Phi(x(\tau)) + \eta\rho^2 P^\perp_{\Phi(x(\tau)),\Gamma} \nabla\lambda_1(t)/2\| = O(\eta\rho^3 + \eta^2\rho^2), t_{\mathrm{ALIGN}} \leq \tau \leq t.$$

By Corollary L.3, let $b(x) = -\partial\Phi(x)\nabla\lambda_1(\nabla^2 L(x))/2$, $p = \eta\rho^2$ and $\epsilon = O(\eta + \rho)$, it holds that

$$\|\Phi(x(\tau)) - X(\eta\rho^2\tau)\|$$
$$= O(\|\Phi(x(t_{\mathrm{ALIGN}})) - \Phi(x_{\mathrm{init}})\| + T_3\eta\rho^2 + (\rho + \eta)T_3)$$
$$= O(\eta\ln(1/\rho)), t_{\mathrm{ALIGN}} \leq \tau \leq t+1$$

This implies $\|x(t+1) - X(\eta\rho^2(t+1))\|_2 \leq \|x(t+1) - \Phi(x(t+1))\|_2 + \|\Phi(x(t+1)) - X(\eta\rho^2(t+1))\|_2 = \tilde{O}(\eta\ln(1/\rho)) < h/2$. Hence $x(t+1) \in K^{h/2}$. Combining with $\mathcal{P}(t)$ holds, we have that $\mathcal{P}(t+1)$ holds. The induction is complete.

Now $\mathcal{P}(\lceil T_3/\eta\rho^2 \rceil)$ is equivalent to our theorem. □

#### I.2.1 ALIGNMENT TO TOP EIGENVECTOR

We will continue to use the notations introduced in Appendix I.1.3.

We further define

$$S = \{t | \|\tilde{x}(t)\| \leq \frac{\eta\lambda_1^2}{2 - \eta\lambda_1}\rho + O(\eta\rho^2)\},$$

$$T = \{t | \|\tilde{x}(t)\| \leq \frac{1}{2}\left(\frac{\eta\lambda_1^2}{2 - \eta\lambda_1} + \frac{\eta\lambda_2^2}{2 - \eta\lambda_2}\right)\rho\},$$

$$U = \{t | \Omega(\rho^2) \leq \|\tilde{x}_1(t)\| \leq \frac{1}{2}\left(\frac{\eta\lambda_1^2}{2 - \eta\lambda_1} + \frac{\eta\lambda_2^2}{2 - \eta\lambda_2}\right)\rho\}.$$

Here the constant in $O$ depends on the constant in $\mathbb{I}_j$ and will be made clear in Lemma I.16.

For $s \in S$, define $\mathrm{next}(s)$ as the smallest integer greater than $s$ in $S$.

**Lemma I.14.** *Under the condition of Theorem I.3, there exist constants $C_1, C_2 < 1$ independent of $\eta$ and $\rho$, if $\|\tilde{x}_1(t)\|_2 \leq \frac{1}{2}\left(\frac{\eta\lambda_1(t)^2}{2-\eta\lambda_1(t)} + \frac{\eta\lambda_2(t)^2}{2-\eta\lambda_2(t)}\right)\rho$ and $x(t) \in (\cap_{j\in[M]}\mathbb{I}_j) \cap K^{7h/8}$, then*

$$\|\tilde{x}(t)\|_2 \geq C_1\frac{\eta\lambda_1^2}{2-\eta\lambda_1}\rho \Rightarrow \|\tilde{x}(t+1)\|_2 \leq C_2\frac{\eta\lambda_1^2}{2-\eta\lambda_1}\rho$$

*Proof of Lemma I.14.* By Lemma I.10, if we write $x'(t+1)$ as shorthand of $\tilde{x}(t)-\eta A(t)\tilde{x}(t)-\eta\rho A^2(t)\frac{\tilde{x}(t)}{\|\tilde{x}(t)\|}$, then $\|\tilde{x}(t+1) - x'(t+1)\| = O(\eta\rho^2)$.

Define $\mathbb{I}_j^{\text{quad}}$ as $\{x|R_j(x) \leq 0\}$. Then we can find a surrogate $x_{\text{sur}}(t)$ such that $x_{\text{sur}}(t) \in (\cap_{j\in[M]}\mathbb{I}_j^{\text{quad}})\cap K^h$ and $\|x_{\text{sur}}(t) - \tilde{x}(t)\|_2 = O(\eta\rho^2)$. We will write $x'_{\text{sur}}(t+1)$ as shorthand of $x_{\text{sur}}(t) - \eta A(t)x_{\text{sur}}(t) - \eta\rho A^2(t)\frac{x_{\text{sur}}(t)}{\|x_{\text{sur}}(t)\|}$.

Let

$$h(t) \triangleq (2-t)\sqrt{\frac{1}{2t}\left(\frac{(\zeta-\Delta)^2}{\zeta^2}+1\right) + (1-\frac{1}{2t}\left(\frac{(\zeta-\Delta)^2}{\zeta^2}+1\right))\max\{\frac{\zeta^2-\mu^2}{\zeta^2}, \frac{(\zeta-\Delta)^2}{\zeta^2}\}}.$$

As $h(1) < 1$, we can choose $C_1 < 1$, such that $h(C_1) < 1$.

We can further choose $C_2 = \max\{(h(C_1)+1)/2, 1-\frac{\mu^2}{3\zeta^2}\} < 1$.

We will discuss by cases

1. If

$$\|\tilde{x}(t)\|_2 \geq \frac{\eta\lambda_1^4}{\lambda_1^2(1-\eta\lambda_D) + (\lambda_1^2 - \lambda_D^2)(1-\eta\lambda_1)}\rho$$

   Then

$$\frac{\|\tilde{x}(t)\|_2}{\frac{\eta\lambda_1^2}{2-\eta\lambda_1}\rho} = \frac{\lambda_1^2(2-\eta\lambda_1)}{\lambda_1^2(1-\eta\lambda_D) + (\lambda_1^2 - \lambda_D^2)(1-\eta\lambda_1)}$$

$$= \frac{\lambda_1^2(2-\eta\lambda_1)}{\lambda_1^2(2-\eta\lambda_1-\eta\lambda_D) - \lambda_D^2(1-\eta\lambda_1)}$$

$$\geq \frac{1}{1-\frac{\lambda_D^2}{\lambda_1^2}\frac{1-\eta\lambda_1}{2-\eta\lambda_1}} \geq 1 + \frac{\lambda_D^2}{\lambda_1^2}\frac{1-\eta\lambda_1}{2-\eta\lambda_1} \geq 1 + \frac{\mu^2}{3\zeta^2}.$$

   In such case we have

$$\|\frac{\tilde{x}(t)}{\|\tilde{x}(t)\|} - \frac{x_{\text{sur}}(t)}{\|x_{\text{sur}}(t)\|}\| = O(\rho).$$

   Then we have $\|x'_{\text{sur}}(t+1) - x'(t+1)\| = O(\eta\rho^2)$. By Lemma H.5, we have that
$$\|\tilde{x}(t+1)\|_2 \leq \|\tilde{x}(t+1) - x'(t+1)\|_2 + \|\tilde{x}(t+1) - x'_{\text{sur}}(t+1)\| + \|x'_{\text{sur}}(t+1)\|$$

$$\leq \max(\frac{\eta\lambda_1^2}{2-\eta\lambda_1}\rho - \eta\rho\frac{\lambda_D^4}{2\lambda_1^2}, \eta\rho\lambda_1^2 - (1-\eta\lambda_1)\|\tilde{x}(t)\|_2) + O(\eta\rho^2)$$

$$\leq \max(1-\frac{\lambda_D^4(2-\eta\lambda_1)}{2\lambda_1^4}, (2-\eta\lambda_1) - (1-\eta\lambda_1)(1+\frac{\mu^2}{3\zeta^2}))\frac{\eta\lambda_1^2}{2-\eta\lambda_1}\rho$$

$$\leq (1-\frac{\mu^2}{3\zeta^2})\frac{\eta\lambda_1^2}{2-\eta\lambda_1}\rho \leq C_2\frac{\eta\lambda_1^2}{2-\eta\lambda_1}\rho.$$

2. If

$$\|\tilde{x}(t)\|_2 \leq \frac{\eta\lambda_1^4}{\lambda_1^2(1-\eta\lambda_D) + (\lambda_1^2 - \lambda_D^2)(1-\eta\lambda_1)}\rho \leq \frac{\eta\lambda_1^2}{1-\eta\lambda_1}\rho.$$

   Then we have

$$\frac{|-\eta\rho\lambda_D^2 + (1-\eta\lambda_D)\|\tilde{x}(t)\|_2|}{\eta\rho\lambda_1^2 - (1-\eta\lambda_1)\|\tilde{x}(t)\|_2} \leq \frac{\lambda_1^2 - \lambda_D^2}{\lambda_1^2}.$$

$$\frac{|\eta\rho\lambda_2^2 - (1-\eta\lambda_2)\|\tilde{x}(t)\|_2|}{\eta\rho\lambda_1^2 - (1-\eta\lambda_1)\|\tilde{x}(t)\|_2} \leq \frac{\lambda_2^2}{\lambda_1^2}.$$

By Lemma H.8,

$$\|x'(t+1)\|_2$$

$$\leq (\eta\rho\lambda_1^2 - (1-\eta\lambda_1)\|\tilde{x}(t)\|_2)\sqrt{\frac{\|\tilde{x}_1^2(t)\|_2}{\|\tilde{x}(t)\|_2^2} + (1 - \frac{\|\tilde{x}_1^2(t)\|_2}{\|\tilde{x}(t)\|_2^2})\max\{\frac{\lambda_1^2 - \lambda_D^2}{\lambda_1^2}, \frac{\lambda_2^2}{\lambda_1^2}\}}$$

$$\leq (\eta\rho\lambda_1^2 - (1-\eta\lambda_1)\|\tilde{x}(t)\|_2)\sqrt{\frac{\|\tilde{x}_1^2(t)\|_2}{\|\tilde{x}(t)\|_2^2} + (1 - \frac{\|\tilde{x}_1^2(t)\|_2}{\|\tilde{x}(t)\|_2^2})\max\{\frac{\zeta^2 - \mu^2}{\zeta^2}, \frac{(\zeta-\Delta)^2}{\zeta^2}\}}.$$

As

$$\|\tilde{x}_1(t)\|_2 \leq \frac{1}{2}\left(\frac{\eta\lambda_1^2}{2 - \eta\lambda_1} + \frac{\eta\lambda_2^2}{2 - \eta\lambda_2}\right)\rho.$$

For $\|\tilde{x}(t)\|_2 \geq \frac{\eta\lambda_1^2}{2-\eta\lambda_1}\rho C_1$,

$$\frac{\|\tilde{x}_1(t)\|_2}{\|\tilde{x}(t)\|_2} \leq \frac{1}{2}\left(\frac{\lambda_2^2(2-\eta\lambda_1)}{\lambda_1^2(2-\eta\lambda_2)} + 1\right)/C_1 \leq \frac{1}{2}\left(\frac{\lambda_2^2}{\lambda_1^2} + 1\right)/C_1 \leq \frac{1}{2C_1}\left(\frac{(\zeta-\Delta)^2}{\zeta^2} + 1\right).$$

After plugging in, we have that

$$\|\tilde{x}(t+1)\|_2 \leq \|x'(t+1)\|_2 + O(\eta\rho^2) \leq h(C_1)\frac{\eta\lambda_1^2}{2-\eta\lambda_1}\rho + O(\eta\rho^2) \leq C_2\frac{\eta\lambda_1^2}{2-\eta\lambda_1}\rho.$$

This concludes the proof. $\square$

**Lemma I.15.** *Under the condition of Theorem I.3, for any $t \geq 0$ satisfying that (1) $x(t) \in (\cap_{j\in[M]}\mathbb{I}_j) \cap K^h$, (2) $t \notin S$, it holds that $t + 1 \in S$.*

*Moreover, if $|\tilde{x}_1(t)| \geq \Omega(\rho^2)$ and $\|\tilde{x}(t)\|_2 \leq \eta\rho\lambda_1^2 - \Omega(\rho^2)$, then it holds that $\|\tilde{x}_1(t+1)\| \geq \Omega(\rho^2)$.*

*Proof of Lemma I.15.* As $t \notin S$, it holds that

$$\|\tilde{x}(t)\| \geq \frac{\eta\lambda_1^2}{2-\eta\lambda_1}\rho + \Theta(\eta\rho^2).$$

By Lemma I.10, if we write $x'(t+1)$ as shorthand of $\tilde{x}(t) - \eta A(t+1)\tilde{x}(t) - \eta\rho A^2(t)\frac{\tilde{x}(t)}{\|\tilde{x}(t)\|}$, then $\|\tilde{x}(t+1) - x'(t+1)\| = O(\eta\rho^2)$.

Define $\mathbb{I}_j^{\text{quad}}$ as $\{x|R_j(x) \leq 0\}$. Then we can find a surrogate $x_{\text{sur}}(t)$ such that $x_{\text{sur}}(t) \in (\cap_{j\in[M]}\mathbb{I}_j^{\text{quad}}) \cap K^h$, and $\|x_{\text{sur}}(t) - \tilde{x}(t)\|_2 = O(\eta\rho^2)$. We will write $x'_{\text{sur}}(t+1)$ as shorthand of $x_{\text{sur}}(t) - \eta A(t)x_{\text{sur}}(t) - \eta\rho A^2(t)\frac{x_{\text{sur}}(t)}{\|x_{\text{sur}}(t)\|}$.

As $\|\tilde{x}(t)\| = \Omega(\eta\rho)$, we have

$$\|\frac{x_{\text{sur}}(t)}{\|x_{\text{sur}}(t)\|} - \frac{\tilde{x}(t)}{\|\tilde{x}(t)\|}\|_2 = O(\rho).$$

Hence we have that $\|\tilde{x}(t+1) - x'_{\text{sur}}(t+1)\| = \|\tilde{x}(t+1) - x'(t+1)\| + \|x'(t+1) - x'_{\text{sur}}(t+1)\| = O(\eta\rho^2)$

Notice we have $\|x_{\text{sur}}(t)\|_2 \geq \frac{\eta\lambda_1^2}{2-\eta\lambda_1}\rho$ for properly chosen function in the definition $S$, hence, by Lemma H.5

$$\|x'_{\text{sur}}(t+1)\|_2 \leq \frac{\eta\lambda_1^2}{2-\eta\lambda_1}\rho.$$

This further implies $t + 1 \in S$.

We also have

$$|\langle x'_{\text{sur}}(t+1), v_1\rangle| = |\langle x_{\text{sur}}(t), v_1\rangle - \eta\lambda_1\langle x_{\text{sur}}(t), v_1\rangle - \eta\rho\lambda_1^2\frac{\langle x_{\text{sur}}(t), v_1\rangle}{\|x_{\text{sur}}(t)\|}|$$

$$= |\langle x_{\text{sur}}(t), v_1\rangle|(\eta\lambda_1 + \frac{\eta\rho\lambda_1^2}{\|x_{\text{sur}}(t)\|} - 1)$$

We will discuss by cases. Let $C$ satisfies that $C = \sqrt{\frac{1}{2}(\lambda_2^4 + \lambda_1^4)}$.

1. If $\|x_{\mathrm{sur}}(t)\| \leq C\eta\rho$, then as we have $\frac{\lambda_1^2}{C} \geq \frac{\sqrt{2}\zeta^2}{\sqrt{\zeta^2 + (\zeta - \Delta)^2}}$.

$$|\langle x'_{\mathrm{sur}}(t+1), v_1 \rangle| \geq |\langle x_{\mathrm{sur}}(t), v_1 \rangle|(\frac{\lambda_1^2}{C} - 1) \geq \Omega(\rho^2).$$

2. If $\|x_{\mathrm{sur}}(t)\| \geq C\eta\rho$, then as $x(t) \in \mathbb{I}_2$, we have that $|\langle x_{\mathrm{sur}}(t), v_1 \rangle| \geq \Omega(\eta\rho)$. Then as $\|x_{\mathrm{sur}}(t)\| \leq \|\tilde{x}(t)\|_2 + O(\eta\rho^2) \leq \lambda_1^2 \eta\rho - \Omega(\rho^2)$, we have that

$$|\langle x'_{\mathrm{sur}}(t+1), v_1 \rangle| \geq |\langle x_{\mathrm{sur}}(t), v_1 \rangle|(\frac{\lambda_1^2 \eta\rho}{\lambda_1^2 \eta\rho - \Omega(\rho^2)} - 1) \geq \Omega(\rho^2).$$

By previous approximation results, we have that $\|\tilde{x}_1(t+1)\| \geq \Omega(\rho^2)$. □

**Lemma I.16.** *Under the condition of Theorem I.3, for any $t \geq 0$ satisfying that (1) $x(t) \in (\cap_{j \in [M]} \mathbb{I}_j) \cap K^{15h/16}$, (2) $t \in S$, it holds that $\mathrm{next}(t)$ is well defined and $\mathrm{next}(t) \leq t + 2$.*

*Proof of Lemma I.16.* Following similar argument in Lemma H.1, we have that $x(t+1) \in (\cap_{j \in [M]} \mathbb{I}_j) \cap K^h$.

If $t + 1 \notin S$, then we can apply Lemma I.15 to show that $t + 2 \in S$. □

**Lemma I.17.** *Under the condition of Theorem I.3, there exists constant $C > 0$ independent of $\eta$ and $\rho$, assuming that (1) $x(t) \in (\cap_{j \in [M]} \mathbb{I}_j) \cap K^{7h/8}$, (2) $t \in S$, (3) $\Omega(\rho^2) \leq \|\tilde{x}_1(t)\|$, then*
$$\|\tilde{x}_1(\mathrm{next}(t))\| \geq \|\tilde{x}_1(t)\| - O(\eta\rho^2).$$

*Proof of Lemma I.17.* This is by standard approximation as in previous proof and Lemma H.9. □

**Lemma I.18.** *Under the condition of Theorem I.3, there exists constant $C > 0$ independent of $\eta$ and $\rho$, assuming that (1) $x(t) \in (\cap_{j \in [M]} \mathbb{I}_j) \cap K^{7h/8}$, (2) $t \in S$ (3) $\Omega(\rho^2) \leq \|\tilde{x}_1(t)\| \leq \frac{1}{2}\left(\frac{\eta\lambda_1^2}{2 - \eta\lambda_1} + \frac{\eta\lambda_2^2}{2 - \eta\lambda_2}\right)\rho$, then*

$$\|\tilde{x}_1(\mathrm{next}(t))\| \geq \min\{(1 + C\eta)\|\tilde{x}_1(t)\|, \frac{1}{2}\left(\frac{\eta\lambda_1^2}{2 - \eta\lambda_1} + \frac{\eta\lambda_2^2}{2 - \eta\lambda_2}\right)\rho\},$$

$$or \ \|\tilde{x}_1(\mathrm{next}(\mathrm{next}(t)))\| \geq \min\{(1 + C\eta)\|\tilde{x}_1(t)\|, \frac{1}{2}\left(\frac{\eta\lambda_1^2}{2 - \eta\lambda_1} + \frac{\eta\lambda_2^2}{2 - \eta\lambda_2}\right)\rho\}.$$

*Proof of Lemma I.18.* In this proof, we will sometime drop the $t$ in $\lambda_k(t)$ or $A(t)$. Applying Lemma I.16, we have $\mathrm{next}(t)$ and $\mathrm{next}(\mathrm{next}(t))$ are well-defined. We can suppose $\|\tilde{x}_1(\mathrm{next}(t))\|_2 \leq \frac{1}{2}\left(\frac{\eta\lambda_1^2}{2 - \eta\lambda_1} + \frac{\eta\lambda_2^2}{2 - \eta\lambda_2}\right)$, else the result holds already.

By assumption, we have $\|\tilde{x}_1(t)\| \geq \Omega(\rho^2)$.

Using Lemma I.10,

$$\|\tilde{x}(t+1) - \tilde{x}(t) + \eta A\tilde{x}(t) + \eta\rho A^2 \frac{\tilde{x}(t)}{\|\tilde{x}(t)\|}\| \leq O(\eta\rho^2).$$

Denote

$$x'(t+1) = \tilde{x}(t) + \eta A\tilde{x}(t) + \eta\rho A^2 \frac{\tilde{x}(t)}{\|\tilde{x}(t)\|},$$

as the one step update of SAM on the quadratic approximation of the general loss.

Now using Lemma I.14 and the induction hypothesis, we have for some $C_1$ and $C_2$ smaller than 1, $\|\tilde{x}(t)\| \geq C_1 \frac{\eta\lambda_1^2}{2 - \eta\lambda_1}\rho \Rightarrow \|x'(t+1)\| \leq C_2 \frac{\eta\lambda_1^2}{2 - \eta\lambda_1}\rho$.

We will discuss by cases,

1. If $\|\tilde{x}(t)\| \leq C_1 \frac{\eta\lambda_1^2}{2 - \eta\lambda_1}\rho$

   If $\mathrm{next}(t) = t + 1$, then

   $$\frac{\|x'_1(t+1)\|}{\|\tilde{x}_1(t)\|} = \frac{\eta\rho\lambda_1^2 - (1 - \eta\lambda_1)\|\tilde{x}(t)\|}{\|\tilde{x}(t)\|} \geq \frac{(2 - C_1) - \eta\lambda_1 + C_1\eta\lambda_1}{C_1} \geq \frac{1}{C_1}$$

As we have $\tilde{x}_1(t) = \Omega(\rho^2)$, we have $x_1'(t+1) = \Omega(\rho^2)$, then as $\|\tilde{x}_1(t+1) - x_1'(t+1)\| = O(\eta\rho^2)$, this implies

$$\|\tilde{x}_1(t+1)\| \geq \|x_1'(t+1)\| - O(\eta\rho^2) \geq \frac{1}{C_1}\|\tilde{x}_1(t+1)\| - O(\eta\rho^2) \geq \frac{1}{2}(\frac{1}{C_1} + 1)\|\tilde{x}_1(t)\|.$$

If $\text{next}(t) = t+2$, define $x'(t+2) = x'(t+1) - \eta A x'(t+1) - \eta\rho A^2 \frac{x'(t+1)}{\|x'(t+1)\|}$, as $\|\tilde{x}_1(t+1)\| = \Omega(\eta\rho)$, by Lemma I.9, we have $\|x'(t+2) - \tilde{x}(t+2)\| = O(\eta\rho^2)$.

$$
\begin{aligned}
\frac{\|x'(t+2)\|}{\|\tilde{x}_1(t)\|} &= \frac{(\eta\rho\lambda_1^2 - (1-\eta\lambda_1)\|\tilde{x}(t)\|)(\eta\rho\lambda_1^2 - (1-\eta\lambda_1)\|x'(t+1)\|)}{\|\tilde{x}(t)\|\|x'(t+1)\|} \\
&\geq \frac{(\eta\rho\lambda_1^2 - (1-\eta\lambda_1)\|\tilde{x}(t)\|)\left(\eta\rho\lambda_1^2 - (1-\eta\lambda_1)\left(\eta\rho\lambda_1^2 - (1-\eta\lambda_1)\|\tilde{x}(t)\|\right)\right)}{\|\tilde{x}(t)\|\left(\eta\rho\lambda_1^2 - (1-\eta\lambda_1)\|\tilde{x}(t)\|\right)} \\
&= \frac{\eta\rho\lambda_1^2 - (1-\eta\lambda_1)\left(\eta\rho\lambda_1^2 - (1-\eta\lambda_1)\|\tilde{x}(t)\|\right)}{\|\tilde{x}(t)\|} \\
&\geq (1-\eta\lambda_1)^2 + \frac{\eta\lambda_1}{C_1}(2 - \eta\lambda_1) \geq 1 + 4C\eta.
\end{aligned}
$$

Combining with $|\tilde{x}_1(t)| \geq \Omega(\rho^2)$, we have that

$$\|\tilde{x}_1(\text{next}(t))\| \geq (1 + C\eta)\|\tilde{x}_1(t)\|$$

2 *Case 2* $\|\tilde{x}(t)\| > C_1 \frac{\eta\lambda_1^2}{2-\eta\lambda_1}\rho$, then $\|\tilde{x}(t+1)\| \leq C_2 \frac{\eta\lambda_1^2}{2-\eta\lambda_1}\rho$, $\text{next}(t) = t+1$

By Lemma I.17, $\|\tilde{x}_1(t+1)\| \geq (1 - C\eta)\|\tilde{x}_1(t)\|$.

As $\|\tilde{x}(\text{next}(t))\| \leq C_2 \frac{\eta\lambda_1^2}{2-\eta\lambda_1}$, similar to the first case,

$$\|\tilde{x}_1(\text{next}(\text{next}(t)))\| \geq (1 + 4C\eta)\|\tilde{x}_1(\text{next}(t))\| \geq (1 + C\eta)\|\tilde{x}_1(t)\|.$$

In conclusion, if $\|\tilde{x}_1(t)\| \leq \frac{1}{2}\left(\frac{\eta\lambda_1^2}{2-\eta\lambda_1} + \frac{\eta\lambda_2^2}{2-\eta\lambda_2}\right)\rho$, we would have there exists $C > 0$

$$\|\tilde{x}_1(\text{next}(t))\| \geq (1 + C\eta)\|\tilde{x}_1(t)\| \text{ or } \|\tilde{x}_1(\text{next}(\text{next}(t)))\| \geq (1 + C\eta)\|\tilde{x}_1(t)\|.$$

$\square$

**Lemma I.19.** *Under the condition of Theorem I.3, there exists constant $T_2 > 0$ independent of $\eta$ and $\rho$, we would have that when $t = t_{\text{ALIGN}} = \lceil T_2 \ln(1/\rho)/\eta \rceil$,*

$$|\langle x(t) - \Phi(x(t)), v_1(x(t))\rangle| = \Theta(\eta\rho),$$
$$\max_{j\in[2:M]} |\langle x(t) - \Phi(x(t)), v_j(x(t))\rangle| = O(\eta\rho^2).$$

*Further if $x(t') \in K^h$ holds for $t' = 0, 1, ..., t_{\text{LOCAL}}$, then for $t$ satisfying $t_{\text{ALIGN}} \leq t \leq t_{\text{LOCAL}}$*

$$|\langle x(t) - \Phi(x(t)), v_1(x(t))\rangle| = \Theta(\eta\rho),$$
$$\max_{j\in[2:M]} |\langle x(t) - \Phi(x(t)), v_j(x(t))\rangle| = O(\eta\rho^2).$$

*Proof of Lemma I.19.* Let $C$ be the constant defined in Lemma I.18.

By Lemma I.15, we can suppose WLOG $\|\tilde{x}_1(0)\| \geq \rho^2$ and $0 \in S$. Define

$$C_1 \triangleq \lceil \log_{1+C\eta}(\frac{\eta\lambda_1^2}{2-\eta\lambda_1}/\rho)\rceil$$

$$C_2 \triangleq C_1 + \lceil \ln_{\max\{1-\frac{\mu^2}{2\zeta^2}, 1-\frac{\Delta^2}{4\zeta^2}\}} \frac{\rho^2}{\zeta^2}\rceil = O(\log(1/\rho)/\eta).$$

We will choose $t_{\text{ALIGNMID}}$ as the minimal $t \in S$, such that $\|\tilde{x}_1(t)\| \geq \frac{1}{2}\left(\frac{\eta\lambda_1^2}{2-\eta\lambda_1} + \frac{\eta\lambda_2^2}{2-\eta\lambda_2}\right)\rho$.

Then by induction and Lemmas I.17 and I.18, we easily have that for $t \leq \min\{C_2 + 1, t_{\text{ALIGNMID}}\}$ and $t \in S$, we have that

$$x(t) \in K^{7h/8} \cap (\cap_j \mathbb{I}_j),$$

$$\|\tilde{x}_1(t)\| \geq \min\{(1 + C\eta)^{t/4}\|\tilde{x}_1(0)\|, \frac{1}{2}\left(\frac{\eta\lambda_1^2}{2 - \eta\lambda_1} + \frac{\eta\lambda_2^2}{2 - \eta\lambda_2}\right)\rho\}$$

$$\text{or} \quad \|\tilde{x}_1(\text{next}(t))\| \geq \min\{(1 + C\eta)^{t/4}\|\tilde{x}_1(0)\|, \frac{1}{2}\left(\frac{\eta\lambda_1^2}{2 - \eta\lambda_1} + \frac{\eta\lambda_2^2}{2 - \eta\lambda_2}\right)\rho\}.$$

The detailed induction is analogous to previous inductive argument and is omitted. If $t_{\text{ALIGNMID}} \geq C_1$, then we have for the minimal $t \geq C_1$ and $t \in S$

$$\|\tilde{x}_1(t)\| \geq \frac{\eta\lambda_1^2}{2 - \eta\lambda_1}\rho.$$

This is a contradiction and we have that $t_{\text{ALIGNMID}} \leq C_1$.

By Lemma I.17, $\|\tilde{x}_1(\text{next}(t))\| \geq \|\tilde{x}_1(t)\| - O(\eta\rho^2)$ for $\|\tilde{x}_1(t)\| \geq \frac{1}{2}\left(\frac{\eta\lambda_1^2}{2 - \eta\lambda_1} + \frac{\eta\lambda_2^2}{2 - \eta\lambda_2}\right)\rho$ and $t \in S$ and then by Lemma I.18,

$$\|\tilde{x}(t)\| \geq \|\tilde{x}_1(t))\| \geq \frac{1}{4}\left(\frac{\eta\lambda_1^2}{2 - \eta\lambda_1} + 3\frac{\eta\lambda_2^2}{2 - \eta\lambda_2}\right)\rho.$$

for $C_2 \geq t \geq t_{\text{ALIGNMID}}$.

We will then show that for $t \geq t_{\text{ALIGNMID}} + C_1$ iteration, $\|P^{(2:D)}\bar{x}(t+1)\| \leq O(\eta\rho^2)$.

For $C_1 \geq t \geq t_{\text{ALIGNMID}}$,

$$1 - \eta\lambda_2 - \eta\rho\frac{\lambda_2^2}{\|\tilde{x}(t)\|} \leq 1 - \eta\lambda_D - \eta\rho\frac{\lambda_D^2}{\|\tilde{x}(t)\|} \leq 1 - \frac{\lambda_D^2}{2\lambda_1^2} \leq 1 - \frac{\mu^2}{\zeta^2}.$$

Notice that,

$$1 - \eta\lambda_2 - \eta\rho\frac{\lambda_2^2}{\|\tilde{x}(t)\|} \geq 1 - \eta\lambda_2 - \eta\rho\frac{\lambda_2^2}{\|\tilde{x}(t)\|}$$

$$\geq 1 - \eta\lambda_2 - \frac{4\lambda_2^2}{\lambda_1^2 + 3\lambda_2^2}(2 - \eta\lambda_2) \geq -1 + \frac{2(\lambda_1^2 - \lambda_2^2)}{\lambda_1^2 + 3\lambda_2^2} \geq -1 + \frac{\Delta^2}{2\zeta^2}$$

Hence,

$$\|P^{(2:D)}(t)x'(t+1)\|_2 \leq \max\{1 - \frac{\mu^2}{\zeta^2}, 1 - \frac{\Delta^2}{2\zeta^2}\}\|P^{(2:D)}(t)\tilde{x}(t)\|_2$$

Now by Lemma K.1 and Theorem K.3,

$$\|P^{(2:D)}(t) - P^{(2:D)}(t+1)\| \leq O(\eta\rho^2)$$

$$\|v_1(t) - v_1(t+1)\| \leq O(\eta\rho^2)$$

$$\|\lambda_1(t) - \lambda_1(t+1)\| \leq O(\eta\rho^2)$$

By Lemma I.10, we have that $\|x'(t+1) - \tilde{x}(t+1)\| = O(\eta\rho^2)$.

Combining the above, it holds that

$$\|P^{(2:D)}(t+1)\tilde{x}(t+1)\| \leq \max\{1 - \frac{\mu^2}{2\zeta^2}, 1 - \frac{\Delta^2}{4\zeta^2}\}\|P^{(2:D)}(t)\tilde{x}(t)\| + O(\eta\rho^2)$$

Hence when $t = t_{\text{ALIGN}} = t_{\text{ALIGNMID}} + C_2$,

$$\|\tilde{x}(t)\| \geq \|\tilde{x}_1(t)\| \geq \Omega(\eta\rho),$$

$$\|P^{(2:D)}(t)\tilde{x}(t)\| \leq O(\eta\rho^2).$$

By $x(t) \in \mathbb{I}_1$, we easily have $\|\tilde{x}_1(t)\| = O(\eta\rho)$. Hence we conclude that

$$\|\tilde{x}_1(t)\| = \Theta(\eta\rho),$$

$$\|P^{(2:D)}(t)\tilde{x}(t)\| = O(\eta\rho^2).$$

The second claim is just another induction similar to previous steps and is omitted as well. $\square$

### I.2.2 TRACKING RIEMANNIAN GRADIENT FLOW

We are now ready to show that $\Phi(x(t))$ will track the solution of Equation 7. The main principal of this proof has been introduced in Section 4.3.

**Lemma I.20.** *Under the condition of Theorem I.3, for any $t$ satisfying that*

$$x(t) \in K^h,$$
$$\|\tilde{x}_1(t)\| = \Theta(\eta\rho),$$
$$\|P^{(2:D)}(t)\tilde{x}(t)\| = O(\eta\rho^2),$$

*it holds that*

$$\|\Phi(x(t+1)) - \Phi(x(t)) + \eta\rho^2 P^{\perp}_{\Phi(x(t)),\Gamma}\nabla\lambda_1(t)/2\| \le O(\eta\rho^3 + \eta^2\rho^2).$$

*Proof of Lemma I.20.* To begin with, we can approximate $\Phi(x(t+1)) - \Phi(x(t))$ by its first order Taylor Expansion, by Lemma F.7,

$$\|\Phi(x(t+1)) - \Phi(x(t)) - \partial\Phi(x(t))(x(t+1) - x(t))\| = O(\|x(t+1) - x(t)\|^2) = O(\eta^2\rho^2).$$

Then by plugging in the update rule and another Taylor Expansion,

$$\|\partial\Phi(x(t))(x(t+1) - x(t)) - \eta\rho\partial\Phi(x(t))\nabla^2 L(x)\frac{\nabla L(x)}{\|\nabla L(x)\|}$$
$$-\eta\rho^2\partial\Phi(x(t))\partial\nabla^2 L(x)[\frac{\nabla L(x)}{\|\nabla L(x)\|}, \frac{\nabla L(x)}{\|\nabla L(x)\|}]/2\|_2 = O(\eta\rho^3).$$

Using Lemma F.3, we have

$$\|\eta\rho\partial\Phi(x(t))\nabla^2 L(x)\frac{\nabla L(x)}{\|\nabla L(x)\|}\| = \eta\rho\|\nabla L(x)\|\|\partial^2\Phi(x(t))\left[\frac{\nabla L(x)}{\|\nabla L(x)\|}, \frac{\nabla L(x)}{\|\nabla L(x)\|}\right]\| = O(\eta\rho\|\nabla L(x)\|).$$

Putting together, we have that

$$\|\Phi(x(t+1)) - \Phi(x(t)) - \eta\rho^2\partial\Phi(x(t))\partial\nabla^2 L(\Phi(x(t)))[\frac{\nabla L(x)}{\|\nabla L(x)\|}, \frac{\nabla L(x)}{\|\nabla L(x)\|}]/2\|$$
$$\le O(\eta^2\rho^2 + \eta\rho^3) + O(\eta\rho\|\nabla L(x)\|).$$

As we have $\|\tilde{x}(t)\| = \Theta(\eta\rho)$, hence by Lemmas F.2 and I.7,

$$\|\Phi(x(t+1)) - \Phi(x(t)) - \eta\rho^2\partial\Phi(x(t))\partial\nabla^2 L(\Phi(x(t)))[\frac{\nabla L(x(t))}{\|\nabla L(x(t))\|}, \frac{\nabla L(x(t))}{\|\nabla L(x(t))\|}]/2\|$$
$$\le O(\eta\rho^3 + \eta^2\rho^2)$$

Finally, we have that

$$\|\eta\rho^2\partial\Phi(x(t))\partial\nabla^2 L(\Phi(x(t)))[\frac{\nabla L(x(t))}{\|\nabla L(x(t))\|}, \frac{\nabla L(x(t))}{\|\nabla L(x(t))\|}]/2$$
$$- \eta\rho^2\partial\Phi(x(t))\partial\nabla^2 L(\Phi(x(t)))[v_1(t), v_1(t)]/2\| \le O(\eta\rho^3)$$

as the angle between $\frac{\nabla L(x)}{\|\nabla L(x)\|}$ and $v_1(t)$ is $O(\rho)$.

By Lemma F.3, it holds that

$$\partial\Phi(x(t))\partial\nabla^2 L(\Phi(x(t)))[v_1(t), v_1(t)] = P^{\perp}_{X,\Gamma}\nabla(\lambda_1(t))$$

Putting together we have that,

$$\|\Phi(x(t+1)) - \Phi(x(t)) + \eta\rho^2 P^{\perp}_{X,\Gamma}\nabla\lambda_1(t)/2\| \le O(\eta\rho^3 + \eta^2\rho^2).$$

It completes the proof. $\qquad\square$

## I.3 PROOF OF THEOREM 4.5

*Proof of Theorem 4.5.* By Theorem I.1, there exists constant $T_1$ independent of $\eta, \rho$, such that for any $T_1' > T_1$ independent of $\eta, \rho$, it holds that

$$\max_{T_1 \ln(1/\eta\rho) \leq \eta t \leq T_1' \ln(1/\eta\rho)} \max_{j \in [M]} R_j(x(t)) = O(\eta\rho^2).$$

$$\max_{T_1 \ln(1/\eta\rho) \leq \eta t \leq T_1' \ln(1/\eta\rho)} \|\Phi(x(t)) - \Phi(x_{\text{init}})\| = O((\eta + \rho) \ln(1/\eta\rho)).$$

By Assumption I.2, there exists step $T_1 \ln(1/\eta\rho) \leq \eta t_{\text{PHASE}} \leq T_1' \ln(1/\eta\rho)$, such that

$$\max_{j \in [M]} R_j(x(t_{\text{PHASE}})) = O(\eta\rho^2),$$

$$\|\Phi(x(t_{\text{PHASE}})) - \Phi(x_{\text{init}})\| = O((\eta + \rho) \ln(1/\eta\rho)),$$

$$|\langle x(t_{\text{PHASE}}) - \Phi(x(t_{\text{PHASE}})), v_1(x(t_{\text{PHASE}})) \rangle| \geq \Omega(\rho^2).$$

$$\|x(t_{\text{PHASE}})\|_2 \leq \lambda_1(t_{\text{PHASE}})\eta\rho - \Omega(\rho^2).$$

Hence by Theorem I.3, if we consider a translated process with $x'(t) = x(t + t_{\text{PHASE}})$, we would have for any $T_3$ such that the solution $X$ of Equation 7 is well defined, we have that for $t = \lceil \frac{T_3}{\eta\rho^2} \rceil$

$$\|\Phi(x'(t)) - X(\eta\rho^2 t)\|_2 = O(\eta \ln(1/\rho)).$$

This implies for $t$ satisfying $X(\eta\rho^2(t - t_{\text{PHASE}}))$ is well-defined,

$$\|\Phi(x(t)) - X(\eta\rho^2(t - t_{\text{PHASE}}))\|_2 = O(\eta \ln(1/\rho)).$$

Finally, as

$$\|X(\eta\rho^2(t - t_{\text{PHASE}})) - X(\eta\rho^2 t)\|_2 \qquad = O(\eta\rho^2 t_{\text{PHASE}}) = O(\rho \ln(1/\eta\rho)) = O(\eta \ln(1/\rho)).$$

We have that

$$\|\Phi(x(t)) - X(\eta\rho^2 t)\|_2 = O(\eta \ln(1/\rho)).$$

The alignment result is a direct consequence of Theorem I.3.

$\square$

## I.4 PROOFS OF COROLLARIES 4.6 AND 4.7

*Proof of Corollary 4.6.* We will do a Taylor expansion on $L_\rho^{\text{Max}}$. By Theorem I.1 and I.3, we have $\|x(\lceil T_3/\eta\rho^2 \rceil)) - X(T_3)\| = \tilde{O}(\eta + \rho)$ and $\|x(\lceil T_3/\eta\rho^2 \rceil)) - \Phi(x(\lceil T_3/\eta\rho^2 \rceil)))\|_2 = O(\eta\rho)$. For convenience, we denote $x(\lceil T_3/\eta\rho^2 \rceil)$ by $x$.

$$R_\rho^{\text{Max}}(x) = \max_{\|v\|_2 \leq 1} \rho v^T \nabla L(x) + \rho^2 v^T \nabla^2 L(x)v/2 + O(\rho^3)$$

Since $\max_{\|v\|_2 \leq 1} \|v^T \nabla L(x)\|_2 = O(\|x - \Phi(x)\|_2) = O(\eta\rho)$, it holds that

$$R_\rho^{\text{Max}}(x) = \rho^2 \max_{\|v\|_2 \leq 1} v^T \nabla^2 L(x)v/2 + O(\eta^2\rho^2 + \rho^3)$$

$$= \rho^2 \lambda_1(\nabla^2 L(x)) + O(\eta^2\rho^2 + \rho^3)$$

$$= \rho^2 \lambda_1(\nabla^2 L(X(T_3))) + \tilde{O}(\eta\rho^2),$$

which completes the proof.

$\square$

*Proof of Corollary 4.7.* We choose $T$ such that $X(T_\epsilon)$ is sufficiently close to $X(\infty)$, such that $\lambda_1(X(T_\epsilon)) \leq \lambda_1(X(\infty)) + \epsilon/2$. By Corollary 4.6 (let $T_3 = T_\epsilon$), we have that for all $\rho, \eta$ such that $\eta \ln(1/\rho)$ and $\rho/\eta$ are sufficiently small, $\|R_\rho^{\text{Max}}(x(\lceil T_\epsilon/(\eta\rho^2) \rceil)) - \rho^2 \lambda_1(X(T_\epsilon))/2\| \leq \tilde{o}(1)$. This further implies $\|R_\rho^{\text{Max}}(x(\lceil T_\epsilon/(\eta\rho^2) \rceil)) - \rho^2 \lambda_1(X(\infty))/2\| \leq \epsilon\rho^2 + o(1)$. We also have $L(x(\lceil T_\epsilon/(\eta\rho^2) \rceil)) - \inf_{x \in U'} L(x) = o(1)$. Then we can leverage Theorem G.6 and Theorem G.3 to get the desired bound. $\square$

### I.5 DERIVATIONS FOR SECTION 4.3

We will first show our derivation of Equation 9.

In Phase II, $x(t)$ is $O(\eta\rho)$-close to the manifold $\Gamma$ and therefore it can be shown that $\|x(t) - \Phi(x(t))\|_2 = O(\eta\rho)$ holds for every step in Phase II. This also implies that $\|x(t+1) - x(t)\|_2 = O(\eta\rho)$ (See Lemma F.7). Using Taylor expansion around $x(t)$, we have that

$$\Phi(x(t+1)) - \Phi(x(t)) = \partial\Phi(x(t))(x(t+1) - x(t)) + O(\|x(t+1) - x(t)\|_2^2)$$
$$= -\eta\partial\Phi(x(t))\nabla L\big(x(t) + \rho\frac{\nabla L(x(t))}{\|\nabla L(x(t))\|_2}\big) + O(\eta^2\rho^2). \tag{31}$$

For any $x \in \mathbb{R}^D$, applying Taylor expansion on $\nabla L\big(x + \rho\frac{\nabla L(x)}{\|\nabla L(x)\|_2}\big)$ around $x$, we have that

$$\nabla L\big(x + \rho\frac{\nabla L(x)}{\|\nabla L(x)\|_2}\big)$$
$$= \nabla L(x) + \rho\nabla^2 L(x)\frac{\nabla L(x)}{\|\nabla L(x)\|_2} + \frac{\rho^2}{2}\partial^2(\nabla L)(x)\big[\frac{\nabla L(x)}{\|\nabla L(x)\|_2}, \frac{\nabla L(x)}{\|\nabla L(x)\|_2}\big] + O(\rho^3). \tag{32}$$

Using Equation 32 with $x = x(t)$, plugging in Equation 31 and then rearranging, we have that

$$\Phi(x(t+1)) - \Phi(x(t)) + \frac{\eta\rho^2}{2}\partial\Phi(x(t))\partial^2(\nabla L)(x(t))\big[\frac{\nabla L(x(t))}{\|\nabla L(x(t))\|_2}, \frac{\nabla L(x(t))}{\|\nabla L(x(t))\|_2}\big]$$
$$= -\eta\partial\Phi(x(t))\nabla L(x(t)) - \eta\rho\partial\Phi(x(t))\nabla^2 L(x(t))\frac{\nabla L(x(t))}{\|\nabla L(x(t))\|_2} + O(\eta^2\rho^2 + \eta\rho^3).$$

By Lemma 3.1, we have that $\partial\Phi(x(t))\nabla L(x(t)) = 0$. Furthermore, by Lemma F.5, we have that $\partial\Phi(\Phi(x(t)))\nabla^2 L(\Phi(x(t))) = 0$. This implies that

$$\partial\Phi(x(t))\nabla^2 L(x(t)) = \partial\Phi(\Phi(x(t)))\nabla^2 L(\Phi(x(t))) + O(\|x(t) - \Phi(x(t))\|_2) = O(\eta\rho).$$

Thus we conclude that

$$\Phi(x(t+1)) - \Phi(x(t)) = -\frac{\eta\rho^2}{2}\partial\Phi(x(t))\partial^2(\nabla L)(x(t))\big[\frac{\nabla L(x(t))}{\|\nabla L(x(t))\|_2}, \frac{\nabla L(x(t))}{\|\nabla L(x(t))\|_2}\big]$$
$$+ O(\eta^2\rho^2 + \eta\rho^3). \tag{9}$$

We will then show our derivation of Equation 10

$$\Phi(x(t+1)) - \Phi(x(t))$$
$$= -\frac{\eta\rho^2}{2}\partial\Phi(x(t))\partial^2(\nabla L)(x(t))\big[\frac{\nabla L(x(t))}{\|\nabla L(x(t))\|_2}, \frac{\nabla L(x(t))}{\|\nabla L(x(t))\|_2}\big] + O(\eta^2\rho^2 + \eta\rho^3)$$
$$= -\frac{\eta\rho^2}{2}\partial\Phi(x(t))\partial^2(\nabla L)(x(t))\big[v_1(\nabla^2 L(x(t))), v_1(\nabla^2 L(x(t)))\big] + O(\eta^2\rho^2 + \eta\rho^3)$$
$$= -\frac{\eta\rho^2}{2}\partial\Phi(x(t))\nabla\lambda_1(\nabla^2 L(x(t))) + O(\eta^2\rho^2 + \eta\rho^3)$$
$$= -\frac{\eta\rho^2}{2}\partial\Phi(\Phi(x(t)))\nabla\lambda_1(\nabla^2 L(\Phi(x(t)))) + O(\eta^2\rho^2 + \eta\rho^3), \tag{10}$$

where the second to last step we use the property of the derivative of eigenvalue (Lemma K.7) and the last step is due to Taylor expansion of $\partial\Phi(\cdot)\nabla\lambda_1(\nabla^2 L(\cdot))$ at $\Phi(x(t))$ and the fact that $\|\Phi(x(t)) - x(t)\| = O(\eta\rho)$.

We will finally show our derivation of Equation 12.

The update of the gradient (Equation 12) can be viewed as an $O(\eta\rho^2)$-perturbed version of the update of the iterate in the quadratic case. Note $O(\eta\rho^2)$ is a higher order term comparing to the other two terms, which are on the order of $\Theta(\eta^2\rho)$ and $\Theta(\eta\rho)$ respectively. By controlling the error terms, the mechanism and analysis of the implicit alignment between Hessian and gradient still apply to the general case. We can also show that once this alignment happens, it will be kept until the end of our analysis, which is $\Theta(\eta^{-1}\rho^{-2})$ steps.

Finally, we derive Equation 12 by Taylor expansion. We first apply Taylor expansion (Equation 32) on the update rule of the iterate of SAM (Equation 3):

$$x(t+1) = x(t) - \eta \nabla L(x(t)) - \eta \rho \nabla^2 L(x(t)) \frac{\nabla L(x(t))}{\|\nabla L(x(t))\|_2} + O(\eta \rho^2). \tag{33}$$

Since phase II happens in an $O(\eta\rho)$-neighborhood of manifold $\Gamma$, we have $\|x(t+1) - x(t)\|_2 = O(\eta\rho)$. Then by Equation 33 and Taylor expansion on $\nabla L(x(t+1))$ at $x(t)$, we have that

$$\nabla L(x(t+1)) = \nabla L(x(t)) - \nabla^2 L(x(t))\big(x(t+1) - x(t)\big) + O(\eta^2 \rho^2)$$

$$= \nabla L(x(t)) - \eta \nabla^2 L(x(t))\big(\nabla L(x(t)) + \rho \nabla^2 L(x(t)) \frac{\nabla L(x(t))}{\|\nabla L(x(t))\|_2}\big) + O(\eta \rho^2). \tag{34}$$

## J   ANALYSIS FOR 1-SAM (PROOF OF THEOREM 5.4)

The goal of this section is to prove the following theorem.

**Theorem 5.4.** *Let $\{x(t)\}$ be the iterates of 1-SAM (Equation 13) and $x(0) = x_{init} \in U$, then under Setting 5.1, for almost every $x_{init}$, for all $\eta$ and $\rho$ such that $(\eta + \rho) \ln(1/\eta\rho)$ is sufficiently small, with probability at least $1 - O(\rho)$ over the randomness of the algorithm, the dynamics of 1-SAM (Equation 13) can be split into two phases:*

- *Phase I (Theorem J.1): 1-SAM follows Gradient Flow with respect to L until entering an $\tilde{O}(\eta\rho)$ neighborhood of the manifold $\Gamma$ in $O(\ln(1/\rho\eta)/\eta)$ steps;*
- *Phase II (Theorem J.2): 1-SAM tracks the solution of Equation 14, $X$, the Riemannian gradient flow with respect to $\mathrm{Tr}(\nabla^2 L(\cdot))$ in an $\tilde{O}(\eta\rho)$ neighborhood of manifold $\Gamma$. Quantitatively, the approximation error between the iterates $x$ and the corresponding limiting flow $X$ is $\tilde{O}(\eta^{1/2} + \rho)$, that is,*

$$\|x(\lceil T_3/(\eta\rho^2)\rceil) - X(T_3)\|_2 = \tilde{O}(\eta^{1/2} + \rho).$$

As mentioned in our proof setups in Appendix E, we will prove Theorem 5.4 under a more general (and weaker) condition, namely Condition E.1 and Assumption 3.2. The only usage of Setting 5.1 in the proof is Theorems 5.2 and E.2, which are restated below.

**Theorem 5.2.** *Loss $L$, set $\Gamma$ and integer $M$ defined in Setting 5.1 satisfy Assumption 3.2.*

**Theorem E.2.** *Setting 5.1 implies Condition E.1.*

**Condition E.1.** *Total loss $L = \frac{1}{M}\sum_{k=1}^{M} L_k$. For each $k \in [M]$, $L_k$ is $\mathcal{C}^4$, and there exists a $(D-1)$-dimensional $\mathcal{C}^2$-submanifold of $\mathbb{R}^D$, $\Gamma_k$, where for all $x \in \Gamma_k$, $x$ is a global minimizer of $L_k$, $L_k(x) = 0$ and $rank(\nabla^2 L_k(x)) = 1$. Moreover, $\Gamma = \cap_{k=1}^{M} \Gamma_k$ for $\Gamma$ defined in Assumption 3.2.*

Analogous to the full-batch setting, we will split the trajectory into two phases.

**Theorem J.1** (Phase I). *Let $\{x(t)\}$ be the iterates defined by SAM (Equation 13) and $x(0) = x_{init} \in U$, then under Assumption 3.2 and E.1, for almost every $x_{init}$, there exists a constant $T_1$, it holds for sufficiently small $(\eta + \rho) \ln 1/\eta\rho$, we have with probability $1 - O(\rho)$, there exists $t \leq T_1 \ln(1/\eta\rho)/\eta$, such that $\|x(t) - \Phi(x(t))\|_2 = O(\eta\rho)$ and $\|\Phi(x_{init}) - \Phi(x(t))\|_2 = \tilde{O}(\eta^{1/2} + \rho)$.*

Theorem J.1 shows that SAM will converges to an $\tilde{O}(\eta\rho)$ neighborhood of the manifold without getting far away from $\Phi(x(0))$, where we can perform a local analysis on the trajectory of $\Phi(x(t))$.

Under Assumptions 3.2 and E.1, we have $\mathrm{Tr}(\nabla^2 L_k(x)) = \lambda_1(\nabla^2 L_k(x))$ is differentiable for $x \in \Gamma_i$. Hence $\mathrm{Tr}(\nabla^2 L(x)) = \sum_{k=1}^{M} \mathrm{Tr}(\nabla^2 L_k(x))$ is also differentiable and we have (14) is well defined for some finite time $T_2$.

**Theorem J.2** (Phase II). *Let $\{x(t)\}$ be the iterates defined by SAM (Equation 13) under Assumptions 3.2 and E.1, assuming (1) $\|x(0) - \Phi(x(0))\|_2 = O(\eta\rho)$ and (2) $\|\Phi(x_{init}) - \Phi(x(0))\|_2 = \tilde{O}(\eta^{1/2} + \rho)$, then for almost every $x(0)$, for any $T_2 > 0$ till which solution of (14) $X$ exists, for sufficiently small $(\eta + \rho) \ln 1/(\eta\rho)$, we have with probability $1 - O(\eta\rho)$, for all $\eta\rho^2 t < T_2$, $\|\Phi(x(t)) - X(\eta\rho^2 t)\|_2 = \tilde{O}(\eta^{1/2} + \rho)$ and $\|x(t) - \Phi(x(t))\|_2 = O(\eta\rho)$.*

Combining Theorems E.2, J.1 and J.2, the proof of Theorem 5.4 is clear and we deferred it to Appendix J.3.

Now we recall our notations for stochastic setting with batch size one. **Notations for Stochastic Setting:** Since $L_k$ is rank-1 on $\Gamma_k$ for each $k \in [M]$, we can write it as $L_k(x) = \Lambda_k(x)w_k(x)w_k^\top(x)$ for any $x \in \Gamma_k$, where $w_k$ is a continuous function on $\Gamma_k$ with pointwise unit norm. Given the loss function $L_k$, its gradient flow is denoted by mapping $\phi_k : \mathbb{R}^D \times [0, \infty) \to \mathbb{R}^D$. Here, $\phi_k(x, \tau)$ denotes the iterate at time $\tau$ of a gradient flow starting at $x$ and is defined as the unique solution of $\phi_k(x, \tau) = x - \int_0^\tau \nabla L_k(\phi_k(x, t))dt$, $\forall x \in \mathbb{R}^D$. We further define the limiting map $\Phi_k$ as $\Phi_k(x) = \lim_{\tau \to \infty} \phi_k(x, \tau)$, that is, $\Phi_k(x)$ denotes the convergent point of the gradient flow starting from $x$. Similar to Definition 3.3, we define $U_k = \{x \in \mathbb{R}^D | \Phi(x) \text{ exists and } \Phi_k(x) \in \Gamma_k\}$ be the attraction set of $\Gamma_i$. We have that each $U_k$ is open and $\Phi_k$ is $\overline{C}^2$ on $U_k$ by Lemma B.15 in Arora et al. (2022).

In this section we will define $K$ as $\{X(t) \mid t \in [0, T_3]\}$ where $X$ is the solution of (14). We will denote $h(K)$ in Lemma E.6 by $h$. Using Theorem D.3, we will assume the update is always well defined.

## J.1 PHASE I (PROOF OF THEOREM J.1)

*Proof of Theorem J.1.* The proof consists of two steps.

1. *Tracking Gradient Flow.* By Lemma J.3, with probability $1 - \rho^2$, there exists step $t_{\text{GF}} = O(1/\eta)$ such that

$$\|x(t_{\text{GF}}) - \Phi(x(t_{\text{GF}}))\|_2 \le h/4.$$
$$\|\Phi(x(t_{\text{GF}})) - \Phi(x_{\text{init}})\|_2 = \tilde{O}(\eta^{1/2} + \rho).$$

2. *Decreasing Loss.* By Lemma J.7, with probability $1 - O(\rho)$, there exists step $t_{\text{DEC}} = t_{\text{GF}} + O(\ln(1/\rho)/\eta) = O(\ln(1/\rho)/\eta)$ such that

$$\|\nabla L(x(t_{\text{DEC}}))\|_2 = O(\rho).$$
$$\|\Phi(x(t_{\text{DEC}})) - \Phi(x_{\text{init}})\|_2 \le \|\Phi(x(t_{\text{DEC}})) - \Phi(x(t_{\text{GF}}))\|_2 + \|\Phi(x(t_{\text{GF}})) - \Phi(x_{\text{init}})\|_2$$
$$= \tilde{O}(\eta^{1/2} + \rho).$$

Then by Lemma J.12, with probability $1 - O(\rho)$, there exists step $t_{\text{DEC2}} = t_{\text{DEC}} + O(\ln(1/\eta\rho)/\eta) = O(\ln(1/\eta\rho)/\eta)$, it holds that

$$\|x(t_{\text{DEC2}}) - \Phi(x(t_{\text{DEC2}}))\|_2 = O(\eta\rho).$$
$$\|\Phi(x(t_{\text{DEC2}})) - \Phi(x_{\text{init}})\|_2 \le \|\Phi(x(t_{\text{DEC2}})) - \Phi(x(t_{\text{DEC}}))\|_2 + \|\Phi(x(t_{\text{DEC}})) - \Phi(x_{\text{init}})\|_2$$
$$= \tilde{O}(\eta^{1/2} + \rho).$$

Concluding, let $T_1$ be the constant satisfying $t_{\text{DEC2}} \le T_1 \ln(1/\eta\rho)/\eta$, then we have for $t = t_{\text{DEC2}} \le T_1 \ln(1/\eta\rho)/\eta$ such that

$$\|x(t) - \Phi(x(t))\|_2 = O(\eta\rho).$$
$$\|\Phi(x(t)) - \Phi(x_{\text{init}})\|_2 = \tilde{O}(\eta^{1/2} + \rho).$$

$\square$

### J.1.1 TRACKING GRADIENT FLOW

Lemma J.3 shows that the iterates $x(t)$ tracks gradient flow to an $O(1)$ neighbor of $\Gamma$.

**Lemma J.3.** *Under condition of Theorem J.1, with probability $1 - O(\rho^2)$, there exists $t_{\text{GF}} = O(1/\eta)$, such that the iterate $x(t_{\text{GF}})$ is $O(1)$ close to the manifold $\Gamma$ and $\Phi(x(t_{\text{GF}}))$ is $\tilde{O}(\eta^{1/2} + \rho)$ is close to $\Phi(x_{init})$. Quantitatively,*

$$L(x(t_{\text{GF}})) \le \frac{\mu h^2}{32}$$
$$\|x(t_{\text{GF}}) - \Phi(x(t_{\text{GF}}))\| \le h/4,$$
$$\|\Phi(x(t_{\text{GF}})) - \Phi(x_{init})\| = \tilde{O}(\eta^{1/2} + \rho).$$

*Proof of Lemma J.3.* Choose $C = \frac{1}{4}\sqrt{\frac{\mu}{\zeta}}$.

There exists $T > 0$, such that

$$\|\phi(x_{\mathrm{init}}, T) - \Phi(x_{\mathrm{init}})\|_2 \le Ch/2\,.$$

Consider

$$x(t+1) = x(t) - \eta \nabla L_k(x(t) + \rho \frac{\nabla L_k\,(x(t))}{\|\nabla L_k\,(x(t))\,\|})$$
$$= x(t) - \eta \nabla L_k(x(t)) + O(\eta\rho)\,.$$

By Theorem L.1, let $b(x) = -\nabla L(x)$, $p = \eta$ and $\epsilon = O(\rho)$, for sufficiently small $\eta$ and $\rho$, the iterates $x(t)$ tracks gradient flow $\phi(x_{\mathrm{init}}, T)$ in $O(1/\eta)$ steps in expectation, Quantitatively, with probability $1 - \rho^2$, for $t_{\mathrm{GF}} = \lceil \frac{T_0}{\eta} \rceil$, we have that

$$\|x(t_{\mathrm{GF}}) - \phi(x_{\mathrm{init}}, T_0)\|_2 = \tilde{O}(\sqrt{p} + \epsilon) \le \tilde{O}(\eta^{1/2} + \rho)\,.$$

This implies $x(t_{\mathrm{GF}}) \in K^h$, hence by Taylor Expansion on $\Phi$,

$$\|\Phi(x(t_{\mathrm{GF}})) - \Phi(x_{\mathrm{init}})\|_2 = \|\Phi(x(t_{\mathrm{GF}})) - \Phi(\phi(x_{\mathrm{init}}, T))\|_2$$
$$\le O(\|x(t_{\mathrm{GF}}) - \phi(x_{\mathrm{init}}, T)\|_2)$$
$$\le \tilde{O}(\eta^{1/2} + \rho)\,.$$

This implies

$$\|x(t_{\mathrm{GF}}) - \Phi(x(t_{\mathrm{GF}}))\|_2 \le \|x(t_{\mathrm{GF}}) - \phi(x_{\mathrm{init}}, T_0)\|_2 + \|\phi(x_{\mathrm{init}}, T_0) - \Phi(x_{\mathrm{init}})\|_2$$
$$+ \|\Phi(x_{\mathrm{init}}) - \Phi(x(t_{\mathrm{GF}}))\|_2$$
$$\le Ch/2 + \tilde{O}(\eta^{1/2} + \rho) \le Ch \le h/4\,.$$

By Taylor Expansion,

$$L(x(t_{\mathrm{GF}})) \le \zeta \|x(t_{\mathrm{GF}}) - \Phi(x(t_{\mathrm{GF}}))\|_2^2/2 \le \frac{\mu h^2}{32}\,.$$

$\square$

### J.1.2 DECREASING LOSS

**Lemma J.4.** *Under condition of Theorem J.1, assuming $x(t_0) \in K^{h/4}$ and for any $t$ satisfying $t_0 \le t \le t_0 + O(\ln(1/\eta\rho)/\eta)$,* $\max_{t_0 \le \tau \le t_0 + O(\ln(1/\eta\rho)/\eta)} L(x(\tau)) \le \frac{\mu h^2}{16}$, *it holds that*

$$x(\tau) \in K^h, \forall t_0 \le \tau \le t\,.$$

*Moreover, we have that*

$$\|\Phi(x(t)) - \Phi(x(t_0))\| = O((\eta + \rho)\ln(1/\eta\rho))\,.$$

*Proof of Lemma J.4.* We will prove by induction. For $\tau = t_0$, the result holds trivially. Suppose the result holds for $t - 1$, then for any $\tau$ satisfying $t_0 \le \tau \le t - 1$, by Lemmas F.1 and F.8,

$$\|\Phi(x(\tau+1)) - \Phi(x(\tau))\| \le \xi \eta \rho \|\nabla L\,(x(\tau))\,\|_2 + \nu \eta \rho^2 + \xi \eta^2 \|\nabla L\,(x(\tau))\,\|_2^2 + \xi \zeta^2 \eta^2 \rho^2$$
$$= O(\eta^2 + \eta\rho)\,.$$

Also by Lemma F.1, $\|x(t) - \Phi(x(t))\|_2 \le h/2\sqrt{2}$, this implies,

$$\mathrm{dist}(K, x(t)) \le \mathrm{dist}(K, x(t_0)) + \|x(t_0) - \Phi(x(t_0))\|_2$$
$$+ \|\Phi(x(t_0)) - \Phi(x(t))\| + \|\Phi(x(t)) - x(t)\|$$
$$\le 0.99h + O(\eta^2(t - t_{\mathrm{GF}})) = 0.99h + O(\eta \ln(1/\eta\rho)) \le h\,.$$

$\square$

**Lemma J.5.** *Under condition of Theorem J.1, if $x(\tau) \in K^h$, then we have that*

$$\mathbb{E}[L(x(\tau+1))|x(\tau)] \leq L(x(\tau)) - \frac{\eta\mu}{2}L(x(\tau)) \,.$$

*Moreover it holds that,*

$$\mathbb{E}[\ln L(x(\tau+1))|x(\tau)] \leq \ln \mathbb{E}[L(x(\tau+1))|x(\tau)] \leq \ln L(x(\tau)) - \frac{\eta\mu}{2} \,.$$

*Proof of Lemma J.5.* By Lemma F.8 and Taylor Expansion,

$$\mathbb{E}[L(x(\tau+1))|x(\tau)]$$

$$=\mathbb{E}\left[L\left(x(\tau) - \eta\nabla L_k[x(\tau) + \rho\frac{\nabla L_k(x(\tau))}{\|\nabla L_k(x(\tau))\|}]\right)|x(\tau)\right]$$

$$\leq\mathbb{E}\left[L(x(\tau)) - \eta\left\langle\nabla L(x(\tau)), \nabla L_k\left(x(\tau) + \rho\frac{\nabla L_k(x(\tau))}{\|\nabla L_k(x(\tau))\|}\right)\right\rangle\right]$$

$$+\mathbb{E}\left[\frac{\zeta\eta^2}{2}\|\nabla L_k[x(\tau) + \rho\frac{\nabla L_k(x(\tau))}{\|\nabla L_k(x(\tau))\|}]\|_2^2\right]$$

$$\leq L(x(\tau)) - \eta\|\nabla L(x(\tau))\|_2^2 + \eta\rho\zeta\|\nabla L(x(\tau))\|_2 + \zeta\eta^2\mathbb{E}[\|\nabla L_k(x(\tau))\|_2^2] + \zeta^3\eta^2\rho^2$$

$$\leq L(x(\tau)) - \frac{\eta}{2}\|\nabla L(x(\tau))\|_2^2$$

$$\leq L(x(\tau)) - \frac{\eta\mu}{2}L(x(\tau)) \,.$$

$\square$

**Lemma J.6.** *Under condition of Theorem J.1, assuming $x(t_0) \in K^{h/4}$ and $L(x(t_0)) \leq \frac{\mu h^2}{32}$, then with probability $1 - O(\rho)$, for any $t$ satisfying $t_0 \leq t \leq t_0 + O(\ln(1/\eta\rho)/\eta)$, it holds that $x(t) \in K^h$. Moreover, we have that*

$$\|\Phi(x(t)) - \Phi(x(t_0))\| = O((\eta + \rho)\ln(1/\eta\rho)) \,.$$

*Proof of Lemma J.6.* By Uniform Bound and Lemma J.4,

$$\mathbb{P}(\exists t_0 \leq t \leq t_0 + O(\ln(1/\eta\rho)/\eta), L(x(t)) \geq \frac{\mu h^2}{16})$$

$$\leq \sum_{t=t_0}^{t_0+O(\ln(1/\eta\rho)/\eta)} \mathbb{P}(L(x(t)) \geq \frac{\mu h^2}{16} \quad \text{and} \quad L(x(\tau)) \leq \frac{\mu h^2}{16}, \forall t_0 \leq \tau \leq t-1)$$

$$\leq \sum_{t=t_0}^{t_0+O(\ln(1/\eta\rho)/\eta)} \mathbb{P}(L(x(t)) \geq \frac{\mu h^2}{16} \quad \text{and} \quad x(\tau) \in K^h, \forall t_0 \leq \tau \leq t-1)$$

Consider each term, and applying uniform bound again,

$$P(L(x(t)) \geq \frac{\mu h^2}{16} \quad \text{and} \quad x(\tau) \in K^h, \forall t_0 \leq \tau \leq t-1)$$

$$\leq \sum_{\tau=t_0}^{t} P(L(x(t)) \geq \frac{\mu h^2}{16} \quad \text{and} \quad L(x(\tau)) \leq \frac{\mu h^2}{32}$$

$$\text{and} \quad \forall t-1 \geq \tau' \geq \tau+1, \frac{\mu h^2}{16} > L(x(\tau')) > \frac{\mu h^2}{32}$$

$$\text{and} \quad \forall t-1 \geq \tau'' \geq \tau, x(\tau'') \in K^h) \,.$$

Then if we consider each term, we have that it is bounded by

$$P(L(x(t)) \geq \frac{\mu h^2}{16} \quad \text{and} \quad \forall t-1 \geq \tau' \geq \tau+1, L(x(\tau')) > \frac{\mu h^2}{32}$$

$$\text{and} \quad \forall t-1 \geq \tau'' \geq \tau, x(\tau'') \in K^h \mid L(x(\tau)) \leq \frac{\mu h^2}{32}) \,.$$

Define a coupled process $\tilde{L}(\tau+1) = \ln L(x(\tau+1))$ and

$$\tilde{L}(\tau') = \begin{cases} \ln L(x(\tau')), & \text{if } \tilde{L}(\tau'-1) = \ln L(x(\tau'-1)) \geq \ln(\frac{\mu h^2}{32}), \\ \tilde{L}(\tau'-1) - \eta\mu/2, & \text{if otherwise.} \end{cases}$$

Then clearly

$$P(L(x(t)) \geq \frac{\mu h^2}{16} \quad \text{and} \quad \forall t \geq \tau' \geq \tau+1, L(x(\tau')) > \frac{\mu h^2}{32}$$

$$\text{and} \quad \forall t \geq \tau'' \geq \tau, x(\tau'') \in K^h \mid L(x(\tau)) \leq \frac{\mu h^2}{32})$$

$$\leq P(\tilde{L}(t) \geq \ln(\frac{\mu h^2}{16})).$$

Consider a fixed $\tau'$ satisfying $\tau+1 \leq \tau' \leq t$. By Lemma J.5, we have that

$$\tilde{L}(x(\tau'+1)) - \tilde{L}(x(\tau')) \leq -\eta\mu/2.$$

Hence $\tilde{L}(t) + \eta\mu t/2$ is a super martingale.

Further it holds that if $L(x(\tau'-1)) \geq (\frac{\mu h^2}{32})$, then

$$L(x(\tau'-1)) - L(x(\tau')) = O(\|x(\tau'-1) - x(\tau')\|) = O(\eta).$$

Using the smoothness at $\log(x)$ at $\frac{\mu h^2}{32}$ which is a positive constant,

$$\|\tilde{L}(\tau'+1) - \tilde{L}(\tau')\| \leq O(\eta) \leq C\eta.$$

Here $C$ is a constant independent of $\eta$. This implies $\tilde{L}(x(\tau+1)) \leq \frac{\mu h^2}{16\sqrt{2}}$

Now by Azuma-Hoeffding bound (Lemma K.4), we have that

$$P(\tilde{L}(t) - \tilde{L}(\tau+1) + (t-\tau-1)\eta\mu/2 > a) \leq 2\exp(-\frac{a^2}{8(t-\tau-1)(C+\mu)^2\eta^2}).$$

With $a = \ln(\frac{\mu h^2}{16\tilde{L}(\tau+1)}) + (t-\tau-1)\eta\mu/2 \geq (\ln 2 + (t-\tau-1)\eta\mu)/2$, we have that

$$P(\tilde{L}(t) > \ln(\frac{\mu h^2}{16})) \leq 2\exp(-\frac{(\ln 2 + (t-\tau-1)\eta\mu)^2}{32(C+\mu)^2\eta^2})$$

$$\leq 2\exp(-\frac{\ln 2(t-\tau-1)\mu}{8(C+\mu)^2\eta})$$

Hence we have

$$\mathbb{P}(\exists t_0 \leq t \leq t_0 + O(\ln(1/\eta\rho)/\eta), L(x(t)) \geq \frac{\mu h^2}{16})$$

$$\leq O(2\exp(-\frac{\ln 2(t-\tau-1)\mu}{8(C+\mu)^2\eta})\ln^2(1/\eta\rho)/\eta^2) \leq \rho.$$

Hence with probability $1-\rho$, $L(x(t)) \leq \frac{\mu h^2}{16}, \forall t_0 \leq t \leq t_0 + O(\ln(1/\eta\rho)/\eta)$, combining with Lemma J.4, we have completed our proof. $\square$

**Lemma J.7.** *Under condition of Theorem J.1, assuming there exists $t_{\text{GF}}$ such that $L(x(t_{\text{GF}})) \leq \frac{\mu h^2}{32}$ and $x(t_{\text{GF}}) \in K^{h/4}$, then with probability $1-O(\rho)$, there exists $t_{\text{DEC}} = t_{\text{GF}} + O(\ln(1/\rho)/\eta)$, such that $x(t_{\text{DEC}})$ is in $O(\rho)$ neighbor of $\Gamma$, quantitatively, we have that*

$$\|\nabla L(x(t_{\text{DEC}}))\|_2 \leq 4\zeta\rho.$$

*Moreover the movement of the projection of $\Phi(x(\cdot))$ on the manifold is bounded,*

$$\|\Phi(x(t_{\text{GF}})) - \Phi(x(t_{\text{DEC}}))\|_2 = O((\eta+\rho)\ln(1/\rho)).$$

*Proof of Lemma J.7.* For simplicity of writing, define $T_1 \triangleq \lceil \frac{2\ln\frac{h^2}{256\rho^3\mu}}{\eta\mu} \rceil = O(\ln(1/\rho)/\eta)$.

By Lemma J.6, we may assume $x(t) \in K^h$ for $t_{\mathrm{GF}} \le t \le T_1 + t_{\mathrm{GF}}$.

Define indicator function as

$$\mathcal{A}(t) = \mathbf{1}[\nabla L\left(x(\tau)\right) \ge 4\zeta\rho, \forall t \ge \tau \ge t_{\mathrm{GF}}].$$

By Lemma J.5, we have that,

$$\mathbb{E}[L(x(t+1))\mathcal{A}(t+1)] \le \mathbb{E}[L(x(t+1))\mathcal{A}(t)] \le (1 - \frac{\eta\mu}{2})\mathbb{E}[L(x(t))\mathcal{A}(t)].$$

We can then conclude that with $T_2 = T_1 + t_{\mathrm{GF}}$, using Lemma F.2,

$$8\mu\rho^2\mathbb{E}\mathcal{A}(T_2+1) \le \mathbb{E}[L(x(T_2+1))\mathcal{A}(T_2+1)] \le (1 - \frac{\eta\mu}{2})^{T_1} L(x(t_{\mathrm{GF}})) \le 8\mu\rho^3.$$

We have

$$\mathbb{E}\mathcal{A}(T_2+1) \le \rho.$$

This implies $\mathcal{A}(T_2+1) = 0$ with probability $1 - O(\rho)$, which indicates the existence of $t_{\mathrm{DEC}}$. The second claim is a direct application of Lemma J.6. $\qquad\square$

**Lemma J.8** (A general version of Lemma 5.5). *Under Assumption 3.2 and Condition E.1, for $x \in K^h$ and $p \in C, \nabla^2 L_k(p) = \Lambda_k(p)w_k(p)w_k(p)^\top$, there exists $s \in \{1, -1\}$,*

$$\frac{\nabla L_k(x)}{\|\nabla L_k(x)\|} = sw_k(p) + O(\|x - p\|_2).$$

*Further if $|w_k^\top(x - p)| \ge \|x - p\|_2^{3/2}$, then $s = \mathrm{sign}(w_k^\top(x - p))$. This implies*

$$\frac{\nabla L_k(x)}{\|\nabla L_k(x)\|}^\top (x - p) \ge sw_k^\top(x - p) - O(\|x - p\|_2^2)$$

$$\ge \|w_k^\top(x - p)\|_2 - O(\|x - p\|_2^{3/2}).$$

*Proof of Lemma J.8.* We will calculate the direction of $\frac{\nabla L_k(x)}{\|\nabla L_k(x)\|}$ using two different approximations and compare them to get our result.

1. According to Lemma F.4,

$$\frac{\nabla L_k(x)}{\|\nabla L_k(x)\|} = \frac{\nabla^2 L_k(\Phi_k(x))(x - \Phi_k(x))}{\|\nabla^2 L_k(\Phi_k(x))(x - \Phi_k(x))\|_2} + O(\|x - \Phi_k(x)\|_2).$$

Suppose $\nabla^2 L_k(\Phi_k(x)) = \Lambda_k(\Phi_k(x))w_k(\Phi_k(x))w_k(\Phi_k(x))^\top$, then

$$\frac{\nabla L_k(x)}{\|\nabla L_k(x)\|} = w_k(\Phi_k(x)) + O(\|x - \Phi_k(x)\|_2)$$

As $\nabla^2 L_k(p) = \Lambda_k(p)w_k(p)w_k(p)^\top$, using Davis-Kahan Theorem K.3, we would have $\exists s \in \{-1, 1\}$, such that $\|w_k(\Phi_k(x)) - sw_k(p)\|_2 \le \zeta\|\Phi_k(x) - p\|_2$.

$$\frac{\nabla L_k(x)}{\|\nabla L_k(x)\|} = sw_k(p) + O(\|\Phi_k(x) - p\|_2 + \|x - p\|_2).$$

According to Lemma F.1, we have $\|x - \Phi_k(x)\|_2 \le \frac{\|\nabla L_k(x)\|_2}{\mu} \le \frac{\zeta\|x-p\|_2}{\mu}$. This implies,

$$\frac{\nabla L_k(x)}{\|\nabla L_k(x)\|} = sw_k(p) + O(\|x - p\|_2). \tag{35}$$

Equation 35 is our first statement.

2. By Taylor expansion at $p$,
$$\nabla L_k(x) = \Lambda_k(x)w_k(p)w_k(p)^\top(x-p) + O(\nu\|x-p\|_2^2).$$
That being said, when $|w_k^\top(x-p)| \geq \|x-p\|_2^{3/2}$, we have
$$\|\nabla L_k(x) - \Lambda_k w_k w_k^\top(x-p)\|_2 \leq O(\|x-p\|_2^2).$$
$$\|\nabla L_k(x)\| \geq \|\Lambda_k w_k w_k^\top(x-p)\|_2 - O(\|x-p\|_2^2) \geq \Omega(\|x-p\|_2^{3/2}).$$
Concluding,
$$\|\frac{\nabla L_k(x)}{\|\nabla L_k(x)\|} - \frac{\Lambda_k w_k w_k^\top(x-p)}{\|\Lambda_k w_k w_k^\top(x-p)\|}\|_2 \leq O(\|x-p\|_2^{1/2})$$
Hence we have
$$\frac{\nabla L_k(x)}{\|\nabla L_k(x)\|} = \text{sign}(w_k^\top(x-p))w_k + O(\|x-p\|_2^{1/2}). \tag{36}$$
Comparing (35) and (36), we have $s = \text{sign}(w_k(p)^\top(x-p))$ when $|w_k^\top(x-p)| \geq \|x-p\|_2^{3/2}$.

$\square$

**Lemma J.9.** *Under condition of Theorem J.1, for any constant $C > 0$ independent of $\eta, \rho$, there exists constant $C_1 > C_2 > 0$ independent of $\eta, \rho$, if $x(t) \in K^h$ and $C_1\eta\rho \leq \|x(t) - \Phi(x(t))\| \leq C\rho$, then we have that*
$$\mathbb{E}_k[\|x(t+1) - \Phi(x(t+1))\|_2 \mid x(t)] \leq \|x(t) - \Phi(x(t))\|_2 - C_2\eta\rho.$$

*Proof of Lemma J.9.* By Lemma F.2, $\|x(t) - \Phi(x(t))\| = O(\rho)$. Hence we have that by Taylor Expansion,
$$x(t+1) = x(t) - \eta\nabla L_k\left(x(t) + \rho\frac{\nabla L_k(x(t))}{\|\nabla L_k(x(t))\|}\right)$$
$$= x(t) - \eta\nabla L_k(x(t)) - \eta\rho\nabla^2 L_k(x(t))\frac{\nabla L_k(x(t))}{\|\nabla L_k(x(t))\|} + O(\eta\rho^2)$$
$$= x(t) - \eta\nabla L_k(x(t)) - \eta\rho\Lambda_k w_k w_k^\top\frac{\nabla L_k(x(t))}{\|\nabla L_k(x(t))\|} + O(\eta\rho^2).$$
Here $\Lambda_k, w_k$ indicates $\Lambda_k(\Phi(x(t))), w_k(\Phi(x(t)))$.

Notice that given $\|x(t) - \Phi(x(t))\| = O(\rho)$, by Lemma F.8, we have that
$$\|\Phi(x(t+1)) - \Phi(x(t))\|_2 = O(\eta\rho^2),$$
$$\|x(t+1) - x(t)\|_2 = O(\eta\rho).$$
This implies $x(t+1) \in K^r$.

Further by Taylor Expansion, $\nabla L_k(x(t)) = \Lambda_k w_k w_k^\top(x(t) - \Phi(x(t))) + O(\rho^2)$.

By Lemma J.8, we have for some $s_k(t) \in \{-1, 1\}$.
$$w_k^\top\frac{\nabla L_k(x(t))}{\|\nabla L_k(x(t))\|} = s_k(t)w_k + O(\|x(t) - \Phi(x(t))\|_2).$$
We also have
$$s_k(t) \neq \text{sign}(w_k^\top(x(t) - \Phi(x(t)))) \Rightarrow \|w_k^\top(x(t) - \Phi(x(t)))\|_2 \leq \|x(t) - \Phi(x(t))\|_2^{3/2}. \tag{37}$$
Concluding,
$$x(t+1) - \Phi(x(t+1))$$
$$= (x(t) - \Phi(x(t))) - \eta\Lambda_k w_k w_k^\top(x(t) - \Phi(x(t))) - \eta\rho\Lambda_k s_k(t)w_k w_k^\top w_k + O(\eta\rho^2).$$
After we take square and expectation,
$$\mathbb{E}[\|x(t+1) - \Phi(x(t+1))\|_2^2 \mid x(t)]$$
$$\leq \|x(t) - \Phi(x(t))\|_2^2 + \frac{2\eta^2}{M}\sum_{k=1}^M \Lambda_k^2 |w_k^\top(x(t) - \Phi(x(t)))|^2 + \frac{2\eta^2\rho^2}{M}\sum_{k=1}^M \Lambda_k^2$$
$$- 2\frac{\eta}{M}\sum_{k=1}^M \Lambda_k |w_k^\top(x(t) - \Phi(x(t)))|^2 - 2\frac{\eta\rho}{M}\sum_{k=1}^M \Lambda_k s_k(t)w_k^\top(x(t) - \Phi(x(t)))$$
$$+ O(\eta\rho^2\|x(t) - \Phi(x(t))\| + \eta^2\rho^3).$$

We will then carefully examine each positive term,

$$\frac{2\eta^2}{M}\sum_{k=1}^{M}\Lambda_k^2|w_k^\top(x(t)-\Phi(x(t)))|^2 = 2M\eta^2(x(t)-\Phi(x(t)))^\top\nabla^2 L(x(t))^2(x(t)-\Phi(x(t)))$$

$$\leq 2M\zeta\eta^2\|x(t)-\Phi(x(t))\|^2 = O(\eta^2\rho^2)\,.$$

$$\frac{2\eta^2\rho^2}{M}\sum_{k=1}^{M}\Lambda_k^2 \leq 2\zeta^2\eta^2\rho^2 = O(\eta^2\rho^2)\,.$$

This implies,

$$\mathbb{E}[\|x(t+1)-\Phi(x(t+1))\|_2^2 \mid x(t)]$$

$$\leq\|x(t)-\Phi(x(t))\|_2^2 - 2\frac{\eta\rho}{M}\sum_{k=1}^{M}\Lambda_k s_k(t)w_k^\top(x(t)-\Phi(x(t)))$$

$$+ O(\eta\rho^2\|x(t)-\Phi(x(t))\| + \eta^2\rho^2)\,.$$

We will now lower bound $\sum_{k=1}^{M}\Lambda_k s_k(t)w_k^\top(x(t)-\Phi(x(t)))$. By Equation 37,

$$\sum_{k=1}^{M}\Lambda_k s_k(t)w_k^\top(x(t)-\Phi(x(t))) \geq \sum_{k=1}^{M}\Lambda_k\|w_k^\top(x(t)-\Phi(x(t)))\|_2 - 2\sum_{k=1}^{M}\Lambda_k\|x(t)-\Phi(x(t))\|_2^{3/2}$$

$$\geq \sum_{k=1}^{M}\Lambda_k\|w_k^\top(x(t)-\Phi(x(t)))\|_2 - O(\|x(t)-\Phi(x(t))\|_2^{3/2})\,.$$

For $\sum_{k=1}^{M}\Lambda_k\|w_k^\top(x(t)-\Phi(x(t)))\|_2$, by Lemma Lemma F.4,

$$\sum_{k=1}^{M}\Lambda_k\|w_k^\top(x(t)-\Phi(x(t)))\|_2 \geq \sqrt{\sum_{k=1}^{M}\Lambda_k^2\|w_k^\top(x(t)-\Phi(x(t)))\|_2^2}$$

$$= \sqrt{(x(t)-\Phi(x(t)))^\top\nabla^2 L(\Phi(x(t)))^2(x(t)-\Phi(x(t)))}$$

$$= \|\nabla^2 L(\Phi(x(t)))(x(t)-\Phi(x(t)))\|_2$$

$$\geq \mu\|\partial\Phi(\Phi(x(t)))(x(t)-\Phi(x(t)))\|_2$$

$$\geq \mu\|x(t)-\Phi(x(t))\|_2 - O(\|x(t)-\Phi(x(t))\|_2^2)\,.$$

Concluding, we have that

$$\sum_{k=1}^{M}\Lambda_k s_k(t)w_k^\top(x(t)-\Phi(x(t))) \geq \mu\|x(t)-\Phi(x(t))\|_2/2\,.$$

So

$$\mathbb{E}[\|x(t+1)-\Phi(x(t+1))\|_2^2 \mid x(t)]$$

$$\leq\|x(t)-\Phi(x(t))\|_2^2 - \frac{\mu\eta\rho}{M}\|x(t)-\Phi(x(t))\|_2$$

$$+ O(\eta\rho^2\|x(t)-\Phi(x(t))\| + \eta^2\rho^2)$$

$$\leq(\|x(t)-\Phi(x(t))\|_2 - C_2\eta\rho)^2\,.$$

The inequality holds if $\|x(t)-\Phi(x(t))\|_2 > C_1\eta\rho$.

Finally by Jenson's Inequality,

$$\mathbb{E}[\|x(t+1)-\Phi(x(t+1))\|_2|x(t)] \leq \|x(t)-\Phi(x(t))\|_2 - C_2\eta\rho\,.$$

$\square$

**Lemma J.10.** *Under condition of Theorem J.1, for any constant $C > 0$ independent of $\eta, \rho$, there exists constant $C_3 > 0$ independent of $\eta, \rho$, if $x(t) \in K^h$ and $\|x(t)-\Phi(x(t))\| \leq C\rho$, then we have that*

$$|\|x(t+1)-\Phi(x(t+1))\|_2 - \|x(t)-\Phi(x(t))\|_2| \leq C_3\eta\rho\,.$$

*Proof of Lemma J.10.* This is a direct application of Lemma F.8. □

**Lemma J.11.** *Under condition of Theorem J.1, assuming $x(t_0) \in K^{h/2}$ and $\|x(t_0) - \Phi(x(t_0))\| \leq f(\eta, \rho)$ for some fixed function $f$ and $f(\eta, \rho) \in \Omega(\eta\rho \ln^2(1/\eta\rho)) \cap O(\rho)$, then with probability $1 - O(\rho)$, for any $t$ satisfying $t_0 \leq t \leq t_0 + O(\ln(1/\eta\rho)/\eta)$, it holds that $\|x(t) - \Phi(x(t)\| \leq 2f(\eta, \rho)$. Moreover, we have that*
$$\|\Phi(x(t)) - \Phi(x(t_0))\| = O((\eta + \rho) \ln(1/\eta\rho)).$$

*Proof of Lemma J.11.* By Lemma J.6, we have that $x(t) \in K^h$ for any $t$ satisfying that $t_0 \leq t \leq t_0 + O(\ln(1/\eta\rho)/\eta)$ and with probability $1 - O(\rho)$ we will suppose this hold for the following deduction.

By Uniform Bound,

$\mathbb{P}(\exists t_0 \leq t \leq t_0 + O(\ln(1/\eta\rho)/\eta), \|x(t) - \Phi(x(t))\| \geq 2f(\eta, \rho))$

$$\leq \sum_{t=t_0}^{t_0 + O(\frac{\ln(1/\eta\rho)}{\eta})} \mathbb{P}(\|x(t) - \Phi(x(t))\| \geq 2f(\eta, \rho)) \quad \text{and} \quad \|x(\tau) - \Phi(x(\tau))\| \leq 2f(\eta, \rho), \forall t_0 \leq \tau \leq t - 1).$$

Consider each term and apply Uniform bound again,

$\mathbb{P}(\|x(t) - \Phi(x(t))\| \geq 2f(\eta, \rho)) \quad \text{and} \quad \|x(\tau) - \Phi(x(\tau))\| \leq 2f(\eta, \rho), \forall t_0 \leq \tau \leq t - 1)$

$$\leq \sum_{\tau=t_0}^{t} \mathbb{P}(\|x(t) - \Phi(x(t))\| \geq 2f(\eta, \rho)) \quad \text{and} \quad \|x(\tau) - \Phi(x(\tau))\| \leq f(\eta, \rho),$$

$$\text{and} \quad f(\eta, \rho) \leq \|x(\tau') - \Phi(x(\tau'))\| \leq 2f(\eta, \rho), \forall \tau + 1 \leq \tau' \leq t - 1).$$

Then if we consider each term, it is bounded by

$$\mathbb{P}(\|x(t) - \Phi(x(t))\| \geq 2f(\eta, \rho))$$
$$\text{and} \quad f(\eta, \rho) \leq \|x(\tau') - \Phi(x(\tau'))\| \leq 2f(\eta, \rho), \forall \tau + 1 \leq \tau' \leq t - 1$$
$$| \|x(\tau) - \Phi(x(\tau))\| \leq f(\eta, \rho)). \tag{38}$$

Now let $C$ be the positive constant satisfying $2f(\eta, \rho) \leq C\rho$, suppose $C_1, C_2$ are the constants corresponds to $C$ in Lemma J.9 and $C_3$ is the constant correspond to $C$ in Lemma J.10. By definition $C_3 > C_2$.

Define a coupled process $\tilde{y}(\tau + 1) = y(\tau + 1)$ and

$$\tilde{y}(\tau') = \begin{cases} \|x(\tau') - \Phi(x(\tau'))\|_2, & \text{if } \tilde{y}(\tau' - 1) = \|x(\tau' - 1) - \Phi(x(\tau' - 1))\|_2 > f(\eta, \rho) \\ \tilde{y}(\tau' - 1) - C_2\eta\rho, & \text{if otherwise.} \end{cases}$$

Now clearly Equation 38 is bounded by $\mathbb{P}(\tilde{y}(t) \geq 2f(\eta, \rho))$.

As $\mathbb{E}[\tilde{y}(\tau')] \leq \tilde{y}(\tau' - 1) - C_2\eta\rho$ by Lemma J.9 and $\|\tilde{y}(\tau') - \tilde{y}(\tau' - 1)\| \leq C_3\eta\rho$ by Lemma J.10. This implies $\|\tilde{y}(\tau')\| - C_2\eta\rho\tau'$ is a super martingale. By Azuma-Hoeffding bound(Lemma K.4), we have

$$P(\tilde{y}(t) \geq \tilde{y}(\tau + 1) - C_2\eta\rho(t - \tau - 1) + h) \leq 2\exp(-\frac{h^2}{4(t - \tau - 1)(C_3 + C_2)^2\eta^2\rho^2}).$$

Choosing $h = C_2\eta\rho(t - \tau - 1) - \|x(\tau + 1) - \Phi(x(\tau + 1))\| + 2f(\eta, \rho)$
$$P(\tilde{y}(t + 1) \geq 2f(\eta, \rho))$$
$$\leq 2\exp(-\frac{(C_2\eta\rho(t - \tau) - \|x(\tau + 1) - \Phi(x(\tau + 1))\| + 2f(\eta, \rho))^2}{8(t - \tau)(C_3 + C_2)^2\eta^2\rho^2})$$
$$\leq 2\exp(-\frac{(C_2\eta\rho(t - \tau) + f(\eta, \rho)/2)^2}{4(t - \tau)(C_3 + C_2)^2\eta^2\rho^2})$$
$$\leq 2\exp(-\frac{C_2 f(\eta, \rho)}{2(C_3 + C_2)^2\eta\rho}) \leq \eta^{10}\rho^{10}.$$

We then have

$$\mathbb{P}(\exists t_0 \leq t \leq t_0 + O(\ln(1/\eta\rho)/\eta), \|x(t) - \Phi(x(t)\| \geq 2f(\eta, \rho)) \leq \rho.$$

□

**Lemma J.12.** *Under condition of Theorem J.1, assuming there exists $t_{\mathrm{DEC}}$ such that $x(t_{\mathrm{DEC}}) \in K^{h/2}$ and $\|\nabla L(x(t_{\mathrm{DEC}}))\| \leq 4\zeta\rho$, then with probability $1 - O(\rho)$, there exists $t_{\mathrm{DEC2}} = t_{\mathrm{DEC}} + O(\ln(1/\eta\rho)/\eta)$, such that $\|x(t_{\mathrm{DEC2}}) - \Phi(t_{\mathrm{DEC2}})\| \leq O(\eta\rho)$.*

*Furthermore, for any $t$ satisfying $t_{\mathrm{DEC2}} \leq t \leq t_{\mathrm{DEC2}} + \Theta(\ln(1/\eta\rho)/\eta)$, we have that $\|\Phi(x(t)) - \Phi(x(t_{\mathrm{DEC}}))\| = O(\rho^2 \ln(1/\eta\rho))$.*

*Proof of Lemma J.12.* We have that $x(t) \in K^h$ (Lemma J.6) and $\|x(t) - \Phi(x(t))\| \leq C\rho$ for some constant $C$ (Lemma J.11) for any $t$ satisfying that $t_{\mathrm{DEC}} \leq t \leq t_{\mathrm{DEC}} + O(\ln(1/\eta\rho)/\eta)$ with probability $1 - O(\rho)$ and we will suppose this holds for the following deduction. The second statement then follows directly from Lemma F.8.

Let $C_1, C_2$ be the constant in Lemma J.9 corresponding to $C$, For simplicity of writing, define $T_1 \triangleq \lceil \frac{C \ln(\frac{C}{C_1 \eta\rho^2})}{C_2 \eta} \rceil = O(\ln(1/\eta\rho)/\eta)$. Define indicator function as
$$\mathcal{A}(t) = \mathbf{1}[\|x(t) - \Phi(x(t))\| \geq C_1 \eta\rho, \forall t \geq \tau \geq t_{\mathrm{GF}}].$$

By Lemma J.9, we have that,
$$\mathbb{E}[\|x(t+1) - \Phi(x(t+1))\|\mathcal{A}(t+1)] \leq \mathbb{E}[\|x(t+1) - \Phi(x(t+1))\|\mathcal{A}(t)]$$
$$\leq \mathbb{E}[\|x(t) - \Phi(x(t))\|\mathcal{A}(t)] - C_2 \eta\rho \mathbb{E}[\mathcal{A}(t)]$$
$$\leq \mathbb{E}[\|x(t) - \Phi(x(t))\|\mathcal{A}(t)](1 - \frac{C_2\eta}{C})$$

We can then conclude that with $T_2 = T_1 + t_{\mathrm{DEC}}$, using Lemma F.2,
$$C_1 \eta\rho \mathbb{E}\mathcal{A}(T_2 + 1) \leq \mathbb{E}[\|x(T_2 + 1) - \Phi(x(T_2 + 1))\|_2 \mathcal{A}(T_2 + 1)]$$

$$\leq (1 - \frac{C_2\eta}{C})^{T_1} \|x(t_{\mathrm{DEC}}) - \Phi(x(t_{\mathrm{DEC}}))\| \leq C_1 \eta\rho^3.$$
This implies $\mathcal{A}(T_2 + 1) = 0$ with probability $1 - O(\rho)$, which indicates the existence of $t_{\mathrm{DEC2}}$. $\qquad\square$

## J.2 PHASE II (PROOF OF THEOREM J.2)

*Proof of Theorem J.2.* We will inductively prove the following induction hypothesis $\mathcal{P}(t)$ holds with probability $1 - O(\eta^3 \rho^3 t)$ for $t \leq T_3/\eta\rho^2 + 1$,
$$x(\tau) \in K^{h/2}, \tau \leq t$$
$$\|x(\tau) - \Phi(x(\tau))\|_2 \leq 2\|x(0) - \Phi(x(0))\|_2 = O(\eta\rho), \tau \leq t$$
$$\|\Phi(x(\tau)) - X(\eta\rho^2 \tau)\| = \tilde{O}(\eta^{1/2} + \rho), \tau \leq t$$

$\mathcal{P}(0)$ holds trivially. Now suppose $\mathcal{P}(t)$ holds, then $x(t+1) \in K^h$. By Lemma J.13, we have that with probability $1 - O(\eta^3 \rho^3)$, $\|x(t+1) - \Phi(x(t+1))\| \leq 2\|x(0) - \Phi(x(0))\|_2 = O(\eta\rho)$.

Now we have
$$2\|x(0) - \Phi(x(0))\|_2 = O(\eta\rho), \tau \leq t + 1.$$
$$x(\tau) \in K^h, \tau \leq t + 1$$

By Lemma J.14, it holds that
$$\|\Phi(x(\tau+1)) - \Phi(x(\tau)) + \eta\rho^2 P^{\perp}_{\Phi(x(\tau)),\Gamma} \nabla \lambda_1 \left( \nabla^2 L_{k_\tau} \left( \Phi(x(\tau)) \right) \right)/2\| \leq \tilde{O}(\eta\rho^3 + \eta^2 \rho^2).$$

As
$$\mathbb{E}_{k_t} P^{\perp}_{\Phi(x(t)),\Gamma} \nabla \lambda_1 \left( \nabla^2 L_{k_t} \left( \Phi(x(t)) \right) \right) = P^{\perp}_{\Phi(x(t)),\Gamma} \nabla \mathrm{Tr}(\nabla^2 L(\Phi(x(t)))).$$
By Theorem L.1, let $b(x) = -\partial\Phi(x)\nabla\mathrm{Tr}(\nabla^2 L(x))$, $b_k(x) = -\partial\Phi(x)\mathrm{Tr}(\nabla^2 L_{k_t}(x))$, $p = \eta\rho^2$ and $\epsilon = O(\eta + \rho)$, it holds that, with probability $1 - O(\eta^3 \rho^3)$,
$$\|\Phi(x(\tau)) - X(\eta\rho^2 \tau)\|$$
$$= O(\|\Phi(x(0)) - \Phi(x_{\mathrm{init}})\| + T_3 \eta\rho^2 + \sqrt{\eta\rho^2 T_3 \log(2eT_3/(\eta^2\rho^4))} + (\rho + \eta)T_3)$$
$$= \tilde{O}(\eta^{1/2} + \rho), \tau \leq t + 1$$
This implies $\|x(t+1) - X(\eta\rho^2(t+1))\|_2 \leq \|x(t+1) - \Phi(x(t+1))\|_2 + \|\Phi(x(t+1)) - X(\eta\rho^2(t+1))\|_2 = \tilde{O}(\eta^{1/2} + \rho) < h/2$. Hence $x(t+1) \in K^{h/2}$. Combining with $\mathcal{P}(t)$ holds with probability $1 - O(\eta^3 \rho^3 t)$, we have that $\mathcal{P}(t+1)$ holds with probability $1 - O(\eta^3 \rho^3 (t+1))$. The induction is complete.

Now $\mathcal{P}(\lceil T_3/\eta\rho^2 \rceil)$ is equivalent to our theorem. $\qquad\square$

### J.2.1 CONVERGENCE NEAR MANIFOLD

**Lemma J.13.** *Under condition of Theorem J.2, assuming $x(t) \in K^h, \forall t_0 \leq t \leq t_0 + O(1/\eta\rho^2)$ and $\|x(t_0) - \Phi(x(t_0))\| \leq f(\eta, \rho)$ for some fixed function $f$ and $f(\eta, \rho) \in \Omega(\eta\rho\ln^2(1/\eta\rho)) \cap O(\rho)$, then with probability $1 - O(\eta^3\rho^3)$, for any $t$ satisfying $t_0 \leq t \leq t_0 + O(1/\eta\rho^2)$, it holds that $\|x(t) - \Phi(x(t))\| \leq 2f(\eta, \rho)$.*

*Proof of Lemma J.13.* The proof is almost identical to Lemma J.11 and is omitted. □

### J.2.2 TRACKING RIEMANNIAN GRADIENT FLOW

**Lemma J.14.** *Under the condition of Theorem J.2, for any $t$ satisfying that $x(t) \in K^h$ and*

$$\|x(t) - \Phi(x(t))\| = O(\eta\rho\ln^2(1/\eta\rho)).$$

*It holds that*

$$\|\Phi(x(t+1)) - \Phi(x(t)) + \eta\rho^2 P^{\perp}_{\Phi(x(t)),\Gamma}\nabla\lambda_1\Big(\nabla^2 L_{k_t}\big(\Phi(x(t)))\big)\Big)/2\| \leq \tilde{O}(\eta\rho^3 + \eta^2\rho^2).$$

*Proof of Lemma J.14.* We will abbreviate $k_t$ by $k$ in this proof.

By Taylor Expansion,

$$x(t+1) = x(t) - \eta\nabla L_k\left(x(t) + \rho\frac{\nabla L_k(x(t))}{\|\nabla L_k(x(t))\|}\right)$$

$$= x(t) - \eta\nabla L_k(x(t)) - \eta\rho\nabla^2 L_k(x(t))\frac{\nabla L_k(x(t))}{\|\nabla L_k(x(t))\|}$$

$$- \eta\rho^2\partial^2(\nabla L_k)[\frac{\nabla L_k(x(t))}{\|\nabla L_k(x(t))\|}, \frac{\nabla L_k(x(t))}{\|\nabla L_k(x(t))\|}]/2 + O(\eta\rho^3).$$

Now as $\|x(t) - \Phi(x(t))\|_2 = \tilde{O}(\eta\rho)$, by Lemma F.8, it implies

$$\|x(t+1) - x(t)\|_2 = O(\eta\rho).$$

Then we have

$$\|\Phi(x(t+1)) - \Phi(x(t)) - \partial\Phi(x(t))(x(t+1) - x(t))\|_2 \leq \xi\|x(t+1) - x(t)\|_2^2 = O(\eta^2\rho^2).$$

Using Lemma F.6, we have

$$\|\eta\partial\Phi(x(t))\nabla L_k(x(t))\|_2 = O(\eta\|x(t) - \Phi(x(t))\|_2^2) = O(\eta^3\rho^2 + \eta\rho^4),$$

$$\|\eta\rho\partial\Phi(x(t))\nabla^2 L_k(x(t))\frac{\nabla L_k(x(t))}{\|\nabla L_k(x(t))\|}\|_2 = O(\eta\rho\|x(t) - \Phi(x(t))\|_2) = \tilde{O}(\eta^2\rho^2 + \eta\rho^3).$$

Hence

$$\|\Phi(x(t+1)) - \Phi(x(t)) + \eta\rho^2\partial\Phi(x(t))\partial^2(\nabla L_k)[\frac{\nabla L_k(x(t))}{\|\nabla L_k(x(t))\|}, \frac{\nabla L_k(x(t))}{\|\nabla L_k(x(t))\|}]/2\|_2 = \tilde{O}(\eta^2\rho^2 + \eta\rho^3).$$

Notice finally that by Lemma J.8,

$$\partial\Phi(x(t))\partial^2(\nabla L_k)[\frac{\nabla L_k(x(t))}{\|\nabla L_k(x(t))\|}, \frac{\nabla L_k(x(t))}{\|\nabla L_k(x(t))\|}]$$

$$= \partial\Phi(\Phi(x(t)))\partial^2(\nabla L_k)[w_k, w_k] + O(\|x(t) - \Phi(x(t))\|_2)$$

$$= P^{\perp}_{\Phi(x(t)),\Gamma}\nabla(\lambda_1(\nabla^2 L_k(\Phi(x(t))))) + O(\|x(t) - \Phi(x(t))\|_2).$$

Hence we have

$$\Phi(x(t+1)) - \Phi(x(t)) = -\eta\rho^2 P^{\perp}_{\Phi(x(t)),\Gamma}\nabla\lambda_1\Big(\nabla^2 L_{k_t}\big(\Phi(x(t)))\big)\Big)/2 + \tilde{O}(\eta^2\rho^2 + \eta\rho^3)$$

This completes the proof. □

### J.3 Proof of Theorem 5.4

*Proof of Theorem 5.4.* By Theorem J.1, there exists constant $T_1$ independent of $\eta, \rho$, such that there exists $t_{\text{PHASE}} \le T_1 \ln(1/\eta\rho)/\eta$, with probability $1 - O(\rho)$, it holds that

$$\|x(t_{\text{PHASE}}) - \Phi(x(t_{\text{PHASE}}))\|_2 = O(\eta\rho).$$

$$\|\Phi(x(t_{\text{PHASE}})) - \Phi(x_{\text{init}})\| = \tilde{O}(\eta^{1/2} + \rho)$$

Hence by Theorem J.2, if we consider a translated process with $x'(t) = x(t + t_{\text{PHASE}})$, we would have for any $T_3$ such that the solution $X$ of Equation 14 is well defined, we have that for $t = \lceil \frac{T_3}{\eta\rho^2} \rceil$

$$\|\Phi(x'(t)) - X(\eta\rho^2 t)\|_2 = O(\eta \ln(1/\rho)).$$

This implies for $t$ satisfying $X(\eta\rho^2(t - t_{\text{PHASE}}))$ is well-defined,

$$\|\Phi(x(t)) - X(\eta\rho^2(t - t_{\text{PHASE}}))\|_2 = \tilde{O}(\eta^{1/2} + \rho).$$

Finally, as

$$\|X(\eta\rho^2(t - t_{\text{PHASE}})) - X(\eta\rho^2 t)\|_2 = O(\eta\rho^2 t_{\text{PHASE}}) = O(\rho \ln(1/\eta\rho)) = \tilde{O}(\rho).$$

We have that

$$\|\Phi(x(t)) - X(\eta\rho^2 t)\|_2 = \tilde{O}(\eta^{1/2} + \rho).$$

We also have

$$\|x(t) - \Phi(x(t))\|_2 = O(\eta\rho).$$

by Theorem J.2. □

### J.4 Proofs of Corollaries 5.6 and 5.7

*Proof of Corollary 5.6.* We will do Taylor expansion on $\mathbb{E}_k[L_{k,\rho}^{\text{Max}}](x)$. By Theorem J.1 and J.2, we have $\|x(\lceil T_3/\eta\rho^2 \rceil) - X(T_3)\|_2 = \tilde{O}(\eta^{1/2} + \rho)$ and $\|\Phi(x(\lceil T_3/\eta\rho^2 \rceil)) - x(\lceil T_3/\eta\rho^2 \rceil)\|_2 = \tilde{O}(\eta^{1/2} + \rho)$. For convenience, we denote $x(\lceil T_3/\eta\rho^2 \rceil)$ by $x$.

$$\mathbb{E}_k[R_{k,\rho}^{\text{Max}}](x) = \max_{\|v\| \le 1} \mathbb{E}_k[\rho v^\top \nabla L_k(x) + \rho^2 v^\top \nabla^2 L_k(x) v/2] + O(\rho^3)$$

Since $\max_{\|v\| \le 1} |v^\top \nabla L_k(x)| = O(\|x - \Phi(x)\|) = \tilde{O}(\eta^{1/2} + \rho)$, it holds that,

$$\mathbb{E}_k[R_{k,\rho}^{\text{Max}}](x) = \rho^2 \mathbb{E}_k[\max_{\|v\| \le 1} v^\top \nabla^2 L(x) v/2] + O\big((\eta^{1/4} + \rho^{1/4})\rho^2\big)$$

$$= \rho^2 \mathbb{E}_k \max_{\|v\| \le 1} [v^\top \nabla^2 L(X(T_3)) v/2] + O\big((\eta^{1/4} + \rho^{1/4})\rho^2\big)$$

$$= \rho^2 \text{Tr}(X(T_3))/2 + O\big((\eta^{1/4} + \rho^{1/4})\rho^2\big)$$

□

*Proof of Corollary 5.7.* We choose $T_\epsilon$ such that $X(T_\epsilon)$ is sufficiently close to $X(\infty)$, such that $\text{Tr}(X(T_\epsilon)) \le \text{Tr}(X(\infty)) + \epsilon/2$. By corollary 5.6 (let $T_3 = T_\epsilon$), we have for all $\rho, \eta$ such that $(\eta + \rho) \ln(1/\eta\rho)$ is sufficiently small, $\|\mathbb{E}_k[R_{k,\rho}^{\text{Max}}](x(\lceil T_\epsilon/(\eta\rho^2) \rceil)) - \rho^2 \text{Tr}(X(T))/2\|_2 \le o(1)$. This further implies $\|\mathbb{E}_k[R_{k,\rho}^{\text{Max}}](x(\lceil T_\epsilon/(\eta\rho^2) \rceil)) - \rho^2 \text{Tr}(X(\infty))/2\|_2 \le \epsilon\rho^2/2 + o(1)$. We also have $L(x(\lceil T_\epsilon/(\eta\rho^2) \rceil)) - \inf_{x \in U'} L(x) = o(1)$. Then we can leverage Theorems G.6 and G.14 to get the desired bound. □

### J.5 Other Omitted Proofs for 1-SAM

We will use $\ell'(y, y_k)$ and $\ell''(y, y_k)$ to denote $\frac{d\ell(y', y_k)}{dy'}|_{y'=y}$ and $\frac{d^2\ell(y', y_k)}{dy'^2}|_{y'=y}$.

**Lemma J.15.** *Under Setting 5.1, fix $k \in [M]$, for any $p$ satisfying $\ell(f_k(p), y_k) = 0$, we have that*

$$\nabla^2 L_k(p) = \ell''(f_k(p), y_k) \nabla f_k(p)(\nabla f_k(p))^\top.$$

*Proof of Lemma J.15.* $\ell(f_k(p), y_k) = 0$ implies $\ell'(f_k(p), y_k) = 0$. Then by Taylor Expansion,

$$
\begin{aligned}
\nabla^2 L_k(p) &= \nabla_p^2 \ell(f_k(p), y_k) \\
&= \partial_p[\ell'(f_k(p), y_k)\nabla f_k(p)] \\
&= \ell''(f_k(p), y_k)\nabla f_k(p)\nabla f_k(p)^\top + \ell'(f_k(p), y_k)\nabla^2 f_k(p) \\
&= \ell''(f_k(p), y_k)\nabla f_k(p)\nabla f_k(p)^\top.
\end{aligned}
$$

This concludes the proof. $\qquad\square$

*Proof of Lemma 5.5.* By Lemma J.15, as $L(p) = \frac{1}{M}\sum_{k=1}^M L_k(p)$, we have

$$
\nabla^2 L(p) = \frac{1}{M}\sum_{k=1}^M \frac{\partial^2 \ell(y', y_k)}{(\partial y')^2}\Big|_{y'=f_k(x)}\nabla f_k(p)\nabla f_k(p)^\top.
$$

By definition of $\Gamma$ in Setting 5.1, we have for any $p \in \Gamma$, $\{\nabla f_k(p)\}_{k=1}^n$ are linearly independent, which implies that $\nabla f_k(p) \neq 0$ for any $p \in \Gamma$.

For any $p \in \Gamma$, as $\frac{\nabla f_k(p)}{\|\nabla f_k(p)\|}$ is well defined and continuous at $p$, there exists a open ball $V$ containing $p$ such that for any $x \in V$, $\|\nabla f_k(x)\|_2 \geq C_1 > 0$ and $\|\nabla[\frac{\nabla f_k(x)}{\|\nabla f_k(x)\|}]\|_2 \leq C_2$ for some constants $C_1$ and $C_2$.

Suppose $\nabla L_k(x) \neq 0$, then as by Taylor Expansion,

$$
\nabla L_k(x) = \ell'(f_k(x), y_k)\nabla f_k(x).
$$

We have $\frac{\nabla L_k(x)}{\|\nabla L_k(x)\|} = \frac{\nabla f_k(x)}{\|\nabla f_k(x)\|} = \frac{\nabla f_k(p)}{\|\nabla f_k(p)\|} + C_2\|x - p\|$, which completes the proof. $\qquad\square$

We note that the alignment result in Lemma 5.5 is not directly used in our proof. Instead, we use its generalized version Lemma J.8 which holds under holds under a more general condition than Setting 5.1, namely Condition E.1.

## K  TECHNICAL LEMMAS

**Lemma K.1** (Corollary 4.3.15 in Horn et al. (2012))**.** *Let $\Sigma, \hat{\Sigma} \in R^{D \times D}$ be symmetric and non-negative with eigenvalues $\lambda_1 \geq ... \geq \lambda_D$ and $\hat{\lambda}_1 \geq ... \geq \hat{\lambda}_D$, then for any $i$,*

$$
|\hat{\lambda}_i - \lambda_i| \leq \|\Sigma - \hat{\Sigma}\|_2
$$

**Definition K.2** (Unitary invariant norms)**.** *A matrix norm $\|\cdot\|_*$ on the space of matrices in $\mathbb{R}^{p \times d}$ is unitary invariant if for any matrix $K \in \mathbb{R}^{p \times d}$, $\|UKW\|_* = \|K\|_*$ for any unitary matrices $U \in \mathbb{R}^{p \times p}, W \in \mathbb{R}^{d \times d}$.*

**Theorem K.3.** *[Davis-Kahan $\sin(\theta)$ theorem (Davis et al., 1970)] Let $\Sigma, \hat{\Sigma} \in \mathbb{R}^{p \times p}$ be symmetric, with eigenvalues $\lambda_1 \geq \ldots \geq \lambda_p$ and $\hat{\lambda}_1 \geq \ldots \geq \hat{\lambda}_p$ respectively. Fix $1 \leq r \leq s \leq p$, let $d \triangleq s - r + 1$ and let $V = (v_r, v_{r+1}, \ldots, v_s) \in \mathbb{R}^{p \times d}$ and $\hat{V} = (\hat{v}_r, \hat{v}_{r+1}, \ldots, \hat{v}_s) \in \mathbb{R}^{p \times d}$ have orthonormal columns satisfying $\Sigma v_j = \lambda_j v_j$ and $\hat{\Sigma}\hat{v}_j = \hat{\lambda}_j \hat{v}_j$ for $j = r, r+1, \ldots, s$. Define $\Delta \triangleq \min\left\{\max\{0, \lambda_s - \hat{\lambda}_{s+1}\}, \max\{0, \hat{\lambda}_{r-1} - \lambda_r\}\right\}$, where $\hat{\lambda}_0 \triangleq \infty$ and $\hat{\lambda}_{p+1} \triangleq -\infty$, we have for any unitary invariant norm $\|\cdot\|_*$,*

$$
\Delta \cdot \|\sin\Theta(\hat{V}, V)\|_* \leq \|\hat{\Sigma} - \Sigma\|_*.
$$

*Here $\Theta(\hat{V}, V) \in \mathbb{R}^{d \times d}$, with $\Theta(\hat{V}, V)_{j,j} = \arccos\sigma_j$ for any $j \in [d]$ and $\Theta(\hat{V}, V)_{i,j} = 0$ for all $i \neq j \in [d]$. $\sigma_1 \geq \sigma_2 \geq \cdots \geq \sigma_d$ denotes the singular values of $\hat{V}^\top V$. $[\sin\Theta]_{ij}$ is defined as $\sin(\Theta_{ij})$.*

**Lemma K.4** (Azuma-Hoeffding Bound)**.** *Suppose $\{Z_n\}_{n \in \mathbb{N}}$ is a super-martingale, suppose $-\alpha \leq Z_{i+1} - Z_i \leq \beta$, then for all $n > 0, a > 0$, we have*

$$
\mathbb{P}(Z_n - Z_0 \geq a) \leq 2\exp(-a^2/(2n(\alpha+\beta)^2))
$$

**Lemma K.5** (Azuma-Hoeffding Bound, Vector Form, Hayes (2003))**.** *Suppose $\{Z_n\}_{n \in \mathbb{N}}$ is a $\mathbb{R}^D$-valued martingale, suppose $\|Z_{i+1} - Z_i\|_2 \leq \sigma$, then for all $n > 0, a > 0$, we have*

$$
\mathbb{P}(\|Z_n - Z_0\|_2 \geq \sigma(1+a)) \leq 2\exp(1 - a^2/2n).
$$

*In other words, for any $0 < \delta < 1$, with probability at least $1 - \delta$, we have that*

$$
\|Z_n - Z_0\|_2 \leq \sigma\left(1 + \sqrt{2n\log\frac{2e}{\delta}}\right) \leq 2\sigma\sqrt{2n\log\frac{2e}{\delta}}.
$$

**Lemma K.6** (Discrete Gronwall Inequality, Borkar (2009)). *Let $\{x(t)\}_{t\in\mathbb{N}}$ be a sequence of nonnegative real numbers, $\{a_n\}_{n\in\mathbb{N}}$ be a sequence of positive real numbers and $C, L > 0$ scalars such that for all $n$,*

$$x(t) \le C + L \sum_{n=0}^{t-1} a_n x(n).$$

*Then for $T_t = \sum_{n=0}^{t} a_n$, it holds that $x(t+1) \le Ce^{LT_t}$.*

**Lemma K.7** (Magnus (1985)). *Let $A : \mathbb{R}^D \to \mathbb{R}^{D\times D}$ be any $\mathcal{C}^1$ symmetric matrix function and $x^* \in \mathbb{R}^D$ satisfying $\lambda_1(A(x^*)) > \lambda_2(A(x^*))$ and $v_1$ be the top eigenvector of $A(x^*)$. It holds that $\nabla\lambda_1(A(x))|_{x=x^*} = \nabla(v_1^\top A(x)v_1)|_{x=x^*}$.*

We then present some of the technical lemmas we required to prove Lemma H.5.

**Lemma K.8.** *If $0 < c < \frac{b-a}{b^2}, a\sqrt{\frac{a^2+2b^2}{2(1-cb)}} \ge \frac{a^2+b^2}{2-ca-cb}$, then $a > \frac{1}{2}b, cb \le \frac{1}{2}$*

*Proof of Lemma K.8.* Notice that

$$ca\sqrt{\frac{a^2+b^2}{1-cb}} \ge ca\sqrt{\frac{a^2+2b^2}{2(1-cb)}} \ge \frac{cb^2+ca^2}{2-cb-ca} \ge \frac{cb^2+ca^2}{2-cb}.$$

So

$$\sqrt{1-cb} + \frac{1}{\sqrt{1-cb}} \ge \sqrt{1+\frac{b^2}{a^2}}$$

As $c < \frac{b-a}{b^2}$, we have $1 > 1 - cb > \frac{a}{b}$.

So

$$\sqrt{\frac{a}{b}} + \sqrt{\frac{b}{a}} \ge \sqrt{1+\frac{b^2}{a^2}}$$

The above inequality implies $a \ge \frac{1}{2}b$. As $c < \frac{b-a}{b^2}$, $cb \le \frac{1}{2}$. $\qquad\square$

**Lemma K.9.** *When $0 < a < b, 0 < c < \frac{b-a}{b^2}$, we have*

$$cb^2 + ca^2(2-cb-\frac{2}{3}ca) - (1-cb)\frac{c(a^2+b^2)}{2-ca-cb} - ca^2(\frac{1}{2}a^2+b^2)\frac{2-ca-cb}{(a^2+b^2)} \le \frac{cb^2}{2-cb}$$

*Proof of Lemma K.9.* Equivalently, we are going to prove

$$(1-cb)b^2\left(\frac{1}{2-ca-cb} - \frac{1}{2-cb}\right) + a^2\frac{1-cb}{2-ca-cb} + a^2(\frac{1}{2}a^2+b^2)\frac{2-ca-cb}{(a^2+b^2)} \ge a^2(2-cb-\frac{2}{3}ca)$$

Further simplifying, we only need to prove

$$\frac{(1-cb)cab^2}{(2-cb)(2-ca-cb)} + a^2\frac{1-cb}{2-ca-cb} \ge \frac{1}{3}ca^3 + \frac{a^4}{2(a^2+b^2)}(2-ca-cb)$$

We have the following auxiliary inequalities,

$$(1-cb)b > a$$

$$\frac{1-cb}{2-ca-cb} \ge \frac{1}{\frac{a+b}{b}+\frac{b-a}{(1-cb)b}} \ge \frac{1}{\frac{a+b}{b}+\frac{b-a}{a}} = \frac{ab}{a^2+b^2} \ge \frac{a^2}{a^2+b^2}$$

$$\frac{1-cb}{a^2} \ge \frac{2-ca-cb}{a^2+b^2}$$

Using the above auxiliary inequalities we have

$$\frac{(1-cb)cab^2}{(2-cb)(2-ca-cb)} + a^2\frac{1-cb}{2-ca-cb} \geq \frac{1}{3}ca^3 + \frac{a^4}{2(a^2+b^2)}(2-ca-cb)$$

$$\Leftarrow \frac{ca^2b}{(2-cb)(2-ca-cb)} + \left(1-\frac{1}{2}(2-ca-cb)\right)\frac{a^2(1-cb)}{2-ca-cb} \geq \frac{1}{3}ca^3$$

$$\Leftarrow \frac{ca^2b}{(2-cb)(2-ca-cb)} + \frac{ca^2(a+b)(1-cb)}{2(2-ca-cb)} \geq \frac{1}{3}ca^3$$

$$\Leftarrow \frac{ca^2b}{(2-cb)(2-ca-cb)} + \frac{ca^2b(1-cb)}{2(2-ca-cb)} \geq \frac{1}{3}ca^2b$$

$$\Leftarrow \frac{1}{(2-cb)^2} + \frac{1-cb}{2(2-cb)} \geq \frac{1}{3}$$

$$\Leftarrow 3(1-cb)(2-cb) + 6 \geq 2(2-cb)^2$$

$$\Leftarrow (cb)^2 - cb + 4 \geq 0$$

$\square$

**Lemma K.10.** *When* $0 < a < b, 0 < c < \frac{b-a}{b^2}, a\sqrt{\frac{a^2+2b^2}{2(1-cb)}} \geq \frac{a^2+b^2}{2-ca-cb}$*, we have*

$$cb^2 + ca^2(2-cb-\frac{2}{3}ca) - (1-cb)cb^2 - ca^2(\frac{1}{2}a^2+b^2)\frac{1}{b^2} \leq \frac{cb^2}{2-cb}$$

*Proof of Lemma K.10.* Equivalently, we are going to prove,

$$cb^3 + a^2(2-cb-\frac{2}{3}ca) \leq \frac{b^2}{2-cb} + \frac{a^2(\frac{1}{2}a^2+b^2)}{b^2}$$

$$\Longleftrightarrow cb^3 + a^2(1-cb-\frac{2}{3}ca) \leq \frac{b^2}{2-cb} + \frac{a^4}{2b^2}$$

We have the auxiliary inequality $\frac{1}{2-cb} > \frac{1}{2} + \frac{cb}{4}$.

Hence

$$cb^3 + a^2(1-cb-\frac{2}{3}ca) \leq \frac{b^2}{2-cb} + \frac{a^4}{2b^2}$$

$$\Leftarrow cb^3 + a^2(1-cb-\frac{2}{3}ca) \leq \frac{b^2}{2} + \frac{a^4}{2b^2} + \frac{cb^3}{4}$$

$$\Leftarrow c(\frac{3b^3}{4} - ba^2 - \frac{2}{3}a^3) \leq \frac{b^2}{2} + \frac{a^4}{2b^2} - a^2$$

1 *Case 1,* If $\frac{3b^3}{4} - ba^2 - \frac{2}{3}a^3 \leq 0$, then

$$c(\frac{3b^3}{4} - ba^2 - \frac{2}{3}a^3) \leq 0 \leq \frac{b^2}{2} + \frac{a^4}{2b^2} - a^2$$

2 *Case 2,* If $\frac{3b^3}{4} - ba^2 - \frac{2}{3}a^3 > 0$, then

$$c(\frac{3b^3}{4} - ba^2 - \frac{2}{3}a^3) \leq \frac{b^2}{2} + \frac{a^4}{2b^2} - a^2$$

$$\Leftarrow \frac{b-a}{b^2}(\frac{3b^3}{4} - ba^2 - \frac{2}{3}a^3) \leq \frac{(b^2-a^2)^2}{2b^2}$$

$$\Leftarrow 2(\frac{3b^3}{4} - ba^2 - \frac{2}{3}a^3) \leq (b-a)(b+a)^2$$

$$\Leftarrow 2(b^3 - ba^2) - (b-a)(b+a)^2 \leq \frac{b^3}{2} + \frac{4a^3}{3}$$

$$\Leftarrow (b-a)(2b(a+b) - (a+b)^2) \leq \frac{b^3}{2} + \frac{4a^3}{3}$$

$$\Leftarrow (b-a)^2(b+a) \leq \frac{b^3}{2} + \frac{4a^3}{3}$$

Using Lemma K.8, $a > \frac{b}{2}, (b-a)^2(b+a) = (b^2-a^2)(b-a) \leq b^2(b-a) \leq \frac{b^3}{2}$

$\square$

**Lemma K.11.** *When* $0 \le a \le b, 0 \le c \le \frac{b-a}{b^2}, b^2 \ge a\sqrt{\frac{a^2+2b^2}{2(1-cb)}} \ge \frac{a^2+b^2}{2-ca-cb}$, *we have*

$$cb^2 + ca^2(2 - cb - \frac{2}{3}ca) - 2ca\sqrt{(b^2 + \frac{1}{2}a^2)(1 - cb)} \le \frac{cb^2}{2 - cb}$$

*Proof of Lemma K.11.* Define

$$F(a) \triangleq a^2(2 - cb - \frac{2}{3}ca) - 2a\sqrt{(b^2 + \frac{1}{2}a^2)(1 - cb)}$$

$$S(c,b) \triangleq \{a | 0 \le a \le b, 0 < c \le \frac{b-a}{b^2}, b^2 \ge a\sqrt{\frac{a^2+2b^2}{2(1-cb)}} \ge \frac{a^2+b^2}{2-ca-cb}\}$$

$$a_{min}(c,b) \triangleq \inf S(c,b)$$
$$a_{max}(c,b) \triangleq \sup S(c,b) \le b - cb^2$$

Consider

$$\frac{dF(a)}{da} = 2a(2 - cb - \frac{2}{3}ca) - \frac{2}{3}ca^2 - 2\sqrt{(b^2 + \frac{1}{2}a^2)(1 - cb)} - a^2\sqrt{\frac{1-cb}{b^2 + \frac{1}{2}a^2}}$$

$$\frac{d^2F(a)}{da^2} = 2(2 - cb - \frac{2}{3}ca) - \frac{4}{3}ca - \frac{4}{3}ca - a\sqrt{\frac{1-cb}{b^2 + \frac{1}{2}a^2}} - 2a\sqrt{\frac{1-cb}{b^2 + \frac{1}{2}a^2}} + \frac{a^3}{2(b^2 + \frac{1}{2}a^2)^{\frac{3}{2}}}\sqrt{1-cb}$$

$$\ge 4 - 2cb - 4ca - 3a\sqrt{\frac{1-cb}{b^2 + \frac{1}{2}a^2}}$$

Define $u \triangleq cb, v \triangleq \frac{a}{b}$, then $u + v \le 1$.

$$\frac{d^2F(a)}{da^2} \ge 4 - 2u - 4uv - 3\sqrt{1-u}\frac{1}{\sqrt{\frac{1}{2} + \frac{1}{v^2}}}$$

$$\ge 4 - 2u - 4u(1-u) - 3\sqrt{1-u}\frac{1}{\sqrt{\frac{1}{2} + \frac{1}{(1-u)^2}}}$$

$$\ge 4u^2 - 6u + 4 - 3\sqrt{1-u}\frac{(1-u)}{\sqrt{\frac{(1-u)^2}{2} + 1}}$$

As $\sqrt{\frac{(1-u)^2}{2} + 1} \ge \sqrt{\frac{(1-u)^2+1}{2}} \ge \sqrt{1-u}$, we have

$$\frac{d^2F(a)}{da^2} \ge 4u^2 - 6u + 4 - 3(1-u) = 4u^2 + 1 - 3u > 0$$

The above inequality shows that $F(a)$ is convex w.r.t to $a$ for $a_{min}(c,b) \le a \le a_{max}(c,b)$. Hence $F(a) \le \max(F(a_{min}(c,b)), F(a_{max}(c,b)))$. Below we use $a_{min}, a_{max}$ as shorthands for $a_{min}(c,b), a_{max}(c,b)$.

For $F(a_{min})$, we have $a_{min}\sqrt{\frac{a_{min}^2+2b^2}{2(1-cb)}} = \frac{a_{min}^2+b^2}{2-ca_{min}-cb}$. This implies

$$2a_{min}\sqrt{(b^2 + \frac{1}{2}a_{min}^2)(1-cb)} = (1-cb)\frac{(a_{min}^2+b^2)}{2-ca_{min}-cb} + a_{min}^2(\frac{1}{2}a_{min}^2 + b^2)\frac{2-ca_{min}-cb}{(a_{min}^2+b^2)}$$

Hence using Lemma K.9,

$$F(a_{min}) = a_{min}^2(2 - cb - \frac{2}{3}ca_{min}) - (1-cb)\frac{c(a_{min}^2+b^2)}{2-ca_{min}-cb} - ca_{min}^2(\frac{1}{2}a_{min}^2 + b^2)\frac{2-ca_{min}-cb}{(a_{min}^2+b^2)}$$

$$\le \frac{1}{c}(\frac{cb^2}{2-cb} - cb^2)$$

For $F(a_{max})$, we know that $a_{max}$ must satisfy at least of the following three equalities and we discuss three cases one by one.

1. $a_{max}\sqrt{\frac{a_{max}^2+2b^2}{2(1-cb)}} = \frac{a_{max}^2+b^2}{2-ca_{max}-cb}$, in this case we simply redo the calculation in Part 1.

2. $b^2 = a_{max}\sqrt{\frac{a_{max}^2+2b^2}{2(1-cb)}}$. This implies

$$2a_{max}\sqrt{(b^2 + \frac{1}{2}a_{max}^2)(1-cb)} = (1-cb)b^2 + a_{max}^2(\frac{1}{2}a_{max}^2 + b^2)\frac{1}{b^2}$$

Hence using Lemma K.10,

$$F(a_{max}) = a_{max}^2(2 - cb - \frac{2}{3}ca_{max}) - (1-cb)cb^2 - ca_{max}^2(\frac{1}{2}a_{max}^2 + b^2)\frac{1}{b^2}$$

$$\leq \frac{1}{c}(\frac{cb^2}{2-cb} - cb^2)$$

3. $cb^2 = b - a_{max}$. Define $v \triangleq \frac{a_{max}}{b}, cb = 1 - v$. Note that $1 - cb = \frac{a_{max}}{b}$ and $b^2 \geq a_{max}\sqrt{\frac{b(a_{max}^2+2b^2)}{2a_{max}}}$. These imply $a_{max}^3 + 2a_{max}b^2 - 2b^3 \leq 0 \Rightarrow a_{max} < 0.9b$. This implies $v \leq 0.9$. By Lemma K.8, $0.5 \leq v$.
   As $v \in [0.5, 0.9]$, it holds that

$$v(1 + v) + \frac{1}{1+v} \leq 2\sqrt{(1 + \frac{v^2}{2})v}.$$

This implies

$$v^2(2 - (1-v) - \frac{2}{3}(1-v)v) - 2v\sqrt{(1+\frac{v^2}{2})v} \leq \frac{-v}{1+v} = \frac{1}{2-cb} - 1$$

Finally,

$$F(a_{max}) = a_{max}^2(2 - cb - \frac{2}{3}ca_{max}) - 2a_{max}\sqrt{(b^2 + \frac{1}{2}a_{max}^2)(1-cb)}$$

$$= b^2\left(v^2(2 - (1-v) - \frac{2}{3}(1-v)v) - 2v\sqrt{(1+\frac{v^2}{2})v}\right) \leq b^2(\frac{1}{2-cb} - 1).$$

In conclusion, it holds that,

$$F(a) \leq \max\left(F(a_{min}(c,b)), F(a_{max}(c,b))\right)$$

$$\leq \frac{1}{c}(\frac{cb^2}{2-cb} - cb^2).$$

$\square$

## L  OMITTED PROOFS ON CONTINUOUS APPROXIMATION

In this section we give a general approximation result (Theorem L.1) between a continuous-time flow (Equation 39) and a discrete-time (stochastic) iterates (Equation 40) in some compact subset of $\mathbb{R}^D$, denoted by $K$. This result is used multiple times in our analysis for full-batch SAM and 1-SAM. [7] Let $b : K \to \mathbb{R}^D$ is a $C_1$-lipschitz function, that is, $\forall x, x' \in K$, it holds that $\|b(x) - b(x')\|_2 \leq C_1 \|x - x'\|_2$. Let $b_k$ be mappings from $K$ to $\mathbb{R}^D$ for $k \in [M]$ satisfying that $b(x) = \frac{1}{M}\sum_{k=1}^{M} b_k(x)$ for all $x \in K$.

We consider the continuous-time flow $X : [0, T] \to K$, which is the unique solution of

$$dX(\tau) = b(X(\tau))d\tau. \tag{39}$$

and the discrete-time iterate $\{x(t)\}_{t\in\mathbb{N}}$ which approximately satisfy

$$x(t+1) \approx x(t) + pb_{k_t}(x(t)), \tag{40}$$

---

[7]Though we believe this approximation result is folklore, we cannot find a reference under the exact setting as ours. For completeness, we provide a quick proof in this section.

where $k_t$ is independently sampled from uniform distribution over $[M]$ for each $t \in \mathbb{N}$ and $x(t)$ is a deterministic function of $k_0, \dots, k_{t-1}$. We use $\mathcal{F}_t$ to denote the $\sigma$-algebra generated by $k_0, \dots, k_{t-1}$ and $\mathcal{F}_*$ to denote the filtration $(\mathcal{F}_t)_{t \in \mathbb{N}}$. Thus $x(t)$ is adapted to filtration $\mathcal{F}_*$. Note $b$ is undefined outside $K$, thus in the analysis we only consider the process stopped immediately leaving $K$, that is, $x^K(t) \triangleq x(\min(t, t_K))$, where $t_K \triangleq \{t' \in \mathbb{N} \mid x(t') \notin K\}$. If $x(t)$ is in $K$ for all $t \geq 0$, then $t_K = \infty$. It is easy to verify that $t_K$ is a stopping time with respect to the filtration $\mathcal{F}_*$. For convenience, we denote $X_K(\tau) = X(\min(\tau, pt_K))$ as the stopped continuous counterpart of $x^K$.

**Theorem L.1.** *Suppose there exist constants $C_2, \epsilon, \epsilon > 0$ satisfying that*

1. $\|b_k(x)\|_2 \leq C_2$, *for any $x \in K$ and $k \in [M]$;*
2. $\|b_k(x) - b(x)\|_2 \leq C_3$, *for any $x \in K$ and $k \in [M]$;*
3. $\left\|b_{k_t}(x(t)) - \frac{x(t+1) - x(t)}{p}\right\|_2 \leq \epsilon$, *for all $t$.*

*Then for any integer $0 \leq k \leq T/p$ and $0 < \delta < 1$, with probability at least $1 - \delta$, it holds that*

$$\max_{0 \leq t \leq T/p} \|x^K(t) - X^K(pt)\| \leq H_{p,\delta} e^{C_1 T},$$

*where $H_{p,\delta} \triangleq \|x(0) - X(0)\|_2 + C_1 C_2 Tp + 2C_3\sqrt{pT \log \frac{2eT}{\delta p}} + \epsilon T$.*

*Proof of Theorem L.1.* Summing up Equation 39 and Equation 40, for any $t \leq t_K$, we have that

$$X(pt) - X(0) = \int_{\tau=0}^{pt} b(X(\tau)) \mathrm{d}\tau, \tag{41}$$

and that

$$x(t) - x(0) = \sum_{t'=0}^{t-1} x(t'+1) - x(t') \tag{42}$$

Denote $\|x(t) - X(pt)\|_2$ by $E_t$, we have that for $t \leq t_K$,

$$E_t - E_0$$
$$\leq \left\| \int_{\tau=0}^{pt} b(X(\tau)) \mathrm{d}\tau - \sum_{t'=0}^{t-1} (x(t'+1) - x(t')) \right\|_2$$
$$\leq \underbrace{\left\| \int_{\tau=0}^{pt} b(X(\tau)) \mathrm{d}\tau - p\sum_{t'=0}^{t-1} b(X(pt')) \right\|_2}_{(A)} + \underbrace{\left\| p\sum_{t'=0}^{t-1} b(X(pt')) - p\sum_{t'=0}^{t-1} b(x(t')) \right\|_2}_{(B)}$$
$$+ \underbrace{\left\| p\sum_{t'=0}^{t-1} b(x(t')) - p\sum_{t'=0}^{t-1} b_{k_{t'}}(x(t')) \right\|_2}_{(C)} + \underbrace{\left\| p\sum_{t'=0}^{t-1} b_{k_{t'}}(x(t')) - \sum_{t'=0}^{t-1} (x(t'+1) - x(t')) \right\|_2}_{(D)}. \tag{43}$$

Below we will proceed by bounding the four terms (A), (B), (C) and (D) in Equation 43.

1. Note that for any $0 \leq \tau \leq \tau' \leq T$, we have that

$$\|X(\tau) - X(\tau')\|_2 = \left\| \int_{s=\tau}^{\tau'} b(X(s)) \mathrm{d}s \right\|_2 \leq \int_{s=\tau}^{\tau'} \|b(X(s))\|_2 \mathrm{d}s \leq (\tau' - \tau)C_2.$$

Thus, by $C_1$-lipschitzness of $b$,

$$(A) = \left\| \int_{\tau=0}^{pt} b(X(\tau)) - b(X(\lfloor \tau/p \rfloor p)) \mathrm{d}\tau \right\|_2 \leq \int_{\tau=0}^{pt} \|b(X(\tau)) - b(X(\lfloor \tau/p \rfloor p))\|_2 \mathrm{d}\tau$$
$$\leq C_1 C_2 p^2 t \leq C_1 C_2 pT.$$

2. By definition of $E_t$ and $C_1$-lipschitzness of $b$, we have that $(B) \leq C_1 p \sum_{t'=0}^{t-1} E_{t'}$.

3. We claim that for any $0 < \delta < 1$, we have that for probability at least $1 - \delta$, it holds that

$$(C) \leq 2C_3\sqrt{2pT\log\frac{2eT}{\delta p}}. \tag{44}$$

Below we prove our claim. We denote $p\sum_{t'=0}^{\min(t,t_K)-1} b(x(t')) - p\sum_{t'=0}^{\min(t,t_K)-1} b_{k_{t'}}(x(t'))$ by $S_t$, which is a martingale with respect to filtration $\mathcal{F}_*$, since $t_K$ is a stopping time. Note

$$\|S_t - S_{t+1}\|_2 \leq \max_{k\in[M],x\in K}\|b(x) - b_k(x)\|_2 \leq C_3,$$

by Azuma-Hoeffding's inequality (vector form, Lemma K.5), it holds that for any $0 \leq t \leq T/p$ and $0 \leq \delta \leq 1$, with probability at least $1 - \delta$,

$$\|S_t\|_2 \leq 2C_3 p\sqrt{2t\log\frac{2e}{\delta}}.$$

Applying an union bound on the above inequality over $t = 0, \ldots, \lfloor T/p \rfloor - 1$, we conclude that with probability at least $1 - \delta$, $(C) \leq 2C_3 p\sqrt{2T/p\log\frac{2eT}{\delta p}} = 2C_3\sqrt{2Tp\log\frac{2eT}{\delta p}}$.

4. We have that

$$(D) \leq p\sum_{t'=0}^{t-1}\left\|b_{k_{t'}}(x(t')) - \frac{x(t'+1) - x(t')}{p}\right\| \leq pt\epsilon \leq \epsilon T.$$

Combining the above upper bounds for (A), (B), (C) and (D), we conclude that for any $0 \leq t \leq \min(T/p, t_K)$,

$$E_t \leq H_{p,\delta} + C_1 p\sum_{t'=0}^{t-1} E_{t'}. \tag{45}$$

Applying the discrete gronwall inequality (Lemma K.6) on Equation 45, we have that

$$E_t \leq H_{p,\delta}e^{C_1 pt} \leq H_{p,\delta}e^{C_1 T},$$

which completes the proof. $\qquad\square$

**Corollary L.2.** *If $\min_{0\leq\tau\leq T}\text{dist}(X(\tau), \mathbb{R}^D \setminus K) > H_{p,\delta}e^{C_1 T}$, then with probability at least $1 - \delta$, $t_K > \lfloor T/p \rfloor$ and therefore*

$$\max_{0\leq t\leq T/p}\|x(t) - X(pt)\| \leq H_{p,\delta}e^{C_1 T}.$$

*Proof of Corollary L.2.* By Theorem L.1, we know with probability at least $1 - \delta$, we have that

$$\max_{0\leq t\leq T/p}\left\|x^K(t) - X^K(pt)\right\| \leq H_{p,\delta}e^{C_1 T}.$$

Therefore $\text{dist}(x^K(t), \mathbb{R}^D \setminus K) \geq \text{dist}(X^K(pt), \mathbb{R}^D \setminus K) - \text{dist}(X^K(pt), x^K(t)) > 0$ for any $0 \leq t \leq T/p$, which implies $x^K(t) \notin \mathbb{R}^D \setminus K$, or equivalently, $x^K(t) \in K$. Thus we conclude that $t_K \geq \lfloor T/p \rfloor$. $\qquad\square$

**Corollary L.3.** *Suppose $M = 1$ and there exist constants $C_2, \epsilon > 0$ satisfying that*

1. $\|b(x)\|_2 \leq C_2$ *for any $x \in K$;*
2. $\left\|b(x) - \frac{x(t+1)-x(t)}{p}\right\| \leq \epsilon$, *for all $x \in K$.*

*Then for any $k \in \mathbb{N}$ such that $kp \leq T$, it holds that*

$$\max_{0\leq t\leq T/p}\left\|x^K(t) - X^K(pt)\right\| \leq H_p e^{C_1 T},$$

*where $H_p \triangleq \|x(0) - X(0)\|_2 + C_1 C_2 Tp + \epsilon T$.*

*Therefore, similar to Corollary L.2, if $\min_{0\leq\tau\leq T}\text{dist}(X(\tau), \mathbb{R}^D \setminus K) > H_p e^{C_1 T}$, then it holds that $t_K > \lfloor T/p \rfloor$ and that*

$$\max_{0\leq t\leq T/p}\|x(t) - X(pt)\| \leq H_{p,\delta}e^{C_1 T}.$$

*Proof of Corollary L.3.* For any $\delta \in (0, 1]$, choosing $C_3 = 0$ and by Theorem L.1, we have that

$$\mathbb{P}\left[\max_{0 \leq t \leq T/p} \left\| x^K(t) - X^K(pt) \right\| \leq H_p e^{C_1 T}\right] \geq 1 - \delta \,.$$

Since $\delta$ can be any number in $(0, 1]$, the above probability is exactly 1. $\qquad\square$

We end this section with a summary of applications of Theorem L.1 and corollary L.3 in our proofs (Table 2).

| Setting | $p$ | $b_k$ | $\epsilon$ |
|---|---|---|---|
| Full-batch SAM, Phase I (Lemma I.4) | $\eta$ | $-\nabla L(\cdot)$ | $\rho$ |
| Full-batch SAM, Phase II (Theorem I.3) | $\eta\rho^2$ | $-\partial\Phi(\cdot)\nabla\lambda_1(\nabla^2 L(\cdot))/2$ | $\rho + \eta$ |
| 1-SAM, Phase I (Lemma J.3) | $\eta$ | $-\nabla L_k(\cdot)$ | $\rho$ |
| 1-SAM, Phase II (Theorem J.2) | $\eta\rho^2$ | $-\partial\Phi(\cdot)\nabla\text{Tr}(\nabla^2 L_k(\cdot))/2$ | $\rho + \eta$ |

Table 2: Summary of applications of Theorem L.1 and corollary L.3 in our analysis

