# OpenReview forum: "How Sharpness-Aware Minimization Minimizes Sharpness?"
_ICLR.cc/2023/Conference — ICLR 2023 poster_

### Official Review · Reviewer_G4Gz · 2022-10-24

**Confidence:** 3
**Correctness:** 3
**Technical Novelty And Significance:** 2
**Empirical Novelty And Significance:** Not applicable
**Recommendation:** 6

**Clarity, Quality, Novelty And Reproducibility:**

Clarity: notation could be improved a lot and there could be an effort from the authors to make the results more accesible and readable to a more general public. There are many typos and things that are not properly defined in the appendix that makes it difficult to review the results. The second paragraph in the introduction is hard to read because at that point the reader doesn't know anything about the content and definitions in the paper that the authors refer to.

Quality: The final results need a lot of technical proofs. The main text feels rushed, there should be more care in the appendix regarding notation.

Originality: the relation between sharpness and the eigenvalues of the Hessian appear to be somewhat known. From the original work of Keskar et al. 2017 (end of page 6):
> Note that Metric 2.1 is closely related to the spectrum of $ \nabla^2 f(x)$. Assuming $\epsilon$ to be small enough, when $A = I$, the value (4) relates to the largest eigenvalue of $ \nabla^2 f(x)$ and when $A$ is randomly
sampled it approximates the Ritz value of $ \nabla^2 f(x)$ projected onto the column-space of $A$.

Also see https://arxiv.org/pdf/2206.10654.pdf (June 2022) page 3:
> While the final SAM procedure (their Algorithm 1) does not directly penalize $\lambda_{\text{max}}$, the idealized regularizer in their Equation 1 is, at a local or global minimum, equivalent to the maximum Hessian eigenvalue, if one assumes that
the training objective is well-approximated by its second-order Taylor approximation.

Hence, this leads me to believe that some (not all) of the results here are already understood. The following is *concurrent work* so it should not be taken into account for the decision regarding this submission, but it illustrates that the convex quadratic case leads to more understandable results while still requiring careful work: https://arxiv.org/pdf/2210.01513.pdf

Lemma 3.3 in https://arxiv.org/pdf/2203.08065.pdf is equivalent to Theorem 4.5

Reproducibility: the work only contains theoretical contributions.

**Strength And Weaknesses:**

The main weakness is that the notation and results are presented with really complicated math, hurting the readability. The results are basically a study on sharpness aware minimization on quadratic functions (in this setting the results are more clear and readable), and the added complexity comes from extrapolating such results to the general case by using a second-order Taylor expansion. I will list some further issues:

1. Definition of $U$ in page 3 should be made clear as it appears in other statements. This is problematic as theorem 4.3 which is supposed to be important, is no longer self-contained which hurts readability and clarity
2. The definition of the "Limiting regularizer", as it contains a limit when $\rho \to 0$, simply has the goal of making the second order Taylor approximation to have a vanishing error. This facilitates the analysis and makes the function behave like a quadratic function $L(x)=x^\top A x$. Indeed all the arguments become simple and clear if one just studies this family of functions. Most of the results presented like Theorems 4.5, 4.6 and 4.7 follow easily in this case. The arguments dealing with the approximation error become highly technical and confusing. This hurts readability. I would say the authors could expose the results more clearly if they just explained the arguments for the case of quadratic functions, and leave all the technicalities of the approximation to the appendix.
3. Another trick the authors use for simplifiyng the analysis is assuming that the solution of the regularized objective is also a solution of the unregularized objective. However this is never clearly stated, and is probably not true except in the limit when $\rho \to 0$ (in which case there is no sharpness-aware minimization). This leads to confusing claims like the ones in the paragraph after equation (5): "The first order term vanishes when we try to minimize the *regularized objective* because every minimizer of the *original loss* has zero gradient". Again, this is potentially false as there is no reason to believe that a solution of the regularized objective is also a solution of the unregularized objective.
4. Also in the paragraph following (5) the authors claim "both the first order terms are the same since $\sup_{\|v\| \leq 1} v^\top \nabla L(x) = \| \nabla L(x) \|_2$", however this is not true as the supremum in equation (4) contains a second order term of the form $v^\top \nabla L(x) v$, hence the supremum is not necessarily the supremum of the first term only. This claim is misleading.
5. Issues with notation: In page 4 a regularizer is defined as a positive function as implied by the notation $R_\rho: \mathbb{R}^D \to \mathbb{R}^+ \cup \{\infty\}$, however immediately after this they discuss regularizers that might be negative? the notation should be fixed.
6. In definition 4.2 it should be noted that $S$ can take the value $\infty$ in some cases. Furthermore there is a typo $\epsilon > 0, \epsilon > 0$.
7. The appendix is rather careless with a lot of notation that is not introduced with care. For example already at the first page of the appendix (page 13) we have in Lemma B.5 some letter $C$ that then does not appear at all in the body of the lemma. After a lot of thinking I assume the authors meant $K$ instead of $C$. But the problem does not stop there. Item 3 in lemma B.5 states that $L$ is $\mu$-PL in some set $K^r$ where $K$ is some compact set. This is clearly false in general. I had to do my best effort to track the issue and indeed $K$ is not an arbitrary compact set $K \subseteq \Gamma$ but has a particular structure that is defined in the reference Arora et al. 2022, but which is not reproduced here. Indeed I had to go to the appendix in that reference to decipher that $K$ here corresponds to $Y$ in the reference, which is defined as "the trajectory of limiting flow Equation (4)", none of which is clarified in the appendix of this work. I believe the authors simply copied and pasted without care. Hence, the beginning of the appendix is already unreadable and hard to review.
8. Lemma B.6 has the same issue where there is some letter $C$ that actually refers to $K$ (I believe).
9. To me, it is not clear how the second order Taylor approximation can be used for $L^\text{asc}$ as in equation (5): Note that this is only possible in a neighborhood of points where $\nabla L (x) \neq 0$, as this term appears as a denominator. However the authors then start arguing that they study points where $\nabla L(x)$ vanishes? I don't think this is really rigorous as equiation (5) assumes that the taylor expansion has an error that is uniformly bounded by the term $\mathcal{O}(\rho^3)$ but in fact this term can depend on $x$.
10. Assumption 4.1 holds trivially as soon as the regularizer is continous, as any continuous function attains a minimum over a compact set. $R^\max$ is continuous. Hence it is trivial that assumption 4.1 holds and it is not necessary to go over the complicated argument in Step 1 in the proof of theorem 4.5 (appendix page 17). The same goes for theorem 4.7.
11. There are some missing citations and some of the results like theorem 4.5 and 4.7 might be well-known, see the clarity/quality/novelty/reproducibility section for details.

Another weakness is that there is no clear impact from the results presented. Even though they are interesting, what practical conclusions can we draw from this? Having a better understanding of SAM could be achieved by studying the case of quadratic functions, but the heavy math to extrapolate the results to the general case through Taylor approximation, at least in my case, does not improve my understanding of what SAM does further. In any case the results hold in the regime where $\rho \to 0$ which means when the regularization is almost negligible. I think this will be of interest only in a small niche in the community. I would say the authors could achieve more impact and more interest if they would rewrite the results with simpler notation and just exposing the core ideas in the context of quadratic functions, while relegating the notation-heavy stuff to the appendix in case someone wants to delve further.

**Summary Of The Paper:**

The paper studies the properties of sharpness aware minimization through a second order Taylor approximation, in the setting where the perturbation radius tends to zero. Under many strong assumptions, some sort of equivalence (in the limit) between different notions of sharpness-aware-minimization and the spectra of the Hessian of the loss function is derived. The stochastic setting is also studied where two common notions of SAM are shown to be equivalent. The main ideas of the proofs follow the arguments in the particular case of quadratic functions, which are then extended to the general case by means of a second order Taylor approximation. Some of the results are in fact presented only for the quadratic case.

**after rebuttal**
Authors have made an effort to address some of my concerns. Although I do not feel all of them have been addressed in full, I am not opposed to it being accepted in current form. I can increase my score to "marginally above acceptance threshold". The main drawback of this paper is that its hard to read and the appendix is so long and full of equations that **it's unrealistic that anyone will review it in full for a conference cycle**. Hence, my score should be seen as reflecting the fact that the main 9 pages present a coherent story with (potentially) significant results, but **trusting their validity is left to the authors and potential readers**. As a result, I will also decrease my confidence.

**Summary Of The Review:**

The paper requires a good effort to improve readability and be more accessible to a wider audience. The appendix was not carefully written which makes it hard to review. Some of the results might be already understood, but there is no mention of the works that already establish some relation between sharpness and the spectra of the Hessian.

---

> ### Author Response · Authors · 2022-11-17
> **Response to Reviewer G4Gz (1)**
>
> We thank Reviewer G4Gz for the careful and detailed review as well as the constructive feedback. Before responding to the detailed questions, we first want to clarify the originality and impact of our work.
>
> **Contributions and Originality:** We stress that our **main contributions** are the **implicit bias of SAM** in both the full-batch setting and stochastic setting with batch size one, but not the explicit bias of various sharpness-aware losses.   We disagree with your comments "The main ideas of the proofs follow the arguments in the particular case of quadratic functions, which are then extended to the general case by means of a second order Taylor approximation." because our analysis for the implicit bias of SAM does not follow from a straightforward extension from quadratic case to the general case. In the quadratic case, the Hessian is always constant and there is no room for implicit bias of penalizing for either trace of Hessian or the largest eigenvalue of Hessian.
>
> With that being said, we agree with the reviewer that limiting form of the explicit bias for worst-direction sharpness and average-direction sharpness in terms of the spectrum of Hessian is more-or-less already known and that they can be derived in a much simpler setting (no need to assume a manifold). We have already added the missing references. We also kindly note that the manifold assumption is mainly needed to derive the explicit bias of ascent-direction sharpness (which is a novel result) and to align with the setting of our implicit bias results so we can conclude that for small perturbation radius $\rho$, full-batch SAM is not only minimizing the top eigenvalue on the manifold of minimizers but also penalizing the original notion of worst-direction sharpness as shown in Corollary 4.10 and 4.11. (Same for stochastic SAM)
>
> ------
>
> **Impact:** Our analysis explains why there could be a difference in generalization quality between SAM with different batch sizes, which is only observed experimentally in previous works but lacks a theoretical explanation. We show that full-batch SAM and stochastic SAM with batch size one correspond to different implicit regularizations, which could be the starting point of future research on understanding the different generalization benefits of different Hessian-based regularizers. Our research also settles the mystery of why SAM can still minimize the worst-discretion sharpness despite the two heuristic approximations made in its derivation.
>
> Finally, we disagree with the reviewer that "In any case the results hold in the regime where $\rho\to 0$ which means when the regularization is almost negligible" in two aspects. First, as shown by our Theorem 4.3, even for arbitrarily small $\rho$, minimizing the regularized loss implies minimizing the limiting regularizer on the manifold, which has a non-trivial regularization effect. Second, so far all the theoretical analysis on SAM uses Taylor expansion, including the original derivation of SAM, which means the analysis is in the regime $\rho\to 0$ because Taylor expansion does not make sense for constant $\rho$. Still, though the analysis only holds rigorously for sufficiently small $\rho$, our toy example in Figure 1 shows that our theoretical prediction actually can happen even in standard training settings where we choose $\rho = 0.2$ for full-batch and stochastic SAM experiments and $\rho = 0.05$ for the ascent-direction Gradient Descent experiment.

---

> > ### Author Response · Authors · 2022-11-17
> > **Response to Reviewer G4Gz (2)**
> >
> > Below we respond to the reviewer's questions one by one.
> >
> > **Q1:** Definition of $U$ on page 3 should be made clear.
> >
> > **A1:** Thank you for your constructive feedback. We've put the definition in a definition environment to highlight it.
> >
> > **Q2:** The results can be better presented in quadratic case.
> >
> > **A2:** The main reason we introduce the manifold setting is to analyze the implicit bias of the particular optimization algorithm, SAM. Please refer to **More Discussions of Manifold Assumption** in our general response for the importance of the assumption for our analysis.
> >
> > **Q3:** It is wrong to assume "the solution of the regularized objective is also a solution of the unregularized objective".
> >
> > **A3:** You are correct that the above is wrong, but we never assumed that in our proof. We do apologize for the confusion caused by the sentence "The first order term vanishes when we try to minimize the regularized objective because every minimizer of the original loss has zero gradient". Here by "vanishes", we meant that when we try to optimize the regularized objective, the solution of the regularized objective will get very close to the minimizer of the unregularized objective and the first order term may not be the leading term, in which sense the first order term "vanishes" and only the second order term in Taylor expansion is kept. Indeed, taking $\epsilon =0$, our Theorem 4.3 implies that any $o(\rho^2)$ approximate minimizer of the regularized loss is also an $o(\rho^2)$ approximate minimizer of the unregularized loss, namely it is $o(\rho)$ close to the manifold of minimizers of unregularized loss. In other words, the first order term is $o(\rho^2)$ and thus smaller than the second order term, which justifies the above explanation of "vanishing".
> >
> > we have modified the corresponding paragraph to avoid confusion.
> >
> > **Q4:** The first order term in Equations (4) and (5) are not the same.
> >
> > **A4:** The above claim is actually correct because taking the second order term into the maximization only changes the first order term by $O(\rho^2)$. To see this, let $v^*=\mathrm{argmax}_{\|v\|\le 1} L(x+\rho v)$, and it holds that $ L(x+\rho v^*)\ge L(x+\rho \nabla L(x) / \|\nabla L(x)\|_2) = L(x) + \rho \|\nabla L(x)\|_2+O(\rho^2)$.  Note that $ L(x+ \rho v^*) = L(x) + \rho \langle v^*, \nabla L(x) \rangle + O(\rho^2)$, we conclude that $\rho\|\nabla L(x)\|_2 -O(\rho^2)\le \rho \langle v^*, \nabla L(x) \rangle\le \rho\|\nabla L(x)\|_2$.
> >
> > We stress that our original argument in this section is intuitive and is not meant to be rigorous. Please see the appendix for rigorous proof.
> >
> > **Q5:** Undefined Notations in Appendix.
> >
> > **A5:** We apologize for the poor writing in the appendix. We've fixed this in the revision.
> >
> > **Q6:** The lemma by Arora et al. (2022) does not apply to an arbitrary compact set $K \in \Gamma$.
> >
> > **A6:** Though the original lemma in Arora et al. (2022) only applies to a compact set with a particular structure, i.e., the finite-time solution of the ODE, it turns out their proof directly applies to a general compact set because the only property of the solution of the ODE in a finite interval is its compactness. We have added a discussion in the revised Appendix C.1.
> >
> > Finally, we want to express our appreciation again for the effort made by Reviewer G4Gz including checking our definitions and proofs carefully and even going back to previous works. This is helpful in terms of improving the quality of the presentation of this paper.

---

> > > ### Author Response · Authors · 2022-11-17
> > > **Response to Reviewer G4Gz (3)**
> > >
> > > **Q7:** The second order Taylor approximation for $L^{\text{asc}}$ is ill-defined for minimizers.
> > >
> > > **A7:** You are correct and we never apply this Taylor expansion at minimizers where the gradient is zero. Note here we don't need the normalized gradient $\frac{\nabla L(x)}{\|\nabla L(x)\|}$ to be Lipschitz in $x$, but instead, we only need $\frac{\nabla L(x)}{\|\nabla L(x)\|}$ and third order derivative of loss $L$ to be upper bounded, which is true when $x$ is chosen from any compact set (excluding those points with zero gradients).
> > >
> > >
> > > **Q8:** Equation (5) is not rigorous as the Taylor expansion in it assumes the error is uniformly $O(\rho^3)$ but it indeed depends on $x$.
> > >
> > > **A8:** The opening argument in Section 4 including Equation (5) is intuitive and is not meant to be rigorous. But Equation (5) can be made rigorous if $x$ is taken from a compact set excluding minimizers, as the setting of Theorem 4.3.
> > >
> > > **Q9:** Assumption 4.1 is trivial for continuous regularizers.
> > >
> > > **A9:** The constant in $O(\cdot)$ is independent of $\rho$ and thus this is a non-trivial assumption.
> > >
> > > **Q10:** Missing citations for theorems like 4.5 and 4.7
> > >
> > > **A10:** Thanks for pointing out these references. We have included them in the revision in the penultimate paragraph in Section 4.1.
> > >
> > > **Q11:** Comparison to concurrent work Bartlett et al. (2022).
> > >
> > > **A11:** Bartlett et al. (2022) prove that on quadratic loss, the iterate of SAM (Equation (11)) and its gradient converges to the top eigenvector of hessian, which is almost the same as our Theorem 4.14. Assuming such alignment for a general loss, the work of Bartlett et al. (2022) shows that the largest eigenvalue of hessian decreases in the next step. This paper also proves such a Hessian-gradient alignment for general loss functions and an end-to-end theorem showing that the largest eigenvalue of Hessian and worst-direction sharpness decrease along the trajectory of SAM (Theorem 4.9), which are not shown in Bartlett et al. (2022). Moreover, this paper also characterizes the implicit bias of stochastic SAM with batch size 1, which is minimizing the average-direction sharpness, while Bartlett et al. (2022) only consider the deterministic case.
> > >
> > >
> > > ------
> > > **References**
> > >
> > > - Sanjeev Arora, Zhiyuan Li, and Abhishek Panigrahi. Understanding gradient descent on edge of stability in deep learning. arXiv preprint arXiv:2205.09745, 2022
> > > - Peter L Bartlett, Philip M Long, and Olivier Bousquet. The dynamics of sharpness-aware minimization: Bouncing across ravines and drifting towards wide minima. arXiv preprint arXiv:2210.01513, 2022.

---

> > ### Comment · Reviewer_G4Gz · 2022-11-17
> > **Acknowledgment of Rebuttal**
> >
> > I have read your rebuttal and I will consider it for updating my review. Some quick comments:
> >
> > 1. Given that you emphasize that your contributions are theoretical and rigorous, I find it surprising that you now say some arguments are supposed to be intuitive and not rigorous. I think it would make things more clear if you state when an argument is supposed to just give some intuition vs being a real argument.
> > 2. I do agree that it appears that the results in the stochastic setting are the main contributions here, it would have been more clear if this was highlighted. Given that those are the last results to be presented, from the perspective of the reader it looks like it is not the case.
> > 3. I pointed out to the work by Bartlett not necessarily asking for a comparison (which would be a great addition) but to show how by studying the apparently "easier" quadratic case, the results are still interesting but are much easier to write rigorously (the notation is cleaner too). This is inline with my main criticism that if the goal is to understand what SAM does, it is more important that the paper is readable by a wide audience rather than to be as general as possible (which complicates notation and makes it harder to go over the proofs)
> >
> > I will try to re evaluate the paper in light of the new edits and your comments.

---

> > > ### Author Response · Authors · 2022-11-19
> > > **Response to Reviewer G4Gz**
> > >
> > > Thank you for your constructive feedback! We will continue to improve the writing quality and clarity in future versions. We agree that the result in the quadratic case admits a much easier and readable proof, which can be found in our section C.4. However, a complete characterization of the sharpness-reduction effect of SAM cannot be done on a quadratic loss, because it has constant Hessian. Thus we decide to work with a general loss.
> > >
> > > Due to the 9-page space limits, we defer the rigorous full proofs of our theorems into the appendix and only sketch our high-level ideas in a non-rigorous way in the main text, which can still be made rigorous with more careful definition. We will adopt your advice on clearly stating state whether an argument is supposed to be intuitive or fully rigorous in future versions.

---

### Official Review · Reviewer_ReRk · 2022-10-26

**Confidence:** 3
**Clarity, Quality, Novelty And Reproducibility:** Good clarity, high quality, and suffi…
**Correctness:** 3
**Technical Novelty And Significance:** 3
**Empirical Novelty And Significance:** 3
**Recommendation:** 8

**Strength And Weaknesses:**

**Strengths:**

* The paper is very well-written. I enjoy reading this paper. Instead of stating the theoretical stuff alone, they elaborate various notions and theorems with the high-level idea throughout the paper.

* Three notions of sharpness, their explicit biases, and correlations to SAM discovered in this paper are interesting and could stimulate future research in this area.

* The toolboxes used in the analysis are interesting, which borrow ideas from the recent literature on edge of stability in SGD.

**Weakness:**

* It is not very clear why assuming the minimizers of loss form a manifold following (Fehrman et al. 2020; Li et al. 2021; Arora et al. 2022) is important. What is the limitation and advantage of this assumption? Please clarify.
* The proof techniques largely follow (Arora et al. 2022). So it would be helpful to clarify the new challenges and how they are being solved in this paper.
* In Section 5.1, it is not clear why the Hessian of each individual loss is rank-1 although I am aware of the Hessian of the total loss is rank-M. Does the individual loss need to be independent?
* This paper is mainly theoretical sound, but it can benefit from more empirical validations beyond the toy example. For example, adding a real simulation to verify different convergence properties of deterministic and stochastic SAM will be useful. If not, adding references that can support the theoretical results is also helpful.

Minor issues:
* Sharpness in stochastic setting seems not well defined.



**Summary Of The Paper:**

Despite the success of SAM in many applications, the existing theory for SAM is still insufficient to explain its impressive empirical performance. To bridge the gap, this paper rigorously proposes three types of sharpness: worst-direction, ascent-direction, and average-direction, which correspond to the notions of sharpness used in the objective, the actual SAM implementation, and the term used in the generalization analysis of SAM, respectively. They demonstrate that these three types of sharpness lead to different explicit biases in the deterministic case. Moreover, they show that the full-batch SAM minimizes the worst-direction sharpness though it adopts the inexact computationally efficient variants in realization, owing to the implicit alignment between the gradient and the maximal eigenvalue of Hessian. Besides, stochastic SAM with batch size 1 minimizes the average-direction sharpness, which directly benefits the generalization performance.

**Summary Of The Review:**

This paper is novel and solid, which solves an open question in the theoretical area and provides a new aspect in sharpness and generalization analysis.

---

> ### Author Response · Authors · 2022-11-17
> **Response to Reviewer ReRk**
>
> We thank the reviewer for the constructive feedback and appreciation for our results. Below we answer your questions one by one.
>
>
> **Q1:** Limitation and advantages of manifold assumption
>
> **A1:** Please see **More Discussions of Manifold Assumption** in our general response.
>
> **Q2:** Clarify the difference in proof techniques between this paper and Arora et al.(2022).
>
> **A2:** First, Arora et al. (2022) only deal with the deterministic case, while our analysis extends to stochastic SAM as well (sec 5). Second, our analysis for the deterministic case is different from that of Arora et al.(2022) in the following two aspects:
>
> - The alignment analysis is more complicated in this paper because we have two hyperparameters, LR $\eta$ and perturbation radius $\rho$, while Arora et al.,(2022) only need to deal with one hyperparameter, LR $\eta$.
>
> - The mechanism of penalizing worst-direction sharpness is different, which can be seen from the dependency of the sharpness-reduction rate over LR $\eta$.  In Arora et al.(2022), normalized GD reduces the sharpness via a second-order effect of GD and thus the sharpness is reduced by $O(\eta^2)$ per step. In our analysis, for fixed small perturbation radius $\rho$, the sharpness is reduced by $O(\rho^2 \eta)$ per step, which is linear in $\eta$.
>
>
>
>
> **Q3:** Clarify why the Hessian of each individual loss is rank 1.  Does the individual loss need to be independent?
>
> **A3:** This is proved in Lemma 5.3.  The proof idea is that under our setting of regression problems in Sec 5, $\nabla^2 L_k(x) = \ell'(f_k(x),y_k) \nabla^2 f_k(x) + \ell''(f_k(x),y_k) \nabla f_k(x) (\nabla f_k(x))^\top $, where $\ell'$ and $\ell''$ denote the first and second order derivative of $\ell$ over its first variable. Thus the Hessian is at most rank-1 when $f_k(x)$ is the minimizer of $\ell(\cdot,y_k)$, because $ \ell'(L_k(x),y_k)=0$ in this case. Finally, $\nabla^2 L_k(x)$ at least rank-1, because by Assumption 3.1, $\sum_{k=1}^M \nabla^2 L_k(x)$ is rank-$M$.
>
>
> The data doesn’t need to be independently sampled. The randomness in the high-probability bounds is with respect to initialization and the randomness in the algorithm (i.e., at each iteration, the training data is uniformly sampled).
>
> However, we forgot to mention in the setting of section 5 that global minimizers in $\Gamma$ can actually interpolate all the data, i.e., $f_k(x)=y_k$ for each $k\in[M]$. We fixed this in the revision.
>
> **Q4:** Sharpness in the stochastic setting is undefined.
>
> **A4:** Thanks for pointing it out. We defined it in the revision in the first paragraph of Section 5.1. We use $L_{k,\rho}^{\text{Max}}$,$L_{k,\rho}^{\text{Asc}}$ to denote the corresponding sharpness-aware
> loss for $L_k$ and $R_{k,\rho}^{\text{Max}}$,$R_{k,\rho}^{\text{Asc}}$ to denote the corresponding sharpness for $L_k$.
>
> ------
> **References**
>
> - Sanjeev Arora, Zhiyuan Li, and Abhishek Panigrahi. Understanding gradient descent on edge of stability in deep learning. arXiv preprint arXiv:2205.09745, 2022

---

### Official Review · Reviewer_pjDp · 2022-10-27

**Confidence:** 3
**Clarity, Quality, Novelty And Reproducibility:** This paper is easy to follow.
**Correctness:** 3
**Technical Novelty And Significance:** 3
**Empirical Novelty And Significance:** 1
**Recommendation:** 6

**Strength And Weaknesses:**

Strength
1. This paper categorizes the SAM in three ways, and they explain each method in detail.
2. The paper claims the novel findings and their explicit bias.
3. There are no experimental results for the real-world dataset, but they provide an intuitive toy experiment to support the claim.

Weakness and Questions
1. There is no experiment for the real-world datset.
2. This paper raises a new perspective to analyze the SAM. It would be more interesting if the paper provided a new approach from a new perspective.
3. This paper claims that batch size=1 SAM corresponds to the average-direction sharpness. In the SAM paper, they introduce the m-sharpness, and m=1 shows a strong performance improvement compared to the full-batch SAM.
I wonder about the author's opinion about the average-direction sharpness that performs better than worst-direction sharpness in real-world experiments.

**Summary Of The Paper:**

Sharpness-aware training has been widely adopted for its generalization performance.
This paper provides the theoretical analysis for sharpness-aware minimization (SAM).
This paper categorizes the SAM in three ways, worst-direction, ascent-direction, and average direction.
Besides, they provide explicit bias for each way.
The paper is clear and theoretically sound.

**Summary Of The Review:**

I carefully read the paper and the main statement of the theorem.

---

> ### Author Response · Authors · 2022-11-17
> **Response to Reviewer Pjdp**
>
> We thank the reviewer for the effort and time devoted to the review. Please see our detailed response below.
>
> **Q1:** Why does SAM with batch size one (average-direction sharpness) perform better than full-batch SAM (worst-direction sharpness) in real-world experiments?
>
> **A1:** It is still open whether the trace of Hessian is always a better regularizer than the top eigenvalue of Hessian, and why it is better (if it is the case). This is an interesting and important question and we leave it for our future work. There are some results demonstrating the good regularization effect of the trace of Hessian for simple models, e.g.,   [Orvieto et al., 2022](https://arxiv.org/abs/2206.04613) and the analysis in Section 6 of [Li et al., 2022](https://openreview.net/forum?id=siCt4xZn5Ve). However, to our best knowledge, there is no separation shown between the regularization effects of the trace of Hessian and the top eigenvalue of Hessian.
>
> ------
> **References**
>
> - Orvieto, Antonio, Anant Raj, Hans Kersting, and Francis Bach. "Explicit Regularization in Overparametrized Models via Noise Injection." arXiv preprint arXiv:2206.04613 (2022).
>
> - Li, Zhiyuan, Tianhao Wang, and Sanjeev Arora. "What Happens after SGD Reaches Zero Loss?--A Mathematical Framework." arXiv preprint arXiv:2110.06914 (2021).

---

### Author Response · Authors · 2022-11-17
**General Comments for All Reviewers**

We thank all the reviewers for their careful and thoughtful reviews. Since more than one reviewer (Reviewers ReRk and G4Gz) have asked about our manifold setting, we provide more discussions on it below.

**More Discussions of Manifold Assumption:**

- **Validity**. Recent empirical studies have shown that there are essentially no barriers in loss landscape
between different minimizers, that is, the set of minimizers is path-connected (Draxler et al.,
2018; Garipov et al., 2018). Motivated by this empirical discovery, we make the assumption below
following Fehrman et al. (2020); Li et al. (2021); Arora et al. (2022), which is theoretically justified
by Cooper (2018) under a generic setting.

- **Importance**. The connectivity of the set of local minimizers implied by the manifold assumption above allows
us to take limits of perturbation radius $\rho\to 0$ while still yielding interesting and insightful implicit
bias results. So far almost all analysis of implicit bias for general model
parameterizations relies on Taylor expansion, e.g. Blanc et al. (2019); Damian et al. (2021); Li et al.
(2021); Arora et al. (2022), so does the derivation of the SAM algorithm Foret et al. (2020); Wu
et al. (2020). Thus it’s crucial to consider small perturbation size $\rho$. On the contrary, if the set of
global minimizers is a set of discrete points, then with a small perturbation radius $\rho$, the implicit bias of
optimizers is not sufficient to drive the iterate from the global minimum to another one.

	It can be shown that for a minimum loss manifold, the rank of Hessian plus the dimension of the manifold is at most the environmental dimension D, and thus our assumption about Hessian rank essentially says the rank is maximal. This assumption is necessary for the analysis to guarantee the differentiability of $\Phi$.

- **Limitation**. As proved by Cooper (2018), the manifold setting holds generically for regression problems with smooth loss. The manifold assumption doesn't hold for networks with ReLU (which is non-smooth) and classification problems with softmax loss (where an overparametrized network may achieve arbitrarily small but non-zero loss). In the latter case, there may not even exist a global minimizer, not to mention a manifold of minimizers.

------
**References**

* Felix Draxler, Kambis Veschgini, Manfred Salmhofer, and Fred Hamprecht. Essentially no barriers in neural network energy landscape. In International conference on machine learning, pp. 1309–1318. PMLR, 2018

* Timur Garipov, Pavel Izmailov, Dmitrii Podoprikhin, Dmitry P Vetrov, and Andrew G Wilson. Loss surfaces, mode connectivity, and fast ensembling of dnns. Advances in neural information processing systems, 31, 2018

* Benjamin Fehrman, Benjamin Gess, and Arnulf Jentzen. Convergence rates for the stochastic gradient descent method for non-convex objective functions. Journal of Machine Learning Research, 21:136, 2021

* Alex Damian, Tengyu Ma, and Jason Lee. Label noise sgd provably prefers flat global minimizers, 2021

* Zhiyuan Li, Tianhao Wang, and Sanjeev Arora. What happens after sgd reaches zero loss? a mathematical framework. In International Conference on Learning Representations, 2021

* Yaim Cooper. The loss landscape of overparameterized neural networks. arXiv preprint arXiv:1804.10200, 2018

* Guy Blanc, Neha Gupta, Gregory Valiant, and Paul Valiant. Implicit regularization for deep neural networks driven by an ornstein-uhlenbeck like process. arXiv preprint arXiv:1904.09080, 2019

* Sanjeev Arora, Zhiyuan Li, and Abhishek Panigrahi. Understanding gradient descent on edge of stability in deep learning. arXiv preprint arXiv:2205.09745, 2022

- Pierre Foret, Ariel Kleiner, Hossein Mobahi, and Behnam Neyshabur. Sharpness-aware minimization for efficiently improving generalization. arXiv preprint arXiv:2010.01412, 2020

- Dongxian Wu, Shu-Tao Xia, and Yisen Wang. Adversarial weight perturbation helps robust generalization. Advances in Neural Information Processing Systems, 33:2958–2969, 2020

---

### Author Response · Authors · 2022-11-17
**Summary of Changes in the Revision**


We have uploaded a revision to address the issues and incorporate the feedback from the reviewers. Besides fixing the typos and polishing the writing (including the proofs in the appendix), we make the following major changes:

* In Section 3, we highlight and clarify the definition of $U$ (in response to **Reviewer G4Gz**).
* In Section 4.1, we add citations to references on the connection between worst- and average-direction sharpness and the spectrum of Hessian pointed out **Reviewer G4Gz** in the penultimate paragraphs.
* In Section 5, we clarify our setting and define stochastic worst- and ascent-direction sharpness formally (in response to **Reviewer ReRk**).
* In Appendix A, we add a comparison with the concurrent work of Bartlett et al. (**in response to Reviewer G4Gz**) and a comparison with Arora et al. in terms of proof techniques (in response to **Reviewer ReRk**).
* In Appendix B, we justify our manifold assumption in response to **Reviewer ReRk and G4Gz**.
* We rewrite Section C.1 (original Section B.1) for more clear notations in response to **Reviewer G4Gz**.

We also change our title to "How Does Sharpness-Aware Minimization Minimize Sharpness?".

------
**References**

- Peter L Bartlett, Philip M Long, and Olivier Bousquet. The dynamics of sharpness-aware minimization: Bouncing across ravines and drifting towards wide minima. arXiv preprint arXiv:2210.01513, 2022.

- Sanjeev Arora, Zhiyuan Li, and Abhishek Panigrahi. Understanding gradient descent on edge of stability in deep learning. arXiv preprint arXiv:2205.09745, 2022

---

### Decision · Program_Chairs · 2023-01-20

**Decision:**

Accept: poster

**Justification For Why Not Higher Score:**

The paper is very technical and it is relevant to a specific optimization algorithm that, while famous, is not actually widely used. That said, I am also fine if this is bumped up.

**Justification For Why Not Lower Score:**

The paper is a clear accept.

**Metareview: Summary, Strengths And Weaknesses:**

The paper proposes a theoretical analysis of Sharpness-Aware Minimization algorithm, characterizing the exact notions of sharpness that the algorithm regularizes.

All the reviewer agreed that this is an interesting paper with solid theoretical results and so the paper should be accepted at the conference.

**Note From Pc:**

if the above contains the word "oral" or "spotlight" please see: "oral" presentation means -> notable-top-5% and "spotlight" means -> notable-top-25%. As stated in our emails, we are disassociating presentation type from AC recommendations